# ReLaSH: Reconstructing Joint Latent Spaces for Efficient Generation of Synthetic Hypergraphs with Hyperlink Attributes

**Feiyan Ma, Shihao Wu, Gongjun Xu, Ji Zhu**
Department of Statistics
University of Michigan
Ann Arbor, MI 48109, USA
`{feiyanma,wshihao,gongjun,jizhu}@umich.edu`

## Abstract

Hypergraph network data, which capture multi-way interactions among entities, have become increasingly prevalent in the big data era, spanning fields such as social science, medical research, and biology. Generating synthetic hyperlinks with attributes from an observed hypergraph has broad applications in data augmentation, simulation, and advancing the understanding of real-world complex systems. This task, however, poses unique challenges due to special properties of hypergraphs, including discreteness, hyperlink sparsity, and the mixed data types of hyperlinks and their attributes, rendering many existing generative models unsuitable. In this paper, we introduce ReLaSH (REconstructing joint LAtent Spaces for Hypergraphs with attributes), a general generative framework for producing realistic synthetic hypergraph data with hyperlink attributes via training a likelihood-based joint embedding model and reconstructing the joint latent space. Given a hypergraph dataset, ReLaSH first embeds the hyperlinks and their attributes into a joint latent space by training a likelihood-based model, and then reconstructs this joint latent space using a distribution-free generator. The generation task is completed by first sampling embeddings from the distribution-free generator and then decoding them into hyperlinks and attributes through the trained likelihood-based model. Compared with existing generative models, ReLaSH explicitly accounts for the unique structure of hypergraphs and jointly models hyperlinks and their attributes. Moreover, the likelihood-based embedding model provides efficiency and interpretability relative to deep black-box architectures, while the distribution-free generator in the joint latent space ensures flexibility. We theoretically demonstrate the consistency and generalizability of ReLaSH. Empirical results on a range of real-world datasets from diverse domains demonstrate the strong performance of ReLaSH, underscoring its broad utility and effectiveness in practical applications.

## 1 Introduction

Hypergraph data capture multi-way interactions among entities, such as co-occurrence, collaboration, and co-functioning, (Benson et al., 2016; Battiston et al., 2020), and have become ubiquitous, spanning areas including biology (Rhodes et al., 2005; Nepusz et al., 2012; Feng et al., 2021), medical research (Johnson et al., 2016; 2023), and the social sciences (Zhu et al., 2019; Ji et al., 2022; Wu et al., 2024). Generating hypergraphs with hyperlink attributes has broad applications in data augmentation (Wei et al., 2022; Zhou et al., 2025), simulation (Nguyen & Le, 2024), and understanding real-world complex systems (Torres et al., 2021). For example, Intensive Care Unit (ICU) records can be viewed as a symptom co-occurrence hypergraph with hyperlink attributes: for each patient profile, the co-occurrence of symptoms and diseases forms a hyperlink, while other patient information constitutes hyperlink attributes. Generating synthetic hyperlinks with attributes from the symptom co-occurrence hypergraph corresponds to generating synthetic patient profiles, enabling applications such as privacy-preserving data sharing across medical centers and patient simulations.

Fig. 1 showcases a synthetic patient profile produced by ReLaSH. The widespread need across fields to generate realistic hypergraphs with hyperlink attributes calls for a general generative model architecture for this task.

| Personal Information | | |
|---|---|---|
| Name: Jane Doe | Gender: ☐Male ☒Female | Religion: Catholic |
| Marital Status: ☐Single ☒Married ☐Divorced ☐Widowed ☐Separated ☐Life Partner | | |
| Ethnicity: ☒White ☐Black ☐Hispanic/Latino☐ Asian ☐ American Indian/Alaska Native ☐Other | | |
| Lifetime: 86.19 yrs | Hospital Stay Time: 14d 19h | ICU Stay Time: 8d 6h |

| Representative Major Diseases | Other Diseases and Complications Record |
|---|---|
| Coronary Atherosclerosis Congestive Heart Failure Chronic Kidney Disease Intracerebral Hemorrhage Dementia | Hyperlipidemia; Hyperpotassemia; Pneumococcus infection; Atrial fibrillation; Primary cardiomyopathies; Long-term (current) use of anticoagulants; Chronic systolic heart failure; Abdominal aneurysm, ruptured; Embolism and thrombosis of iliac artery; Chronic obstructive asthma; Chronic airway obstruction; Noninfectious gastroenteritis and colitis; Hemorrhage of gastrointestinal tract; Acute kidney failure; Sinoatrial node dysfunction; Hematoma complicating a procedure; Personal history of malignant neoplasm of breast. |

| Total Diseases: 23 | |

Figure 1: An example of synthetic ICU medical record forms generated from ReLaSH, trained on a symptom co-occurrence hypergraph from Johnson et al. (2016), which includes 3,000 ICU patient profiles and 2,230 distinct disease and symptom codes. The disease combinations in this synthetic record reflect the characteristics of an aged, medically complex ICU patient, where the co-occurrence of symptoms often leads to the development of new syndromes. For example, anticoagulant use in the setting of atrial fibrillation increases the risk of intracerebral hemorrhage and gastrointestinal bleeding (Lopes et al., 2017; Scridon & Balan, 2023), and they co-occur on the record.

Generative models are trained to learn the distribution of real-world observations and to generate novel yet realistic samples (Kingma & Welling, 2013; Goodfellow et al., 2014). Recent research has witnessed many powerful generative architectures, including variational autoencoders (VAEs) (Kingma & Welling, 2013), generative adversarial networks (GANs) (Goodfellow et al., 2014; Arjovsky et al., 2017), flow-based models (Dinh et al., 2016; 2014; Kingma & Dhariwal, 2018), score-based and diffusion models (Sohl-Dickstein et al., 2015; Ho et al., 2020; Song et al., 2020), and autoregressive models (Van Den Oord et al., 2016; 2017), with notable successes in tasks including image generation (Karras et al., 2019; Dhariwal & Nichol, 2021; Rombach et al., 2022), audio generation (van den Oord et al., 2016; Kong et al., 2021), and speech synthesis (Shen et al., 2018). However, most of these popular models are designed for continuous data and do not directly apply to discrete structures such as hypergraphs. Another thread of work studies generative modeling for discrete data, including extending GANs for text and audio generation (Yu et al., 2017; Nie et al., 2018), and diffusion models for categorical variables (Austin et al., 2021; Hoogeboom et al., 2021). These approaches, however, often suffer from computational and storage limitations and typically do not account for the special structure of hypergraphs.

On the other hand, existing graph generative models (You et al., 2018; Chen et al., 2023b) primarily focus on generating graphs that capture pairwise relations and do not extend directly to hypergraph generation. Prior work has considered hyperlink representation learning and generation (Jo et al., 2021; Wu et al., 2025), but these models do not incorporate hyperlink attributes. Representation learning on pairwise graphs with edge attributes has also been studied (Wang et al., 2024), yet these methods do not apply to generating hypergraphs that capture multi-way interactions with hyperlink attributes. A thread of recent work has studied attributed hypergraphs from different perspectives. Badalyan et al. (2024) study structure and inference in hypergraphs with node attributes, where the primary goal is to understand the observed hypergraph for tasks such as community detection and link prediction, rather than generating synthetic data. Chun et al. (2025) focus on generating synthetic hyperlinks on a fixed set of nodes and node attributes, aiming to realistically capture the interplay between structure and attributes in hyperlink generation, but do not extend to generating new hyperlink attributes. Gailhard et al. (2025) propose a hierarchical coarsening-expansion framework to jointly generate hypergraph topology and node attributes; in generation, their method produces new nodes, node attributes, and hyperlinks, whereas our goal is to generate new hyper-

links and hyperlink attributes on a fixed set of nodes (for example, generating new combinations of symptoms as hyperlinks over a fixed symptom set and corresponding attributes to form new medical records). A recent survey (Stoian et al., 2025) summarizes advances in tabular data generation, which can be applied to our tasks, as these methods handle general mixed-type data, but they do not scale well and also ignore the special structural properties of hypergraphs. Moreover, most existing generative modeling approaches for such tasks lack theoretical guarantees. A principled, theoretically grounded generative modeling framework for hypergraphs with hyperlink attributes is therefore greatly needed.

In this work, we introduce ReLaSH (REconstructing joint LAtent Spaces for Hypergraphs with attributes), a generative model architecture for hypergraphs with hyperlink attributes. Given an observed hypergraph with hyperlink attributes, ReLaSH first trains a likelihood-based joint embedding model and embeds the observed hypergraph together with hyperlink attributes into a joint low-dimensional latent space, then reconstructs this joint latent space using a distribution-free generator. Synthetic hyperlinks with attributes are then generated by first sampling embeddings from the distribution-free generator in the joint latent space and then decoding the embeddings via the trained likelihood-based model. Below, we summarize our main contributions:

1. Methodologically, we introduce ReLaSH, a generative model architecture for generating realistic synthetic hypergraphs with hyperlink attributes. ReLaSH consists of a likelihood-based joint embedding model and a distribution-free generator in the joint latent space. The likelihood-based joint embedding model provides efficiency and interpretability relative to deep architectures, while the distribution-free generator offers flexibility in the latent space.

2. Theoretically, we show that the KL divergence between the distributions of true samples and generated ones can be decomposed into three parts. The first two parts correspond to errors arising from training the likelihood-based joint embedding model, whereas the third depends on the discrepancy between the distributions of true and generated embeddings in the joint latent space. Notably, this analysis shows that ReLaSH circumvents the curse of ambient dimensionality in high-dimensional hypergraphs by exploiting the special structure of hypergraphs through the likelihood model, so that the overall error rate is dominated by errors from the relatively low-dimensional latent space rather than the original high-dimensional problem.

3. Numerically, we evaluate ReLaSH on three tasks: generating synthetic medical records from the MIMIC-III dataset (Johnson et al., 2016); generating new recipes from the ERRN dataset[1]; and generating reference author lists with top keywords from a co-citation dataset (Ji et al., 2022; Ke et al., 2023). In all tasks, ReLaSH efficiently generates realistic hyperlinks with reasonable hyperlink attributes and achieves superior performance compared with competing methods, demonstrating its broad utility and advantages for this task. Additional simulation results further demonstrate the effectiveness of ReLaSH.

The remainder of the paper is organized as follows. Section 2 introduces the ReLaSH framework. Section 3 presents the theoretical results. Section 4 reports numerical experiments on three real-world hypergraph generation tasks. Section 5 concludes with a discussion and outlines potential directions for future research. Additional materials, including additional theoretical results, algorithm details, simulation study results, experimental settings, and proofs, are provided in the appendix.

## 2 RECONSTRUCTING THE JOINT LATENT SPACE FOR HYPERGRAPHS

### 2.1 NOTATION

For positive real numbers $a$ and $b$, we define $a \vee b = \max(a, b)$ and $a \wedge b = \min(a, b)$. Let $\|A\|_F$ be the Frobenius norm of matrix $A$, $\|A\|_2$ be the spectral norm, and $A_{ij}$ denote its element at the $i$-th row and $j$-th column. For two sequences of positive real numbers $a_n$ and $b_n$, we write $a_n = O(b_n)$ or $a_n \lesssim b_n$ if there exist constants $N$ and $C$ such that $a_n \leq Cb_n$ for all $n > N$. For random variable sequences $X_n$ and $Y_n$, we write $X_n = O_p(Y_n)$ if for any $\varepsilon > 0$, there exists a constant $C_\varepsilon > 0$ such that $\sup_n \mathbb{P}(|X_n| \geq C_\varepsilon |Y_n|) < \varepsilon$. For two probability distribution $P, Q$

---

[1] https://www.kaggle.com/datasets/hugodarwood/epirecipes/data

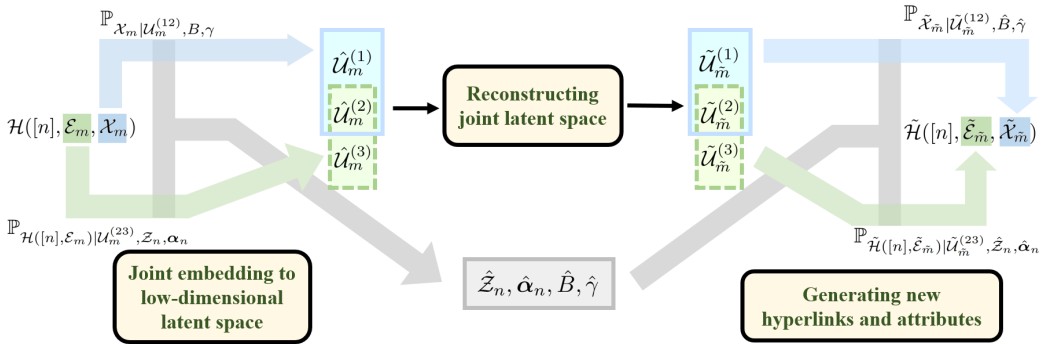

Figure 2: The general pipeline of ReLaSH.

defined on the same sample space $\mathcal{X}$, we denote the KL-divergence from $Q$ to $P$ as $\mathrm{d}_{\mathrm{KL}}(P\|Q) = \sum_{x\in\mathcal{X}} P(x)\log(P(x)/Q(x))$.

## 2.2 SETUP AND THE GENERAL RELASH

We denote an observed hypergraph with hyperlink attributes by $\mathcal{H}(\mathcal{V}_n, \mathcal{E}_m, \mathcal{X}_m)$, where $\mathcal{V}_n = \{v_1, \ldots, v_n\}$ is the set of $n$ nodes in the hypergraph, $\mathcal{E}_m = \{e_1, \ldots, e_m\}$ is the set of observed hyperlinks, and $\mathcal{X}_m = \{x_1, \ldots, x_m\}$ collects the attributes associated with each of the $m$ hyperlinks. For simplicity, let $\mathcal{V}_n = [n] = \{1, \ldots, n\}$; each hyperlink is then a subset of $[n]$ indexing the nodes that form it. Given $\mathcal{H}([n], \mathcal{E}_m, \mathcal{X}_m)$, the goal is to generate a synthetic hypergraph $\tilde{\mathcal{H}}([n], \tilde{\mathcal{E}}_{\tilde{m}}, \tilde{\mathcal{X}}_{\tilde{m}})$, where $\tilde{m}$ denotes the number of generated hyperlinks, $\tilde{\mathcal{E}}_{\tilde{m}} = \{\tilde{e}_1, \ldots, \tilde{e}_{\tilde{m}}\}$ is the set of generated hyperlinks, and $\tilde{\mathcal{X}}_{\tilde{m}} = \{\tilde{x}_1, \ldots, \tilde{x}_{\tilde{m}}\}$ is the corresponding set of attributes.

In this section, we introduce the general ReLaSH framework for the hypergraph generation task. Fig. 2 presents the pipeline of ReLaSH. In brief, ReLaSH first jointly embeds the hyperlinks and their attributes into a latent space by training a likelihood-based model; it then reconstructs this joint latent space via a distribution-free generator; and finally generates new hyperlinks with attributes by decoding sampled embeddings from the joint latent space using the trained likelihood-based model. Below, we describe these three steps in sequence. Sections 2.3 and 2.4 introduce the specific likelihood-based model and the latent-space generator used in this paper, respectively.

**Embedding hyperlinks with attributes to the joint latent space.** To build the likelihood-based embedding model, we partition the dimensions of the joint latent space into three blocks with dimensions $k_1$, $k_2$, and $k_3$, corresponding respectively to attributes only, attributes and hyperlinks jointly, and hyperlinks only. We associate each node $i$ with a latent embedding $z_i \in \mathbb{R}^{k_2+k_3}$ and a degree parameter $\alpha_i \in \mathbb{R}$ to capture heterogeneity in node popularity. The overall rate of $\alpha_i$'s, i.e., $\bar{\alpha}_{m,n} = n^{-1}\sum_{i=1}^n \alpha_i$ serves to control hyperlink sparsity in the hypergraph (Wu et al., 2024): the smaller $\bar{\alpha}_{m,n}$, the sparser the hyperlinks. The attributes are associated with a latent loading matrix $B \in \mathbb{R}^{p\times(k_1+k_2)}$ and an intercept vector $\gamma \in \mathbb{R}^p$, where $p$ is the number of attributes. We also associate each hyperlink and its attributes with a joint latent embedding $u = (u^{(1)\top}, u^{(2)\top}, u^{(3)\top})^\top \in \mathbb{R}^{k_1+k_2+k_3}$ drawn from an unknown distribution $\mathbb{P}_U$; for convenience, we write $u^{(12)} = (u^{(1)\top}, u^{(2)\top})^\top$ and $u^{(23)} = (u^{(2)\top}, u^{(3)\top})^\top$. Let $\mathcal{U}_m = \{u_1, \ldots, u_m\}$ denote the joint embeddings of the observed hypergraph $\mathcal{H}([n], \mathcal{E}_m, \mathcal{X}_m)$, and let $\mathcal{Z}_n = \{z_1, \ldots, z_n\}$ and $\alpha_n = (\alpha_1, \ldots, \alpha_n)$. Then $\mathcal{U}_m$ indeed collects $m$ realizations from $\mathbb{P}_U$. Consider a random hypergraph $\mathcal{H}_{m,n} := \mathcal{H}([n], \{E_1, \ldots, E_m\}, \{X_1, \ldots, X_m\})$, of which $\mathcal{H}([n], \mathcal{E}_m, \mathcal{X}_m)$ is a single realization. Let $\mathbb{P}_{\mathcal{H}_{m,n}|\mathcal{U}_m, \mathcal{Z}_n, \alpha_n, B, \gamma}$ denote the distribution of $\mathcal{H}_{m,n}$ conditional on $\mathcal{U}_m$, $\mathcal{Z}_n$, $\alpha_n$, $B$, and $\gamma$. This defines a probability measure over the product space $(\mathcal{P}([n]) \times \mathcal{A})^m$, where $\mathcal{P}([n]) = \{e : e \subseteq [n]\}$ is the power set of $[n]$, and $\mathcal{A}$ is the attribute space. Given all embeddings and parameters, we consider the factorization

$$\mathbb{P}_{\mathcal{H}_{m,n}|\mathcal{U}_m, \mathcal{Z}_n, \alpha_n, B, \gamma} = \prod_{j=1}^m \mathbb{P}_{\mathcal{H}([n], \{E_j\}, \{X_j\})|u_j, \mathcal{Z}_n, \alpha_n, B, \gamma},$$

which implies that dependency in node co-occurrence and attributes are characterized by the embeddings, and different hyperlinks are conditionally independent; similar assumptions are widely adopted in the literature on latent space network models (Ma et al., 2020) and hypergraphs (Ke et al., 2019). Additionally, decomposing the latent space into $(k_1, k_2, k_3)$ dimensions permits a factorization of the joint likelihood, where

$$\mathbb{P}_{\mathcal{H}([n], \{E_j\}, \{X_j\}) | u_j, \mathcal{Z}_n, \alpha_n, B, \gamma} = \mathbb{P}_{\mathcal{H}([n], \{E_j\}) | u_j^{(23)}, \mathcal{Z}_n, \alpha_n} \cdot \mathbb{P}_{X_j | u_j^{(12)}, B, \gamma}.$$

Here, $\mathbb{P}_{\mathcal{H}([n], \{E\}) | u^{(23)}, \mathcal{Z}_n, \alpha_n}$ is the hyperlink-generation model and $\mathbb{P}_{X | u^{(12)}, B, \gamma}$ is the attribute generation model; dependence between hyperlinks and their attributes is captured via the shared latent embedding $u^{(2)}$. Under a specified likelihood, ReLaSH obtains embeddings $(\hat{\mathcal{U}}_m, \hat{\mathcal{Z}}_n, \hat{\alpha}_n, \hat{B}, \hat{\gamma})$ via optimizing a joint loss from the likelihood model. Section 2.3 introduces a specific likelihood model for this task.

**Reconstructing the joint latent space.** The joint embeddings $\hat{\mathcal{U}}_m$ constitute an *estimated sample* of latent characteristics of the observed hyperlinks and attributes in the joint embedding space. In the second step, ReLaSH trains a distribution-free generator on this estimated sample. Examples of such generators include normalizing flows (Kingma & Dhariwal, 2018), kernel density estimation (Silverman, 2018), and score-based generative models (Song et al., 2020). Section 2.4 specifies the score-based generator and its implementation used in this paper. ReLaSH then produces $\tilde{\mathcal{U}}_m$ from the generator and separates it by dimension to obtain $\tilde{\mathcal{U}}_m^{(12)}$ and $\tilde{\mathcal{U}}_m^{(23)}$.

**Generating new hyperlinks with attributes.** Finally, given the generated embeddings $\tilde{\mathcal{U}}_m^{(12)}$ and $\tilde{\mathcal{U}}_m^{(23)}$, ReLaSH decodes them stochastically through the fitted likelihood models to obtain a synthetic hypergraph $\tilde{\mathcal{H}}([n], \tilde{\mathcal{E}}_{\tilde{m}}, \tilde{\mathcal{X}}_{\tilde{m}})$, with hyperlinks drawn from $\mathbb{P}_{\tilde{\mathcal{H}}([n], \tilde{\mathcal{E}}_{\tilde{m}}) | \tilde{\mathcal{U}}_m^{(23)}, \hat{\mathcal{Z}}_n, \hat{\alpha}_n}$ and attributes drawn from $\mathbb{P}_{\tilde{\mathcal{X}}_{\tilde{m}} | \tilde{\mathcal{U}}_m^{(12)}, \hat{B}, \hat{\gamma}}$, respectively.

## 2.3 A Joint Embedding Approach and its Identifiability Conditions

The embedding approach in the first step of ReLaSH can be flexibly chosen and designed domain-adaptively. In this section, we specify a joint embedding likelihood model implemented in this paper. Specifically, for each $e \subseteq [n]$, let $\mathbb{P}_{\mathcal{H}([n], \{E\}) | u^{(23)}, \mathcal{Z}_n, \alpha_n}(E = e) = \prod_{i \in e} p_i(u^{(23)}) \prod_{i \notin e} (1 - p_i(u^{(23)}))$ with $p_i(u^{(23)}) = \sigma(u^{(23)\top} z_i + \alpha_i)$, where $\sigma(\cdot) = \exp(\cdot)/(1 + \exp(\cdot))$ is the sigmoid function. The distribution $\mathbb{P}_{X | u^{(12)}, B, \gamma}$ is specified by $x = \gamma + Bu^{(12)} + \epsilon$, where $\epsilon$ is a $p$-dimensional vector of independent, mean-zero random errors with sub-Gaussian tails (Vershynin, 2018) chosen to accommodate different attribute types (e.g., Gaussian for continuous attributes and Bernoulli for binary attributes). Our choice of embedding models is motivated by prior work with theoretical support on hyperlink generation (Wu et al., 2024) and joint attribute modeling in graphs (Zhang et al., 2022; Li et al., 2025), yet our setting differs substantially and therefore requires new analysis and justification. These analyses are presented in Section 3 and in Appendix A.

To obtain embeddings from the observed hypergraph $\mathcal{H}([n], \mathcal{E}_m, \mathcal{X}_m)$, we propose to optimize a joint loss based on the likelihood models. Specifically, let

$$\ell_H = -\log \mathbb{P}_{\mathcal{H}([n], \mathcal{E}_m) | \mathcal{U}_m^{(23)}, \mathcal{Z}_n, \alpha_n} = -\sum_{j=1}^{m} \sum_{i=1}^{n} [\mathbf{1}_{\{i \in e_j\}} \theta_{ji}^H - \log\{1 + \exp(\theta_{ji}^H)\}]$$

with $\theta_{ji}^H = u_j^{(23)\top} z_i + \alpha_i$ and $\ell_A = \sum_{j=1}^{m} \|x_j - \gamma - Bu_j^{(12)}\|_2^2$. We optimize the joint loss $\ell(U, Z, B, \alpha, \gamma) = \ell_H + \lambda \ell_A$, where $\lambda > 0$ is a weight parameter that balances the contributions from each part. To ensure identifiability during the embedding and estimation procedure, we need to impose additional structural constraints. Define the node embedding matrix as $Z = (z_1, \cdots, z_n)^\top \in \mathbb{R}^{n \times (k_2 + k_3)}$, and let $B = (B_1 \ B_2) \in \mathbb{R}^{p \times (k_1 + k_2)}$. The identifiability of the joint embedding model, i.e., the distribution of $\mathbb{P}_U$ and $Z, \alpha, B, \gamma$ is defined as follows.

**Definition 1.** *(Identifiability of the joint embedding model.) The joint embedding model is identifiable if for any two sets of model parameters $(\mathbb{P}_U, Z, \alpha, \gamma, B)$ and $(\mathbb{P}'_U, Z', \alpha', \gamma', B')$,*

$$\alpha + ZU^{(23)} \stackrel{d}{=} \alpha' + Z'U^{(23)'} \quad \text{and} \quad \gamma + BU^{(12)} \stackrel{d}{=} \gamma' + B'U^{(12)'}$$

*imply $(\mathbb{P}_U, Z, \alpha, \gamma, B) = (\mathbb{P}_{U'}, Z', \alpha', \gamma', B')$.*

The following theorem ensures the identifiability of the embedding parameters under a set of identifiability conditions.

**Theorem 1.** *Under the following conditions: (C1)* $\mathbb{E}_{\mathbb{P}_U}[U] = 0$*; (C2)* $\mathbb{E}_{\mathbb{P}_U}[U^{(23)}U^{(23)\top}] = \frac{1}{n}Z^\top Z$ *is a diagonal matrix with distinct positive diagonal elements; (C3)* $\mathbb{E}_{\mathbb{P}_U}[U^{(1)}U^{(1)\top}] = \frac{1}{p}B_1^\top B_1$ *is a diagonal matrix with distinct positive diagonal elements; (C4)* $\mathbb{E}_{\mathbb{P}_U}[U^{(1)}U^{(2)\top}] = 0_{k_1 \times k_2}$*, the joint embedding model is identifiable according to Definition 1.*

Such identifiability conditions have been widely considered in the literature; see, e.g., (Wu et al., 2025; Li et al., 2025). In general, if the true $(\mathbb{P}_U, Z, B, \alpha, \gamma)$ does not satisfy the conditions in Theorem 1, we can apply a unique transformation to enforce the constraints while keeping the joint distribution of hyperlinks and attributes unchanged. Therefore, we jointly embed the hypergraph and attributes by minimizing $\ell(U, Z, B, \alpha, \gamma)$ under identifiability constraints designed based on Theorem 1. Full algorithmic details are provided in Appendix B.1.

### 2.4 SCORE-BASED JOINT EMBEDDING SPACE RECONSTRUCTION

In this section, we specify the score-based generator in the joint latent embedding space implemented in this work. Our construction follows score-based generative modeling through stochastic differential equations (SDEs) (Song et al., 2020): a forward SDE gradually perturbs the data into Gaussian noise, and a reverse-time SDE generates samples from noise using score functions learned via denoising score matching (Hyvärinen & Dayan, 2005; Song et al., 2020). A key difference in our setup is that we train score functions using unobserved embeddings rather than observed data.

Specifically, we define a continuous-time forward diffusion process initialized from $\hat{\mathcal{U}}_m$:

$$\mathrm{d}U_t = -U_t\,\mathrm{d}t + \sqrt{2}\,\mathrm{d}W_t, \qquad U_0 \sim \hat{\mathcal{U}}_m, \tag{1}$$

where $\{W_t\}_{t \in [0,T]}$ is a standard $k$-dimensional Wiener process with $k = k_1 + k_2 + k_3$. Let $N = T/h$ denote the number of discretization steps with step size $h$. Let $s_\theta : \mathbb{R}^k \times [0,T] \to \mathbb{R}^k$ be the score network, a multilayer perceptron (MLP) with parameters $\theta$, to approximate the score function. The parameter $\theta$ is learned via the denoising score matching objective (Vincent, 2011), constructed using the learned embeddings (Wu et al., 2025):

$$\ell(\theta) = \sum_{l=0}^{N} \lambda_l \cdot \frac{1}{m} \sum_{u_0 \in \hat{\mathcal{U}}_m} \mathbb{E}_{u_{lh}|u_0}\left[\left\| s_\theta(u_{lh}, lh) - \nabla_{u_{lh}} \log p_{lh}(u_{lh}|u_0) \right\|_2^2\right],$$

where $\{u_{lh}\}_{l=0}^{N}$ are diffused samples in the forward process (1) at time $lh$, $\nabla \log p_t(\cdot)$ is the score (gradient of the log-density of $U_t$), $p_{lh}(u_{lh}|u_0)$ is the marginal density of the forward process given $u_0$, and $\lambda_k \approx 1/\mathbb{E}\{\|\nabla_{u_{lh}} \log p_{lh}(u_{lh} \mid u_0)\|_2^2\}$ are nonnegative weights that balance different time steps. To construct the score-based generator, for each $l \in \{0, 1, \ldots, N-1\}$ and $t \in [lh, (l+1)h]$, we hold the score network fixed at the grid time and obtain

$$\mathrm{d}U_t^{\leftarrow} = \left(U_t^{\leftarrow} + 2\,s_{\hat{\theta}}(U_{T-lh}^{\leftarrow}, T - lh)\right)\mathrm{d}t + \sqrt{2}\,\mathrm{d}\tilde{W}_t,$$

where $\tilde{W}_t$ is a standard $k$-dimensional Wiener process independent of $W_t$. To sample $\tilde{m}$ embeddings, we simulate the above reverse-time SDE $\tilde{m}$ times, each initialized at $t = 0$ with $U_0^{\leftarrow} \sim \mathcal{N}(0, I_k)$. Collecting the terminal states yields the synthetic embeddings $\tilde{\mathcal{U}}_{\tilde{m}} = \{\tilde{u}_1, \ldots, \tilde{u}_{\tilde{m}}\}$.

## 3 THEORETICAL RESULTS

Our theoretical analysis in this section considers the following setup. The node embeddings $\mathcal{Z}_n = \{z_1, \ldots, z_n\}$, the degree parameters $\alpha_n = (\alpha_1, \ldots, \alpha_n)^\top$, and $(B, \gamma)$ are treated as fixed parameters. The hyperlink embeddings $\mathcal{U}_m = \{u_1, \ldots, u_m\}$ consist of $m$ i.i.d. draws from the hyperlink-embedding distribution $\mathbb{P}_U$. Conditioned on $(\mathcal{U}_m, \mathcal{Z}_n, \alpha_n, B, \gamma)$, the observed hypergraph is one realization from $\mathbb{P}_{\mathcal{H}_{m,n}|\mathcal{U}_m, \mathcal{Z}_n, \alpha_n, B, \gamma}$. Our goal is to understand how close the distribution of synthetic hyperlinks and attributes generated by ReLaSH is their true distribution under this model. Let $\mathbb{P}_{(E,X,U)}$ denote the joint distribution of a hyperlink $E$, its associated attribute $X$, and the corresponding joint latent embedding $U$. Let $\mathbb{P}_{\tilde{U}}$ denote the marginal distribution of a hyperlink embedding $\hat{U}$ sampled from the latent space generator trained on $\hat{\mathcal{U}}_m$, and let $\mathbb{P}_{(\tilde{E},\tilde{X},\tilde{U})}$ denote the joint

distribution of $\tilde{U}$, and the hyperlink and attributes generated from $\mathbb{P}_{\mathcal{H}([n],\{E_j\},\{X_j\})|\tilde{U},\hat{\mathcal{Z}}_n,\hat{\alpha}_n,\hat{B},\hat{\gamma}}$. Note that the estimated embeddings comes from one realization of $\mathbb{P}_{\mathcal{H}_{m,n}|\mathcal{U}_m,\mathcal{Z}_n,\alpha_n,B,\gamma}$, $\mathbb{P}_{\tilde{U}}$ and $\mathbb{P}_{(\tilde{E},\tilde{X},\tilde{U})}$ are indeed defined conditioned on $\mathcal{U}_m$, $\mathcal{Z}_n$, $\alpha_n$, $B$, and $\gamma$.

**Lemma 1.** *If $\mathcal{U}_m$, $\mathcal{Z}_n$, $\alpha_n$, $B$, and $\gamma$ are available and replace $(\hat{\mathcal{U}}_m, \hat{\mathcal{Z}}_n, \hat{\alpha}_n, \hat{B}, \hat{\gamma})$ in ReLaSH, we have $d_{KL}(\mathbb{P}_{(E,X,U)} \| \mathbb{P}_{(\tilde{E},\tilde{X},\tilde{U})}) = d_{KL}(\mathbb{P}_U \| \mathbb{P}_{\tilde{U}})$.*

Lemma 1 states that the generative error for high-dimensional hyperlinks and attributes can be reduced to the generative error of low-dimensional hyperlink embeddings, under an ideal scenario where the true latent embeddings are available to use. In practical scenarios where these embeddings are unobserved and need to be learned from the data, the error from the embedding algorithm needs to be considered. Conditioned on $\mathcal{U}_m$, let $\mathbb{P}'_{(\tilde{E},\tilde{X},\tilde{U})}$ denote a random measure for the joint distribution of a generated hyperlink embedding $\tilde{U}$ and its associated hyperlink $\tilde{E}$ and attribute $\tilde{X}$, using the estimated parameters $\hat{\mathcal{Z}}_n$, $\hat{B}$, $\hat{\alpha}$, and $\hat{\gamma}$. More formally, this measure is $\mathbb{P}'_{(\tilde{E},\tilde{X},\tilde{U})|\hat{\mathcal{U}}_m,\hat{\mathcal{Z}}_n,\hat{B},\hat{\alpha},\hat{\gamma}}$, but we use the simpler notation for clarity. The randomness of this distribution arises from the observed hypergraph $\mathcal{H}([n], \mathcal{E}_m, \mathcal{X}_m)$ given $(\mathcal{U}_m, \mathcal{Z}_n, B, \alpha, \gamma)$, which further induces randomness in the learned embeddings $(\hat{\mathcal{U}}_m, \hat{\mathcal{Z}}_n, \hat{B}, \hat{\alpha}, \hat{\gamma})$ and consequently in the distribution of $(\tilde{E}, \tilde{X}, \tilde{U})$. Similarly, define $\mathbb{P}'_{\tilde{U}} := \mathbb{P}'_{\tilde{U}|\hat{\mathcal{U}}_m,\hat{\mathcal{Z}}_n,\hat{B},\hat{\alpha},\hat{\gamma}}$ as the random measure on $\tilde{U}$ conditioned on the learned embeddings. The next theorem shows that the generation error can be decomposed into three parts.

**Theorem 2.** *The KL-divergence between the true distribution $\mathbb{P}_{(E,X,U)}$ and the generated distribution $\mathbb{P}_{(\tilde{E},\tilde{X},\tilde{U})}$ admits the following decomposition:*

$$d_{KL}(\mathbb{P}_{(E,X,U)} \| \mathbb{P}_{(\tilde{E},\tilde{X},\tilde{U})}) = \Delta_{(\mathcal{Z}_n,B,\alpha,\gamma)\text{-estimation}} + \Delta_{\mathbb{P}_U\text{-estimation}} + \Delta_{\text{latent-reconstruction}},$$

*where the exact forms of the three components are given in the proof of the theorem (Appendix C).*

As their names indicate, the first error term depends on the estimation error of $(\mathcal{Z}_n, B, \alpha, \gamma)$, the second on the recovery of the joint latent embedding distribution, and the third on the reconstruction of the joint latent space. Next, we analyze these three terms under the specific embedding approach in Section 2.3 and the score-based generator in Section 2.4. We first analyze $\Delta_{(\mathcal{Z}_n,B,\alpha,\gamma)}$. Following the discussion in Wu et al. (2024) on how the sparsity parameter $\bar{\alpha}_{m,n}$ affects the hypergraph embedding procedure, we introduce the following assumption on $\bar{\alpha}_{m,n}$.

**Assumption 1** (Hyperlink sparsity). *As $m, n, p \to \infty$, $\exp(\bar{\alpha}_{m,n}) \gtrsim \log(m \vee n)/(m \wedge n)$.*

When $m \asymp n$, this sparsity scaling is consistent with the sufficient order in Proposition 2.2 and the necessary order in Proposition 2.1 of Wu et al. (2024), up to a logarithmic factor. Further, we consider the following assumption on the embedding space.

**Assumption 2** (Embedding space). *Define $\sigma_{m,n,p} = \{(m \wedge n \wedge p) \exp(\bar{\alpha}_{m,n})\}^{-1/2}$. The minimum eigenvalue of $\mathbb{E}_{\mathbb{P}_U}[UU^\top]$ is lower bounded by a constant, and eigenvalues of $\mathbb{E}_{\mathbb{P}_U}[UU^\top]$ are distinct with their gaps lower bounded by a constant. For the $m$ realizations of $\mathbb{P}_U$ in $\mathcal{U}_m$, it holds that that $\|m^{-1} \sum_{j=1}^m u_j\|_F = O(\sigma_{m,n,p})$, and $\|m^{-1} \sum_{j=1}^m u_j u_j^\top - \mathbb{E}_{\mathbb{P}_U}[UU^\top]\|_F = O(\sigma_{m,n,p}^2)$, $\|m^{-1} \sum_{j=1}^m u_j^{(2)} u_j^{(1)\top}\|_F = O(\sigma_{m,n,p}^2)$, $\|m^{-1} \sum_{j=1}^m u_j^{(2)} u_j^{(3)\top}\|_F = O(\sigma_{m,n,p}^2)$.*

Assumption 2 ensures that the latent embedding space is well-conditioned and identifiable by requiring a non-degenerate, spectrally separated covariance structure, while also guaranteeing that the empirical moments of the embeddings concentrate around their population counterparts to enable consistent estimation. Note that $\exp(\bar{\alpha}_{m,n}) \ll 1$ as $m, n \to \infty$. This concentration requirement is easily satisfied and is weaker than the concentration results available for many popular multivariate distributions (Vershynin, 2018) when $m \asymp n \asymp p$, which cover a wide range of real data scenario. With Assumptions 1 and 2, we have the following theorem on $\Delta_{(\mathcal{Z}_n,B,\alpha,\gamma)}$.

**Theorem 3.** *Suppose that Assumptions 1 and 2 hold, and $\lambda \asymp \exp(\bar{\alpha}_{m,n})$, then as $(m, n) \to \infty$, the rate of estimation-related error satisfies*

$$\frac{1}{(n \vee p)} \Delta_{(\mathcal{Z}_n,B,\alpha,\gamma)\text{-estimation}} = O_p\left(\frac{\log(m \vee n)}{\min\{m,n,p\}}\right).$$

*Consequently, when $m \asymp n \asymp p$, we have $n^{-1}\Delta_{(\mathcal{Z}_n,B,\alpha,\gamma)\text{-estimation}} = O_p(\log n/n)$, thus the error in final generative performance from estimating $\mathcal{Z}_n, B, \alpha, \gamma$ shrinks as fast as $\log n \cdot n^{-1}$.*

Theorem 3 implies that if $m \asymp n \asymp p$, the error introduced by estimating $(Z, B, \alpha, \gamma)$ is asymptotically negligible. We defer the analysis of $\Delta_{\mathbb{P}_U \text{ estimation}}$ to Appendix A due to page constraints, as it requires defining a discretization over the support of $\mathbb{P}_U$. To study $\Delta_{\text{latent-reconstruction}}$, we follow the theoretical development of Chen et al. (2022), noting that our score networks are trained on learned embeddings rather than observed data. Let $p^{\text{e}}$ denote the marginal density of the joint embeddings $\hat{U}$, and let $p_t^{\text{e}}$ denote the law of $U_t$ in the forward process (1). Under the following assumptions (Block et al., 2020; Lee et al., 2022; Chen et al., 2022), we bound the generative error in the joint embedding space.

**Assumption 3.** *The learned score network $\mathbf{s}_{\hat{\theta}}(u, t)$ satisfies for any $1 \leq k \leq N$, $\mathbb{E}_{U \sim p_{kh}^{\text{e}}} \|\nabla \log p_{kh}^{\text{e}}(U) - \mathbf{s}_{\hat{\theta}}(U, kh)\|^2 \leq \varepsilon_0^2$. The distribution of estimated embeddings has a bounded second moment, i.e., $M_U = \mathbb{E}_{p^{\text{e}}} \|\hat{U}\|^2 \leq \infty$. For $t \in [0, T]$, $\nabla \log p_t^{\text{e}}$ is L-Lipschitz.*

Assumption 3 requires that the learned score network approximates the true score function consistently in an $L_2$ sense. The boundedness assumption of its second moment is automatically satisfied due to the constraint in the embedding algorithm. Together with the $L$-Lipshchitz condition, these requirements provide the regularity needed to control sampling and approximation errors in the diffusion dynamics as follows.

**Proposition 1** (Theorem 2 in Chen et al. (2022)). *Under Assumption 3, if $L \geq 1, h \leq 1$ and $T \geq 1$. We have $\Delta_{\text{latent-reconstruction}} \lesssim (M_U + K)e^{-T} + T\varepsilon_0^2 + N^{-1}KT^2L^2$, where $K = k_1 + k_2 + k_3$. Then by choosing $T = \log((M_U + K)/\varepsilon_0^2)$ and $N = \Omega(KTL^2/\varepsilon_0^2)$, we have $\Delta_{\text{latent-reconstruction}} = O(T\varepsilon_0^2)$.*

The first error term quantifies the distance between $U_T$ and the standard Gaussian distribution in the forward diffusion process, which decays exponentially in $T$. The third term accounts for errors arising from discretizing the SDE. Regarding the score-approximation error $\varepsilon_0^2$, Chen et al. (2023a) studied a specific neural network construction and demonstrated that the upper bound on the sample complexity of score estimation is exponential in the score network dimension. This sample-complexity result for score estimation highlights the curse of dimensionality in high-dimensional generative modeling tasks. By jointly embedding the hypergraph and attribute data into a low-dimensional continuous space and reconstructing this joint latent space, we avoid training a high-dimensional score network or other high-dimensional distribution-free generators, thereby significantly improving efficiency.

## 4 EXPERIMENTS

In this section, we empirically evaluate ReLaSH for generating hypergraphs with hyperlink attributes on three datasets. We first describe the experimental setup and then report the numerical results. Additional simulation study and implementation details are provided in Appendix B.

### 4.1 EXPERIMENT SETUP

**Datasets.** We use three datasets: (i) the co-citation hypergraph data from MADStat (Ji et al., 2022) and MADStaText (Ke et al., 2023), where authors are nodes, co-cited authors in a statistical journal paper form hyperlinks, and TF-IDF values (Sparck Jones, 1972) of words with respect to the corresponding paper abstracts are hyperlink attributes; (ii) the recipe hypergraph dataset[2], where food ingredients are nodes, the set of ingredients in a recipe forms

| | $\Delta_{\mathcal{H}_v} \downarrow$ | $\Delta_{\mathcal{X}_m} \downarrow$ | $\Delta_{\mathcal{X}_v} \downarrow$ | FED $\downarrow$ | a-FED $\downarrow$ |
|---|---|---|---|---|---|
| ReLaSH-$(7, 0, 2)$ | 3.260 | **2.989** | **1.435** | 0.532 | 1.084 |
| ReLaSH$_c$-$(7, 0, 2)$ | 27.794 | **2.989** | **1.435** | **0.013** | **0.193** |
| ReLaSH-$(7, 0, 16)$ | **2.624** | 3.681 | 1.655 | 11.738 | 2.047 |
| ReLaSH$_c$-$(7, 0, 16)$ | 6.230 | 3.681 | 1.655 | 9.049 | 1.574 |
| Gau-Diff | 4.268 | 3.497 | 1.719 | 39.731 | 4.387 |
| RealNVP | 3.958 | 33.240 | 2.526 | 27.685 | 2.843 |
| WGAN | 3.506 | 10.534 | 2.176 | 21.053 | 2.654 |
| VAE | 48.450 | 11.499 | 4.134 | 9.374 | 1.376 |

Table 1: Results for patient profile generation. Scales of $\Delta_{\mathcal{H}_v}$, $\Delta_{\mathcal{X}_m}$, $\Delta_{\mathcal{X}_v}$, FED, a-FED are $10^{-4}, 10^{-3}, 10^{-1}, 10^{-2}, 10^{-1}$, respectively.

---

[2] https://www.kaggle.com/datasets/hugodarwood/epirecipes/data

a hyperlink, and metadata such as cuisine type constitute hyperlink attributes; and (iii) the symptom co-occurrence hypergraph from MIMIC-III (Johnson et al., 2016), where medical symptoms are nodes, symptoms co-occurring in a patient profile forms a hyperlink, and the remaining patient notes are hyperlink attributes. Details of these hypergraph datasets are provided in Appendix B.4.

**Baselines.** We compare ReLaSH with 9 methods that can be used to produce synthetic hyperlinks with attributes: Gau-Diff (Song et al., 2020), RealNVP (Dinh et al., 2016), WGAN (Arjovsky et al., 2017), VAE (Kingma & Welling, 2013), ForestDiffusion (Jolicoeur-Martineau et al., 2024), TabPFGen (Ma et al., 2024), CTAB-GAN (Zhao et al., 2021), CTAB-GAN+ (Zhao et al., 2024), and CTGAN (Xu et al., 2019). For the tabular data generation baselines (ForestDiffusion, TabPF-Gen, CTAB-GAN, CTAB-GAN+, CTGAN), these methods do not scale to the patient-profile and co-citation generation tasks, as summarized in Table 1 of the survey (Stoian et al., 2025), and are therefore discarded from those two comparisons. We refer readers to Appendix B.7 for more details and discussion of these methods.

## 4.2 GENERATING SYNTHETIC HYPERLINKS WITH ATTRIBUTES

**Evaluation metrics.** The task aims to generate realistic hyperlinks with attributes that preserve properties of the observed hypergraph from different models. Following Wu et al. (2025), we evaluate performance using the RMSE of the hypergraph node covariances ($\Delta_{\mathcal{H}_v}$), the attribute means ($\Delta_{\mathcal{X}_m}$), and the attribute covariances ($\Delta_{\mathcal{X}_v}$), and FED (Fréchet Embedding Distance), a generalization of the FID (Fréchet Inception Distance) used in evaluating visual generation tasks (Heusel et al., 2017), adapted to the hypergraph generation setting. In addition, we report a-FED, a further variant of FED that adjusts for the potential bias of FED when training the embedding machine. Details of these metrics are provided in Appendix B. For each metric, a lower value indicates better performance.

| | $\Delta_{\mathcal{H}_v}\downarrow$ | $\Delta_{\mathcal{X}_m}\downarrow$ | $\Delta_{\mathcal{X}_v}\downarrow$ | FED $\downarrow$ | a-FED $\downarrow$ |
|---|---|---|---|---|---|
| ReLaSH-$(2,2,2)$ | 1.996 | **8.578** | 1.887 | 1.246 | 0.931 |
| ReLaSH$_c$-$(2,2,2)$ | 3.890 | **8.578** | 1.887 | 5.481 | 1.935 |
| ReLaSH-$(2,7,8)$ | 1.914 | 8.817 | 1.886 | 1.060 | 0.791 |
| ReLaSH$_c$-$(2,7,8)$ | 2.772 | 8.817 | 1.886 | **0.947** | 0.706 |
| ReLaSH-$(8,8,8)$ | **1.626** | 8.608 | 1.887 | 1.454 | 0.871 |
| ReLaSH$_c$-$(8,8,8)$ | 2.816 | 8.608 | 1.887 | 6.451 | 2.042 |
| Gau-Diff | 1.672 | 10.016 | **1.824** | 5.060 | 1.503 |
| RealNVP | 1.668 | 12.646 | 1.863 | 3.948 | 1.256 |
| WGAN | 2.247 | 8.671 | 1.885 | 1.253 | **0.700** |
| VAE | 9.972 | 9.425 | 1.889 | 1.358 | 0.904 |

Table 2: Results for the co-citation hypergraph generation task. Scales of $\Delta_{\mathcal{H}_v}$, $\Delta_{\mathcal{X}_m}$, $\Delta_{\mathcal{X}_v}$, FED, a-FED are $10^{-3}, 10^{-2}, 10^{-1}, 10^{-1}, 10^{-1}$, respectively.

**Results & discussion.**

The results are summarized in Tables 1, 2, and 3. In these tables, ReLaSH-$(k_1, k_2, k_3)$ denotes ReLaSH with joint latent-space dimensions $(k_1, k_2, k_3)$, and ReLaSH$_c$-$(k_1, k_2, k_3)$ further denotes that, during generation, a calibration step is applied at the end so that the node degree sequence of the generated hypergraph matches that of the observed hypergraph. For each metric, the best result is highlighted in bold and the second-best is underlined. Across the three

| | $\Delta_{\mathcal{H}_v}\downarrow$ | $\Delta_{\mathcal{X}_m}\downarrow$ | $\Delta_{\mathcal{X}_v}\downarrow$ | FED $\downarrow$ | a-FED $\downarrow$ |
|---|---|---|---|---|---|
| ReLaSH-$(5,0,2)$ | 1.978 | 2.236 | 0.894 | 0.293 | 0.356 |
| ReLaSH$_c$-$(5,0,2)$ | 7.504 | 2.236 | 0.894 | 0.182 | 0.248 |
| ReLaSH-$(5,0,6)$ | 2.129 | 2.174 | **0.820** | **0.003** | **0.048** |
| ReLaSH$_c$-$(5,0,6)$ | 3.583 | 2.174 | **0.820** | 0.191 | 0.258 |
| ReLaSH-$(5,0,16)$ | 2.355 | 1.533 | 1.112 | 0.766 | 0.847 |
| ReLaSH$_c$-$(5,0,16)$ | 1.847 | 1.533 | 1.112 | 0.180 | 0.255 |
| Gau-Diff | 2.375 | 2.154 | 4.256 | 0.802 | 0.828 |
| RealNVP | 2.484 | **1.146** | 3.562 | 0.909 | 0.997 |
| WGAN | 2.208 | 21.428 | 1.351 | 0.907 | 0.928 |
| VAE | 21.587 | 9.883 | 5.180 | 11.553 | 10.285 |
| CTGAN | 2.519 | 28.799 | 4.983 | 0.847 | 0.865 |
| ForestDiffusion | 1.886 | 8.211 | 2.073 | 0.848 | 0.303 |
| TabPFGen | **1.565** | 1.915 | 1.205 | 0.297 | 0.884 |
| CTAB-GAN | 2.552 | 19.367 | 3.858 | 0.925 | 0.947 |
| CTAB-GAN+ | 2.488 | 8.330 | 3.821 | 0.898 | 0.902 |

Table 3: Results for recipe generation. Scales of $\Delta_{\mathcal{H}_v}$, $\Delta_{\mathcal{X}_m}$, $\Delta_{\mathcal{X}_v}$, FED and a-FED are $10^{-3}, 10^{-2}, 10^{-2}, 10^{-1}, 10^{-1}$, respectively.

tasks, ReLaSH exhibits robust and outstanding performance in terms of the quality of the generated hyperlinks and their attributes. Figure 1 (in the Introduction) and Figure 3 show examples of generated samples for these three tasks. The rationale for the generated medical record is described in the Introduction; below we discuss the other two cases.

The left half of Fig. 3 presents a generated reference–author list with top keywords by ReLaSH-$(2, 7, 8)$. Fan, Hastie, Tibshirani, and Zou are among the main contributors to high-dimensional statistics, variable selection, and penalization methods, notably through their contributions to the development of approaches such as the LASSO (Tibshirani, 1996) and SCAD (Fan & Li, 2001). Keywords like "lasso," "penalty," "oracle," "sparse," and "high-dimension" directly reference the domain of penalized approaches in sparse regression. The cited authors and keywords are highly congruent, reflecting current trends and leading contributors in high-dimensional regression.

The right half of Fig. 3 shows a generated recipe named "Mediterranean Fisherman's Bean Stew," produced by ReLaSH$_c$-$(5, 0, 16)$. By comparing against meals with similar ingredient combinations in the training set, we confirm that no identical recipe exists in the source data, meaning that the generated cuisine is genuinely novel rather than a memorized replication. The dish resembles a Mediterranean or Iberian-style fish and bean stew, similar to Spanish or Portuguese coastal cuisine. Its high-protein, low-fat profile is consistent with ingredients like lean fish and legumes, while saffron, fennel, and wine reflect authentic regional flavor. These results highlight ReLaSH as a powerful and efficient method for generating hypergraphs with attributes, even when trained on relatively small datasets, supporting applications such as the creation of new recipes. Additional remarks and extended examples are provided in Appendix B.4.

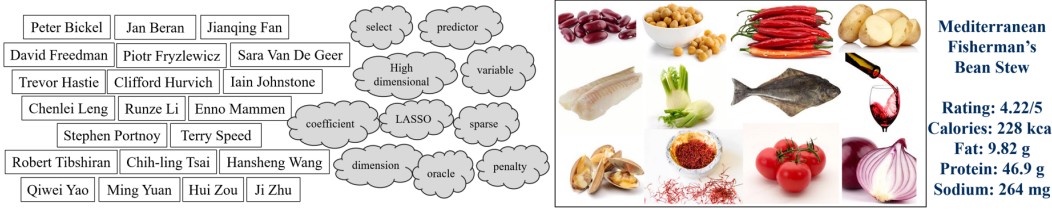

Figure 3: Examples of synthetic reference author list (left) and synthetic recipe (right) from ReLaSH.

## 5 CONCLUSION

We introduce ReLaSH, a general generative framework for hypergraphs with hyperlink attributes by bridging a likelihood-based joint embedding model with a distribution-free latent space generator. By embedding hyperlinks and their attributes into a shared latent space and reconstructing that space prior to decoding, ReLaSH explicitly accounts for the discrete nature of hypergraph structure, hyperlink sparsity, and mixed data types, while avoiding heavy training on the original high-dimensional data. Our analysis presents the consistency and generalizability for the framework, and experiments across diverse real-world datasets demonstrate its strong empirical performance, highlighting ReLaSH as a practical tool for the hypergraph generation task.

This work opens several directions for future research. First, extending ReLaSH to dynamic and temporal hypergraphs, weighted hyperlinks, and richer attribute modalities would broaden its applicability while introducing new challenges. Second, conditional generation (e.g., conditioning on subsets of nodes, attributes, or constraints) could enable targeted simulation and counterfactual analysis. Third, tighter theoretical results, such as uncertainty quantification for generated structures, would further strengthen its theoretical guarantees. We view ReLaSH as a step toward reliable and flexible generative modeling for hypergraph data.

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

APPENDIX

We provide additional theoretical results, experimental details with extended numerical results, and proofs in the appendix. Appendix A presents additional theoretical results. Appendix B collects supplementary materials for experiments, including additional numerical results on simulated and real-world data and details of the evaluation metrics and implementations. Proofs of the theoretical results are in Appendix C.

## A  ADDITIONAL THEORETICAL RESULTS

We start with a discussion of the identifiability of the embedding model introduced in Section 2.3. First, we emphasize that the identifiability conditions in Theorem 1 are sufficient but not necessary. For instance, (C3) may be replaced by (C3*) $B_1$ contains a $k_1 \times k_1$ unit lower-triangular matrix, or (C3**) $p^{-1} B_1^\top B_1 = I_{k_1}$. The identifiability conditions in Theorem 1 are chosen according to the literature (Wu et al., 2025). Other identifiability conditions can be adopted in our framework as well. Additionally, we address identifiability up to column sign flips by fixing the sign of all coordinates in the first hyperlink embedding vector and adjusting the estimators accordingly. In the following, we specify the constraint sets for the embedding and the parameter estimation procedure. Let the degree parameter be $\alpha = (\alpha_1, \cdots, \alpha_n)^\top \in \mathbb{R}^n$, the hyperlink embedding matrix be $U_m = (u_1, \cdots, u_m)^\top = (U_m^{(1)} \ U_m^{(2)} \ U_m^{(3)}) \in \mathbb{R}^{m \times K}$, where $K = k_1 + k_2 + k_3$. Furthermore, denote the block submatrices of $U_m$ as $U_m^{(12)} = (U_m^{(1)} \ U_m^{(2)})$ and $U_m^{(23)} = (U_m^{(2)} \ U_m^{(3)})$. Under the parametrization conditions outlined in Theorem 1, we define the feasible region:

$$
\begin{aligned}
\mathcal{F}(\Theta) := \{ (U_m, Z, B, \alpha, \gamma) \mid & \Theta = [\Theta^H, \Theta^X], \Theta^H = \mathbf{1}_m \alpha^\top + U_m^{(23)} Z^\top, \Theta^X = \mathbf{1}_m \gamma^\top + U_m^{(12)} B^\top, \\
& \max\{\|\alpha - \bar{\alpha}_{m,n} \mathbf{1}_n\|_\infty, \max_{i \in [n]} \|z_i\|_2\} \le M_1, \max\{\|\gamma\|_\infty, \max_{i \in [p]} \|B_{i*}\|_2\} \le M_2, \\
& \max_{j \in [m]} \|u_j\|_2 \le (M_1 \wedge M_2), \text{ and } -C_{m,n} \le \bar{\alpha}_{m,n} \le -C' C_{m,n} \}.
\end{aligned}
$$
(2)

where $M_1, M_2 > 0$, $C' \in (0, 1)$, and the boundary parameter $C_{m,n}$, which may diverge slowly as $m, n \to \infty$, accounts for the sparsity of hyperlinks. An ideal choice of $C_{m,n}$ would satisfy $\bar{\alpha}_{m,n} \asymp -C_{m,n}$ and $\exp(\bar{\alpha}_{m,n}) \asymp \exp(-C_{m,n})$. Following Proposition 3.2 in Wu et al. (2024), we suggest setting $C_{m,n} = -C'' \log\left(\sum_{j=1}^m |e_j|/(mn)\right)$ for some constant $C'' > 1$.

In the embedding step, we attain $(\hat{U}, \hat{Z}, \hat{B}, \hat{\alpha}, \hat{\gamma})$ by solving:

$$
\min_{(U_m, Z, B, \alpha, \gamma) \in \mathcal{F}(\Theta)} \ell(U, Z, B, \alpha, \gamma).
$$
(3)

In what follows, we present additional theoretical results that characterize the error rates of the embedding and estimation procedure.

**Theorem 4** (F-consistent estimation of $\Theta^H, \Theta^X$). *Let $\Theta^* = [\Theta^{H*} \ \Theta^{X*}]$ be the true parameters, and $\hat{\Theta} = [\hat{\Theta}^H \ \hat{\Theta}^X]$ be the estimated version derived by the optimizers $(\hat{U}, \hat{Z}, \hat{B}, \hat{\alpha}, \hat{\gamma})$. Under Assumption 1 and the condition that $(U^*, Z^*, B^*, \alpha^*, \gamma^*) \in \mathcal{F}(\Theta)$, we have*

$$
\|\hat{\Theta} - \Theta^*\|_F = O_p \left( \frac{\sqrt{(m \vee n) \exp(\bar{\alpha}_{m,n}) \log(m \vee n) + 4\lambda^2 (m \vee p)}}{(\exp(-C_{m,n}) \wedge \lambda)} \right).
$$

**Remark 1.** *Theorem 4 implies that if $m \asymp n \asymp p$, $\exp(\bar{\alpha}_{m,n}^*) \asymp \exp(-C_{m,n})$, and with Assumption 1, the optimizers achieve F-consistent, i.e. $\frac{1}{\sqrt{m(n+p)}} \|\hat{\Theta} - \Theta^*\|_F = o_p(1)$, implies that for any set of $\mathcal{U}_m$ from the distribution $\mathbb{P}_U$, the estimation is precise.*

**Remark 2.** *We conducted simulation experiments regarding the embedding procedure, as detailed in Appendix B.2, to validate the theoretical error rate.*

**Remark 3.** *The error bound in Theorem 4 depends jointly on the sparsity parameter of the hypergraph $\bar{\alpha}_{m,n}$ and the regularization weight $\lambda$. Consequently, $\lambda$ should be tuned in accordance with the observed sparsity in order to balance these two sources of error. Simulation results presented in Appendix B.2 further indicate that while the tuning of $\lambda$ does not substantially affect the accuracy of estimation, it may influence the stability of the gradient descent procedure.*

**Remark 4.** *Moreover, when $m \asymp n \asymp p$ and $\lambda \asymp \exp(\bar{\alpha}^*_{m,n}) \asymp \exp(-C_{m,n})$, the error rate in Theorem 4 is dominated by the hypergraph embedding error rate (Wu et al., 2024), and the error from the attribute part is of smaller order.*

**Corollary 1.** *(F-consistent estimation of the embedding and parameters) Let $(\hat{U}, \hat{Z}, \hat{B}, \hat{\alpha}, \hat{\gamma})$ be the optimizers, and $(U^*, Z^*, B^*, \alpha^*, \gamma^*)$ be the true embeddings and parameters satisfying the constraints in the feasible region $\mathcal{F}(\Theta)$. Denote*

$$\delta_{m,n,p} = \frac{\sqrt{(m \vee n)\exp(\bar{\alpha}_{m,n})\log(m \vee n) + 4\lambda^2(m \vee p)}}{\sqrt{m}\left(\exp(-C_{m,n}) \wedge \lambda\right)}.$$

*As $(m,n) \to \infty$ and with Assumption 1, 2, we have F-consistent error of the embedding and parameters that $\|\hat{\alpha} - \alpha^*\|_2 = O_p(\delta_{m,n,p})$, $\|\hat{U} - U^*\|_F = O_p\left(\sqrt{m(n+p)/(np)}\delta_{m,n,p}\right)$, $\|\hat{Z} - Z^*\|_F = O_p(\delta_{m,n,p})$, $\|\hat{\gamma} - \gamma^*\|_2 = O_p(\delta_{m,n,p})$, $\|\hat{B} - B^*\|_F = O_p(\delta_{m,n,p})$.*

To address identifiability, we apply the transformations in Remark 5, whose details are deferred to Appendix C, and which introduce an additional source of error in the embedding procedure. The following theorem characterizes this error.

**Theorem 5.** *Let $\mathcal{H}([n], \mathcal{E}_m, \mathcal{X}_m)$ be the hypergraph generated from underlying embeddings $\mathcal{U}_m$, $\mathcal{Z}_n$, and parameters $\alpha$, $\gamma$, $B$, where $\mathcal{U}_m$ are $m$ realizations from $\mathbb{P}_U$. Let $(\hat{U}, \hat{Z}, \hat{B}, \hat{\alpha}, \hat{\gamma})$ be the optimizers, then with Assumption 1, 2 and as $(m,n) \to \infty$, we have $\|\hat{\alpha} - \alpha\|_2 = O_p(\delta_{m,n,p})$, $\|\hat{U} - U_m\|_F = O_p\left(\sqrt{m(n+p)/(np)}\delta_{m,n,p}\right)$, $\|\hat{Z} - Z\|_F = O_p(\delta_{m,n,p})$, $\|\hat{\gamma} - \gamma\|_2 = O_p(\delta_{m,n,p})$, $\|\hat{B} - B\|_F = O_p(\delta_{m,n,p})$.*

Next, we discuss $\Delta_{\mathbb{P}_U}$ via a discretization strategy (Wu et al., 2025). In general, analyzing the joint embedding-related error $\Delta_{\mathbb{P}_U\text{-estimation}}$ is challenging, as it requires comparing the distribution of the estimated embeddings based on $m$ observations, $\mathcal{U}_m = \{u_1, \ldots, u_m\}$, from the continuous latent distribution $\mathbb{P}_U$. To address this, we adopt a discretization strategy to bridge these two quantities and analyze the error term step by step. We first introduce Assumption 4 and analyze $\Delta_{\mathbb{P}_U\text{-estimation}}$ under these conditions.

**Assumption 4** (Support of $\mathbb{P}_U$). *The support of $\mathbb{P}_U$ satisfies $supp(\mathbb{P}_U) \subset \{u \in \mathbb{R}^k : \|u\|_\infty \leq (M_1 \wedge M_2)\}$ for the constant $M_1, M_2$ in 2.*

Let $U^{\text{dis}}$ denote a discretized version of $U$, with distribution $p_{U^{\text{dis}}} = d\mathbb{P}_{U^{\text{dis}}}/d\mu(U^{\text{dis}})$, defined as $p_{U^{\text{dis}}}(u^{\text{dis}}) = \int_{[u^{\text{dis}} - \frac{1}{2\gamma_{m,n,p}}, u^{\text{dis}} + \frac{1}{2\gamma_{m,n,p}}) \cap supp(\mathbb{P}_U)} p_U(u)\, d\mu(u)$ for any $u^{\text{dis}} \in \mathcal{A}_{C, \gamma_{m,n,p}^{-1}} = \{a \in \mathbb{R}^k : \|a\|_\infty \leq C, \ a_i \in \frac{1}{\gamma_{m,n,p}} \cdot \mathbb{Z} \ \forall i \in [k]\}$, where $C = M_1 \wedge M_2$, and $\gamma_{m,n,p}$ is a sequence diverging to $\infty$ as $m, n, p \to \infty$. Since $\mathbb{P}_{U^{\text{dis}}}$ and $\mathbb{P}_U$ are defined on different sample spaces, they are not directly comparable. To address this, we introduce the random vector $U^{\text{pc}}$, defined on the same sample space as $U$, whose density is piecewise constant.

Without loss of generality, let $supp(\mathbb{P}_U) = [-C - (2\gamma_{m,n,p})^{-1}, C + (2\gamma_{m,n,p})^{-1})^k$, which forms a hypercube centered at 0 with side length $2C + (\gamma_{m,n,p})^{-1}$ in each coordinate. This hypercube can be perfectly partitioned into finitely many disjoint intervals of length $\gamma_{m,n,p}^{-1}$ per coordinate. Define $\mathbb{P}_{U^{\text{pc}}}$ as follows: for any $u \in supp(\mathbb{P}_U)$, let $u^{\text{dis}}(u) \in \mathcal{A}_{C, \gamma_{m,n,p}^{-1}}$ satisfy $u \in [u^{\text{dis}}(u) - (2\gamma_{m,n,p})^{-1}, u^{\text{dis}}(u) + (2\gamma_{m,n,p})^{-1})$, and define the probability density of $U^{\text{pc}}$ as $p_{U^{\text{pc}}}(u) := \frac{d\mathbb{P}_{U^{\text{pc}}}}{d\mu(U^{\text{pc}})}(u) = \gamma_{m,n,p}^k \int_{[u^{\text{dis}}(u) - (2\gamma_{m,n,p})^{-1}, u^{\text{dis}}(u) + (2\gamma_{m,n,p})^{-1})} p_U(u)\, d\mu(u)$.

**Lemma 2.** *$U^{pc}$ is a well-defined random variable on $supp(\mathbb{P}_U)$ in the sense that its probability density $p_{U^{pc}}(\cdot)$ integrated to 1.*

Next, we show that the distance between $p_{U^{\text{pc}}}(u)$ and $p_U(u)$ is uniformly bounded, which quantifies the distance between $\mathbb{P}_{U^{\text{pc}}}$ and $\mathbb{P}_U$.

**Theorem 6.** *Suppose $p_U(u)$ is L-Lipschitz continuous on its support. Then, for any $u \in supp(\mathbb{P}_U)$, there is $|p_{U^{pc}}(u) - p_U(u)| \leq L\sqrt{k}\,\gamma_{m,n,p}^{-1}$.*

In our setting, the hypergraph with $m$ hyperedges is constructed on the $m$ observations $\mathcal{U}_m$. Let $\mathbb{P}_{\mathcal{U}_m^{\text{dis}}}$ denote the empirical distribution of $m$ realizations $\mathcal{U}_m^{\text{dis}} = \{u_1^{\text{dis}}, u_2^{\text{dis}}, \cdots, u_m^{\text{dis}}\}$, defined by

$p_{\mathcal{U}_m^{\text{dis}}}(u) = \#\{u \in \mathcal{U}_m\}/m, \quad \forall u \in \mathcal{A}_{C,\gamma_{m,n,p}^{-1}}$. We can then show that the distance between $\mathbb{P}_{U^{\text{dis}}}$ and the empirical distribution $\mathbb{P}_{\mathcal{U}_m^{\text{dis}}}$ based on $m$ realizations is bounded as in Lemma 3.

**Lemma 3.** *Let $\mathcal{U}_m$ be a collection of $m$ realizations from $\mathbb{P}_{U^{\text{dis}}}$. Let $p_{\max} = \max_u p_{U^{\text{dis}}}(u)$ and $p_{\min} = \min_u p_{U^{\text{dis}}}(u)$. For any sequence of $\varepsilon_{m,n,p}$ satisfying $p_{\max} > \varepsilon_{m,n,p} \gg p_{\max}\sqrt{k\log(2C\gamma_{m,n,p})/(mp_{\min})}$ and $m \gg k\log(2C\gamma_{m,n,p})/p_{\min}$ as $m,n,p \to \infty$, we have $\mathbb{P}\{\forall u \in \mathcal{A}_{C,\gamma_{m,n,p}^{-1}}, |p_{U^{\text{dis}}}(u) - p_{\mathcal{U}_m^{\text{dis}}}(u)| \le \varepsilon_{m,n,p}\} \to 1$ as $m,n,p \to \infty$.*

When $\mathcal{U}_m$ is replaced by $\mathcal{U}_m^{\text{dis}}$, the corresponding estimators must be adjusted. We project the estimators $\{\hat{u}_1, \cdots, \hat{u}_m\}$ onto $\mathcal{A}_{C,\gamma_{m,n,p}^{-1}}$ by defining $\hat{u}_j^{\text{dis}} = \arg\min_{x \in \mathcal{A}_{C,\gamma_{m,n,p}^{-1}}} \|x - \hat{u}_j\|$. Then $\hat{u}_1^{\text{dis}}, \hat{u}_2^{\text{dis}}, \cdots, \hat{u}_m^{\text{dis}}$ have a probability mass function defined as $p_{\hat{\mathcal{U}}_m^{\text{dis}}}(u) = \#\{u \in \hat{\mathcal{U}}_m\}/m$. To better understand the distance between the distributions of $\mathcal{U}_m^{\text{dis}}$ and $\hat{\mathcal{U}}_m$, we require tighter entry-wise consistency results beyond the average error rate of the estimators, which is left as future work.

# B  SUPPLEMENTARY FOR NUMERICAL RESULTS

## B.1  ALGORITHMS

To jointly embed the hypergraph and its corresponding attributes by minimizing the loss function $\ell(U, Z, B, \alpha, \gamma)$, we consider using a projected gradient descent algorithm similar to (Ma et al., 2020), which is commonly employed for solving constrained optimization problems. The algorithm is summarized in Algorithm 1 below.

---

**Algorithm 1** Projected Gradient Descent for Joint Embedding

---

**Require:** Initial embeddings $U_{(0)}$, $Z_{(0)}$, $\alpha_{(0)}$, initial parameters $B_{(0)},\gamma_{(0)}$, observed hypergraph connection matrix $H$ and attribute matrix $X$, learning rate $\eta$, likelihood weight parameter $\lambda$, maximum number of iterations $T$

1:  **for** $t = 1$ to $T$ **do**
2:  Compute $\Theta_{(t-1)}^X = \mathbf{1}_m\gamma_{(t-1)}^\top + U_{(t-1)}^{(12)}B^\top$, and $\Theta_{(t-1)}^H = \mathbf{1}_m\alpha_{(t-1)}^\top + U_{(t-1)}^{(23)}Z^\top$
3:  $U_{(t)}^{(1)} = U_{(t-1)}^{(1)} - \eta_{U^{(1)}}\nabla_{U^{(1)}}\ell_{(t-1)} = U_{(t-1)}^{(1)} + \lambda\eta_{U^{(1)}}(X - f_A'(\Theta_{(t-1)}^X))B_{1,(t-1)}$
4:  $U_{(t)}^{(2)} = U_{(t-1)}^{(2)} - \eta_{U^{(2)}}\nabla_{U^{(2)}}\ell_{(t-1)} = U_{(t-1)}^{(2)} + \eta_{U^{(2)}}\{\lambda(X - f_A'(\Theta_{(t-1)}^X))B_{2,(t-1)} + (H - \sigma(\Theta_{(t-1)}^H)Z_{2,(t-1)})\}$
5:  $U_{(t)}^{(3)} = U_{(t-1)}^{(3)} - \eta_{U^{(3)}}\nabla_{U^{(3)}}\ell_{(t-1)} = U_{(t-1)}^{(3)} + \eta_{U^{(3)}}(H - \sigma(\Theta_{(t-1)}^H))Z_{3,(t-1)}$
6:  $Z_{(t)} = Z_{(t-1)} - \eta_Z\nabla_Z\ell_{(t-1)} = Z_{(t-1)} + \eta_Z(H - \sigma(\Theta_{(t-1)}^H))B_{2,(t-1)}$
7:  $\alpha_{(t)} = \alpha_{(t-1)} - \eta_\alpha\nabla_\alpha\ell_{(t-1)} = \alpha_{(t-1)} + \eta_\alpha(H - \sigma(\Theta_{(t-1)}^H))^\top\mathbf{1}_m$
8:  $B_{(t)} = B_{(t-1)} - \eta_B\nabla_B\ell_{(t-1)} = B_{(t-1)} + \lambda\eta_B(X - f_A'(\Theta_{(t-1)}^X))U_{(t-1)}^{(2)}$
9:  $\gamma_{(t)} = \gamma_{(t-1)} - \eta_\gamma\nabla_\gamma\ell_{(t-1)} = \gamma_{(t-1)} + \lambda\eta_\gamma(X - f_A'(\Theta_{(t-1)}^X))^\top\mathbf{1}_m$
10:  Project the parameters and embeddings $(U_{(t)}, Z_{(t)}, \alpha_{(t)}, B_{(t)})$ to the constraint set, with the transformation in Remark 5.
11:  **end for**
12:  **return** $(U_{(T)}, Z_{(T)}, \alpha_{(T)}, B_{(T)}, \gamma_{(T)})$ as $(\hat{U}, \hat{Z}, \hat{\alpha}, \hat{B}, \hat{\gamma})$.

---

For the initial values $(U_{(0)}, Z_{(0)}, \alpha_{(0)}, B_{(0)},\gamma_{(0)})$ and choices of step sizes, we adapt the initialization method based on universal singular value thresholding (Chatterjee, 2015) and step size choice proposed by (Ma et al., 2020). The initialization algorithm is shown in Algorithm 2. Also, Moreover, the step sizes in Algorithm 1 are set as:

$$\eta_\alpha = \eta_\gamma = \frac{\eta}{2m}, \eta_Z = \frac{\eta}{\|Z_{(0)}\|_{\text{op}}^2}, \eta_B = \frac{\eta}{\|B_{(0)}\|_{\text{op}}^2}, \eta_{U^{(i)}} = \frac{\eta}{\|U_{(0)}^{(i)}\|_{\text{op}}^2} \text{for } i = 1, 2, 3.$$

Moreover, to select the dimensions $(k_1, k_2, k_3)$ of the joint latent space, we propose the *"hunt then trim" (HTT)* algorithm. HTT consists of three steps. In the first and second steps, it hunts for the latent space dimensions that generate the hypergraph, and then the dimensions that are orthogonal

---

**Algorithm 2** Initialization by Singular Value Thresholding

---

**Require:** Hypergraph matrix $H$, hyperlink attribute matrix $X$, latent embedding dimensions $(k_1, k_2, k_3)$.

1: Let $\sum_i \sigma_i u_i v_i^\top$ be the singular value decomposition of $H$, and denote $\tilde{P} = \sum_{\sigma_i \geq \tau_H} \sigma_i u_i v_i^\top$ as the low-rank approximation of $H$. Elementwisely project $\tilde{P}$ into $[\frac{1}{2}e^{-M_1}, \frac{1}{2}]$ to obtain $\bar{P}$, and take $\bar{\Theta}^H = \text{logit}(\bar{P})$.

2: Take $\alpha_{(0)} = \frac{1}{m}\bar{\Theta}^H \mathbf{1}_m$, and let $U'\Lambda'V'^\top$ as the singular value decomposition of $\bar{\Theta}^H - \alpha_{(0)}\mathbf{1}_m^\top = J_m\bar{\Theta}^H$, then take $U_{(0)}^{(23)} = \sqrt[4]{\frac{m}{n}}U'_{k_2+k_3}\Lambda'^{1/2}_{k_2+k_3}$, $Z_{(0)} = \sqrt[4]{\frac{n}{m}}V'_{k_2+k_3}\Lambda'^{1/2}_{k_2+k_3}$.

3: Treat $X$ as a noisy version of $\Theta^X$, denoted as $\bar{\Theta}^X$.

4: Take $\gamma_{(0)} = \frac{1}{m}\bar{\Theta}^X \mathbf{1}_m$, and regress $J_m\bar{\Theta}^X$ on $U_{(0)}^{(2)}$ to obtain $B_{2,(0)}$. Then take the residual as $R = J_m\bar{\Theta}^X - U_{(0)}^{(2)}B_{2,(0)}^\top$, denote its singular value decomposition as $\tilde{U}\tilde{\Lambda}\tilde{V}^\top$, and denote $B_{1,(0)} = \sqrt[4]{\frac{p}{m}}\tilde{V}_{k_1}\tilde{\Lambda}^{1/2}_{k_1}$, $U_{(0)}^{(1)} = \sqrt[4]{\frac{m}{p}}\tilde{U}_{k_1}\tilde{\Lambda}^{1/2}_{k_1}$ to satisfy the identifiability conditions.

5: **return** $(U_{(0)}, Z_{(0)}, \alpha_{(0)}, B_{(0)}, \gamma_{(0)})$

---

to those but drive attribute generation. In the third step, it then trims the dimensions that drive hypergraph formation but (almost) the attributes. The algorithm is summarized in Algorithm 3 below.

## B.2 ADDITIONAL SIMULATION RESULTS OF THE EMBEDDING PROCEDURE

In this section, we present additional simulation results of the embedding procedure.

Specifically, for a $(k_2+k_3)$-dimensional latent space, we randomly divide the $n$ nodes into $(k_2+k_3)$ nearly equal-sized groups, where the group sizes may differ by at most 1. The node embeddings $\mathcal{Z}_n$ are then independently generated as follows. For node $i$ belonging to the $t$-th group, its embedding $z_i$ is drawn from the truncated Gaussian distribution $\mathcal{N}_{[-1,1]}(\mathbf{1}_{k_2+k_3} - e_t, \Sigma_z)$, where $e_t$ is the unit vector with the $t$-th element being 1 and the others 0, and $[-1,1]$ indicates truncation on each coordinate.

The hyperlink embeddings $\mathcal{U}_m$ are generated from a Gaussian distribution $\mathcal{N}_{[-1,1]}(0, \Sigma_U)$, ensuring that the identifiability condition $\sum_{j=1}^m u_j = \mathbf{0}_k$ is approximately preserved. For the regression parameters $(B, \gamma)$, we generate $\gamma_i$ i.i.d. from $\text{Uniform}([-1,1])$, and each column $B_i$ independently sampled from $\mathcal{N}(0, \Sigma_B)$. The variance-covariance matrices $\Sigma_B, \Sigma_z, \Sigma_U$ are defined such that the $(i,j)$-th entry is $0.2\rho^{|i-j|}$ with $\rho \in \{0, 0.5\}$. The degree heterogeneity parameters $\alpha_i$ are generated from the uniform distribution $[\bar{\alpha}_{m,n} - 1, \bar{\alpha}_{m,n} + 1]$ for the sparsity parameter $\bar{\alpha}_{m,n}$ specified in each experimental setting.

We first examine how the sample size and latent dimension jointly affect estimation accuracy. We set $k_1 = k_2 = k_3$ with $k \in \{6, 9, 12\}$, $\bar{\alpha}^*_{m,n} = -3$, and $m = 10n = 10p$ with $n \in \{100, 200, \ldots, 1000\}$. The weight parameter $\lambda$ is chosen such that $\lambda = 0.2$, matching the order of $\exp(\alpha^*_{m,n})$. Figure 4, 5 shows the relative Frobenius error of $\Theta^H, \Theta^X, \Theta$ as a function of $n$ for different latent dimensions $k$. The error decreases approximately at the rate $(m \wedge n \wedge p)^{-1/2} = n^{-1/2}$, in agreement with the bound in Theorem 4. Increasing $k$ leads to higher estimation errors: intuitively, a larger latent dimension increases model complexity, which amplifies variance in estimation. This dependence on $k$ does not explicitly appear in the bound of Theorem 4 (which assumes fixed $k$), but follows from its proof in Appendix C. When the coordinates of the embeddings are more correlated (when we set $\rho = 0.5$), the error remains of similar magnitude, with a mild increase as $k$ increases.

To further investigate the scaling behavior, we plot the logarithm of the estimation error against $\log(m \wedge n \wedge p)$ in Figures 6 and 7. In both cases, the relationship appears approximately linear, which aligns with the $n^{-1/2}$ rate predicted by Theorem 4. As shown in Table 4, the observed linear trend, with a slope of approximately $-0.5$, is consistent with the theoretical bounds, which suggest that the estimation errors for $\Theta^X, \Theta_H$, and $\Theta$ are of order $O_p(n^{-1/2})$.

---

**Algorithm 3** Hunt-Then-Trim: Selection of Joint Latent Space Dimensions $(k_1, k_2, k_3)$.

---

**Require:** Hypergraph matrix $H$, hyperlink attribute matrix $X$.
1: **Step 1: Selection of $k_2 + k_3$ via Cross-Validation**
2: **for** each candidate dimension $k_{23} = k_2 + k_3$ **do**
3:    Split population hypergraph into training (80%) and testing (20%) sets.
4:    Perform embedding procedure on training set to obtain $\mathcal{U}_{\text{train}}^{k_{23}}$.
5:    Perform embedding procedure on testing set to obtain $\mathcal{U}_{\text{test}}^{k_{23}}$.
6:    Compute Fréchet Distance: $\text{FD}_{k_{23}} = \text{FD}(\mathcal{U}_{\text{train}}^{k_{23}}, \mathcal{U}_{\text{test}}^{k_{23}})$.
7: **end for**
8: Select $k_2 + k_3 = \arg\min_{k_{23}} \text{FD}_{k_{23}}$.
9: Obtain embedded vectors $\{\hat{u}_j^{(23)}\}_{j \in [m]}$ per embedding procedure with selected $k_{23}$.
10: **Step 2: Regression and Sequential Hypothesis Testing for $k_1$**
11: Let $X = (x_1, \ldots, x_m)^\top$, $U^{(23)} = (\hat{u}_1^{(23)}, \ldots, \hat{u}_m^{(23)})^\top$.
12: Regress $X$ over $U^{(23)}$:
13:    $X = 1_n \mu^\top + U^{(23)} B^\top + \mathcal{E}$.
14: Obtain fitted values $\hat{X}, \hat{U}, \hat{\mu}, \hat{B}$ and residuals $\hat{\mathcal{E}} = X - \hat{X}$.
15: **for** $l = 0$ **to** $k_{\max}$ **do**
16:    Test $H_0 : k_1 = l$ vs $H_1 : l + 1 \leq k_1 \leq k_{\max}$.
17:    Compute $r(l) = (\phi_{l+1} - \phi_{k_{\max}+1})/(\phi_{k_{\max}+2})$, where $\phi_l$ is the $l$th largest eigenvalue of $\text{Cov}(\hat{\mathcal{E}})$.
18:    **If** the observed $r(l)$ exceeds the threshold from the joint Tracy–Widom distribution, **reject** $H_0$, and continue testing $H_0 : k_1 = l + 1$.
19:    **Else** accept $H_0$ and set the estimated $k_1 = l$.
20: **end for**
21: **Step 3: Simultaneous Estimation of $k_2$ and $k_3$**
22: **for** $1 \leq l \leq k_2 + k_3$ **do**
23:    Let $B_{\cdot,l}$ denote the $l$-th column of the loading matrix $\hat{B}$.
24:    Test $H_0 : B_{\cdot,l} = 0_{p \times 1}$ versus $H_1 : B_{\cdot,l} \neq 0_{p \times 1}$.
25:    Compute $s(l) = \left(2 \sum_{j=1}^p V_{jl}^2\right)^{-1/2} \sum_{j=1}^p \left(m B_{jl}^2 - V_{jl}\right)$, where $V_{jl} = (U^{(23)\top} U^{(23)})_{ll}^{-1} (\text{Cov}(\hat{\mathcal{E}}))_{jj}$.
26:    Compare $s(l)$ to the standard normal distribution.
27:    If $s(l)$ significant, infer $\mathcal{E}_{\cdot,l} \neq 0$ (belonging to $U^{(3)}$), else $U^{(2)}$.
28: **end for**
29: Simultaneously estimate $U^{(2)}, U^{(3)}, k_2, k_3$ according to the hypothesis testing results.
    **return** Dimensions $(k_1, k_2, k_3)$ of the joint latent space.

---

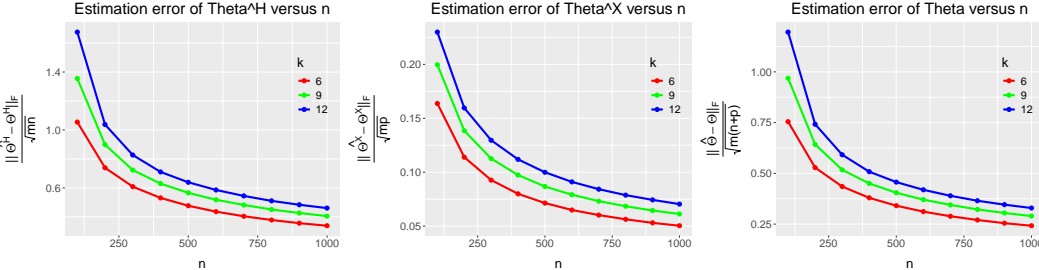

Figure 4: Estimation error under $\rho = 0$ versus sample size based on 30 Monte Carlo repetitions.

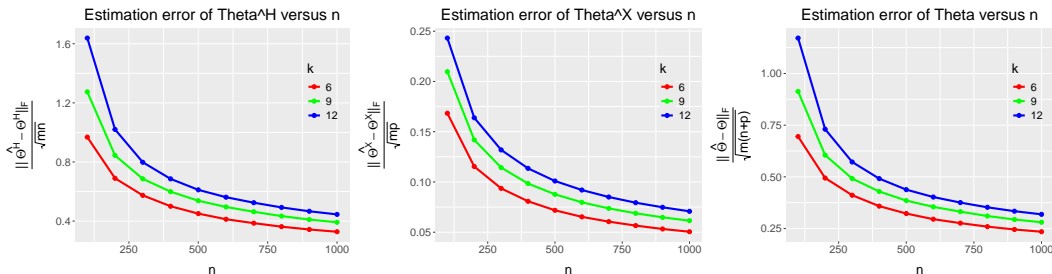

Figure 5: Estimation error under $\rho = 0.5$ versus sample size based on 30 Monte Carlo repetitions.

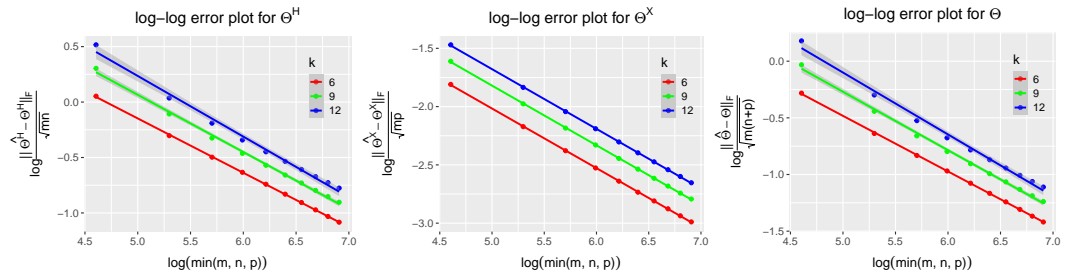

Figure 6: Log–log plot of estimation error versus sample size for $\rho = 0$, based on 30 independent Monte Carlo repetitions.

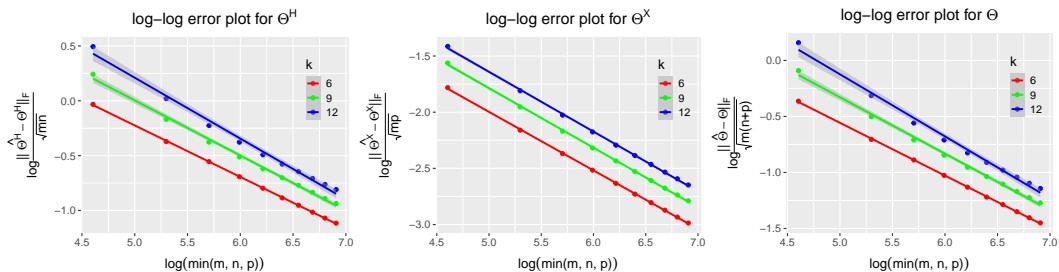

Figure 7: Log–log plot of estimation error versus sample size for $\rho = 0.5$, based on 30 independent Monte Carlo repetitions.

| Error Type | $k$ | $\rho = 0$ | | $\rho = 0.5$ | |
| --- | --- | --- | --- | --- | --- |
| | | Slope | 95% CI of Slope | Slope | 95% CI of Slope |
| $\Theta^H$ | 6 | -0.4902 | $[-0.4951, -0.4854]$ | -0.4697 | $[-0.4744, -0.4650]$ |
| | 9 | -0.5158 | $[-0.5365, -0.4951]$ | -0.5028 | $[-0.5262, -0.4795]$ |
| | 12 | -0.5470 | $[-0.5852, -0.5087]$ | -0.5546 | $[-0.5948, -0.5145]$ |
| $\Theta^X$ | 6 | -0.5117 | $[-0.5150, -0.5085]$ | -0.5212 | $[-0.5270, -0.5153]$ |
| | 9 | -0.5125 | $[-0.5163, -0.5087]$ | -0.5300 | $[-0.5387, -0.5213]$ |
| | 12 | -0.5128 | $[-0.5165, -0.5090]$ | -0.5321 | $[-0.5418, -0.5223]$ |
| $\Theta$ | 6 | -0.4907 | $[-0.4955, -0.4860]$ | -0.4711 | $[-0.4758, -0.4663]$ |
| | 9 | -0.5157 | $[-0.5360, -0.4954]$ | -0.5035 | $[-0.5265, -0.4806]$ |
| | 12 | -0.5462 | $[-0.5837, -0.5088]$ | -0.5541 | $[-0.5935, -0.5147]$ |

Table 4: Slope and confidence intervals for $\rho = 0$ (left) and $\rho = 0.5$ (right).

From the perspective of the error bound derived in Theorem 4, we regress the estimation error of $\Theta^H$ on $\frac{\log{(m \wedge n)}}{(m \vee n)}$ using a log–log plot, as illustrated in Figure 8. It is noteworthy that the regression based on this term performs better than directly regressing on $(m \wedge n)$, with the slope of nealy $0.5$.

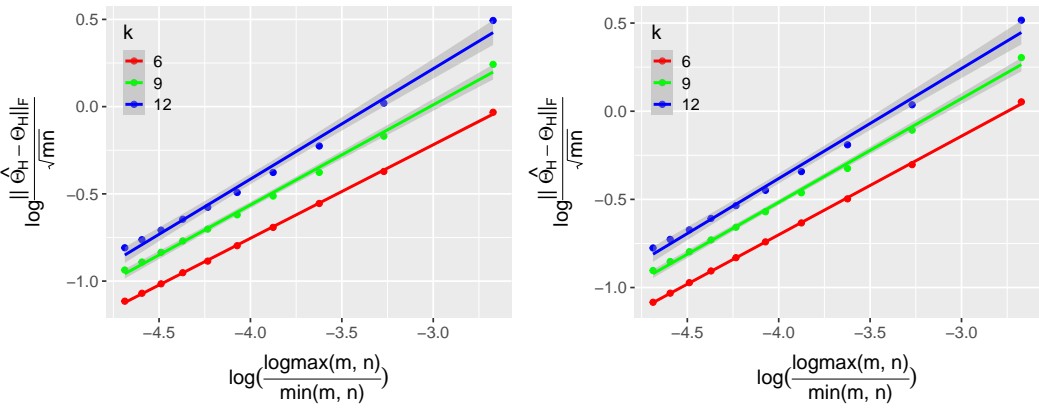

Figure 8: Log–log plot of estimation error of $\Theta^H$ versus $\frac{\log{(m \vee n)}}{(m \wedge n)}$ for $\rho = 0$ (left) and $\rho = 0.5$ (right), based on 30 independent Monte Carlo repetitions.

| $k$ | Slope ($\rho = 0$) | 95% CI ($\rho = 0$) | Slope ($\rho = 0.5$) | 95% CI ($\rho = 0.5$) |
|---|---|---|---|---|
| 6 | 0.5596 | [0.5522, 0.5669] | 0.5361 | [0.5285, 0.5438] |
| 9 | 0.5886 | [0.5621, 0.6150] | 0.5738 | [0.5443, 0.6032] |
| 12 | 0.6240 | [0.5773, 0.6707] | 0.6327 | [0.5838, 0.6817] |

Table 5: Slope and confidence intervals for $\rho = 0$ (left) and $\rho = 0.5$ (right).

To be more precise, our analysis focuses on two aspects: (i) the effect of varying $p$ on the estimation error of $\Theta^X$, and (ii) the effect of varying $n$ on the estimation error of $\Theta^H$. Preliminary experiments suggest that the correlation parameter $\rho$ in hyperlink embeddings has a negligible effect on the results. Therefore, we fix $\rho = 0.5$ for all subsequent experiments, as this correlation introduced is commonly encountered in various scenarios. Additionally, we scale $k$ to $k \in \{6, 12, 24\}$ to demonstrate the robustness of our algorithm in handling more complex data.

We fix $m = 5000$, $n = 500$, $\lambda = 0.2$, $\rho = 0.5$, $\bar{\alpha}_{m,n} = -3$, and vary $p \in \{100, 200, \ldots, 1000\}$ to estimate the errors of $\Theta^H$ and $\Theta^X$ (Figure 9). The results confirm our theoretical expectation: the estimation error of $\Theta^H$ remains generally unchanged as $p$ varies, because the associated error term does not depend on $p$ and the attribute dimension has minimal impact on hypergraph structure estimation. In contrast, the estimation error of $\Theta^X$ decreases with increasing $p$, in an approximately power-law pattern. This is intuitive—higher-dimensional attribute information facilitates more accurate recovery of the underlying attribute-based structure.

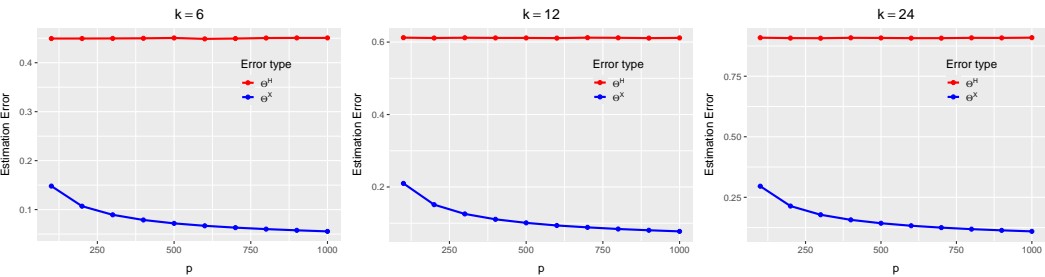

Figure 9: Estimation error versus attribute dimension $p$, averaged over 30 Monte Carlo repetitions.

To further examine this scaling relationship, we plot the estimation error of $\Theta^X$ against $p$ on a log–log scale (Figure 10). The observed linear trend confirms the power-law behavior. The figure also shows the 95% confidence interval for the slope and for the expected value of the error.

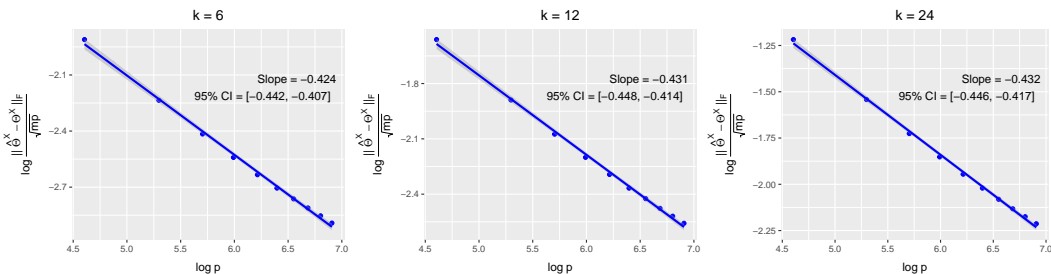

Figure 10: Log–log plot of estimation error versus $p$, based on 30 independent Monte Carlo repetitions.

Similarly, we exanmine the estimation error of both $\Theta^H$ and $\Theta^X$ as $n$ varies in the set $\{100, 200, \ldots, 1000\}$, while fixing $m = 5000$, $p = 500$, $\bar{\alpha}_{m,n} = -3$, $\rho = 0.5$, and $\lambda = 0.2$. As shown in Figure 11, the error associated with the attribute-based part, $\Theta^X$, remains relatively unchanged as the node size $n$ increases. In contrast, the error associated with hypergraph estimation, $\Theta^H$, exhibits a clear power-law trend.

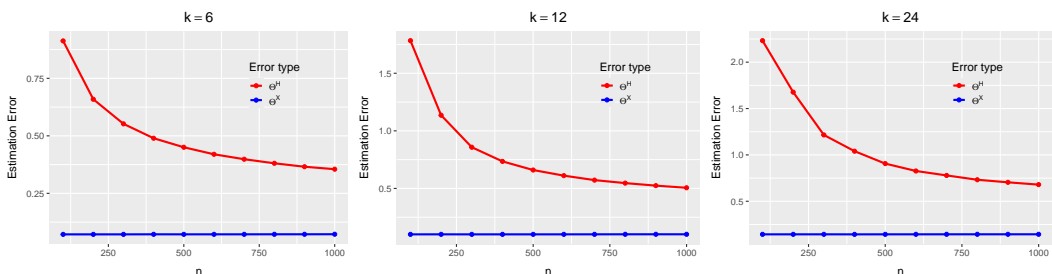

Figure 11: Estimation error versus node size $n$, averaged over 30 Monte Carlo repetitions.

To gain further insight, we plot the estimation error on a log–log scale, as shown in Figure 12. This plot illustrates that the error associated with hypergraph modeling ($\Theta^H$) follows a power-law trend with respect to $n$. The figure also shows the 95% confidence interval for the slope and for the expected value of the error.

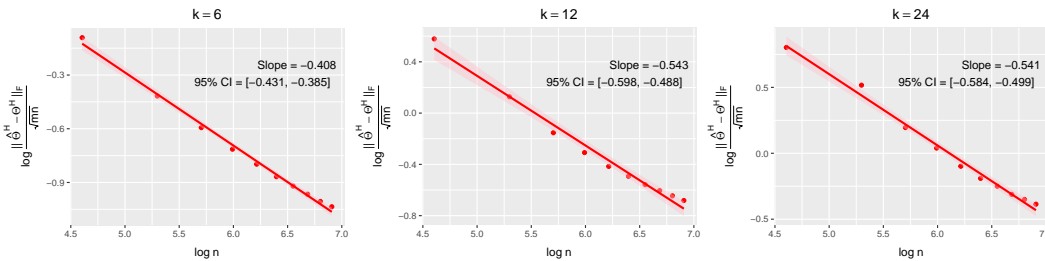

Figure 12: Log–log plot of estimation error versus $n$, based on 30 independent Monte Carlo repetitions.

We next explore the influence of the sparsity parameter $\bar{\alpha}_{m,n}$ under varying latent dimensions. We set $k \in \{6, 9, 12\}$, $m = 10n = 5000$, $\lambda = 0.2$, and $-\bar{\alpha}_{m,n} \in \{0.5, 1, \ldots, 3.5, 4\}$. To focus on sparse hypergraph settings, we narrow the possible interval of degree parameters by setting $\alpha_i$ i.i.d. $\sim \text{Uniform}[\bar{\alpha}_{m,n} - 0.5, \bar{\alpha}_{m,n} + 0.5]$ for any $i \in [n]$. Figure 13 shows that the estimation errors of $\Theta^H$ and $\Theta$ increase approximately proportionally to $\exp(-\bar{\alpha}_{m,n})$, which aligns with

the findings of (Wu et al., 2024), while the error of $\Theta^X$ remains stable. This is reasonable because changes in the hypergraph sparsity will not affect the attribute component $\Theta^X$. The slower growth rate for $\Theta^H$ compared to $\Theta$ is also expected, because of the stabilizing contribution from $\Theta^X$.

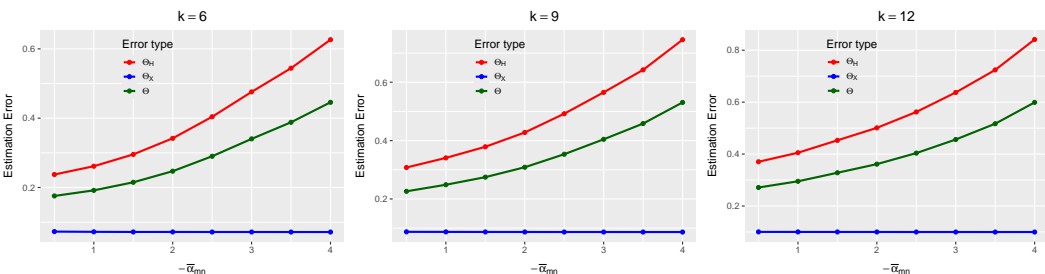

Figure 13: Estimation error under $\rho = 0$ versus sparsity parameter $\alpha_{m,n}$ based on 30 Monte Carlo repetitions.

To further confirm the exponential scaling, we plot log-error of $\Theta^H, \Theta$ versus $-\bar{\alpha}_{m,n}$ in Figure 14 . The fitted slopes confirm the exponential dependence predicted by Theorem 4.

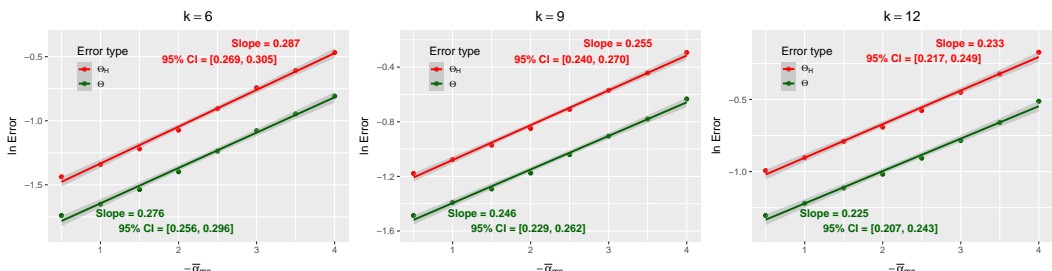

Figure 14: Log–log plot of estimation error versus sparsity parameter $\bar{\alpha}_{m,n}$ for $\rho = 0$, based on 30 independent Monte Carlo repetitions.

Finally, we investigate the influence of the weight parameter $\lambda$ on estimation performance. We fix $k \in \{6, 9, 12\}$, $n = 500$, $\bar{\alpha}_{m,n} = -2$, and vary $\lambda \in \{0.05, 0.1, \ldots, 1.25\}$. To better visualize potential differences, we consider two sample sizes, $m \in \{500, 2000\}$. The results are presented in Figures 15 and 16. Across both settings, the estimation errors remain largely unchanged as $\lambda$ varies, suggesting that the choice of $\lambda$ has minimal impact on accuracy.

From a practical standpoint, we include $\lambda$ primarily as a stability-enhancing tuning parameter. Specifically, a smaller $\lambda$ allows the algorithm to converge with a larger step size $\eta$, leading to faster execution without degrading estimation accuracy.

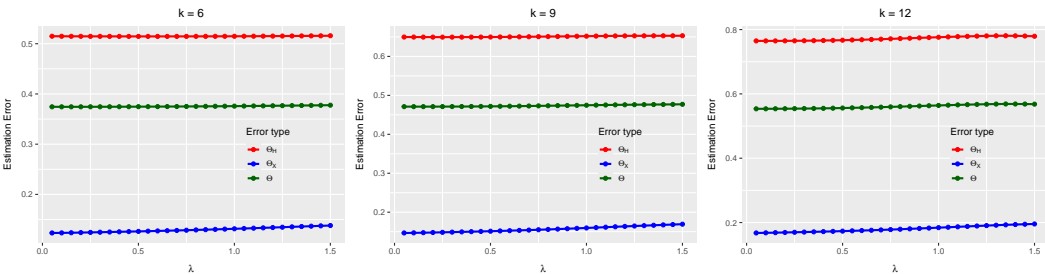

Figure 15: Estimation error versus varying $\lambda$ values for $m = 500$, averaged over 30 Monte Carlo repetitions.

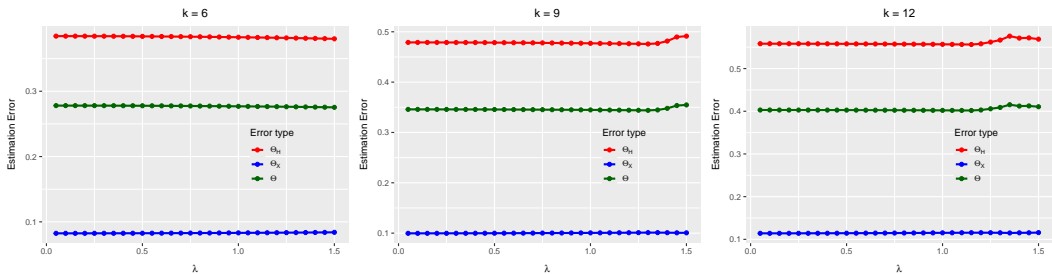

Figure 16: Estimation error versus varying $\lambda$ values for $m = 2000$, averaged over 30 Monte Carlo repetitions.

## B.3 ADDITIONAL SIMULATION RESULTS OF THE SYNTHETIC DATA ANALYSIS

### B.3.1 SIMULATION RESULTS FOR GENERATION

We conduct simulation studies on synthetic datasets to assess the performance of the whole pipeline of the proposed method ReLaSH. To demonstrate the ability of our algorithm to recover the underlying structure of a given hypergraph and its associated attributes, we compare it with other generative models, i.e. Gaussian Diffusion Models (Gau-Diff), Generative Adversarial Networks (GANs), RealNVP, and Variational Autoencoders (VAE). Regarding these methods, we treat each hyperedge as a binary vector and concatenate it with the attribute vector. To ensure fairness in the comparison, we calibrate these algorithms to align with the 0-1 valued hyperlinks. Details of these methods are provided in the Appendix B.7.

We assess the generative performance of different methods using the Root Mean Squared Error (RMSE) of the means and variance-covariances of node co-occurrence and attributes vector. Additionally, we defined a task-specific error metric called the Fréchet Embedding Distance to evaluate the performance of each method. Details of error metrics can be found in Appendix B.6, B.5. The settings of simulation experiments and results of error metrics are shown in details in Appendix B.3.

Regarding the settings of synthetic data analysis, specifically, for a $(k_2 + k_3)$-dimensional latent space, we randomly divide the $n$ nodes into $(k_2 + k_3)$ nearly equal-sized groups, where the group sizes may differ by at most 1. The node embeddings $\mathcal{Z}_n$ are then independently generated as follows. For node $i$ belonging to the $t$-th group, its embedding $z_i$ is drawn from the truncated Gaussian distribution $\mathcal{N}_{[-1,1]}(\mathbf{1}_{k_2+k_3} - e_t, \Sigma_z)$, where $e_t$ is the unit vector with the $t$-th element being 1 and the others 0, and $[-1, 1]$ indicates truncation on each coordinate.

The hyperlink embeddings $\mathcal{U}_m$ are generated from a Gaussian distribution $\mathcal{N}_{[-1,1]}(0, \Sigma_U)$, ensuring that the identifiability condition $\sum_{j=1}^{m} u_j = \mathbf{0}_k$ is approximately preserved. For the regression parameters $(B, \gamma)$, we generate $\gamma_i$ i.i.d. from $\mathrm{Uniform}([-1, 1])$, and each column $B_i$ independently sampled from $\mathcal{N}(0, \Sigma_B)$. The variance-covariance matrices $\Sigma_B, \Sigma_z, \Sigma_U$ are defined such that their diagonal entry is 0.2, with other entries being all 0. The degree heterogeneity parameters $\alpha_i$ are generated from the uniform distribution $[\bar{\alpha}_{m,n} - 1, \bar{\alpha}_{m,n} + 1]$ for the sparsity parameter $\bar{\alpha}_{m,n}$ specified in each experimental setting. Once all parameters and embeddings are generated, we can construct the hyperlinks according to the hyperlink generation model. The hyperlink-related attributes are generated as $X_{jt} \sim N\left(\left(1_m\gamma^\top + U^{(12)}B^\top\right)_{jt}, 1\right)$, where the generation process is independent for each $j \in [m]$ and $t \in [p]$. That is, Gaussian noise is introduced during the generation, which is commonly observed in real data structures.

We assess the generative performance of different methods using the Root Mean Squared Error (RMSE) of the means and variance-covariances of node co-occurrence and attributes vector. Additionally, we defined a task-specific error metric called the Fréchet Embedding Distance to evaluate the performance of each method. Details of error metrics can be found in Appendix B.6, B.5. Each time we generate $\tilde{m} = 32m$ new hyperlinks together with attributes by each generative model.

We take the settings of $k \in \{6, 12, 24, 48\}$ and $m, n \in \{200, 400, 800, 1600\}$, and limit the evaluation to ReLaSH (and the calibrated version ReLaSH$_c$), Gau-Diff, RealNVP, and WGAN due to

the poor performance of VAE and constraints on computational resources and time. The runtime required by the VAE is approximately four times longer than that of other competing methods. In the real data analysis section, we will demonstrate its relatively poor performance, particularly in terms of both the first and second-order moment errors for the hypergraph and attributes estimation. The RMSE results are provided in Table 6, 7, 8, 9 respectively, and the FED results are provided in Table 10, 11. The FED for each method is calculated using Algorithm 4, where the same maximum iteration steps are imposed for all methods, and identical early stopping conditions are applied.

Generally, we observe that as $m$ increases, the error metrics generally decrease due to the increasing data size, which allows for better utilization of available information, resulting in improved performance.

| Embedding Dimension $k = 6$ ($k_1 = k_2 = k_3 = 2$) | | | | | | | | | | |
|---|---|---|---|---|---|---|---|---|---|---|
| $m$ | Method | Objects | $n = p = 200$ | | $n = p = 400$ | | $n = p = 800$ | | $n = p = 1600$ | |
| | | | Mean | Cov | Mean | Cov | Mean | Cov | Mean | Cov |
| 200 | ReLaSH | Hypergraph | 0.0242 | 0.0069 | 0.0242 | **0.0053** | 0.0243 | **0.0042** | 0.0242 | **0.0041** |
| | | Attributes | 0.0740 | **0.0428** | **0.0703** | **0.0409** | 0.0772 | **0.0351** | **0.0732** | **0.0351** |
| | ReLaSH$_c$ | Hypergraph | 0.0243 | 0.0374 | 0.0240 | 0.0385 | 0.0241 | 0.0378 | 0.0239 | 0.0405 |
| | | Attributes | 0.0740 | **0.0428** | **0.0703** | **0.0409** | 0.0772 | **0.0351** | **0.0732** | **0.0351** |
| | Gau Diff | Hypergraph | 0.0243 | **0.0054** | 0.0240 | **0.0053** | 0.0241 | 0.0053 | 0.0239 | 0.0052 |
| | | Attributes | 0.0873 | 0.0683 | 0.1071 | 0.0567 | 0.1025 | 0.0564 | 0.0965 | 0.0548 |
| | RealNVP | Hypergraph | 0.0243 | **0.0054** | 0.0240 | **0.0053** | 0.0241 | 0.0052 | 0.0239 | 0.0052 |
| | | Attributes | **0.0731** | 0.0575 | 0.0707 | 0.0545 | **0.0769** | 0.0563 | 0.0734 | 0.0549 |
| | WGAN | Hypergraph | 0.0243 | 0.0143 | 0.0240 | 0.0108 | 0.0241 | 0.0165 | 0.0239 | 0.0221 |
| | | Attributes | 0.3485 | 0.1697 | 0.1425 | 0.1495 | 0.0892 | 0.0919 | 0.0813 | 0.0864 |
| 400 | ReLaSH | Hypergraph | 0.0178 | **0.0047** | 0.0201 | **0.0039** | 0.0171 | **0.0032** | 0.0180 | **0.0030** |
| | | Attributes | **0.0472** | **0.0292** | **0.0511** | **0.0265** | **0.0535** | **0.0241** | **0.0519** | **0.0246** |
| | ReLaSH$_c$ | Hypergraph | 0.0176 | 0.0414 | 0.0200 | 0.0436 | 0.0169 | 0.0432 | 0.0178 | 0.0434 |
| | | Attributes | **0.0472** | **0.0292** | **0.0511** | **0.0265** | **0.0535** | **0.0241** | **0.0519** | **0.0246** |
| | Gau Diff | Hypergraph | 0.0176 | 0.0050 | 0.0200 | 0.0051 | 0.0169 | 0.0051 | 0.0178 | 0.0051 |
| | | Attributes | 0.0712 | 0.0676 | 0.0911 | 0.0559 | 0.0929 | 0.0547 | 0.0878 | 0.0541 |
| | RealNVP | Hypergraph | 0.0176 | 0.0051 | 0.0200 | 0.0051 | 0.0169 | 0.0051 | 0.0178 | 0.0051 |
| | | Attributes | 0.0476 | 0.0547 | 0.0512 | 0.0526 | 0.0539 | 0.0542 | 0.0520 | 0.0540 |
| | WGAN | Hypergraph | 0.0176 | 0.0131 | 0.0200 | 0.0133 | 0.0169 | 0.0138 | 0.0178 | 0.0175 |
| | | Attributes | 0.2332 | 0.1714 | 0.0790 | 0.1279 | 0.0634 | 0.0639 | 0.0587 | 0.0588 |
| 800 | ReLaSH | Hypergraph | 0.0138 | **0.0037** | 0.0130 | **0.0028** | 0.0125 | **0.0023** | 0.0127 | **0.0021** |
| | | Attributes | 0.0359 | **0.0238** | **0.0411** | **0.0207** | **0.0363** | **0.0178** | **0.0381** | **0.0184** |
| | ReLaSH$_c$ | Hypergraph | 0.0137 | 0.0427 | 0.0129 | 0.0441 | 0.0122 | 0.0461 | 0.0125 | 0.0470 |
| | | Attributes | 0.0359 | **0.0238** | **0.0411** | **0.0207** | **0.0363** | **0.0178** | **0.0381** | **0.0184** |
| | Gau Diff | Hypergraph | 0.0137 | 0.0047 | 0.0129 | 0.0050 | 0.0122 | 0.0051 | 0.0125 | 0.0051 |
| | | Attributes | 0.0544 | 0.0788 | 0.0944 | 0.0579 | 0.0960 | 0.0529 | 0.0896 | 0.0556 |
| | RealNVP | Hypergraph | 0.0137 | 0.0050 | 0.0129 | 0.0050 | 0.0122 | 0.0051 | 0.0125 | 0.0051 |
| | | Attributes | **0.0352** | 0.0648 | 0.0414 | 0.0545 | 0.0367 | 0.0521 | 0.0384 | 0.0555 |
| | WGAN | Hypergraph | 0.0137 | 0.0085 | 0.0129 | 0.0088 | 0.0122 | 0.0111 | 0.0125 | 0.0195 |
| | | Attributes | 0.0789 | 0.1812 | 0.0614 | 0.0958 | 0.0537 | 0.0506 | 0.0491 | 0.0406 |
| 1600 | ReLaSH | Hypergraph | 0.0089 | **0.0033** | 0.0091 | **0.0022** | 0.0091 | **0.0017** | 0.0088 | **0.0016** |
| | | Attributes | 0.0278 | **0.0185** | **0.0286** | **0.0160** | **0.0262** | **0.0140** | **0.0274** | **0.0138** |
| | ReLaSH$_c$ | Hypergraph | 0.0084 | 0.0439 | 0.0088 | 0.0449 | 0.0090 | 0.0458 | 0.0087 | 0.0469 |
| | | Attributes | 0.0278 | **0.0185** | **0.0286** | **0.0160** | **0.0262** | **0.0140** | **0.0274** | **0.0138** |
| | Gau Diff | Hypergraph | 0.0084 | 0.0046 | 0.0088 | 0.0049 | 0.0090 | 0.0050 | 0.0087 | 0.0050 |
| | | Attributes | 0.0516 | 0.0741 | 0.0879 | 0.0581 | 0.0820 | 0.0552 | 0.0831 | 0.0547 |
| | RealNVP | Hypergraph | 0.0084 | 0.0048 | 0.0088 | 0.0049 | 0.0090 | 0.0050 | 0.0087 | 0.0050 |
| | | Attributes | **0.0277** | 0.0578 | 0.0287 | 0.0537 | 0.0263 | 0.0543 | 0.0275 | 0.0545 |
| | WGAN | Hypergraph | 0.0084 | 0.0084 | 0.0088 | 0.0095 | 0.0090 | 0.0136 | 0.0087 | 0.0205 |
| | | Attributes | 0.0587 | 0.1883 | 0.0437 | 0.0590 | 0.0458 | 0.0452 | 0.0426 | 0.0358 |

Table 6: RMSE results for means (columns with "Mean") & covariances (columns with "Cov") of ReLaSH, ReLaSH$_c$, Gau-Diff, RealNVP and WGAN when latent dimension is $k = 6$. Each value comes from the mean of 20 repetitions.

We use bold to highlight the best result in each experimental setting and underline the second-best result. For the RMSE of the hyperlink vector mean, note that all calibrated methods yield the same result, so we do not emphasize these results. It is important to note that the RMSE for the calibrated

| | | | Embedding Dimension $k = 12$ ($k_1 = k_2 = k_3 = 4$) | | | | | | | |
|---|---|---|---|---|---|---|---|---|---|---|
| $m$ | Method | Objects | $n = p = 200$ | | $n = p = 400$ | | $n = p = 800$ | | $n = p = 1600$ | |
| | | | Mean | Cov | Mean | Cov | Mean | Cov | Mean | Cov |
| 200 | ReLaSH | Hypergraph | 0.0351 | 0.0137 | 0.0252 | **0.0081** | 0.0298 | **0.0097** | 0.0302 | **0.0099** |
| | | Attributes | **0.0782** | **0.0583** | **0.0748** | **0.0510** | 0.0782 | **0.0502** | 0.0736 | **0.0461** |
| | ReLaSH$_c$ | Hypergraph | 0.0310 | 0.0367 | 0.0251 | 0.0306 | 0.0263 | 0.0336 | 0.0285 | 0.0280 |
| | | Attributes | **0.0782** | **0.0583** | **0.0748** | **0.0510** | 0.0782 | **0.0502** | 0.0736 | **0.0461** |
| | Gau Diff | Hypergraph | 0.0310 | **0.0130** | 0.0251 | 0.0130 | 0.0263 | 0.0132 | 0.0285 | 0.0130 |
| | | Attributes | 0.0907 | 0.0852 | 0.1151 | 0.0800 | 0.1074 | 0.0753 | 0.0970 | 0.0797 |
| | RealNVP | Hypergraph | 0.0310 | 0.0133 | 0.0251 | 0.0130 | 0.0263 | 0.0132 | 0.0285 | 0.0130 |
| | | Attributes | 0.0797 | 0.0765 | 0.0772 | 0.0786 | **0.0780** | 0.0754 | **0.0734** | 0.0799 |
| | WGAN | Hypergraph | 0.0310 | 0.0212 | 0.0251 | 0.0126 | 0.0263 | 0.0158 | 0.0285 | 0.0188 |
| | | Attributes | 0.3304 | 0.2025 | 0.2052 | 0.1078 | 0.1106 | 0.0942 | 0.0816 | 0.0719 |
| 400 | ReLaSH | Hypergraph | 0.0190 | **0.0081** | 0.0175 | **0.0057** | 0.0176 | **0.0047** | 0.0179 | **0.0044** |
| | | Attributes | **0.0534** | **0.0427** | **0.0514** | **0.0339** | **0.0553** | **0.0339** | **0.0554** | **0.0335** |
| | ReLaSH$_c$ | Hypergraph | 0.0184 | 0.0357 | 0.0182 | 0.0356 | 0.0179 | 0.0397 | 0.0178 | 0.0389 |
| | | Attributes | **0.0534** | **0.0427** | **0.0514** | **0.0339** | **0.0553** | **0.0339** | **0.0554** | **0.0335** |
| | Gau Diff | Hypergraph | 0.0184 | 0.0128 | 0.0182 | 0.0129 | 0.0179 | 0.0129 | 0.0178 | 0.0129 |
| | | Attributes | 0.0792 | 0.0863 | 0.0960 | 0.0741 | 0.1055 | 0.0783 | 0.0905 | 0.0803 |
| | RealNVP | Hypergraph | 0.0184 | 0.0131 | 0.0182 | 0.0130 | 0.0179 | 0.0129 | 0.0178 | 0.0129 |
| | | Attributes | 0.0555 | 0.0780 | 0.0525 | 0.0720 | 0.0557 | 0.0782 | **0.0554** | 0.0804 |
| | WGAN | Hypergraph | 0.0184 | 0.0152 | 0.0182 | 0.0131 | 0.0179 | 0.0184 | 0.0178 | 0.0296 |
| | | Attributes | 0.1838 | 0.1588 | 0.0822 | 0.1146 | 0.0640 | 0.0834 | 0.0610 | 0.0628 |
| 800 | ReLaSH | Hypergraph | 0.0121 | **0.0053** | 0.0123 | **0.0039** | 0.0126 | **0.0035** | 0.0129 | **0.0030** |
| | | Attributes | 0.0402 | **0.0301** | **0.0380** | **0.0262** | **0.0366** | **0.0244** | 0.0402 | **0.0236** |
| | ReLaSH$_c$ | Hypergraph | 0.0126 | 0.0364 | 0.0122 | 0.0408 | 0.0126 | 0.0449 | 0.0128 | 0.0471 |
| | | Attributes | 0.0402 | **0.0301** | 0.0380 | **0.0262** | 0.0366 | **0.0244** | 0.0402 | **0.0236** |
| | Gau Diff | Hypergraph | 0.0126 | 0.0125 | 0.0122 | 0.0127 | 0.0126 | 0.0129 | 0.0128 | 0.0131 |
| | | Attributes | 0.0618 | 0.0862 | 0.0950 | 0.0785 | 0.0994 | 0.0758 | 0.0865 | 0.0767 |
| | RealNVP | Hypergraph | 0.0126 | 0.0129 | 0.0122 | 0.0128 | 0.0126 | 0.0129 | 0.0128 | 0.0131 |
| | | Attributes | **0.0394** | 0.0731 | 0.0385 | 0.0760 | 0.0368 | 0.0755 | **0.0399** | 0.0766 |
| | WGAN | Hypergraph | 0.0126 | 0.0103 | 0.0122 | 0.0118 | 0.0126 | 0.0171 | 0.0128 | 0.0385 |
| | | Attributes | 0.0779 | 0.1672 | 0.0505 | 0.0615 | 0.0502 | 0.0519 | 0.0486 | 0.0469 |
| 1600 | ReLaSH | Hypergraph | 0.0098 | **0.0046** | 0.0099 | **0.0029** | 0.0102 | **0.0026** | 0.0105 | **0.0020** |
| | | Attributes | **0.0284** | **0.0221** | **0.0259** | **0.0192** | **0.0247** | **0.0172** | 0.0236 | **0.0168** |
| | ReLaSH$_c$ | Hypergraph | 0.0095 | 0.0405 | 0.0096 | 0.0419 | 0.0099 | 0.0401 | 0.0101 | 0.0399 |
| | | Attributes | **0.0284** | **0.0221** | **0.0259** | **0.0192** | **0.0247** | **0.0172** | 0.0236 | **0.0168** |
| | Gau Diff | Hypergraph | 0.0095 | 0.0123 | 0.0096 | 0.0128 | 0.0099 | 0.0130 | 0.0101 | 0.0872 |
| | | Attributes | 0.0547 | 0.0877 | 0.0850 | 0.0779 | 0.0836 | 0.0789 | **0.0128** | 0.0794 |
| | RealNVP | Hypergraph | 0.0095 | 0.0128 | 0.0096 | 0.0129 | 0.0099 | 0.0130 | 0.0101 | 0.0128 |
| | | Attributes | 0.0286 | 0.0735 | 0.0262 | 0.0753 | 0.0277 | 0.0785 | 0.0298 | 0.0793 |
| | WGAN | Hypergraph | 0.0095 | 0.0082 | 0.0096 | 0.0089 | 0.0099 | 0.0095 | 0.0101 | 0.0098 |
| | | Attributes | 0.0570 | 0.0885 | 0.0450 | 0.0620 | 0.0467 | 0.0597 | 0.0402 | 0.0610 |

Table 7: RMSE results for means (columns with "Mean") & covariances (columns with "Cov") of ReLaSH, ReLaSH$_c$, Gau-Diff, RealNVP and WGAN when latent dimension is $k = 12$. Each value comes from the mean of 20 repetitions.

| | | | Embedding Dimension $k = 24$ ($k_1 = k_2 = k_3 = 8$) | | | | | | | |
| $m$ | Method | Objects | $n = p = 200$ | | $n = p = 400$ | | $n = p = 800$ | | $n = p = 1600$ | |
| | | | Mean | Cov | Mean | Cov | Mean | Cov | Mean | Cov |
| 200 | ReLaSH | Hypergraph | 0.0329 | **0.0140** | 0.0389 | **0.0133** | 0.0345 | **0.0149** | 0.0371 | **0.0128** |
| | | Attributes | **0.0808** | **0.0709** | 0.0817 | **0.0703** | 0.0790 | **0.0653** | **0.0834** | **0.0685** |
| | ReLaSH$_c$ | Hypergraph | 0.0245 | 0.0179 | 0.0264 | 0.0191 | 0.0319 | 0.0187 | 0.0246 | 0.0240 |
| | | Attributes | **0.0808** | **0.0709** | 0.0817 | **0.0703** | 0.0790 | **0.0653** | **0.0834** | **0.0685** |
| | Gau Diff | Hypergraph | 0.0245 | 0.0284 | 0.0264 | 0.0289 | 0.0319 | 0.0296 | 0.0246 | 0.0296 |
| | | Attributes | 0.0971 | 0.1089 | 0.1198 | 0.1104 | 0.1071 | 0.1131 | 0.0978 | 0.1117 |
| | RealNVP | Hypergraph | 0.0245 | 0.0288 | 0.0264 | 0.0289 | 0.0319 | 0.0296 | 0.0246 | 0.0296 |
| | | Attributes | 0.0818 | 0.1012 | **0.0815** | 0.1102 | **0.0789** | 0.1134 | 0.0839 | 0.1119 |
| | WGAN | Hypergraph | 0.0245 | 0.0257 | 0.0264 | 0.0212 | 0.0319 | 0.0257 | 0.0246 | 0.0309 |
| | | Attributes | 0.2503 | 0.2018 | 0.1372 | 0.1167 | 0.2365 | 0.1291 | 0.3456 | 0.1334 |
| 400 | ReLaSH | Hypergraph | 0.0185 | **0.0093** | 0.0180 | **0.0077** | 0.0177 | **0.0071** | 0.0208 | **0.0063** |
| | | Attributes | **0.0563** | **0.0541** | **0.0554** | **0.0492** | **0.0584** | **0.0470** | **0.0580** | **0.0481** |
| | ReLaSH$_c$ | Hypergraph | 0.0186 | 0.0204 | 0.0195 | 0.0271 | 0.0182 | 0.0315 | 0.0191 | 0.0402 |
| | | Attributes | **0.0563** | **0.0541** | **0.0554** | **0.0492** | **0.0584** | **0.0470** | **0.0580** | **0.0481** |
| | Gau Diff | Hypergraph | 0.0186 | 0.0285 | 0.0195 | 0.0287 | 0.0182 | 0.0295 | 0.0191 | 0.0288 |
| | | Attributes | 0.0799 | 0.1146 | 0.0974 | 0.1080 | 0.0990 | 0.1083 | 0.0789 | 0.1092 |
| | RealNVP | Hypergraph | 0.0186 | 0.0289 | 0.0195 | 0.0288 | 0.0182 | 0.0296 | 0.0191 | 0.0288 |
| | | Attributes | 0.0576 | 0.1073 | 0.0560 | 0.1072 | 0.0585 | 0.1084 | 0.0581 | 0.1094 |
| | WGAN | Hypergraph | 0.0186 | 0.0193 | 0.0195 | 0.0183 | 0.0182 | 0.0263 | 0.0191 | 0.0334 |
| | | Attributes | 0.1951 | 0.1184 | 0.0774 | 0.0948 | 0.0717 | 0.0840 | 0.2848 | 0.1141 |
| 800 | ReLaSH | Hypergraph | 0.0147 | **0.0073** | 0.0130 | **0.0058** | 0.0142 | **0.0050** | 0.0127 | **0.0044** |
| | | Attributes | 0.0418 | **0.0402** | 0.0404 | **0.0364** | **0.0400** | **0.0339** | **0.0431** | **0.0334** |
| | ReLaSH$_c$ | Hypergraph | 0.0145 | 0.0128 | 0.0128 | 0.0344 | 0.0140 | 0.0430 | 0.0128 | 0.0448 |
| | | Attributes | 0.0418 | **0.0402** | 0.0404 | **0.0364** | **0.0400** | **0.0339** | **0.0431** | **0.0334** |
| | Gau Diff | Hypergraph | 0.0145 | 0.0281 | 0.0128 | 0.0282 | 0.0140 | 0.0289 | 0.0128 | 0.0300 |
| | | Attributes | 0.0725 | 0.1139 | 0.1022 | 0.1123 | 0.1035 | 0.1110 | 0.0757 | 0.1069 |
| | RealNVP | Hypergraph | 0.0145 | 0.0287 | 0.0128 | 0.0283 | 0.0140 | 0.0289 | 0.0128 | 0.0300 |
| | | Attributes | **0.0407** | 0.1044 | **0.0402** | 0.1116 | 0.0405 | 0.1111 | **0.0431** | 0.1070 |
| | WGAN | Hypergraph | 0.0145 | 0.0128 | 0.0128 | 0.0269 | 0.0140 | 0.0523 | 0.0128 | 0.0459 |
| | | Attributes | 0.0645 | 0.0865 | 0.0572 | 0.0744 | 0.0523 | 0.0617 | 0.0645 | 0.0757 |
| 1600 | ReLaSH | Hypergraph | 0.0098 | **0.0058** | 0.0115 | **0.0056** | 0.0097 | **0.0057** | 0.0095 | **0.0053** |
| | | Attributes | 0.0286 | **0.0314** | 0.0281 | **0.0309** | **0.0275** | **0.0332** | **0.0272** | **0.0298** |
| | ReLaSH$_c$ | Hypergraph | 0.0095 | 0.0328 | 0.0112 | 0.0342 | 0.0091 | 0.0309 | 0.0088 | 0.0301 |
| | | Attributes | 0.0286 | **0.0314** | 0.0281 | **0.0309** | **0.0275** | **0.0332** | **0.0272** | **0.0298** |
| | Gau Diff | Hypergraph | 0.0095 | 0.0280 | 0.0112 | 0.0272 | 0.0091 | 0.0279 | 0.0088 | 0.0283 |
| | | Attributes | 0.0637 | 0.1276 | 0.0618 | 0.1173 | 0.0628 | 0.1121 | 0.0597 | 0.1083 |
| | RealNVP | Hypergraph | 0.0095 | 0.0287 | 0.0112 | 0.0282 | 0.0091 | 0.0263 | 0.0088 | 0.0251 |
| | | Attributes | **0.0281** | 0.1172 | **0.0280** | 0.1132 | 0.0285 | 0.1104 | 0.0276 | 0.1036 |
| | WGAN | Hypergraph | 0.0095 | 0.0105 | 0.0112 | 0.0224 | 0.0091 | 0.0300 | 0.0088 | 0.0666 |
| | | Attributes | 0.0496 | 0.1053 | 0.0458 | 0.0648 | 0.0441 | 0.0579 | 0.0457 | 0.0466 |

Table 8: RMSE results for means (columns with "Mean") & covariances (columns with "Cov") of ReLaSH, ReLaSH$_c$, Gau-Diff, RealNVP and WGAN when latent dimension is $k = 24$. Each value comes from the mean of 20 repetitions.

| | | | Embedding Dimension $k = 48$ $(k_1 = k_2 = k_3 = 16)$ | | | | | | | |
|---|---|---|---|---|---|---|---|---|---|---|
| $m$ | Method | Objects | $n = p = 200$ | | $n = p = 400$ | | $n = p = 800$ | | $n = p = 1600$ | |
| | | | Mean | Cov | Mean | Cov | Mean | Cov | Mean | Cov |
| 200 | ReLaSH | Hypergraph | 0.0810 | **0.0260** | 0.0815 | **0.0260** | 0.0990 | **0.0299** | 0.0840 | 0.0368 |
| | | Attributes | 0.0863 | **0.1137** | **0.0807** | **0.1043** | **0.0938** | **0.1034** | **0.0872** | **0.1000** |
| | ReLaSH$_c$ | Hypergraph | 0.0281 | 0.0289 | 0.0263 | 0.0282 | 0.0268 | 0.0314 | 0.0291 | 0.0382 |
| | | Attributes | 0.0863 | **0.1137** | **0.0807** | **0.1043** | **0.0938** | **0.1034** | **0.0872** | **0.1000** |
| | Gau Diff | Hypergraph | 0.0281 | 0.0533 | 0.0263 | 0.0526 | 0.0268 | 0.0559 | 0.0291 | 0.0551 |
| | | Attributes | 0.0992 | 0.1575 | 0.1199 | 0.1581 | 0.1241 | 0.1543 | 0.1047 | 0.1550 |
| | RealNVP | Hypergraph | 0.0281 | 0.0538 | 0.0263 | 0.0526 | 0.0268 | 0.0559 | 0.0291 | 0.0551 |
| | | Attributes | **0.0850** | 0.1523 | 0.0808 | 0.1590 | 0.0938 | 0.1551 | 0.0874 | 0.1553 |
| | WGAN | Hypergraph | 0.0281 | 0.0290 | 0.0263 | 0.0427 | 0.0268 | 0.0393 | 0.0291 | **0.0288** |
| | | Attributes | 0.1421 | 0.2809 | 0.1414 | 0.5243 | 0.1201 | 0.3009 | 0.0936 | 0.1969 |
| 400 | ReLaSH | Hypergraph | 0.0418 | **0.0180** | 0.0305 | 0.0139 | 0.0198 | **0.0136** | 0.0214 | **0.0104** |
| | | Attributes | **0.0590** | **0.0819** | 0.0640 | **0.0717** | **0.0682** | 0.0697 | 0.0593 | **0.0678** |
| | ReLaSH$_c$ | Hypergraph | 0.0179 | 0.0176 | 0.0195 | **0.0136** | 0.0194 | 0.0159 | 0.0174 | 0.0220 |
| | | Attributes | **0.0590** | **0.0819** | 0.0640 | **0.0717** | **0.0682** | **0.0697** | 0.0593 | **0.0678** |
| | Gau Diff | Hypergraph | 0.0179 | 0.0529 | 0.0195 | 0.0530 | 0.0194 | 0.0556 | 0.0174 | 0.0549 |
| | | Attributes | 0.0944 | 0.1667 | 0.1114 | 0.1557 | 0.1099 | 0.1501 | 0.0834 | 0.1539 |
| | RealNVP | Hypergraph | 0.0179 | 0.0535 | 0.0195 | 0.0531 | 0.0194 | 0.0556 | 0.0174 | 0.0549 |
| | | Attributes | 0.0601 | 0.1623 | **0.0628** | 0.1567 | 0.0686 | 0.1510 | **0.0590** | 0.1542 |
| | WGAN | Hypergraph | 0.0179 | 0.0247 | 0.0195 | 0.0370 | 0.0194 | 0.0255 | 0.0174 | 0.0226 |
| | | Attributes | 0.0939 | 0.2347 | 0.1011 | 0.2433 | 0.0767 | 0.1589 | 0.0691 | 0.1597 |
| 800 | ReLaSH | Hypergraph | 0.0209 | **0.0108** | 0.0139 | **0.0079** | 0.0136 | **0.0085** | 0.0130 | **0.0062** |
| | | Attributes | 0.0460 | **0.0579** | **0.0452** | **0.0511** | **0.0481** | **0.0505** | 0.0472 | **0.0500** |
| | ReLaSH$_c$ | Hypergraph | 0.0137 | 0.0127 | 0.0122 | 0.0190 | 0.0138 | 0.0252 | 0.0129 | 0.0273 |
| | | Attributes | 0.0460 | **0.0579** | **0.0452** | **0.0511** | **0.0481** | **0.0505** | 0.0472 | **0.0500** |
| | Gau Diff | Hypergraph | 0.0137 | 0.0530 | 0.0122 | 0.0523 | 0.0138 | 0.0554 | 0.0129 | 0.0553 |
| | | Attributes | 0.0816 | 0.1655 | 0.1116 | 0.1514 | 0.1022 | 0.1536 | 0.0727 | 0.1548 |
| | RealNVP | Hypergraph | 0.0137 | 0.0539 | 0.0122 | 0.0524 | 0.0138 | 0.0554 | 0.0129 | 0.0554 |
| | | Attributes | **0.0459** | 0.1583 | 0.0457 | 0.1526 | 0.0485 | 0.1544 | **0.0428** | 0.1550 |
| | WGAN | Hypergraph | 0.0137 | 0.0208 | 0.0122 | 0.0251 | 0.0138 | 0.0107 | 0.0129 | 0.0183 |
| | | Attributes | 0.0811 | 0.1395 | 0.0663 | 0.1663 | 0.0671 | 0.0933 | 0.0551 | 0.0754 |
| 1600 | ReLaSH | Hypergraph | 0.0173 | **0.0097** | 0.0102 | **0.0086** | 0.0097 | **0.0080** | 0.0090 | **0.0054** |
| | | Attributes | 0.0328 | **0.0525** | **0.0322** | **0.0491** | **0.0326** | **0.0495** | 0.0318 | **0.0472** |
| | ReLaSH$_c$ | Hypergraph | 0.0105 | 0.0127 | 0.0085 | 0.0169 | 0.0095 | 0.0184 | 0.0087 | 0.0204 |
| | | Attributes | 0.0328 | **0.0525** | **0.0322** | **0.0491** | **0.0326** | **0.0495** | 0.0318 | **0.0472** |
| | Gau Diff | Hypergraph | 0.0105 | 0.0526 | 0.0085 | 0.0521 | 0.0095 | 0.0555 | 0.0087 | 0.0559 |
| | | Attributes | 0.0653 | 0.1609 | 0.1061 | 0.1555 | 0.0978 | 0.1520 | 0.0811 | 0.1525 |
| | RealNVP | Hypergraph | 0.0105 | 0.0535 | 0.0085 | 0.0522 | 0.0095 | 0.0555 | 0.0087 | 0.0559 |
| | | Attributes | **0.0319** | 0.1510 | 0.0385 | 0.1570 | 0.0333 | 0.1528 | **0.0301** | 0.1527 |
| | WGAN | Hypergraph | 0.0105 | 0.0121 | 0.0085 | 0.0185 | 0.0095 | 0.0098 | 0.0087 | 0.0367 |
| | | Attributes | 0.0856 | 0.0912 | 0.0824 | 0.1415 | 0.0582 | 0.0710 | 0.0479 | 0.0547 |

Table 9: RMSE results for means (columns with "Mean") & covariances (columns with "Cov") of ReLaSH, ReLaSH$_c$, Gau-Diff, RealNVP and WGAN when latent dimension is $k = 48$. Each value comes from the mean of 20 repetitions.

| $m$ | Method | $n = p = 200$ | $n = p = 400$ | $n = p = 800$ | $n = p = 1600$ |
|---|---|---|---|---|---|
| | | Embedding Dimension $k = 6$ ($k_1 = k_2 = k_3 = 2$) | | | |
| | ReLaSH | **0.3540** | 0.2598 | **0.1561** | **0.1452** |
| | ReLaSH$_c$ | 0.3624 | **0.1843** | 0.1884 | 0.2111 |
| 200 | Gau Diff | 0.3608 | 0.3245 | 0.4331 | 0.5995 |
| | RealNVP | 0.3817 | 0.3280 | 0.3791 | 0.5638 |
| | WGAN | 0.4278 | 0.2789 | 0.1673 | 0.1933 |
| | ReLaSH | **0.1642** | 0.1756 | **0.1268** | **0.1341** |
| | ReLaSH$_c$ | 0.2977 | 0.2965 | 0.3499 | 0.2740 |
| 400 | Gau Diff | 0.1764 | 0.1496 | 0.2627 | 0.3947 |
| | RealNVP | 0.1775 | **0.1467** | 0.2819 | 0.4273 |
| | WGAN | 0.2202 | 0.2426 | 0.2867 | 0.2766 |
| | ReLaSH | 0.2199 | 0.1862 | **0.0792** | **0.0720** |
| | ReLaSH$_c$ | 0.3112 | 0.3379 | 0.3688 | 0.3982 |
| 800 | Gau Diff | **0.1725** | **0.0892** | 0.1561 | 0.2866 |
| | RealNVP | 0.1778 | 0.1143 | 0.1682 | 0.2949 |
| | WGAN | 0.2215 | 0.1901 | 0.2582 | 0.2293 |
| | ReLaSH | 0.1670 | **0.0766** | 0.1381 | **0.1225** |
| | ReLaSH$_c$ | 0.1666 | 0.1721 | 0.2752 | 0.5741 |
| 1600 | Gau Diff | 0.1634 | 0.0793 | 0.1344 | 0.2172 |
| | RealNVP | **0.1334** | 0.1842 | **0.0995** | 0.1994 |
| | WGAN | 0.1618 | 0.2282 | 0.1962 | 0.1737 |

| $m$ | Method | $n = p = 200$ | $n = p = 400$ | $n = p = 800$ | $n = p = 1600$ |
|---|---|---|---|---|---|
| | | Embedding Dimension $k = 12$ ($k_1 = k_2 = k_3 = 4$) | | | |
| | ReLaSH | **0.9794** | 0.6914 | 0.6056 | 0.6034 |
| | ReLaSH$_c$ | 1.0132 | **0.6824** | **0.5676** | **0.5067** |
| 200 | Gau Diff | 1.1343 | 0.9088 | 1.3026 | 1.4973 |
| | RealNVP | 1.3139 | 1.0591 | 1.2460 | 1.5199 |
| | WGAN | 1.6482 | 1.4013 | 0.7556 | 1.3379 |
| | ReLaSH | 0.7072 | 0.4392 | **0.2884** | **0.2246** |
| | ReLaSH$_c$ | 0.6191 | **0.4158** | 0.3129 | 0.2952 |
| 400 | Gau Diff | **0.6172** | 0.4947 | 0.6941 | 0.9002 |
| | RealNVP | 0.7296 | 0.5701 | 0.8043 | 0.9341 |
| | WGAN | 0.8856 | 0.6497 | 0.3500 | 0.2412 |
| | ReLaSH | 0.4775 | 0.3718 | **0.2037** | **0.2398** |
| | ReLaSH$_c$ | 0.4546 | **0.3552** | 0.3431 | 0.3174 |
| 800 | Gau Diff | **0.4071** | 0.3575 | 0.5120 | 0.7380 |
| | RealNVP | 0.4533 | 0.4206 | 0.4720 | 0.7130 |
| | WGAN | 0.4418 | 0.5541 | 0.3687 | 0.3399 |
| | ReLaSH | 0.3595 | 0.3217 | 0.2843 | **0.2204** |
| | ReLaSH$_c$ | 0.3745 | **0.2854** | **0.2679** | 0.2589 |
| 1600 | Gau Diff | **0.3529** | 0.3321 | 0.3795 | 0.5534 |
| | RealNVP | 0.4275 | 0.3232 | 0.3423 | 0.5177 |
| | WGAN | 0.3845 | 0.3974 | 0.4582 | 0.4952 |

Table 10: FED results for ReLaSH, ReLaSH$_c$, Gau-Diff, RealNVP and WGAN when latent dimension is $k = 6, 12$. Each value comes from the mean of 20 repetitions.

| $m$ | Method | $n=p=200$ | $n=p=400$ | $n=p=800$ | $n=p=1600$ |
|---|---|---|---|---|---|
| | | Embedding Dimension $k=24$ ($k_1=k_2=k_3=8$) | | | |
| | ReLaSH | 3.7876 | **2.3795** | **1.7776** | 1.4888 |
| | ReLaSH$_c$ | 4.0077 | 2.5164 | 1.8760 | **1.3280** |
| 200 | Gau Diff | 3.9593 | 3.4421 | 3.2590 | 3.2990 |
| | RealNVP | 4.1992 | 3.7454 | 3.2969 | 3.3342 |
| | WGAN | **3.0621** | 1.9154 | 1.5121 | 1.1528 |
| | ReLaSH | 2.3870 | **1.3333** | 0.8480 | **0.5496** |
| | ReLaSH$_c$ | 2.3198 | 1.3818 | **0.8407** | 0.6347 |
| 400 | Gau Diff | **2.0714** | 1.7287 | 1.8749 | 1.7820 |
| | RealNVP | 2.3455 | 1.7374 | 1.8147 | 1.8179 |
| | WGAN | 2.5758 | 1.7842 | 1.5667 | 0.8110 |
| | ReLaSH | 1.6894 | 1.0641 | **0.6340** | **0.4274** |
| | ReLaSH$_c$ | 1.5742 | 0.9923 | 0.7313 | 0.6584 |
| 800 | Gau Diff | **1.5422** | 1.1280 | 1.2945 | 1.5734 |
| | RealNVP | 1.7499 | 1.2599 | 1.3955 | 1.5954 |
| | WGAN | 1.8389 | **0.9427** | 0.9342 | 0.5240 |
| | ReLaSH | 1.2875 | 0.9731 | **0.5482** | **0.3740** |
| | ReLaSH$_c$ | 1.1986 | 1.0372 | 0.8542 | 0.6594 |
| 1600 | Gau Diff | 1.4199 | 1.0037 | 1.4529 | 1.5630 |
| | RealNVP | 1.4597 | 1.6384 | 1.3294 | 1.2934 |
| | WGAN | **0.9762** | **0.9205** | 0.7708 | 0.7188 |

| $m$ | Method | $n=p=200$ | $n=p=400$ | $n=p=800$ | $n=p=1600$ |
|---|---|---|---|---|---|
| | | Embedding Dimension $k=48$ ($k_1=k_2=k_3=16$) | | | |
| | ReLaSH | **15.3227** | **11.4013** | 9.8524 | **9.1217** |
| | ReLaSH$_c$ | 16.9249 | 11.9277 | **9.1791** | 9.2782 |
| 200 | Gau Diff | 17.5939 | 14.7371 | 13.4529 | 14.4237 |
| | RealNVP | 18.0527 | 15.0858 | 13.5804 | 14.4640 |
| | WGAN | 15.9366 | 12.1945 | 10.5906 | 9.4059 |
| | ReLaSH | 9.2837 | **4.2397** | **2.3041** | 1.7503 |
| | ReLaSH$_c$ | 9.7989 | 4.4413 | 2.4005 | **1.6694** |
| 400 | Gau Diff | **9.1772** | 5.3424 | 4.3290 | 4.3520 |
| | RealNVP | 9.8230 | 5.6104 | 4.4242 | 4.3550 |
| | WGAN | 8.4115 | 3.8713 | 2.2187 | 1.6203 |
| | ReLaSH | 6.1817 | 2.9811 | 1.7799 | 1.0864 |
| | ReLaSH$_c$ | 6.0745 | **2.9594** | **1.7735** | 0.9799 |
| 800 | Gau Diff | **5.9200** | 3.3151 | 3.1729 | 3.0181 |
| | RealNVP | 6.2765 | 3.6356 | 3.2323 | 3.0597 |
| | WGAN | 5.1394 | 2.3107 | 1.4140 | **0.9230** |
| | ReLaSH | **3.5762** | 1.7204 | **0.9726** | **0.5382** |
| | ReLaSH$_c$ | 4.2942 | **1.6382** | 1.0038 | 0.6389 |
| 1600 | Gau Diff | 4.5040 | 2.2186 | 2.3887 | 2.5871 |
| | RealNVP | 4.6260 | 2.6337 | 2.3902 | 2.6417 |
| | WGAN | 3.7429 | 1.6919 | 1.0535 | 0.7648 |

Table 11: FED results for ReLaSH, ReLaSH$_c$, Gau-Diff, RealNVP and WGAN when latent dimension is $k=24, 48$. Each value comes from the mean of 20 repetitions.

methods reflects only the error between the training data and the benchmarks, while for ReLaSH, the RMSE accounts for two sources of error: the error between the training data and the benchmarks, as well as the error between the generated data and the training data. As a result, the RMSE for ReLaSH is generally larger than that of the calibrated methods. In terms of the RMSE of the mean attribute vector, ReLaSH consistently performs the best, while RealNVP performs well occasionally, and WGAN shows slight improvement as the data scale increases.

In terms of RMSE of covariances and FED, ReLaSH generally outperforms the other three generation approaches. Generally, Gau-Diff requires the most memory, while RealNVP is the most time-consuming method among the four. WGAN performs comparably to ReLaSH under limited computational resources, but it yields worse results in terms of error metrics. As the latent dimension $k$, or the values of $m$ and $n$, increase (i.e., as the synthetic data structure becomes more complex), the number of epochs required for achieving similar results to ReLaSH by the other methods increases substantially, leading to a corresponding increase in running time. However, ReLaSH constructs the diffusion model in a low-dimensional continuous embedding space, thereby avoiding the need to train a high-dimensional score network. This significantly reduces sample complexity and enhances both efficiency and accuracy.

### B.3.2 Simulation Results for Dimension Selection

To demonstrate the effectiveness of our proposed latent dimension selection algorithm HTT (see Algorithm 3 for details), we conducted a simulation study with data size $n = p = 500$, $m = 5000$ and true latent dimensions $k_1 = k_2 = k_3 = 4$. All embeddings and parameters were generated according to the simulation settings described in Section B.3 of the Appendix. We applied the HTT pipeline to the simulation data across 30 repetitions, and the results are presented in Table 12.

| $k_1$ | $k_2$ | $k_3$ | Frequency |
|---|---|---|---|
| 4 | 4 | 4 | 27 |
| 4 | 3 | 4 | 1 |
| 4 | 5 | 4 | 2 |

Table 12: Result of latent dimension selection across 30 repetitions.

The results indicate that, in most cases, the HTT algorithm successfully recovers the true latent dimensions. We further apply this algorithm in our real data analyses.

### B.3.3 Sensitivity Analysis

In this section, we conduct some sensitivity analysis to better understand the robustness and tuning requirements of ReLaSH.

For the latent dimensions $k_1$, $k_2$, and $k_3$, we conduct a sensitivity analysis with $m = n = p = 400$ and $k_1 = k_2 = k_3 = 4$. In this setting, all true embeddings and parameters are generated with the same simulation settings as described in Section B.3 of the Appendix. We vary the latent dimensions and present the resulting embedding and generation error metrics across 20 repetitions in Table 13, 14, 15, 16.

| $k_1$ | $k_2$ | $k_3$ | $\Delta_{\mathcal{H}_{\mathrm{m}}} \downarrow$ | $\Delta_{\mathcal{H}_{\mathrm{v}}} \downarrow$ | $\Delta_{\mathcal{X}_{\mathrm{m}}} \downarrow$ | $\Delta_{\mathcal{X}_{\mathrm{v}}} \downarrow$ | FED $\downarrow$ | ReLaSH$_{\mathrm{c}}$ $\Delta_{\mathcal{H}_{\mathrm{v}}} \downarrow$ | ReLaSH$_{\mathrm{c}}$ FED $\downarrow$ |
|---|---|---|---|---|---|---|---|---|---|
| 2 | 2 | 2 | 0.0186 | 0.0074 | 0.0683 | 0.0522 | 0.4984 | 0.0472 | 0.5527 |
| 4 | 4 | 4 | **0.0175** | **0.0057** | **0.0514** | **0.0339** | **0.4392** | **0.0356** | **0.4158** |
| 6 | 6 | 6 | 0.0181 | 0.0064 | 0.0581 | 0.0449 | 0.5239 | 0.0405 | 0.5204 |

Table 13: Result of setting different $k_1 = k_2 = k_3$. Each value comes from the mean of 20 repetitions.

The results indicate that, in general, generation performance is optimal when the true latent space dimensions are used. Although mis-specifying the latent dimension can introduce additional error, the resulting performance remains comparable to baseline methods such as RealNVP. Furthermore, the observed fluctuations in error metrics are consistent with changes in the latent dimensions. For instance, when $k_1$ and $k_2$ are fixed and $k_3$ is varied, error metrics associated with hyperlinks tend to

| $k_1$ | $k_2$ | $k_3$ | $\Delta_{\mathcal{H}_m}\downarrow$ | $\Delta_{\mathcal{H}_v}\downarrow$ | $\Delta_{\mathcal{X}_m}\downarrow$ | $\Delta_{\mathcal{X}_v}\downarrow$ | FED$\downarrow$ | ReLaSH$_c$ $\Delta_{\mathcal{H}_v}\downarrow$ | ReLaSH$_c$ FED$\downarrow$ |
|---|---|---|---|---|---|---|---|---|---|
| 4 | 2 | 4 | 0.0182 | 0.0072 | 0.0639 | 0.0553 | 0.4672 | **0.0304** | 0.4821 |
| 4 | 4 | 4 | **0.0175** | 0.0057 | **0.0514** | **0.0339** | **0.4392** | 0.0356 | **0.4158** |
| 4 | 8 | 4 | 0.0179 | **0.0053** | 0.0576 | 0.0451 | 0.4404 | 0.0398 | 0.4927 |

Table 14: Result of setting different $k_2$. Each value comes from the mean of 20 repetitions.

| $k_1$ | $k_2$ | $k_3$ | $\Delta_{\mathcal{H}_m}\downarrow$ | $\Delta_{\mathcal{H}_v}\downarrow$ | $\Delta_{\mathcal{X}_m}\downarrow$ | $\Delta_{\mathcal{X}_v}\downarrow$ | FED$\downarrow$ | ReLaSH$_c$ $\Delta_{\mathcal{H}_v}\downarrow$ | ReLaSH$_c$ FED$\downarrow$ |
|---|---|---|---|---|---|---|---|---|---|
| 4 | 4 | 2 | 0.0182 | 0.0069 | 0.0563 | 0.0432 | 0.4782 | 0.0428 | 0.5476 |
| 4 | 4 | 4 | **0.0175** | 0.0057 | 0.0514 | **0.0339** | **0.4392** | **0.0356** | **0.4158** |
| 4 | 4 | 8 | 0.0180 | **0.0048** | **0.0497** | 0.0392 | 0.4625 | 0.0388 | 0.4624 |

Table 15: Result of setting different $k_3$. Each value comes from the mean of 20 repetitions.

increase, while those related to attributes fluctuate only slightly. Conversely, varying $k_1$ while keeping $k_2$ and $k_3$ fixed causes greater changes in attribute-related errors. Notably, incorrect selection of $k_2$ leads to elevated errors in both hyperlink-related and attribute-related metrics, which is expected since $u^{(2)}$ is shared by both components.

Additionally, the FED metric appears more sensitive to these changes than the moment-based error metrics, likely due to its more involved computational process. Nonetheless, our simulation results demonstrate that moderate deviations in latent dimension selection do not substantially degrade performance, and the results remain on par with those of baseline methods.

Furthermore, we conduct ablations to demonstrate the effectiveness of the proposed tri-partite partitioning strategy compared to a simpler, unified latent space. A unified $k$-dimensional latent space is a specially case of the tri-partite partition with the triple being $(0, k, 0)$. The tri-partite partition $(k_1, k_2, k_3)$ is motivated by the factorization of the joint likelihood: $u^{(1)}$ (attribute-only), $u^{(3)}$ (hyperlink-only), and $u^{(2)}$ (shared), allowing us to separate but still couple the hypergraph and its attributes for generative modeling. Using a unified $k$-dimensional latent space would mix attribute-only and hyperlink-only variation and requires stronger assumptions to recover interpretable components; in our framework, the block structure gives clean orthogonality conditions to distinguish latent effects driving the generation of hypergraph and attributes, separately and jointly.

In ablation studies shown in Table 17, 18, 19, where a unified latent space of total dimension $k = k_1 + k_2 + k_3$ is employed, the tri-partite model demonstrates superior performance, particularly on attribute-related metrics and FED, which are notably sensitive to latent dimension selection. Additionally, the tri-partite model exhibits more stable training dynamics.

## B.4 ADDITIONAL RESULTS FOR REAL DATA ANALYSIS

In this section, we provide additional results on real data analysis to further assess the quality of the synthetic outcomes.

### B.4.1 CO-CITATION HYPERGRAPH WITH ABSTRACT ATTRIBUTES

The MADStat dataset we utilized contains citation information for over 83,000 papers published across 36 journals between 1975 and 2015, while MADStaText includes the abstracts of all these papers. We focus on the top $n = 1,000$ authors most frequently cited during this period, resulting in a dataset of 35,143 papers that cite at least two of these top authors. Given the large size of this dataset, among all the methods discussed previously, only ReLaSH can effectively handle such

| $k_1$ | $k_2$ | $k_3$ | $\Delta_{\mathcal{H}_m}\downarrow$ | $\Delta_{\mathcal{H}_v}\downarrow$ | $\Delta_{\mathcal{X}_m}\downarrow$ | $\Delta_{\mathcal{X}_v}\downarrow$ | FED$\downarrow$ | ReLaSH$_c$ $\Delta_{\mathcal{H}_v}\downarrow$ | ReLaSH$_c$ FED$\downarrow$ |
|---|---|---|---|---|---|---|---|---|---|
| 2 | 4 | 4 | 0.0179 | 0.0072 | 0.0648 | 0.0582 | 0.4872 | 0.0397 | 0.5623 |
| 4 | 4 | 4 | **0.0175** | **0.0057** | **0.0514** | 0.0339 | **0.4392** | **0.0356** | **0.4158** |
| 8 | 4 | 4 | 0.0176 | 0.0063 | 0.0587 | **0.0322** | 0.4613 | 0.0384 | 0.4729 |

Table 16: Result of setting different $k_1$. Each value comes from the mean of 20 repetitions.

| $k_1$ | $k_2$ | $k_3$ | $\Delta_{\mathcal{H}_\mathrm{m}}\downarrow$ | $\Delta_{\mathcal{H}_\mathrm{v}}\downarrow$ | $\Delta_{\mathcal{X}_\mathrm{m}}\downarrow$ | $\Delta_{\mathcal{X}_\mathrm{v}}\downarrow$ | FED $\downarrow$ | ReLaSH$_\mathrm{c}$ $\Delta_{\mathcal{H}_\mathrm{v}}\downarrow$ | ReLaSH$_\mathrm{c}$ FED $\downarrow$ |
|---|---|---|---|---|---|---|---|---|---|
| 2 | 2 | 2 | 0.0201 | 0.0039 | 0.0511 | 0.0265 | 0.1756 | 0.0436 | 0.2965 |
| 0 | 6 | 0 | 0.0209 | 0.0041 | 0.0562 | 0.0384 | 0.6432 | 0.0412 | 0.5261 |

Table 17: Result of setting a unified latent space of total dimension $k = k_1 + k_2 + k_3$. Each value comes from the mean of 20 repetitions.

| $k_1$ | $k_2$ | $k_3$ | $\Delta_{\mathcal{H}_\mathrm{m}}\downarrow$ | $\Delta_{\mathcal{H}_\mathrm{v}}\downarrow$ | $\Delta_{\mathcal{X}_\mathrm{m}}\downarrow$ | $\Delta_{\mathcal{X}_\mathrm{v}}\downarrow$ | FED $\downarrow$ | ReLaSH$_\mathrm{c}$ $\Delta_{\mathcal{H}_\mathrm{v}}\downarrow$ | ReLaSH$_\mathrm{c}$ FED $\downarrow$ |
|---|---|---|---|---|---|---|---|---|---|
| 4 | 4 | 4 | 0.0175 | 0.0057 | 0.0514 | 0.0339 | 0.4392 | 0.0356 | 0.4158 |
| 0 | 12 | 0 | 0.0181 | 0.0050 | 0.0566 | 0.0698 | 1.0164 | 0.0383 | 1.2767 |

Table 18: Result of setting a unified latent space of total dimension $k = k_1 + k_2 + k_3$. Each value comes from the mean of 20 repetitions.

a large volume of data, owing to the dimension reduction procedure it offers. Consequently, we further narrow the scope by selecting the top $m = 2,000$ papers that cite the greatest number of these authors, and use the top $5,000$ such papers as the population data. This approach ensures that the resulting hypergraph remains both dense and informative. To construct the co-citation hypergraph, we treat each top author as a node, with each paper forming a hyperlink. Specifically, if a paper cites a top author, the corresponding node appears in the hyperlink representing that paper. As noted in (Ji et al., 2022), co-citing two authors in an article suggests that they likely share common research interests. In our application, this co-citation setting provides a hypergraph with sufficient density, which justifies the construction of the co-citation hypergraph as described above.

To construct the attributes corresponding to each hyperlink, we use the abstract data of each paper. We consider the pre-processed corpus of 2,106 words from the dictionary generated by Ke et al. (2023), and further refine this list by removing words that do not appear in the top $m = 2,000$ papers. This leaves us with $p = 2,039$ words for analysis. To construct the $p$-dimensional attribute vector for each hyperlink, the $t$-th entry of the vector is set to the TF-IDF value (Sparck Jones, 1972) of the $t$-th word in relation to the abstract of the corresponding paper ($1 \leq t \leq p$).

We use each approach to generate $\tilde{m} = 32m$ hyperlinks and corresponding attributes. The performance of each method is evaluated using the RMSE of the sample means and covariances of the generated hyperlinks, compared against those of the overall population consisting of the top 5,000 most-cited papers from the selected authors. We also assess the performance by comparing ReLaSH with other calibrated methods (i.e., ReLaSH$_\mathrm{c}$, Gau-Diff, RealNVP, WGAN, VAE) in terms of the RMSE of the hypergraph node means, denoted as $(\Delta_{\mathcal{H}_\mathrm{m}})$, as summarized in Table 20.

Additionally, the Fréchet Embedding Distance (FED) is computed with respect to the chosen latent dimension, with the dimension selection procedure outlined in Appendix B.6. Based on the cross-validation results presented in Table 21, we select $k_1 = k_2 = k_3 = 2$ and generate benchmarks for the embedding machine $\mathcal{E}^\mathcal{T}$ using the population data.

To evaluate the coherence and plausibility of the synthesized statistical research articles, we randomly sampled several papers from the result generated by ReLaSH-$(2, 7, 8)$, as shown in Table 22. We then analyzed the relationship between their cited authors and their top 10 TF-IDF keywords. The analysis examined whether the referenced scholars are thematically aligned with the identified keywords and what research areas such combinations are likely to represent in contemporary statistical literature.

| $k_1$ | $k_2$ | $k_3$ | $\Delta_{\mathcal{H}_\mathrm{m}}\downarrow$ | $\Delta_{\mathcal{H}_\mathrm{v}}\downarrow$ | $\Delta_{\mathcal{X}_\mathrm{m}}\downarrow$ | $\Delta_{\mathcal{X}_\mathrm{v}}\downarrow$ | FED $\downarrow$ | ReLaSH$_\mathrm{c}$ $\Delta_{\mathcal{H}_\mathrm{v}}\downarrow$ | ReLaSH$_\mathrm{c}$ FED $\downarrow$ |
|---|---|---|---|---|---|---|---|---|---|
| 8 | 8 | 8 | 0.0180 | 0.0077 | 0.0554 | 0.0492 | 1.3333 | 0.0271 | 1.3818 |
| 0 | 24 | 0 | 0.0184 | 0.0072 | 0.0625 | 0.0781 | 2.7461 | 0.0304 | 2.9824 |

Table 19: Result of setting a unified latent space of total dimension $k = k_1 + k_2 + k_3$. Each value comes from the mean of 20 repetitions.

| | ReLaSH-(2,2,2) | ReLaSH-(8,8,8) | ReLaSH-(2,7,8) | Calibrated methods |
|---|---|---|---|---|
| $\Delta_{\mathcal{H}_{\mathrm{m}}} \downarrow$ | 3.20 | 2.59 | 3.68 | 3.57 |

Table 20: $\Delta_{\mathcal{H}_{\mathrm{m}}}$ for different generation methods for the co-citation hypergraph generation task. The scale of $\Delta_{\mathcal{H}_{\mathrm{m}}}$ is $10^{-2}$.

| Latent dimension | FD by cross validation |
|---|---|
| $(k_1, k_2, k_3) = (2, 2, 2)$ | **0.257** |
| $(k_1, k_2, k_3) = (4, 4, 4)$ | 0.329 |
| $(k_1, k_2, k_3) = (8, 8, 8)$ | 0.627 |

Table 21: Cross-Validation results for the co-citationship hypergraph.

Table 22: Citation authors and top keywords in abstracts of sample papers. Authors are sorted alphabetically by surname.

| Paper ID | Cited Authors | Top Words |
|---|---|---|
| 56801 | Bradley Efron; Frank Critchley; Alan Gelfand; Robert Kohn; Samuel Kotz; Nan Laird; Faming Liang; Gareth Roberts; G. A. Young | posterior, Bayesian, prior, Markov chain, density, MCMC, mixture, compute, Bayes, approximation |
| 53347 | Kani Chen; Anthony C. Davison; Peter Diggle; Theo Gasser; Harry Kesten; Charles Kooperberg; Jerry Lawless; Youngjo Lee; Danyu Y. Lin; Ian W. Mckeague; Jing Qin; Bernard W. Silverman; Liuquan Sun; Yanqing Sun; Lee-jen Wei; Liugen Xue; Grace Y. Yi; Guosheng Yin; Zhiliang Ying | hazard, proportion, survival, censor, failure, cox, event, maximum, time, cure |
| 55762 | Michael Akritas; Anestis Antoniadis; Raymond Carroll; Marie Davidian; Bert Van Es; Jianqing Fan; Peter Hall; Wolfgang Härdle; M. C. Jones; Takeaki Kariya; Tze Leung Lai; Doug Nychka; Byeong U. Park; John Rice; Joseph Romano; Naisyin Wang | asymptotic, non-parametric, confident, construct, smooth, test, empirical, derivative, kernel, rate |

The cited authors in article 56801 are widely acknowledged for their seminal contributions to Bayesian statistics, posterior inference, and computational algorithms. In particular, many have played pivotal roles in advancing Markov chain Monte Carlo (MCMC) methods, mixture modeling, and the efficient computation of Bayesian posterior distributions. This constellation of expertise strongly suggests that the article represents a significant methodological advancement in Bayesian computational techniques, with emphasis on novel sampling algorithms and strategies for posterior approximation. Moreover, the close alignment between the cited authors and the article's keywords further underscores its relevance to cutting-edge research in statistical computation and contemporary Bayesian analysis.

Regarding article 53347, the cited authors are leading figures in survival analysis and event history modeling. Their collective expertise spans proportional hazards modeling, analysis of censored and failure-time data, and the application of maximum likelihood estimation within survival contexts. The principal keywords centralize themes and methods that are foundational to survival data analysis, particularly in biostatistics. This synergy of authorship and thematic focus strongly indicates that the article introduces methodological innovations in survival modeling—potentially centering on the Cox model, cure rate models, or advanced approaches for the analysis of censored and time-to-event data.

Finally, the referenced authors in article 55762 have made foundational contributions to nonparametric statistics, asymptotic theory, and smoothing techniques. The associated keywords point to research on nonparametric inference, construction of confidence sets, and the theoretical analysis of estimator properties. The intersection of these elements compellingly suggests that the article is devoted to advancing nonparametric estimation methods, with attention to empirical and asymptotic results, and the development of new confidence regions for smooth estimators.

Overall, the pairing of cited authors and prominent keywords in these synthesized articles is highly consistent with the structure and topical alignment found in authentic statistical methodology papers. Each article demonstrates an internally coherent thematic structure, where the cited scholars are authoritative within the research area denoted by the keywords. Therefore, our generative method, ReLaSH, can support the prediction of emerging trends in the statistical community and facilitate the identification of co-citation relationships between specialists and their research interests.

### B.4.2 Recipe Hypergraph with Nutrition Attributes

The *"Epicurious – Recipes with Rating and Nutrition"*[3] dataset contains 17,736 recipes information lifted from http://www.epicurious.com/recipes-menus, each accompanied by user ratings, nutritional information, and ingredient lists. Naturally, we represent each ingredient as a node and each recipe as a hyperedge consisting of the corresponding ingredients. Ratings and nutritional variables ("calories", "sodium", "protein", and "fat") are used as hyperedge attributes. If an ingredient is part of a particular recipe, the hyperedge representing that recipe is connected to the node representing the ingredient.

Note that the attributes in the recipe dataset vary in both scale and interpretation, so we apply z-score standardization to the attribute set before training the generative models to ensure comparability across features. The summary of attributes are listed in Table 23.

|  | Ratings | Calories | Protein | Fat | Sodium |
|---|---|---|---|---|---|
| Mean | 4.192 | 430.04 | 17.46 | 22.90 | 455.90 |
| Std. Error | 0.598 | 257.24 | 16.41 | 17.37 | 429.33 |

Table 23: Means and standard errors for recipe attributes.

For pre-processing, we consolidated ingredient names with the same or highly similar meanings (e.g., "green onion" and "spring onion", "chili pepper" and "hot pepper"), and removed ingredients appearing in only a single recipe. To mitigate issues caused by missing information, we retained only recipes containing at least seven ingredients, resulting in a population dataset of 789 recipes with 200 distinct ingredients. From this population, we randomly selected 300 recipes as the training dataset, where each recipe is associated with five hyperedge attributes. We use each approach to generate $\tilde{m} = 32m$ hyperlinks and corresponding attributes.

Specifically, in the recipe dataset, the attribute dimension is relatively small ($p = 5$) compared to the size of the hypergraph node set ($n = 200$), and all attributes are continuous variables. Therefore, in the embedding procedure of ReLaSH, we set $k_1 = 5$, $k_2 = 0$, while the attribute embedding reduces to an identity mapping. Consequently, after embedding the hypergraph into a $k_1$-dimensional latent space, we concatenate the latent embedding of each hyperedge with its corresponding attributes, and then train diffusion models on the concatenated data. Intuitively, the latent variables obtained through embedding represent underlying structural features of each hyperedge, whereas the attributes provide interpretable features for each recipe. Thus, concatenating the two and performing generation constitutes a natural modeling strategy, which further demonstrates the flexibility of our proposed method.

We also assess the performance by comparing ReLaSH with other calibrated methods (i.e., ReLaSH$_c$, Gau-Diff, RealNVP, WGAN, VAE) in terms of the RMSE of the hypergraph node means, denoted as ($\Delta_{\mathcal{H}_m}$), as summarized in Table 24.

|  | ReLaSH-(5,0,2) | ReLaSH-(5,0,6) | ReLaSH-(5,0,16) | Calibrated methods |
|---|---|---|---|---|
| $\Delta_{\mathcal{H}_m} \downarrow$ | 9.38 | 13.01 | 8.33 | 8.27 |

Table 24: $\Delta_{\mathcal{H}_m}$ for different generation methods for the recipe hypergraph generation task. The scale of $\Delta_{\mathcal{H}_m}$ is $10^{-3}$.

Similarly, the Fréchet Embedding Distance (FED) is computed with respect to the chosen latent dimension. Based on the cross-validation results presented in Table 25, we select latent space di-

---

[3]https://www.kaggle.com/datasets/hugodarwood/epirecipes/data

mensions $(5, 0, 2)$ and generate benchmarks for the embedding machine $\mathcal{E}^{\mathcal{T}}$ using the population data.

| Latent dimension | FD by cross validation |
|---|---|
| $(k_1, k_2, k_3) = (5, 0, 2)$ | **0.194** |
| $(k_1, k_2, k_3) = (5, 0, 4)$ | 0.427 |
| $(k_1, k_2, k_3) = (5, 0, 8)$ | 13.630 |
| $(k_1, k_2, k_3) = (5, 0, 16)$ | 14.064 |

Table 25: Cross-Validation results for the recipe hypergraph.

We examine individual recipes synthesized by ReLaSH$_c$-$(5, 0, 16)$, which generally performs the best among all the generation models, to emphasize the quality of generation. Table 26 presents several randomly selected samples. For interpretability, we assign descriptive names corresponding to real-world cuisines, and all generated recipes resemble authentic dishes. Each sample demonstrates ingredient combinations and nutritional profiles that are both plausible and balanced, particularly for seafood-forward, legume-based, or mixed salads and stews from the Mediterranean and Northern Europe. The coherent ingredient pairings and reasonable macronutrient compositions highlight the applicability of the generated results to realistic culinary settings.

Table 26: Generated recipe samples from ReLaSH$_c$-$(5, 0, 16)$. All nutritional values are reported per entire recipe portion.

| Recipe Name | Rating (out of 5) | Calories (kcal) | Fat (g) | Protein (g) | Sodium (mg) | Ingredients |
|---|---|---|---|---|---|---|
| Mediterranean Fisherman's Bean Stew | 4.22 | 228 | 9.82 | 46.9 | 264 | bean, chickpea, chile pepper, clam, cod, fennel, halibut, lentil, potato, saffron, tomato, onion, wine |
| Hearty Mediterranean Legume Soup | 4.21 | 430 | 16.36 | 32.96 | 482 | carrot, celery, cinnamon, legume, lemon, parsley, pasta, potato |
| Scandinavian Creamy Seafood Chowder | 3.74 | 237 | 10.98 | 17.75 | 1117 | cilantro, citrus, clam, cod, dill, milk, cream, onion |
| Sesame Tuna Mediterranean Salad | 4.40 | 526 | 36.24 | 22.4 | 518 | sesame, spice, tomato, tuna, pepper |
| Mediterranean Mussel | 4.33 | 778 | 49.77 | 63 | 623 | basil, fruit, grape, mussel, parsley, pecan, tomato, spirits |

The "Hearty Mediterranean Legume Soup" can be characterized as a Mediterranean vegetable and legume soup, recalling Italian *minestrone* or Middle Eastern *harira*. With pasta, root vegetables, beans, and aromatic herbs, the nutrition profile is realistic for a hearty soup, rich in protein and complex carbohydrates yet moderate in fat.

The "Scandinavian Creamy Seafood Chowder" is a Scandinavian-inspired creamy seafood chowder, featuring cod, clam, dill, and citrus. Its nutritional composition is plausible for a chowder: adequate protein, moderate fat, and elevated sodium due to seafood and seasoning.

The "Sesame Tuna Mediterranean Salad" represents a fusion-style salad that combines Mediterranean and Asian influences, pairing tuna with sesame, tomato, and peppers. The higher fat content is attributable to sesame, while protein derives from tuna, reflecting the nutritional profile of contemporary main-course salads.

Finally, the "Mediterranean Mussel" is a classical Mediterranean mussel cuisine, integrating mussels, pecans, tomato, and basil, with spirits to enhance flavor. Its high protein and fat levels are consistent with shellfish and nut components, while the addition of fruit and herbs signals modern, health-oriented culinary trends.

To provide a comprehensive illustration of the cuisines, we construct a plot that aggregates the ingredients of each recipe, as shown in Figure 17.

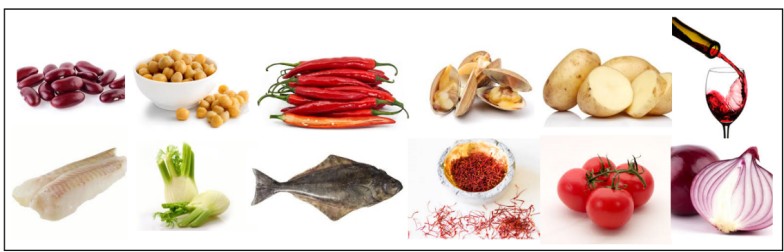

Mediterranean
Fisherman's Bean Stew

Rating: 4.22
Calories: 228 kcal
Fat: 9.82 g
Protein: 46.9 g
Sodium: 264 mg

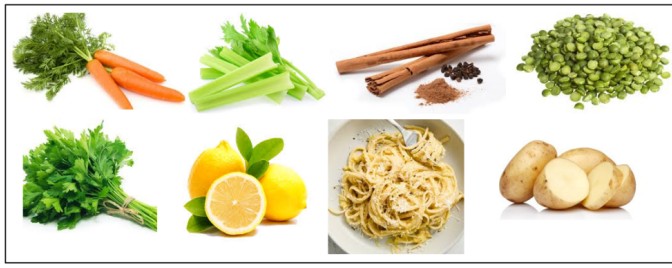

Hearty Mediterranean
Legume Soup

Rating: 4.21
Calories: 430 kcal
Fat: 16.36 g
Protein: 32.96 g
Sodium: 482 mg

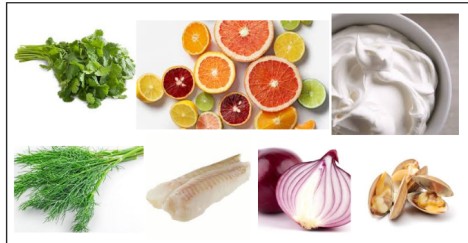

Scandinavian Creamy
Seafood Chowder

Rating: 3.74
Calories: 237 kcal
Fat: 10.98 g
Protein: 17.75 g
Sodium: 1117 mg

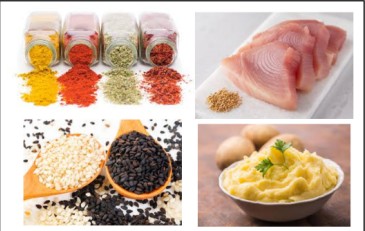

Sesame Tuna
Mediterranean Salad

Rating: 4.40
Calories: 526 kcal
Fat: 36.24 g
Protein: 22.4 g
Sodium: 518 mg

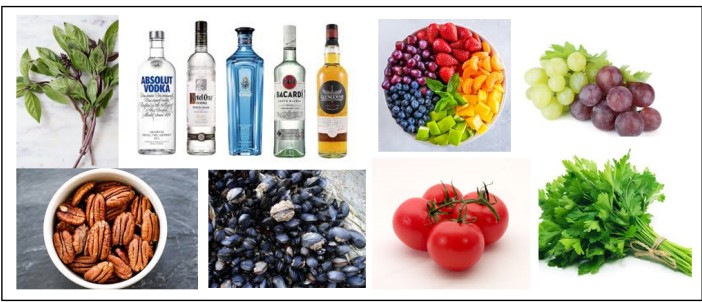

Modern Mediterranean
Mussel & Fruit Salad

Rating: 4.33
Calories: 778 kcal
Fat: 49.77 g
Protein: 63 g
Sodium: 623 mg

Figure 17: Illustration of generated cuisines.

### B.4.3 Medical Hypergraph with Patient Attributes

We also apply ReLaSH and other competitive approaches to a symptom co-occurrence hypergraph constructed from electronic medical records of ICU patients. Specifically, we use the Medical Information Mart for Intensive Care (MIMIC-III; (Johnson et al., 2016)) dataset, which contains clinical data from over 45,000 patients at Beth Israel Deaconess Medical Center in Boston between 2001 and 2012.

In MIMIC-III, we focus on more than 10,000 patients who experienced an ICU stay and for whom a death record is available. To construct the population hypergraph, we consider 4,951 informative patient records with more than 15 co-occurring diseases—an indicator of severe health conditions, covering 2,230 distinct diseases. From this population, we randomly sample 2,000 records as the training set, and retain all $4,951$ patient profiles as the population set for evaluation.

In this hypergraph, each node represents a symptom, and each hyperlink corresponds to a patient profile, with the incident nodes being the symptoms recorded for that patient. The attribute set of each hyperlink is derived from patient metadata, including lifetime, ethnicity, marital status, religion, gender, ICU length of stay, and overall hospital length of stay. Among these, ethnicity, marital status, religion, and gender are categorical variables, which we encode as integers. We use each approach to generate $\tilde{m} = 32m$ hyperlinks and corresponding attributes. Since the attribute dimension is substantially smaller than the hyperlink dimension, we adopt the same identity mapping strategy as in the recipe example for generation in ReLaSH and ReLaSH$_c$.

We also assess the performance by comparing ReLaSH with other calibrated methods (i.e., ReLaSH$_c$, Gau-Diff, RealNVP, WGAN, VAE) in terms of the RMSE of the hypergraph node means, denoted as $(\Delta_{\mathcal{H}_m})$, as summarized in Table 27.

| | ReLaSH-(7,0,2) | ReLaSH-(7,0,16) | Caliberated methods |
|---|---|---|---|
| $\Delta_{\mathcal{H}_m} \downarrow$ | 2.52 | 5.75 | 1.14 |

Table 27: $\Delta_{\mathcal{H}_m}$ for different generation methods for the patient profile generation task. The scale of $\Delta_{\mathcal{H}_m}$ is $10^{-3}$.

Similarly, the Fréchet Embedding Distance (FED) is computed with respect to the chosen latent dimension. Based on the cross-validation results presented in Table 28, we select latent space dimensions $(7, 0, 2)$ and generate benchmarks for the embedding machine $\mathcal{E}^{\mathcal{T}}$ using the population data.

| Latent dimension | FD by cross validation |
|---|---|
| $(7, 0, 2)$ | **0.030** |
| $(7, 0, 4)$ | 0.223 |
| $(7, 0, 8)$ | 0.243 |
| $(7, 0, 16)$ | 0.365 |

Table 28: Cross-Validation results for the symptom co-occurrence hypergraph.

Furthermore, we examine the patient profiles generated by ReLaSH-$(7, 0, 2)$ using the MIMIC-III dataset. For categorical variables such as marital status, ethnicity, religion, and gender, we calibrated the generated outputs against the original training data to preserve the marginal proportions of each category. Examination of continuous variables including ICU stay duration, hospital stay duration, and patients' lifetime data revealed no outliers or abnormal values, indicating that the generated data are clinically reasonable. We randomly sampled 10 patient profiles from the generated set, as shown in Table 29.

Table 29: Sampled Patients: Personal Information and Major Diseases

| ID | Numeric Attributes | Number of Diseases | Demographics | Representative Major Diseases |
|---|---|---|---|---|
| 1 | ICU stay: 8.25 days Lifetime: 86.19 years Hospital Stay: 354.94 hrs | 23 | Catholic White Married Female | Coronary Atherosclerosis; Congestive Heart Failure; Chronic Kidney Disease; Intracerebral Hemorrhage; Dementia |
| 2 | ICU stay: 7.88 days Lifetime: 74.42 years Hospital Stay: 413.36 hrs | 20 | Protestant Quaker White Widowed Male | Atrial Fibrillation; Congestive Heart Failure; Hypertension; Acute Kidney Failure; Diabetes Mellitus |
| 3 | ICU stay: 6.54 days Lifetime: 75.72 years Hospital Stay: 314.68 hrs | 16 | Not Specified White Single Male | Malignant Neoplasm; Chronic Kidney Disease; Cerebral Embolism With Infarction; Heart Failure |
| 4 | ICU stay: 14.16 days Lifetime: 77.55 years Hospital Stay: 512.29 hrs | 36 | Episcopalian White Married Male | Diabetes Mellitus (Uncontrolled); Chronic Kidney Disease (Stage V); Heart Failure; Sepsis; Hepatic Transplant Complications |
| 5 | ICU stay: 7.05 days Lifetime: 57.35 years Hospital Stay: 362.37 hrs | 16 | Not Specified White Single Male | Graft-Versus-Host Disease; Atrial Fibrillation; Acute Vascular Insufficiency of Intestine; Implant Infection |
| 6 | ICU stay: 8.84 days Lifetime: 67.53 years Hospital Stay: 563.22 hrs | 22 | Catholic Black Married Female | Diabetes Mellitus (With Ketoacidosis); Acute On Chronic Diastolic Heart Failure; Sepsis; Thrombocytopenia |
| 7 | ICU stay: 16.95 days Lifetime: 70.73 years Hospital Stay: 515.13 hrs | 21 | Not Specified White Single Male | Secondary Malignant Neoplasms (Intra-Abdominal, Nervous System, Bone, Other Sites); Tuberculosis Of Ureter; Dementia |
| 8 | ICU stay: 2.86 days Lifetime: 41.29 years Hospital Stay: 341.52 hrs | 31 | Protestant Quaker White Widowed Male | Atrial Fibrillation; Congestive Heart Failure; Acute Kidney Failure; Subdural Hemorrhage; Septic Shock |
| 9 | ICU stay: 8.80 days Lifetime: 66.34 years Hospital Stay: 654.82 hrs | 28 | Catholic White Married Female | Malignant Neoplasm of Bronchus and Lung; Heart Failure; Chronic Pain Syndrome; Hepatitis C; Severe Sepsis |
| 10 | ICU stay: 4.00 days Lifetime: 80.60 years Hospital Stay: 136.56 hrs | 13 | Buddhist Asian Divorced Female | Diabetes Mellitus; Chronic Kidney Disease (Stage V); Acute Diastolic Heart Failure; Acute Respiratory Failure; Dysthymic Disorder |

It is worth noting that the number of diseases per patient is relatively high, since during preprocessing we restricted the dataset to ICU patients with at least 15 documented conditions. This reflects the severe health status typical of such cohorts. The demographic characteristics of the generated patients also appear consistent with real-world ICU populations.

To illustrate, we closely examine the set of co-occurring diseases for the first generated patient profile. The disease combination observed is highly representative of older, medically complex ICU patients. Chronic conditions such as hyperlipidemia, hypertensive chronic kidney disease, coronary atherosclerosis, atrial fibrillation, cardiomyopathy, chronic systolic heart failure, and dementia collectively suggest substantial long-term cardiovascular, renal, and neurologic impairment. Superimposed acute complications—including intracerebral hemorrhage, abdominal aortic aneurysm rupture, gastrointestinal bleeding, acute kidney injury, and embolic or thrombotic events—are well-known, life-threatening occurrences frequently encountered in critically ill patients with multiple comorbidities.

Importantly, there is considerable overlap between pre-existing chronic conditions and their acute manifestations. Similarly, congestive heart failure and chronic kidney disease exacerbate one another (the so-called "cardiorenal syndrome"), and this interaction is frequently observed in ICU cohorts. Infectious complications (e.g., pneumococcal infection) and procedural complications (e.g., procedure-related hematoma) are also prevalent in this context. Respiratory comorbidities (such as chronic obstructive asthma or airway obstruction), gastrointestinal conditions, and prior malignancy further reflect the clinical complexity of elderly ICU patients with multimorbidity and high risk of adverse outcomes.

Taken together, the coexistence of chronic diseases, acute organ failures, infections, and iatrogenic complications is highly characteristic of ICU populations, particularly among older adults. The generated case therefore mirrors realistic clinical scenarios commonly observed in critical care cohorts such as MIMIC-III.

### B.4.4 SENSITIVITY ANALYSIS

As addressed in Appendix B.7.1, in the real-data experiments, ReLaSH/ReLaSHc use Forest Diffusion, while Gau-Diff uses a standard diffusion model. This inconsistency in score learners may confound the comparison. To address this concern, we conducted two additional experiments on all three real-world datasets: (1) applying standard diffusion to the joint latent space reconstruction step in ReLaSH/ReLaSHc, and (2) applying ForestDiffusion directly to the entire dataset. The error metrics for these experiments are presented below. In the table, (S.D.) indicates that standard Gaussian diffusion was used for the joint latent space reconstruction step; otherwise, ForestDiffusion was employed. The results are shown in Table 30, 31, 32.

| | $\Delta_{\mathcal{H}_v} \downarrow$ | $\Delta_{\mathcal{X}_m} \downarrow$ | $\Delta_{\mathcal{X}_v} \downarrow$ | FED $\downarrow$ |
|---|---|---|---|---|
| ReLaSH-$(5, 0, 16)$ | 2.355 | **1.533** | **1.112** | 0.766 |
| ReLaSH$_c$-$(5, 0, 16)$ | **1.847** | **1.533** | **1.112** | **0.180** |
| ReLaSH-$(5, 0, 16)$ (S.D.) | 2.067 | 1.832 | 2.854 | 0.659 |
| ReLaSH$_c$-$(5, 0, 16)$ (S.D.) | 2.462 | 1.832 | 2.854 | 0.473 |
| Gau-Diff | 2.375 | 2.154 | 4.256 | 0.802 |
| ForestDiffusion | 1.886 | 8.211 | 2.073 | 0.848 |

Table 30: Results for the recipe generation task. Scales of $\Delta_{\mathcal{H}_v}$, $\Delta_{\mathcal{X}_m}$, $\Delta_{\mathcal{X}_v}$, FED are $10^{-3}$, $10^{-2}$, $10^{-2}$, $10^{-1}$, respectively.

| | $\Delta_{\mathcal{H}_v} \downarrow$ | $\Delta_{\mathcal{X}_m} \downarrow$ | $\Delta_{\mathcal{X}_v} \downarrow$ | FED $\downarrow$ |
|---|---|---|---|---|
| ReLaSH-$(8, 8, 8)$ | **1.626** | **8.608** | 1.887 | **1.454** |
| ReLaSH$_c$-$(8, 8, 8)$ | 2.816 | **8.608** | 1.887 | 6.451 |
| ReLaSH-$(8, 8, 8)$ (S.D.) | 1.925 | 11.836 | 1.889 | 2.648 |
| ReLaSH$_c$-$(8, 8, 8)$ (S.D.) | 2.816 | 11.836 | 1.889 | 6.027 |
| Gau-Diff | 1.672 | 10.016 | **1.824** | 5.060 |
| ForestDiffusion | OOM | OOM | OOM | OOM |

Table 31: Results for the co-citation hypergraph generation task. Scales of $\Delta_{\mathcal{H}_v}$, $\Delta_{\mathcal{X}_m}$, $\Delta_{\mathcal{X}_v}$, FED are $10^{-3}$, $10^{-2}$, $10^{-1}$, $10^{-1}$, respectively.

The error of the three generation tasks generally tends to increase slightly when the distribution-free generator used in the latent space reconstruction step is standard Gaussian diffusion rather than

| | $\Delta_{\mathcal{H}_v} \downarrow$ | $\Delta_{\mathcal{X}_m} \downarrow$ | $\Delta_{\mathcal{X}_v} \downarrow$ | FED $\downarrow$ |
|---|---|---|---|---|
| ReLaSH-$(7,0,2)$ | **3.260** | 2.989 | 1.435 | 0.532 |
| ReLaSH$_c$-$(7,0,2)$ | 27.794 | 2.989 | 1.435 | **0.013** |
| ReLaSH-$(7,0,2)$ (S.D.) | 4.085 | **2.849** | **1.385** | 0.835 |
| ReLaSH$_c$-$(7,0,2)$ (S.D.) | 17.266 | **2.849** | **1.385** | 0.548 |
| Gau-Diff | 4.268 | 3.497 | 1.719 | 39.731 |
| ForestDiffusion | OOM | OOM | OOM | OOM |

Table 32: Results for the patient profile generation task. Scales of $\Delta_{\mathcal{H}_v}, \Delta_{\mathcal{X}_m}, \Delta_{\mathcal{X}_v}$, FED are $10^{-4}$, $10^{-3}, 10^{-1}, 10^{-2}$, respectively.

ForestDiffusion. This is expected, as ForestDiffusion has a more complex structure and typically achieves lower error across the entire process. Nonetheless, using the ReLaSH pipeline still outperforms directly applying standard diffusion for the whole dataset, highlighting the effectiveness of the ReLaSH pipeline. However, it is important to note that ForestDiffusion encounters out-of-memory (OOM) errors on the patient profile and co-citation datasets due to their high dimensionality when processed as tabular data. In contrast, within ReLaSH, both standard diffusion and ForestDiffusion exhibit comparable, fast runtimes. ForestDiffusion is optimized for low-dimensional tabular data, with the largest tested feature size being 90 (Jolicoeur-Martineau et al., 2024), while the patient profile has 2,237 features and the co-citation dataset has 3,039. These results also illustrate a subtle difference in ReLaSH/ReLaSHc performance depending on the type of distribution-free generator used for joint latent space reconstruction. Because the joint latent space is low-dimensional, distribution-free generators such as ForestDiffusion tend to perform well. This finding also underscores the role of our dimension-reduction technique in the pipeline, which greatly improves both efficiency and accuracy in our generative model.

## B.5 FRÉCHET EMBEDDING DISTANCE (FED)

In generative modeling, Fréchet Inception Distance (FID) is a widely used metric for evaluating the quality of generated samples, particularly in visual tasks. FID compares the feature representations (e.g., from the coding layers) of two datasets (e.g., real images and AI-generated images) using a pre-trained Inception model, such as Inception-v3. The difference between the two populations is measured by calculating the Fréchet distance, assuming that both distributions follow a Gaussian distribution. This distance is computed using the means and covariances of the respective distributions. While FID has proven effective for evaluating generative models in vision-related tasks, its applicability is limited in other domains, as it relies on pre-trained Inception models specifically trained on visual data.

To overcome this limitation, FID has been adapted into specialized variants for different domains. For instance, Fréchet Audio Distance (FAD) (Kilgour et al., 2018) has been proposed for evaluating music enhancement algorithms, Fréchet Video Distance (FVD) (Unterthiner et al., 2019) is used for generative models of video, and Fréchet ChemNet Distance (FCD) (Preuer et al., 2018) is used for evaluating AI-generated molecules. These variants maintain the core idea of FID, but they are customized to better suit the unique characteristics of their respective domains.

In a similar vein, we extend the idea of FID to our own task by introducing Fréchet Embedding Distance (FED). FED generalizes FID for generative tasks where no pre-trained models are available for evaluation. Specifically, for a given generative task $T$, we define a "true" dataset $\mathcal{X}^{\text{true}} = \{x_i^{\text{true}}\}_{i \in [n_{\text{true}}]}$ representing real data, and a generated dataset $\mathcal{X}^{\text{gen}} = \{x_i^{\text{gen}}\}_{i \in [n_{\text{gen}}]}$ sampled from the generative model being evaluated. To compute FED, we introduce an embedding machine $\mathcal{E}^T : \mathcal{X} \to \mathbb{R}^K$, which maps the data points into a continuous embedding space that captures the essential features of the original data. Once the data points are mapped into this embedding space, FED calculates the distance between the distributions of the true and generated datasets as

$$\text{FED}(\mathcal{X}^{\text{true}}, \mathcal{X}^{\text{gen}}) = \|\mu^{\text{true}} - \mu^{\text{gen}}\|_2^2 + \text{Tr}(\Sigma^{\text{true}} + \Sigma^{\text{gen}} - 2(\Sigma^{\text{true}}\Sigma^{\text{gen}})^{1/2}),$$

where $\mu^{\text{true}}, \mu^{\text{gen}}$ are the sample means of $\{\mathcal{E}^T(x_i^{\text{true}})\}_{i \in [n_{\text{true}}]}, \{\mathcal{E}^T(x_i^{\text{gen}})\}_{i \in [n_{\text{gen}}]}$, and $\Sigma^{\text{true}}, \Sigma^{\text{gen}}$ are the sample covariance matrices of $\{\mathcal{E}^T(x_i^{\text{true}})\}_{i \in [n_{\text{true}}]}, \{\mathcal{E}^T(x_i^{\text{gen}})\}_{i \in [n_{\text{gen}}]}$. which provides a flexible and robust evaluation metric for generative models in various tasks.

Thus, similar to how FID has been adapted for different domains, we introduce FED as a flexible and robust evaluation metric for generative models across a variety of tasks. FED generalizes the concept of FID by eliminating the dependence on pre-trained models. Furthermore, it proves particularly useful in scenarios where data distributions are too complex to compare directly, such as the high-dimensional hyperedges together with attributes explored in this study.

To avoid the possible invalidity problem emerging from embedding the entire population hypergraph (containing the training set) to obtain reference parameters, we split the training dataset from the overall population and using the remaining data to train the embedding machine $\mathcal{E}^{\mathcal{T}}$, thereby ensuring that the evaluator and evaluated models are independent. In our experiments, this data splitting does not change the final selected rank of the embedding machine.

For the patient profile and recipe datasets, the a-FED values is very similar to FED, as the training sets were constructed by randomly sampling from the population. For the co-citation dataset, there is slightly more variation in the FED values, since we constructed the training hypergraph using the top $m = 2,000$ most cited papers and defined the population as the top $5,000$ most cited papers. Nevertheless, our method consistently outperforms other baseline methods under these independent evaluation settings. Additionally, we observed that generally, the value of a-FED is slightly higher than FED, as reducing the amount of information available for training the embedding machine leads to an increase in error.

Furthermore, the approach of evaluating synthetic data using a pre-trained embedding machine $\mathcal{E}^{\mathcal{T}}$ is inspired by the Inception Score (IS) and Fréchet Inception Distance (FID), both of which use an Inception model pre-trained on a large subset of ImageNet to assess image generative models (Szegedy et al., 2016). While using a Fréchet-style distance may introduce some dependency between the evaluator and the generator, this effect is mitigated when the training set is randomly selected. In hypergraph settings, hyperlink sparsity can limit the effectiveness of splitting the dataset into disjoint training and testing sets. Using a larger training set that includes the data used to train the generative models for the embedding machine may introduce some bias in evaluation, but it also significantly improves its ability to capture the characteristics of the full population.

## B.6 EVALUATION METRIC CALCULATION

In this subsection, we provide a detailed explanation of how we compute the quantities used in the evaluation steps for both the simulation experiments and real data analysis.

Given an observed hypergraph $\mathcal{H}([n], \mathcal{E}_m, \mathcal{X}_m)$, where $\mathcal{E}_m = \{e_1, e_2, \cdots, e_m\}$ and $\mathcal{X}_m = \{x_1, x_2, \cdots, x_m\}$, we encode each hyperlink as a binary vector $e_j \in \{0,1\}^n$, where the $i$-th element of $e_j$ indicates whether node $i$ is connected to the hyperlink. We then concatenate $e_j$ with the corresponding attribute vector $x_j$ to form an $(n+p)$-dimensional vector $h_j = (e_j \quad x_j)$, which encodes the complete information of the $j$-th hyperlink in the observed hypergraph. Consequently, the dataset used to train the generative models (i.e. Gau-Diff, WGAN, RealNVP, VAE) compared with our method is $\mathcal{D}^{\text{true}} = \{h_j\}_{j=1}^m$.

Similarly, for the generated hypergraph $\tilde{\mathcal{H}}([n], \tilde{\mathcal{E}}_{\tilde{m}}, \tilde{\mathcal{X}}_{\tilde{m}})$, where $\tilde{\mathcal{E}}_{\tilde{m}} = \{\tilde{e}_1, \cdots, \tilde{e}_{\tilde{m}}\}$ and $\tilde{\mathcal{X}}_{\tilde{m}} = \{\tilde{x}_1, \cdots, \tilde{x}_{\tilde{m}}\}$, we encode the hyperlinks in the same manner as the observed hypergraph, resulting in the dataset $\mathcal{D}^{\text{gen}} = \{\tilde{h}_j\}_{j=1}^{\tilde{m}}$.

Also, for the random hyperlink $E$ and its random attributes $X$, we denote $H = (E \quad X)$ as the encoded random vector, in the same manner as above.

### B.6.1 CALCULATION OF FED

To calculate the Fréchet Embedding Distance (FED), we first define an embedding machine $\mathcal{E}^{\mathcal{T}}$ and optimize its parameters using the gradient descent method. For each hyperlink data point $h_j$, we aim to minimize the following loss function:

$$\ell_j(u_j) = \ell_{j,H}(u_j) + \lambda \ell_{j,X}(u_j) = -\sum_{i=1}^{n}\left[\log(1 + \exp(u_j^{(23)\top}z_i + \alpha_i)) - h_{ji}(u_j^{(23)\top}z_i + \alpha_i)\right]$$

$$- \lambda \sum_{i=1}^{p}\left[\frac{1}{2}(u_j^{(12)\top}b_i + \gamma_i)^2 - x_{ji}(u_j^{(12)\top}b_i + \gamma_i)\right],$$

where $z_i, b_i, \alpha_i, \gamma_i$ represent the true node embeddings and associated parameters. We optimize this loss using the gradient descent approach as in Algorithm 4, which updates $\{u_j\}_{j=1}^{\tilde{m}}$ during each iteration. The initialization steps are the same as in Algorithm 2. This procedure allows us to obtain the embeddings $\{\mathcal{E}^{\mathcal{T}}(h_j)\}_{j=1}^{m}$ for the observed hypergraph and $\{\mathcal{E}^{\mathcal{T}}(\tilde{h}_j)\}_{j=1}^{\tilde{m}}$ for the generated hypergraph. Finally, we compute the FED between the two datasets according to its definition, i.e.

$$\text{FED}(\mathcal{D}_{\text{true}}, \mathcal{D}_{\text{gen}}) := \text{FID}(\{\mathcal{E}^{\mathcal{T}}(h_j)\}_{j=1}^{m}, \{\mathcal{E}^{\mathcal{T}}(\tilde{h}_j)\}_{j=1}^{\tilde{m}}).$$

---

**Algorithm 4** Projected Gradient Descent for Embedding Machine $\mathcal{E}^{\mathcal{T}}$ in FED Calculation

---

**Require:** Initialized $U_{(0)}$, known embeddings and parameters $\{\alpha, \gamma, Z, B\}$, generated hypergraph matrix $H$ and according attribute matrix $X$, learning rate $\eta$, likelihood weight parameter $\lambda$, maximum number of iterations $T$.

1: **for** $t = 1$ to $T$ **do**
2: $\quad$ Calculate $\Theta_{(t-1)}^{H} = \alpha\mathbf{1}_m^\top + U_{(t-1)}^{(23)\top}Z$, and $\Theta_{(t-1)}^{X} = \gamma\mathbf{1}_m^\top + U_{(t-1)}^{(12)\top}B$
3: $\quad U_{(t)}^{(1)} = U_{(t-1)}^{(1)} - \eta_{U^{(1)}}\nabla_{U^{(1)}}\ell_{(t-1)} = U_{(t-1)}^{(1)} + \lambda\eta_{U^{(1)}}(X - f_A'(\Theta_{(t-1)}^{X}))B_{1,(t-1)}$
4: $\quad U_{(t)}^{(2)} = U_{(t-1)}^{(2)} - \eta_{U^{(2)}}\nabla_{U^{(2)}}\ell_{(t-1)} = U_{(t-1)}^{(2)} + \eta_{U^{(2)}}\{\lambda(X - f_A'(\Theta_{(t-1)}^{X}))B_{2,(t-1)} + (H -$
$\quad \sigma(\Theta_{(t-1)}^{H})Z_{2,(t-1)})\}$
5: $\quad U_{(t)}^{(3)} = U_{(t-1)}^{(3)} - \eta_{U^{(3)}}\nabla_{U^{(3)}}\ell_{(t-1)} = U_{(t-1)}^{(3)} + \eta_{U^{(3)}}(H - \sigma(\Theta_{(t-1)}^{H}))Z_{3,(t-1)}$
6: $\quad$ Project the embedding $U_{(t)}$ to the constraint set, with the transformation in Remark 5.
7: **end for**
8: **return** $U_{(T)}$ as $\{\mathcal{E}^{\mathcal{T}}(h_j)\}_{j\in\tilde{m}}$ for further FED calculation.

---

For the synthetic data analysis, we directly utilize the true node embeddings and associated parameters $(z_i, b_i, \alpha_i, \gamma_i)$ from the data generation procedure. In contrast, for the real data analysis, we construct the reference node embeddings and corresponding parameters $(z_i, b_i, \alpha_i, \gamma_i)$ by embedding the population hypergraph together with its attributes. For instance, in the co-citatioship dataset, we analyze a training set of $2,000$ papers, which is sampled from a larger population of $5,000$ papers. To obtain a fair comparison, we embed the entire population dataset to derive the reference embeddings and associated parameters $(z_i, b_i, \alpha_i, \gamma_i)$.

To select the latent space for the embedding machine, we use cross-validation to determine the optimal latent dimension. For each chosen dimension $k$, we split the population hypergraph into a training set and a testing set, with 80% of the nodes and attributes used for training and the remaining 20% for testing. We then perform the embedding procedure on both the training and testing sets to estimate the hypergraph embeddings, denoted as $\mathcal{U}_{\text{train}}^{k}$ and $\mathcal{U}_{\text{test}}^{k}$. Next, we calculate the Fréchet Distance (FD) between $\mathcal{U}_{\text{train}}^{k}$ and $\mathcal{U}_{\text{test}}^{k}$, denoted as $\text{FD}_k$, and compare the FD values across different latent dimensions $k$. The dimension that minimizes the FD is selected as the optimal latent dimension for the embedding machine $\mathcal{E}^{\mathcal{T}}$, and the corresponding $(\hat{z}_i, \hat{b}_i, \hat{\alpha}_i, \hat{\gamma}_i)$ are used as benchmarks for the embedding machine.

The cross-validation results for the three real datasets are shown in Appendix B.4 respectively, indicating that $k = 2$ or $k_1 = k_2 = k_3 = 2$ is selected for all three datasets. This choice is based on the fact that low-dimensional latent spaces yield the smallest error, thus it provides a much fairer benchmark for comparing different generative models.

### B.6.2 MEANS AND COVARIANCES

As outlined in the setup of simulated data, we assume that the hyperlink embeddings $\{u_j\}_{j=1}^{m}$ are drawn from a distribution $\mathbb{P}_U$. Using this assumption, we can compute the expected occurrences of

the nodes as well as the covariances between the occurrences of the nodes by considering $\mathbb{E}(E)$ and $\text{cov}(E_i, E_j)$, and the expected mean $\mathbb{E}(X)$ and covariances of attributes $\text{cov}(X_i, X_j)$.

For $\mathbb{E}(H) = \mathbb{E}(E \quad X)$, we have

$$\mathbb{E}(H_i) = \mathbb{E}_{\mathbb{P}_U}[\mathbb{E}(H_i|U)] = \mathbb{E}_{\mathbb{P}_U}[\mathbb{E}(E_i|U)]\mathbf{1}_{(i \leq n)} + \mathbb{E}_{\mathbb{P}_U}[\mathbb{E}(X_i|U)]\mathbf{1}_{(i > n)}$$
$$= \mathbb{E}_{\mathbb{P}_U}[\sigma(z_i^\top U^{(23)} + \alpha_i)]\mathbf{1}_{(i \leq n)} + \mathbb{E}_{\mathbb{P}_U}[(b_i^\top U^{(12)} + \gamma_i)]\mathbf{1}_{(i > n)},$$

where $H_i$ is the $i$-th coordinate of $H$.

Similarly, for $\text{cov}(E_i, E_j)$ and $\text{cov}(X_i, X_j)$, we have

$$\text{cov}(E_i, E_j) = \mathbb{E}_{\mathbb{P}_U}[\text{cov}(E_i, E_j \mid U)] + \text{cov}_{\mathbb{P}_U}(\mathbb{E}_{\mathbb{P}_U}[E_i \mid U], \mathbb{E}_{\mathbb{P}_U}[E_j \mid U])$$
$$= \text{cov}_{\mathbb{P}_U}(\mathbb{E}_{\mathbb{P}_U}[E_i \mid U], \mathbb{E}_{\mathbb{P}_U}[E_j \mid U])$$
$$= \mathbb{E}_{\mathbb{P}_U}[\sigma(z_i^\top U^{(23)} + \alpha_i)\sigma(z_j^\top U^{(23)} + \alpha_j)]$$
$$- \mathbb{E}_{\mathbb{P}_U}[\sigma(z_i^\top U^{(23)} + \alpha_i)]\mathbb{E}_{\mathbb{P}_U}[\sigma(z_j^\top U^{(23)} + \alpha_j)],$$

and

$$\text{cov}(X_i, X_j) = \mathbb{E}_{\mathbb{P}_U}[\text{cov}(X_i, X_j \mid U)] + \text{cov}_{\mathbb{P}_U}(\mathbb{E}_{\mathbb{P}_U}[X_i \mid U], \mathbb{E}_{\mathbb{P}_U}[X_j \mid U])$$
$$= \text{cov}_{\mathbb{P}_U}(\mathbb{E}_{\mathbb{P}_U}[X_i \mid U], \mathbb{E}_{\mathbb{P}_U}[X_j \mid U])$$
$$= \mathbb{E}_{\mathbb{P}_U}[(b_i^\top U^{(12)} + \gamma_i)][(b_j^\top U^{(12)} + \gamma_j)]$$
$$- \mathbb{E}_{\mathbb{P}_U}[(b_i^\top U^{(12)} + \gamma_i)]\mathbb{E}_{\mathbb{P}_U}[(b_j^\top U^{(12)} + \gamma_j)].$$

In our evaluations on real-world datasets, these expectations are approximated by their empirical counterparts, as illustrated below.

### B.6.3 COMPARISON WITH BENCHMARKS OF MEANS, COVARIANCES, AND FEDS

Since directly computing the expectations $\mathbb{E}(H_i)$ and covariances $\text{cov}(H_i, H_j)$ based on the distribution $\mathbb{P}_U$ is generally challenging, in synthetic data analysis, we approximate them using Monte Carlo integration with $N = 20,000$ samples. Specifically, for a given set of parameters and node embeddings $(B, Z, \alpha, \gamma)$, we generate a test set of embeddings $\mathcal{U}^{\text{test}} = \{u_j^{\text{test}}\}_{j=1}^N$, and then generate a hypergraph with attributes $\mathcal{H}^{\text{test}}([n], \mathcal{E}_N, \mathcal{X}_N)$ according to the hypergraph generation model. We encode $\mathcal{H}^{\text{test}}([n], \mathcal{E}_N, \mathcal{X}_N)$ into hyperlink sets $\{h_j^{\text{test}}\}_{j=1}^N$, and take the sample mean $\widehat{\mathbb{E}}(E), \widehat{\mathbb{E}}(X)$ as an approximation of $\mathbb{E}(E), \mathbb{E}(X)$, while the sample variance $\widehat{\text{cov}}(E, E), \widehat{\text{cov}}(X, X)$ provides an approximation of $\text{cov}(H, H), \text{cov}(X, X)$. In real data analysis, we directly utilize the sample mean and variance of population data as an approximation of $\text{cov}(H, H), \text{cov}(X, X)$.

To calculate the RMSE of the means and variance-covariances of node co-occurrence and attributes vector from the generated hypergraph $\widetilde{\mathcal{H}}([n], \widetilde{\mathcal{E}}_{\tilde{m}}, \widetilde{\mathcal{X}}_{\tilde{m}})$, we first encode the generated hypergraph as vectors $\{\tilde{e}_j\}_{j=1}^{\tilde{m}}, \{\tilde{x}_j\}_{j=1}^{\tilde{m}}$, then calculate the sample means and sample covariances of both populations as $\widetilde{\mathbb{E}}(E), \widetilde{\mathbb{E}}(X)$ and $\widetilde{\text{cov}}(E, E), \widetilde{\text{cov}}(X, X)$. We then calculate the RMSEs for each population by computing:

$$\frac{1}{n}\|\widetilde{\text{cov}}(E, E) - \widehat{\text{cov}}(E, E)\|_F, \quad \frac{1}{\sqrt{n}}\|\widehat{\mathbb{E}}(E) - \widetilde{\mathbb{E}}(E)\|_2,$$

$$\frac{1}{p}\|\widetilde{\text{cov}}(X, X) - \widehat{\text{cov}}(X, X)\|_F, \quad \frac{1}{\sqrt{p}}\|\widehat{\mathbb{E}}(X) - \widetilde{\mathbb{E}}(X)\|_2.$$

Since we are comparing our method for mixed data types with other generative models that focus on continuous data, we perform calibrations on such methods (e.g. RealNVP, WGAN, Gau-Diff, VAE, and so on) to ensure a fair comparison. Recall that the hyperlink data is encoded as binary vectors in the first $n$ dimensions, while the generative models produce $(n + p)$-dimensional continuous vectors. To map the first $n$ dimensions of the continuous vectors to hyperlinks, we incorporate a calibration step based on the observed hypergraph $\mathcal{H}([n], \mathcal{E}_m, \mathcal{X}_m)$. Let $\{[y_{1,j}^{\text{gen}}, y_{2,j}^{\text{gen}}]\}_{j=1}^{\tilde{m}}$ represent the generated $(n + p)$-dimensional vectors, where $y_{1,j}^{\text{gen}} \in \mathbb{R}^n$ and $y_{2,j}^{\text{gen}} \in \mathbb{R}^p$ for each $j \in [\tilde{m}]$.

Denote the $i$-th coordinate of $y_{1,j}^{\text{gen}}$ by $y_{1,ji}^{\text{gen}}$, and let $\tau_i$ be a threshold for each $i = 1, 2, \ldots, n$. Define $\tilde{y}_{1,ji}^{\text{gen}} = \mathbf{1}_{\{y_{1,ji}^{\text{gen}} \geq \tau_i\}}$, where $\mathbf{1}$ is the indicator function. The thresholds $\{\tau_i\}_{i=1}^n$ are selected such that the following condition holds for each $i \in [n]$:

$$\frac{1}{\tilde{m}} \sum_{j=1}^{\tilde{m}} \tilde{y}_{1,ji}^{\text{gen}} = \frac{1}{m} \sum_{j=1}^{m} e_{ji}.$$

This calibration step effectively decreases the error from the hypergraph part, as it mimics the distribution of node co-occurrence in the observed hypergraph. The rationale behind is to ensure that the node degree sequence of the generated hypergraph matches that of the observed hypergraph. Finally, we replace the $y_{1,j}^{\text{gen}}$'s with their calibrated versions $\tilde{y}_{1,j}^{\text{gen}}$'s, which are binary-valued variables that represent a hyperlink. The calibrated encoded vector becomes $\mathcal{D}^{\text{gen}} = \{\tilde{h}_j\}_{j=1}^{\tilde{m}} := \{[\tilde{y}_{1,j}^{\text{gen}}, y_{2,j}^{\text{gen}}]\}_{j=1}^{\tilde{m}}$, representing a hypergraph with $\tilde{m}$ hyperlinks, where the $j$-th hyperlink is associated with an attribute vector $y_{2,j}^{\text{gen}}$.

In this way, we can also introduce a variant of ReLaSH, denoted ReLaSH$_c$ (calibrated ReLaSH), which generates hyperlinks not by sampling from the connection probability matrix $\tilde{P} = \text{logit}(\tilde{\Theta}^H)$, but by calibrating $\tilde{P}$ with respect to the training data as described above. Consequently, ReLaSH$_c$ achieves performance comparable to competing generative models in terms of the RMSE of hyperlink vector means, and in both synthetic and real data experiments, we observe that ReLaSH$_c$ attains state-of-the-art results on certain error metrics in specific settings.

For the FED, we generate $\mathcal{D}^{\text{test}} = \{\mathcal{E}^{\mathcal{T}}(h_j^{\text{test}})\}_{j=1}^N$ as the ground truth. We then compute the FED between the test dataset and the generated dataset using the formula FED$(\mathcal{D}^{\text{test}}, \mathcal{D}^{\text{gen}})$.

## B.7 DETAILS AND IMPLEMENTATION OF THE GENERATIVE MODELS

In this subsection, we describe the details of the implementation of the methods used in our simulation studies and real data analysis.

### B.7.1 RELASH AND GAU-DIFF

A diffusion model consists of a forward process and a reverse process. In the forward process, Gaussian noise is gradually added to the original data, eventually transforming it into pure noise. In the reverse process, denoising neural networks are trained to remove the noise and recover new samples from the data distribution.

Specifically, for both the ReLaSH and the Gau-Diff, we consider the Ornstein-Uhlenbeck process, which is described by the following Stochastic Differential Equation (SDE) in the forward process:

$$\mathrm{d}U_t = -U_t \, \mathrm{d}t + \sqrt{2} \, \mathrm{d}W_t, \tag{4}$$

where $\{W_t\}_{t \in [0,T]}$ is a standard Wiener process. Under mild conditions, as noise is gradually added to the data over time, the resulting perturbed realizations will approach a standard multivariate Gaussian distribution for sufficiently large $T$. The reverse process, which generates new realizations from the noisy output, is given by another SDE:

$$\mathrm{d}U_t^\leftarrow = (U_t^\leftarrow + 2 \log p_{T-t}(U_t^\leftarrow)) \, \mathrm{d}t + \sqrt{2} \, \mathrm{d}\tilde{W}_t, \tag{5}$$

where $\nabla \log p_t(\cdot)$ is the score function at time $t$. Since direct sampling from this SDE is computationally infeasible, we discretize the process with a step size $h > 0$ and train deep neural networks to estimate the score functions at $T/h$ discrete time steps.

The Ornstein–Uhlenbeck forward process specified can also be expressed as $U_t = e^{-t}U_0 + \sqrt{1 - e^{-2t}} z$, with $z \sim \mathcal{N}(0, I_d)$. The conditional score has a closed-form expression, i.e., $\nabla_{u_t} \log p_t(u_t|u_0) = -z/\sigma_t$ with $\sigma_t^2 = 1 - e^{-2t}$. Plugging this into the general objective, the loss simplifies to $\ell(\boldsymbol{\theta}) = \frac{1}{2} \mathbb{E}_{u_0,t,z} \left[ \left\| \mathbf{s}_{\boldsymbol{\theta}}(u_t, t) \sigma_t + z \right\|_2^2 \right]$, which is the form used in our implementation.

In Gau-Diff, we train a diffusion model on the encoded $(n + p)$-dimensional vectors $\{h_j\}_{j=1}^m$ with the Ornstein-Ulhenbeck process as shown above. In ReLaSH, diffusion models are trained on the $k$-dimensional continuous spaces, with the same architecture as in Gau-Diff without the calibration step. We refer to the diffusion model architecture as outlined in `https://github.com/yang-song/score_sde`. The score neural network is set as a 5-layer MLP, which is enough to avoid overfitting during generation of tabular data. For optimization, we use Adam (Kingma & Ba, 2014) and follow the schedule therein. The models are all trained for fixed epochs for ReLaSH, while trained for a larger fixed epochs for Gau-Diff, with fixed batch size of 128. The difference in required epochs comes from the fact that the diffusion model in ReLaSH only works on a population of $k$-dimensional vectors, while the diffusion model in Gau-Diff works on a population of $(n + p)$-dimensional vectors with $k \ll n + p$. Therefore, the dimension reduction performed by our algorithm successfully boosts the speed of hypergraph generation and enhances performance in all error types.

In the application to real datasets, we employ the Forest Diffusion method proposed by Jolicoeur-Martineau et al. (2024), which leverages XGBoost, a widely used Gradient Boosted Tree (GBT) technique, instead of neural networks to learn the score function. Owing to its substantial computational cost, Forest Diffusion is applied only to the ReLaSH and ReLaSH$_c$ methods, as both train the generative model in the embedded latent space of relatively low dimension. Forest Diffusion also provides an R implementation, which ensures a smooth integration with our entirely R-based pipeline. In contrast, Gau-Diff requires training directly on the full dataset of dimension $(n + p)$, for which we instead adopt the standard diffusion model described earlier with a simple architecture. This comparison highlights the scalability advantage of our dimension reduction framework.

### B.7.2   GANs

Generative Adversarial Networks (GANs) (Goodfellow et al., 2020) are a class of generative models based on a game-theoretic framework between two neural networks: the generator and the discriminator. The generator aims to produce synthetic samples from random noise, while the discriminator attempts to distinguish between real samples from the dataset and those produced by the generator.

However, this setup is often plagued by issues such as vanishing gradients and mode collapse, particularly when the discriminator becomes too strong, leaving the generator with poor learning signals. To address these challenges during training, we employ the Wasserstein GAN (WGAN) (Arjovsky et al., 2017) in our experiments. WGAN introduces the Wasserstein-1 distance as the divergence metric between real and generated data distributions, providing smoother gradients and significantly improved training stability, with the loss formulated as:

$$\min_G \max_{f \in \text{Lip}_1} \mathbb{E}_{x \sim p_{\text{data}}}[f(x)] - \mathbb{E}_{z \sim p(z)}[f(G(z))],$$

where $f$ is a 1-Lipschitz function parameterized by the discriminator. Unlike the classic GAN discriminator, the WGAN critic outputs real-valued scores for how "real" a sample appears, rather than a probability. To enforce the Lipschitz constraint, weight clipping (or, in improved versions, gradient penalty) is used during training.

For all simulation experiments, the WGAN is trained for a fixed number of epochs, ranging from 1500 to 15000, with a consistent learning rate and a latent dimension fixed at 50 to handle synthetic datasets of varying complexity across all settings. In each real data experiment, we adjust the number of training epochs based on the dataset's complexity, considering limited computational resources, and have saved the specific settings for each dataset to ensure reproducibility. We observe that WGAN exhibits stable convergence under these configurations and is comparable to ReLaSH in terms of run time. We apply the early stopping mechanism which monitors the generator's loss, halting training if there is no improvement for patience epochs to prevent overfitting.

### B.7.3   VAE

Variational Autoencoders (VAEs) represent a class of generative models that fuse principles from autoencoders and probabilistic graphical models (Kingma & Welling, 2013). VAEs learn an approximate posterior distribution over latent variables by maximizing a lower bound on the ELBO. The model consists of an encoder network that parameterizes a variational distribution over the latent space, and a decoder network that reconstructs data from latent codes. The loss function includes

both a reconstruction term and a regularization (KL divergence) term encouraging the latent distribution to remain close to a chosen prior, commonly the standard normal distribution.

In our experiments on real world datasets, we observe that VAE training is significantly more time-consuming compared to WGAN and ReLaSH. Furthermore, the quality of generated samples, as evaluated by our downstream metrics, systematically lags behind that of the other generative approaches in our comparative study. As a result, we exclude VAE from large-scale experiments involving scalable $m, n, p, K$ to maintain feasibility and focus our analysis on more tractable and performant alternatives.

### B.7.4 Flow-based Generative Models

Flow-based generative models provide exact data likelihood evaluation and support efficient sampling by learning a sequence of invertible (bijective) transformations between the data space and a simple latent space. The central concept is to represent a complex data distribution by mapping observed data $x$ to latent variable $z$ via an invertible function $f$, so that $x = f^{-1}(z)$. The density of data points can then be computed exactly using the change of variables formula

$$\log p(x) = \log p(z) + \log |J_{f^{-1}}(z)|,$$

where $J_{f^{-1}}$ is the Jacobian matrix of the inverse mapping. Typical flow-based models include NICE (Dinh et al., 2014), RealNVP (Dinh et al., 2016), Glow (Kingma & Dhariwal, 2018), Masked Autoregressive Flows (MAF) (Germain et al., 2015), PixelRNN (Van Den Oord et al., 2016) and so on. NICE and RealNVP use affine coupling layers and can be flexibly applied to continuous vector data of arbitrary dimensionality, while Glow, i-ResNet, PixelRNN, and some hierarchical models are specifically designed for image generation and utilize architectural components that exploit spatial locality and dependencies within images. These designs are less suitable for plain vectors where such spatial structure is absent, making direct extension to general vector-valued data challenging and sometimes inefficient.

In this work, we select RealNVP because of its modular structure and its generalizability to vector-based data. For all experiments, the RealNVP is trained for a fixed number of epochs with a consistent learning rate. For optimization, we use Adam (Kingma & Ba, 2014) and follow the schedule therein. To enhance the robustness of the algorithm and prevent overflow of values, we apply standardization in the training and generating process. Also, we apply the early stopping mechanism which monitors the generator's loss, halting training if there is no improvement for patience epochs to prevent overfitting. The run time of RealNVP is about double of Gau-Diff, while the metric results are quite similar to the compared methods.

### B.7.5 Tabular Data Generative Model Baselines

To the best of our knowledge, existing generative models designed for hypergraphs do not handle our data-generation tasks, whereas tabular data generative models can be applied. Therefore, in addition to including well-established models such as GAN, VAE, diffusion models, and RealNVP, we have also incorporated several recent and high-performing tabular data generative models in our comparisons.

As summarized in Table 1 of (Stoian et al., 2025), most state-of-the-art tabular data synthesis models were previously tested on datasets with fewer than 100 features, and their training and sampling processes are generally time- and memory-intensive. Nevertheless, many of these models can handle mixed data types, which aligns well with the nature of our datasets. For our baseline comparisons, we include ForestDiffusion (Jolicoeur-Martineau et al., 2024), TabPFGen (Ma et al., 2024), CTAB-GAN (Zhao et al., 2021), CTAB-GAN+ (Zhao et al., 2024), and CTGAN (Xu et al., 2019), all using their official implementations and default parameters.

To assess the computational resources required, we first trained each method on the recipe hypergraph, which contains 205 features. TabPFGen and CTGAN require approximately twice the runtime of ReLaSH, while CTAB-GAN requires about four times as long. CTAB-GAN+ and ForestDiffusion are the most computationally demanding, requiring several hours for training and generation. The error metrics for these experiments are shown in Table 3.

Among the models, TabPFGen has good performance and is comparable to ReLaSH/ReLaSH, while the other tabular generative models perform similarly to the standard WGAN baseline. However, according to the official implementation, TabPFGen is limited to handling tabular datasets with at most 500 features, whereas both our patient profile and co-citationship datasets exceed 2,000 dimensions—demonstrating the advantages of ReLaSH in terms of scalability. All the other reviewed baseline models fail to complete training on these larger datasets due to timeout or out-of-memory errors, highlighting the efficiency of ReLaSH for hypergraph data generation tasks.

## C  PROOFS OF MAIN RESULTS

*Proof of Theorem 1.* Suppose two sets of model parameters $(\mathbb{P}_U, Z, \alpha, \gamma, B)$, and $(\mathbb{P}_{U'}, Z', \alpha', \gamma', B')$ yield the same distribution for hyperlinks and attributes, i.e.,

$$\alpha + ZU^{(23)} \overset{d}{=} \alpha' + Z'U^{(23)'} \text{ and } \gamma + BU^{(12)} \overset{d}{=} \gamma' + B'U^{(12)'}.$$

First, by taking the mean on both sides, we obtain $\alpha = \alpha'$ and $\gamma = \gamma'$, thus we have

$$ZU^{(23)} \overset{d}{=} Z'U^{(23)'} \text{ and } BU^{(12)} \overset{d}{=} B'U^{(12)'}.$$

Next, we demonstrate that $\mathbb{P}_U$ and $Z, B$ are identifiable up to sign permutations. Suppose there exists an invertible matrix $A$ such that $U^{(23)'} \overset{d}{=} A^\top U^{(23)}$, and define $\mathbb{E}_{\mathbb{P}_U}[U^{(23)}U^{(23)\top}] = \frac{1}{n}Z^\top Z = D$ and $\mathbb{E}_{\mathbb{P}_{U'}}[U^{(23)'}U^{(23)'\top}] = \frac{1}{n}Z'^\top Z' = D'$, both of which are diagonal matrices with distinct positive diagonal elements. Thus, we have:

$$A^{-1}DA^{-\top} = \frac{1}{n}A^{-1}Z^\top ZA^{-\top} = \frac{1}{n}Z'^\top Z' = E_{\mathbb{P}_{U'}}[U^{(23)'}U^{(23)'\top}]$$
$$= A^\top \mathbb{E}_{\mathbb{P}_U}[U^{(23)}U^{(23)\top}]A = A^\top DA,$$

which implies that

$$D = AA^\top DAA^\top \triangleq (MD^{\frac{1}{2}})(MD^{\frac{1}{2}})^\top,$$

where $M = AA^\top$. By the Cholesky decomposition, we conclude that $MD^{\frac{1}{2}} = D^{\frac{1}{2}}W$ for some orthogonal matrix $W$, meaning that $W = D^{-\frac{1}{2}}MD^{\frac{1}{2}}$, and $I = W^\top W = D^{-\frac{1}{2}}M^\top MD^{\frac{1}{2}}$, i.e. $M^2 = I$, and $M$ is positive semi-definite. Therefore, $M = AA^\top = I$, $A$ must be an orthogonal matrix. Furthermore, since $A^\top DA = D'$, where both $D$ and $D'$ are positive diagonal matrices with distinct elements, we deduce that $A$ must also be a sign permutation matrix. Therefore, we conclude that $U^{(23)'} \overset{d}{=} A^\top U^{(23)}$ and $Z' = ZA^{-\top}$, i.e., $\mathbb{P}_U$ and $Z$ are identifiable up to sign permutations. To resolve the identifiability issue up to column sign flips during estimation, we fix the sign of all coordinates in the first row of $U_m$.

Next, suppose there exists an invertible matrix $F$ such that $U^{(12)'} \overset{d}{=} F^\top U^{(12)}$ and $B' = BF^{-\top}$. Let $F = \begin{pmatrix} F_{11} & F_{12} \\ F_{21} & F_{22} \end{pmatrix}$, so that:

$$U^{(1)'} \overset{d}{=} F_{11}^\top U^{(1)} + F_{21}^\top U^{(2)}, \text{ and } U^{(2)'} \overset{d}{=} F_{12}^\top U^{(1)} + F_{22}^\top U^{(2)}.$$

Since we have already shown that $U^{(2)} \overset{d}{=} U^{(2)'}$, it follows that

$$0 = \mathbb{E}_{\mathbb{P}_{U'}}[U^{(1)'}U^{(2)'\top}] = F_{11}^\top \mathbb{E}_{\mathbb{P}_U}[U^{(1)}U^{(2)\top}] + F_{21}^\top \mathbb{E}_{\mathbb{P}_U}[U^{(2)}U^{(2)\top}] = F_{21}^\top \mathbb{E}_{\mathbb{P}_U}[U^{(2)}U^{(2)\top}],$$

$$\mathbb{E}_{\mathbb{P}_{U'}}[U^{(2)'}U^{(2)'\top}] = F_{12}^\top \mathbb{E}_{\mathbb{P}_U}[U^{(1)}U^{(2)\top}] + F_{22}^\top \mathbb{E}_{\mathbb{P}_U}[U^{(2)}U^{(2)\top}] = F_{22}^\top \mathbb{E}_{\mathbb{P}_U}[U^{(2)}U^{(2)\top}],$$

thus, $F_{21} = 0_{k_2 \times k_1}$, $F_{22} = I_{k_2}$, and $F_{12} = 0_{k_1 \times k_2}$.

For the loading matrices $B$ and $B'$, we have:

$$\begin{pmatrix} B_{11} & B_{21} \\ B_{12} & B_{22} \end{pmatrix} = B = B'F^\top = \begin{pmatrix} B'_{11} & B'_{21} \\ B'_{12} & B'_{22} \end{pmatrix} \begin{pmatrix} F_{11}^\top & 0 \\ 0 & I_{k_2} \end{pmatrix} = \begin{pmatrix} B'_{11}F_{11}^\top & B'_{21} \\ B'_{12}F_{11}^\top & B'_{22} \end{pmatrix},$$

which implies that $B'_{21} = B_{21}$ and $B'_{22} = B_{22}$.

Under condition (C3), we assume that $B'_1 = \begin{pmatrix} B'_{11} \\ B'_{12} \end{pmatrix}$ and $B_1 = \begin{pmatrix} B_{11} \\ B_{12} \end{pmatrix}$ both contain unit lower triangular matrices, i.e., $B_{11}$ and $B'_{11}$ are both unit lower triangular matrices. We conclude that $F_{11}$ is a unit upper triangular matrix because $B_{11} = B'_{11}F_{11}^\top$. By checking each of the diagonal elements of $B_{11}$ and $B'_{11}F_{11}^\top$, we can conclude that all the off-diagonal elements of $F_{11}$ should be 0, i.e. $F_{11} = I_{k_1}$. This leads us to conclude that $U^{(3)}$ and $B$ are both identifiable, and the strict identifiability holds for all parameters.

On the other hand, if the identifiability condition is (C3) $\frac{1}{p}B^\top B = I_k$, it follows that $F_{11} = I_3$. It's also easy to show that with (C3**), $F_{11}$ will be a sign permutation matrix, and the identifiability can be resolved by arranging the sign of the first column in each matrix. In practice, we use (C3) or (C3**) instead of (C3*) because it offers a more natural estimation procedure.

$\square$

**Remark 5.** *In the case where the true parameters $(U_m, Z, B, \alpha, \gamma)$ do not satisfy the conditions in Theorem 1, we can apply the following transformation w.r.t. the observations.*

*Firstly, compute $\Theta^H$ and $\Theta^X$ as defined respectively. Define $\alpha^* = \alpha + \frac{1}{m}ZU_m^{(23)\top}\mathbf{1}_m$ and $\gamma^* = \gamma + \frac{1}{m}BU_m^{(12)\top}\mathbf{1}_m$, and let $U' = J_m U_m$.*

*Secondly, we transform $(U_m, Z, B)$ such that $\Theta^X$ and $\Theta^H$ remain unchanged and the orthogonality condition $U_m^{(2)} \perp U_m^{(1)}, U_m^{(3)}$ is satisfied by orthogonal transformation. Specifically, let*

$$U^{(1)''} = (I_m - U^{(2)'}(U^{(2)'\top}U^{(2)'})^{-1}U^{(2)'\top})U^{(1)'},$$

$$U^{(3)''} = (I_m - U^{(2)'}(U^{(2)'\top}U^{(2)'})^{-1}U^{(2)'\top})U^{(3)'}.$$

*To preserve the matrix product, we update $B_2'^\top = B_2^\top + (U^{(2)'\top}U^{(2)'})^{-1}U^{(2)'\top}U^{(1)'}B_1^\top$ and $Z_2'^\top = Z_2^\top + (U^{(2)'\top}U^{(2)'})^{-1}U^{(2)'\top}U^{(3)'}Z_3^\top$. Then combine $U^{(23)''} = \begin{bmatrix} U^{(2)'} & U^{(3)''} \end{bmatrix}$.*

*At this stage, we need to rotate $(U_m, Z, B)$ to satisfy the conditions (C2) and (C3). Let $\mathcal{V} = diag(\sigma_1^2, \sigma_2^2, \ldots, \sigma_{k_2+k_3}^2)$, where $\sigma_1^2 > \sigma_2^2 > \cdots > \sigma_{k_2+k_3}^2$ are the eigenvalues of*

$$\frac{1}{mn}(Z'^\top Z')^{1/2}(U^{(23)''\top}U^{(23)''})(Z'^\top Z')^{1/2},$$

*and let $\Gamma$ be the $(k_2 + k_3) \times (k_2 + k_3)$ matrix whose columns are the corresponding eigenvectors. Define $U^{(23)*} = U^{(23)''}G$ and $Z^* = Z'G^{-\top}$, where $G = (Z'^\top Z'/n)^{1/2}\Gamma\mathcal{V}^{-1/4}$.*

*Similarly, let $\mathcal{W} = diag(\delta_1^2, \delta_2^2, \ldots, \delta_{k_1}^2)$, where $\delta_1^2 > \delta_2^2 > \cdots > \delta_{k_1}^2$ are the eigenvalues of*

$$\frac{1}{mp}(B_1^\top B_1)^{1/2}(U^{(1)''\top}U^{(1)''})(B_1^\top B_1)^{1/2},$$

*and let $D$ be the $k_1 \times k_1$ matrix whose columns are the corresponding eigenvectors. Define $U^{(1)*} = U^{(1)''}K$ and $B_1^* = B_1 K^{-\top}$, where $K = (B_1^\top B_1/p)^{1/2}D\mathcal{W}^{-1/4}$. Finally, we take $B^* = [B_1^* \quad B_2']$, and $U^* = \begin{bmatrix} U^{(1)*} & U^{(23)*} \end{bmatrix}$.*

*Thus, we obtain the transformed parameters $(U^*, B^*, Z^*, \alpha^*, \gamma^*)$, and the transformed parameters of distributions $(\Theta^{X*}, \Theta^{H*}) = (\Theta^X, \Theta^H)$, meaning that the distribution of hyperlinks and attributes remains unchanged after the transformation.*

**Remark 6.** *In general, if the true $\mathbb{P}_U$ and $Z, B, \alpha, \gamma$ do not satisfy these constraints, a transformation of $\mathbb{P}_U, Z, B, \alpha, \gamma$ can also be made so that the constraints are met, and the distribution of hyperlink and attributes remains unchanged.*

*Firstly, define $\bar{\alpha} = \alpha + Z\mathbb{E}_{\mathbb{P}_U}[U^{(23)}]$ and $\bar{\gamma} = \gamma + B\mathbb{E}_{\mathbb{P}_U}[U^{(12)}]$, and let $U' = U - \mathbb{E}_{\mathbb{P}_U}[U]$.*

Secondly, we transform $(\mathbb{P}_U, Z, B)$ such that the distribution for hyperlinks and attributes remain unchanged and the orthogonality conditions $\mathbb{E}_{\mathbb{P}_U}[U^{(2)}U^{(1)\top}] = 0_{k_2 \times k_1}$, $\mathbb{E}_{\mathbb{P}_U}[U^{(2)}U^{(3)\top}] = 0_{k_2 \times k_3}$ are satisfied by orthogonal transformation. Specifically, let

$$U^{(1)\prime\prime} = U^{(1)\prime} - \mathbb{E}_{\mathbb{P}_U}[U^{(1)\prime}U^{(2)\prime\top}](\mathbb{E}_{\mathbb{P}_U}[U^{(2)\prime}U^{(2)\prime\top}])^{-1}U^{(2)\prime},$$
$$U^{(3)\prime\prime} = U^{(3)\prime} - \mathbb{E}_{\mathbb{P}_U}[U^{(3)\prime}U^{(2)\prime\top}](\mathbb{E}_{\mathbb{P}_U}[U^{(2)\prime}U^{(2)\prime\top}])^{-1}U^{(2)\prime}.$$

To preserve the matrix product, we update $B_2'^{\top} = B_2^{\top} + (\mathbb{E}_{\mathbb{P}_U}[U^{(2)\prime}U^{(2)\prime\top}])^{-1}\mathbb{E}_{\mathbb{P}_U}[U^{(2)\prime}U^{(1)\prime\top}]B_1^{\top}$ and $Z_2'^{\top} = Z_2^{\top} + (\mathbb{E}_{\mathbb{P}_U}[U^{(2)\prime}U^{(2)\prime\top}])^{-1}\mathbb{E}_{\mathbb{P}_U}[U^{(2)\prime}U^{(3)\prime\top}]Z_3^{\top}$.

At this stage, we need to rotate $(\mathbb{P}_U, Z, B)$ to satisfy the conditions (C2) and (C3). Let $\mathcal{V} = \text{diag}(\sigma_1^2, \sigma_2^2, \ldots, \sigma_{k_2+k_3}^2)$, where $\sigma_1^2 > \sigma_2^2 > \cdots > \sigma_{k_2+k_3}^2$ are the eigenvalues of

$$\frac{1}{n}(Z'^{\top}Z')^{1/2}(\mathbb{E}_{\mathbb{P}_U}[U^{(23)\prime\prime}U^{(23)\prime\prime\top}])(Z'^{\top}Z')^{1/2},$$

and let $\Gamma$ be the $(k_2 + k_3) \times (k_2 + k_3)$ matrix whose columns are the corresponding eigenvectors. Define $\bar{U}^{(23)} = G^{\top}U^{(23)\prime\prime}$ and $\bar{Z} = Z'G^{-\top}$, where $G = (Z'^{\top}Z'/n)^{1/2}\Gamma\mathcal{V}^{-1/4}$.

Similarly, let $\mathcal{W} = \text{diag}(\delta_1^2, \delta_2^2, \ldots, \delta_{k_1}^2)$, where $\delta_1^2 > \delta_2^2 > \cdots > \delta_{k_1}^2$ are the eigenvalues of

$$\frac{1}{p}(B_1^{\top}B_1)^{1/2}(\mathbb{E}_{\mathbb{P}_U}[U^{(1)\prime\prime}U^{(1)\prime\prime\top}])(B_1^{\top}B_1)^{1/2},$$

and let $D$ be the $k_1 \times k_1$ matrix whose columns are the corresponding eigenvectors. Define $\bar{U}^{(1)} = K^{\top}U^{(1)\prime\prime}$ and $\bar{B}_1 = B_1 K^{-\top}$, where $K = (B_1^{\top}B_1/p)^{1/2}D\mathcal{W}^{-1/4}$.

Thus, we obtain the transformed model parameters $(\mathbb{P}_{\bar{U}}, \bar{B}, \bar{Z}, \bar{\alpha}, \bar{\gamma})$, keeping the distribution of hyperlinks and attributes unchanged after the transformation.

*Proof of Lemma 1.* In the following, we slightly abuse the notation by letting $\mu(x)$ denote the counting measure when $x$ is discrete, the Lebesgue measure when $x$ is continuous, and the product of the counting measure and the Lebesgue measure when $x$ involves both discrete and continuous components.

We can see that

$$d_{\text{KL}}(\mathbb{P}_{(E,X,U)}\|\mathbb{P}_{(\tilde{E},\tilde{X},\tilde{U})}) = \mathbb{E}_{\mathbb{P}_{(E,X,U)}}\left[\log\left\{\frac{\frac{d\mathbb{P}_{(E,X,U)}(E,X,U)}{d\mu(E,X,U)}(E,X,U)}{\frac{d\mathbb{P}_{(\tilde{E},\tilde{X},\tilde{U})}(E,X,U)}{d\mu(\tilde{E},\tilde{X},\tilde{U})}(E,X,U)}\right\}\right]$$

$$= \mathbb{E}_{\mathbb{P}_{(E,X,U)}}\left[\log\left\{\frac{\frac{d\mathbb{P}_{\mathcal{H}([n],\{E\},\{X\})|U,\mathcal{Z}_n,\alpha,B,\gamma}}{d\mu(\mathcal{H}([n],\{E\},\{X\}))}(E,X,U)\frac{d\mathbb{P}_U(U)}{d\mu(U)}(U)}{\frac{d\mathbb{P}_{\mathcal{H}([n],\{\tilde{E}\},\{\tilde{X}\})|\tilde{U},\mathcal{Z}_n,\alpha,B,\gamma}}{d\mu(\mathcal{H}([n],\{\tilde{E}\},\{\tilde{X}\}))}(E,X,U)\frac{d\mathbb{P}_{\tilde{U}}(U)}{d\mu(\tilde{U})}(U)}\right\}\right]$$

$$= \mathbb{E}_{\mathbb{P}_{(E,X,U)}}\left[\log\left\{\frac{\frac{d\mathbb{P}_U(U)}{d\mu(U)}(U)}{\frac{d\mathbb{P}_{\tilde{U}}(U)}{d\mu(\tilde{U})}(U)}\right\}\right] = d_{\text{KL}}(\mathbb{P}_U\|\mathbb{P}_{\tilde{U}}).$$

$\square$

*Proof of Theorem 2.* We're going to show that

$$d_{\text{KL}}(\mathbb{P}_{(E,X,U)}\,\|\,\mathbb{P}_{(\tilde{E},\tilde{X},\tilde{U})}) = \Delta_{(\mathcal{Z}_n,B,\alpha,\gamma)\text{-estimation}} + \Delta_{\mathbb{P}_U\text{-estimation}} + \Delta_{\text{latent-reconstruction}},$$

where the three components are given by:

$$\Delta_{(\mathcal{Z}_n,B,\alpha,\gamma)\text{-estimation}} = \mathbb{E}_{\mathbb{P}_{(E,X,U)}}\left[\log\left\{\frac{\frac{d\mathbb{P}_{\mathcal{H}([n],\{E\},\{X\})|U,\mathcal{Z}_n,B,\alpha,\gamma}}{d\mu(\mathcal{H}([n],\{E\},\{X\}))}(E,X,U)}{\frac{d\mathbb{P}_{\mathcal{H}([n],\{\tilde{E}\},\{\tilde{X}\})|\tilde{U},\hat{\mathcal{Z}}_n,\hat{B},\hat{\alpha},\hat{\gamma}}}{d\mu(\mathcal{H}([n],\{\tilde{E}\},\{\tilde{X}\}))}(E,X,U)}\right\}\right],$$

$$\Delta_{\mathbb{P}_U\text{-estimation}} = \mathbb{E}_{\mathbb{P}_U}\left[\log\left\{\frac{\frac{d\mathbb{P}_U}{d\mu(U)}(U)}{\frac{d\mathbb{P}_{\hat{\mathcal{U}}_m}}{d\mu(\hat{U})}(U)}\right\}\right] + \mathbb{E}_{\mathbb{P}_U}\left[\log\left\{\frac{\frac{d\mathbb{P}_{\hat{\mathcal{U}}_m}}{d\mu(\hat{U})}(U)}{\frac{d\mathbb{P}_{\tilde{U}}'}{d\mu(\tilde{U})}(U)}\right\}\right] - \mathbb{E}_{\mathbb{P}_{\hat{\mathcal{U}}_m}}\left[\log\left\{\frac{\frac{d\mathbb{P}_{\hat{\mathcal{U}}_m}}{d\mu(\hat{U})}(U)}{\frac{d\mathbb{P}_{\tilde{U}}'}{d\mu(\tilde{U})}(U)}\right\}\right],$$

$$\Delta_{\text{latent-reconstruction}} = \mathbb{E}_{\mathbb{P}_{\hat{\mathcal{U}}_m}}\left[\log\left\{\frac{\frac{d\mathbb{P}_{\hat{\mathcal{U}}_m}}{d\mu(\hat{U})}(U)}{\frac{d\mathbb{P}_{\tilde{U}}'}{d\mu(\tilde{U})}(U)}\right\}\right].$$

Here, $\mathbb{P}_{\hat{\mathcal{U}}_m}$ denotes the marginal distribution of the estimated attribute embeddings $\{\hat{u}_1, \cdots, \hat{u}_m\}$ given the observed $\mathcal{U}_m$, and we assume absolute continuity of all log-ratios with respect to a common base measure $\mu$.

Note that

$$
\begin{aligned}
&\mathrm{d}_{\mathrm{KL}}(\mathbb{P}_{(E,X,U)} \| \mathbb{P}'_{(\tilde{E},\tilde{X},\tilde{U})}) \\
&= \mathbb{E}_{\mathbb{P}_{(E,X,U)}} \left[ \log \left\{ \frac{\frac{d\mathbb{P}_{(E,X,U)}}{d\mu(E,X,U)}(E,X,U)}{\frac{d\mathbb{P}'_{(\tilde{E},\tilde{X},\tilde{U})}}{d\mu(\tilde{E},\tilde{X},\tilde{U})}(E,X,U)} \right\} \right] \\
&= \mathbb{E}_{\mathbb{P}_{(E,X,U)}} \left[ \log \left\{ \frac{\frac{d\mathbb{P}_{\mathcal{H}([n],\{E\},\{X\})|U,\mathcal{Z}_n,B,\alpha,\gamma}}{d\mu(\mathcal{H}([n],\{E\},\{X\}))}(E,X,U) \frac{d\mathbb{P}_U}{d\mu(U)}(U)}{\frac{d\mathbb{P}_{\mathcal{H}([n],\{\tilde{E}\},\{\tilde{X}\})|\tilde{U},\hat{\mathcal{Z}}_n,\hat{B},\hat{\alpha},\hat{\gamma}}}{d\mu(\mathcal{H}([n],\{\tilde{E}\},\{\tilde{X}\}))}(E,X,U) \frac{d\mathbb{P}'_{\tilde{U}}}{d\mu(\tilde{U})}(U)} \right\} \right] \\
&= \mathbb{E}_{\mathbb{P}_{(E,X,U)}} \left[ \log \left\{ \frac{\frac{d\mathbb{P}_{\mathcal{H}([n],\{E\},\{X\})|U,\mathcal{Z}_n,B,\alpha,\gamma}}{d\mu(\mathcal{H}([n],\{E\},\{X\}))}(E,X,U)}{\frac{d\mathbb{P}_{\mathcal{H}([n],\{\tilde{E}\},\{\tilde{X}\})|\tilde{U},\hat{\mathcal{Z}}_n,\hat{B},\hat{\alpha},\hat{\gamma}}}{d\mu(\mathcal{H}([n],\{\tilde{E}\},\{\tilde{X}\}))}(E,X,U)} \right\} \right] \\
&\quad + \mathbb{E}_{\mathbb{P}_{(E,X,U)}} \left[ \log \left\{ \frac{\frac{d\mathbb{P}_U}{d\mu(U)}(U)}{\frac{d\mathbb{P}'_{\tilde{U}}}{d\mu(\tilde{U})}(U)} \right\} \right].
\end{aligned}
$$

Here, we denote

$$
\Delta_{(\mathcal{Z}_n,B,\alpha,\gamma)\text{-estimation}} = \mathbb{E}_{\mathbb{P}_{(E,X,U)}} \left[ \log \left\{ \frac{\frac{d\mathbb{P}_{\mathcal{H}([n],\{E\},\{X\})|U,\mathcal{Z}_n,B,\alpha,\gamma}}{d\mu(\mathcal{H}([n],\{E\},\{X\}))}(E,X,U)}{\frac{d\mathbb{P}_{\mathcal{H}([n],\{\tilde{E}\},\{\tilde{X}\})|\tilde{U},\hat{\mathcal{Z}}_n,\hat{B},\hat{\alpha},\hat{\gamma}}}{d\mu(\mathcal{H}([n],\{\tilde{E}\},\{\tilde{X}\}))}(E,X,U)} \right\} \right]
$$

since $\frac{d\mathbb{P}_{\mathcal{H}([n],\{E\},\{X\})|U,\mathcal{Z}_n,B,\alpha,\gamma}}{d\mu(\mathcal{H}([n],\{E\},\{X\}))}(E,X,U)$ and $\frac{d\mathbb{P}_{\mathcal{H}([n],\{\tilde{E}\},\{\tilde{X}\})|\tilde{U},\hat{\mathcal{Z}}_n,\hat{B},\hat{\alpha},\hat{\gamma}}}{d\mu(\mathcal{H}([n],\{\tilde{E}\},\{\tilde{X}\}))}(E,X,U)$ only differs in the estimated version and the true $(\mathcal{Z}_n,B,\alpha,\gamma)$ with the same $(E,X,U)$ plug-in.

To decompose the second term, we consider

$$
\begin{aligned}
\mathbb{E}_{\mathbb{P}_{(E,X,U)}} \left[ \log \left\{ \frac{\frac{d\mathbb{P}_U}{d\mu(U)}(U)}{\frac{d\mathbb{P}'_{\tilde{U}}}{d\mu(\tilde{U})}(U)} \right\} \right] &= \mathbb{E}_{\mathbb{P}_{(E,X,U)}} \left[ \log \left\{ \frac{\frac{d\mathbb{P}_U}{d\mu(U)}(U)}{\frac{d\mathbb{P}_{\hat{\mathcal{U}}_m}}{d\mu(\hat{U})}(U)} \right\} \right] + \mathbb{E}_{\mathbb{P}_{(E,X,U)}} \left[ \log \left\{ \frac{\frac{d\mathbb{P}_{\hat{\mathcal{U}}_m}}{d\mu(\hat{U})}(U)}{\frac{d\mathbb{P}'_{\tilde{U}}}{d\mu(\tilde{U})}(U)} \right\} \right] \\
&= \mathbb{E}_{\mathbb{P}_U} \left[ \log \left\{ \frac{\frac{d\mathbb{P}_U}{d\mu(U)}(U)}{\frac{d\mathbb{P}_{\hat{\mathcal{U}}_m}}{d\mu(\hat{U})}(U)} \right\} \right] + \mathbb{E}_{\mathbb{P}_U} \left[ \log \left\{ \frac{\frac{d\mathbb{P}_{\hat{\mathcal{U}}_m}}{d\mu(\hat{U})}(U)}{\frac{d\mathbb{P}'_{\tilde{U}}}{d\mu(\tilde{U})}(U)} \right\} \right] \\
&\quad - \mathbb{E}_{\mathbb{P}_{\hat{\mathcal{U}}_m}} \left[ \log \left\{ \frac{\frac{d\mathbb{P}_{\hat{\mathcal{U}}_m}}{d\mu(\hat{U})}(U)}{\frac{d\mathbb{P}'_{\tilde{U}}}{d\mu(\tilde{U})}(U)} \right\} \right] + \mathbb{E}_{\mathbb{P}_{\hat{\mathcal{U}}_m}} \left[ \log \left\{ \frac{\frac{d\mathbb{P}_{\hat{\mathcal{U}}_m}}{d\mu(\hat{U})}(U)}{\frac{d\mathbb{P}'_{\tilde{U}}}{d\mu(\tilde{U})}(U)} \right\} \right].
\end{aligned}
$$

The first three terms primarily depend on the distance between $\mathbb{P}_U$ and $\mathbb{P}_{\hat{\mathcal{U}}_m}$, and are collectively denoted as $\Delta_{\mathbb{P}_U\text{-estimation}}$. The final term corresponds to the diffusion error on observations of $\mathbb{P}_{\hat{\mathcal{U}}_m}$, and is denoted as $\Delta_{\text{latent-reconstruction}}$.

$\square$

*Proof of Theorem 4.* Denote $H = \{h_{ji}\}_{j \in [n], i \in [m]}$ and $X = \{x_{ji}\}_{j \in [n], i \in [m]}$ as the hypergraph connection matrix and the attribute matrix, respectively, where $h_{ji} = \mathbf{1}_{\{i \in e_j\}}$ and $x_{ji}$ represents the $i$-th attribute associated with hyperlink $e_j$. We consider minimizing the objective loss function

$L(U, Z, B, \alpha, \gamma)$, and the optimal solution $(\hat{U}, \hat{Z}, \hat{B}, \hat{\alpha}, \hat{\gamma})$ implies that

$$
\begin{aligned}
0 &\le L(U^*, Z^*, B^*, \alpha^*, \gamma^*) - L(\hat{U}, \hat{Z}, \hat{B}, \hat{\alpha}, \hat{\gamma}) \\
&= \sum_{j=1}^{m} \sum_{i=1}^{n} h_{ji}(\hat{\Theta}_{ji}^{H} - \Theta_{ji}^{H*}) - (f_H(\hat{\Theta}_{ji}^{H}) - f_H(\Theta_{ji}^{H*})) \\
&\quad + \lambda \left[ \sum_{j=1}^{m} \sum_{i=1}^{n} x_{ji}(\hat{\Theta}_{ji}^{X} - \Theta_{ji}^{X*}) - (f_X(\hat{\Theta}_{ji}^{X}) - f_X(\Theta_{ji}^{X*})) \right] \\
&= \sum_{j=1}^{m} \sum_{i=1}^{n} (\hat{\Theta}_{ji}^{H} - \Theta_{ji}^{H*})(h_{ji} - f_H'(\Theta_{ji}^{H*})) - \frac{1}{2} f_H''(\tilde{\Theta}_{ji}^{H})(\Theta_{ji}^{H*} - \hat{\Theta}_{ji}^{H})^2 \\
&\quad + \lambda \left[ \sum_{j=1}^{m} \sum_{i=1}^{n} (\hat{\Theta}_{ji}^{X} - \Theta_{ji}^{X*})(x_{ji} - f_X'(\Theta_{ji}^{X*})) - \frac{1}{2} f_X''(\tilde{\Theta}_{ji}^{X})(\Theta_{ji}^{X*} - \hat{\Theta}_{ji}^{X})^2 \right],
\end{aligned}
$$

where $\tilde{\Theta}_{ji}^{X} = a_{ji}\hat{\Theta}_{ji}^{X} + (1 - a_{ji})\Theta_{ji}^{X*}$, and $\tilde{\Theta}_{ji}^{H} = a_{ji}\hat{\Theta}_{ji}^{H} + (1 - a_{ji})\Theta_{ji}^{H*}$ for some $a_{ji}, b_{ji} \in [0, 1]$. Here, we have $f_H(x) = \log(1 + \exp(x))$, and $f_X(x) = \frac{1}{2}x^2$. Denote $E_H = H - f_H'(\Theta^{H*})$, $E_X = X - f_X'(\Theta^{X*})$. Since we require that both true parameters and estimated ones should belong to the feasible parameter space $\mathcal{F}(\Theta)$, we have the above equation transferred into:

$$
\begin{aligned}
&\frac{1}{2} \min \left( \min_{i,j,\tilde{\Theta}^H \in \mathcal{F}(\Theta)} f_H''(\tilde{\Theta}_{ji}^{H}), \lambda \right) \|\hat{\Theta} - \Theta^*\|_F^2 \\
&\le \frac{1}{2} \min_{i,j,\tilde{\Theta}^H \in \mathcal{F}(\Theta)} f_H''(\tilde{\Theta}_{ji}^{H}) \|\hat{\Theta}^H - \Theta^{H*}\|_F^2 + \frac{\lambda}{2} \|\hat{\Theta}^X - \Theta^{X*}\|_F^2 \\
&\le \; <E_H, \hat{\Theta}^H - \Theta^{H*}> + \lambda <E_X, \hat{\Theta}^X - \Theta^{X*}> \\
&\le \|E_H\|_2 \mathrm{rank}(\hat{\Theta}^H - \Theta^{H*}) \|\hat{\Theta}^H - \Theta^{H*}\|_F + \lambda \|E_X\|_2 \mathrm{rank}(\hat{\Theta}^X - \Theta^{X*}) \|\hat{\Theta}^X - \Theta^{X*}\|_F \\
&\le \sqrt{2(k_2 + k_3 + 1)\|E_H\|_2^2 + 2(k_1 + k_2 + 1)\lambda^2 \|E_X\|_2^2} \sqrt{\|\hat{\Theta}^H - \Theta^{H*}\|_F^2 + \|\hat{\Theta}^X - \Theta^{X*}\|_F^2} \\
&= \sqrt{2(k_2 + k_3 + 1)\|E_H\|_2^2 + 2(k_1 + k_2 + 1)\lambda^2 \|E_X\|_2^2} \|\hat{\Theta} - \Theta^*\|_F.
\end{aligned}
$$

Note that $\min_{i,j,\tilde{\Theta}^H \in \mathcal{F}(\Theta)} f_H''(\tilde{\Theta}_{ji}^{H}) \gtrsim \exp(-C_{m,n})$, we should consider the bound of $\|E_H\|_2, \|E_X\|_2$.

For $\|E_H\|_2$, we prove that $\mathbb{P}(\|E_H\|_2 \gtrsim \sqrt{(m \vee n) \exp(\bar{\alpha}_{m,n}) \log((m \wedge n)/\varepsilon)}) \le \varepsilon$. Let

$$
\tilde{E}_H = \begin{pmatrix} 0_{m \times m} & E_H \\ E_H^\top & 0_{n \times n} \end{pmatrix},
$$

we have $\lambda_{\max}(\tilde{E}_H^\top \tilde{E}_H) = \lambda_{\max}(E_H^\top E_H)$, so that $\|E_H\|_2 = \|\tilde{E}_H\|_2$. We introduce Lemma 4 to bound it.

**Lemma 4** (Theorem 5 from (Chung & Radcliffe, 2011)). *Let $X_1, X_2, \cdots, X_m$ be independent random $n \times n$ Hermitian matrices. Moreover, assume that $\|X_i - \mathbb{E}(X_i)\|_2 \le M$ for all $i$, and put $\nu^2 = \|\sum Var(X_i)\|_2$. Let $X = \sum_i X_i$, then for any $a > 0$,*

$$
P(\|X - \mathbb{E}(X)\|_2 > a) \le 2n \exp \left( -\frac{a^2}{2\nu^2 + 2Ma/3} \right).
$$

Let $E^{j,i}$ be the $(m + n) \times (m + n)$ matrix with 1 in the $(j, i)$ and $(i, j)$ positions and 0 elsewhere. Therefore, we represent $\tilde{E}_H = \sum_{j=1}^{m} \sum_{i=1}^{n} Y^{j,m+i} = \sum_{j=1}^{m} \sum_{i=1}^{n} (h_{ji} - \sigma(\Theta_{ji}^{H*})) E^{j,m+i}$. Note that $\mathbb{E}Y^{j,m+i} = 0_{(m+n) \times (m+n)}$, $\|Y^{j,m+i}\|_2 \le 1$, and $\mathbb{E}(Y^{j,m+i})^2 = (\sigma(\Theta_{ji}^{H*}) - \sigma(\Theta_{ji}^{H*})^2)(E^{j,j} + E^{m+i,m+i})$, thus

$$
\begin{aligned}
\nu^2 &= \|\sum_{j=1}^{m} \sum_{i=1}^{n} \mathbb{E}(Y^{j,m+i})^2\|_2 = \max\{\max_{j \in [m]} \sum_{i \in [n]} \sigma(\Theta_{ji}^{H*}) - \sigma(\Theta_{ji}^{H})^2, \max_{i \in [n]} \sum_{j \in [m]} \sigma(\Theta_{ji}^{H*}) - \sigma(\Theta_{ji}^{H})^2\} \\
&\le (m \vee n) \max_{j,i} \sigma(\Theta_{ji}^{H*}).
\end{aligned}
$$

For any $\varepsilon > 0$, we take $a = \sqrt{3(m \vee n)\max_{j,i}\sigma(\Theta_{ji}^{H*})\log(4(m \vee n)/\varepsilon)}$, and note that $\exp(\bar{\alpha}_{m,n}) \gtrsim (m \wedge n)^{-1}\log((m \wedge n))$ implies that there exist large enough $m, n$ s.t. $\exp(\bar{\alpha}_{m,n}) \gtrsim (m \wedge n)^{-1}\log((m \wedge n)/\varepsilon)$, and $\max_{j,i}\sigma(\Theta_{ji}^{H*}) \geq \frac{4}{3}(m \wedge n)^{-1}\log(4(m \wedge n)/\varepsilon)$. Therefore, $(m \vee n)\max_{j,i}\sigma(\Theta_{ji}^{H*}) \geq 2\sqrt{3(m \vee n)\max_{j,i}\sigma(\Theta_{ji}^{H*})\log(4(m \vee n)/\varepsilon)}/3$, and Lemma 4 implies that

$$\mathbb{P}(\|\tilde{E}_H\|_2 \gtrsim \sqrt{3(m \vee n)\max_{j,i}\sigma(\Theta_{ji}^{H*})\log(4(m \vee n)/\varepsilon)})$$

$$\leq 2n\exp(-\frac{3(m \vee n)\max_{j,i}\sigma(\Theta_{ji}^{H*})\log(4(m \vee n)/\varepsilon)}{2(m \vee n)\max_{j,i}\sigma(\Theta_{ji}^{H*}) + 2\sqrt{3(m \vee n)\max_{j,i}\sigma(\Theta_{ji}^{H*})\log(4(m \vee n)/\varepsilon)}/3})$$

$$\leq 4(m \vee n)\exp(-\log(4(m \vee n)/\varepsilon)) = \varepsilon.$$

Moreover, note that $\exp(\bar{\alpha}_{m,n}) \asymp \max_{j,i}\sigma(\Theta_{ji}^{H*})$, there is

$$\mathbb{P}(\|\tilde{E}_H\|_2 \gtrsim \sqrt{(m \vee n)\exp(\bar{\alpha}_{m,n})\log(4(m \vee n)/\varepsilon)}) \leq \varepsilon,$$

then the desired result follows.

For $\|E_X\|_2$, note that the noise term is sub-Gaussian with zero mean; therefore Theorem 4.4.3 from (Vershynin, 2018) implies that

$$\mathbb{P}(\|E_X\|_2 \geq CK(\sqrt{m} + \sqrt{p} + t)) \leq 2\exp(-t^2)$$

for some constant $C$, and we also have $K = \max_{i,j}\|X_{ji}\|_{\psi_2} = \max_{t \in [p]}\|\varepsilon_t\|_{\psi_2}$ as a bounded constant, since the noise term is assumed to be sub-Gaussian, therefore we have $\|E_X\|_2 = O_p(\sqrt{m} + \sqrt{p})$.

In all, with the assumption that $\exp(\bar{\alpha}_{m,n}) \asymp \exp(-C_{m,n})$, we arrive at the result that

$$\|\hat{\Theta} - \Theta^*\|_F = O_p\left(\frac{\sqrt{(m \vee n)\exp(\bar{\alpha}_{m,n})\log(m \vee n) + 4\lambda^2(m \vee p)}}{(\exp(-C_{m,n}) \wedge \lambda)}\right).$$

$\square$

*Proof of Theorem 3.* By taking the hyperlink and attribute $E, X$, and the corresponding embedding $U$ as random parts of the random function, we have:

$$\ell_1(z_i, \alpha_i) := \ell_1(z_i, \alpha_i \mid E, X, U) = \mathbf{1}_{i \in E}(U^{(23)\top}z_i + \alpha_i) - \log(1 + \exp(U^{(23)\top}z_i + \alpha_i)),$$

$$\ell_2(b_i, \gamma_i) := \ell_2(b_i, \gamma_i \mid E, X, U) = x_i(\gamma_i + U^{(12)\top}b_i) - \frac{1}{2}(\gamma_i + U^{(12)\top}b_i)^2,$$

where $b_i$ denotes the $i$-th row of matrix $B$, so as the $\hat{b}_i$, and $x_i$ denotes the $i$-th element of the attribute vector $X$.

Then we expand two error terms to the second order:

$$\ell_1(z_i, \alpha_i) - \ell_1(\hat{z}_i, \hat{\alpha}_i) = \nabla\ell_1(\hat{z}_i, \hat{\alpha}_i)^\top \begin{pmatrix} z_i - \hat{z}_i \\ \alpha_i - \hat{\alpha}_i \end{pmatrix} + \frac{1}{2}\begin{pmatrix} z_i - \hat{z}_i \\ \alpha_i - \hat{\alpha}_i \end{pmatrix}^\top \nabla^2\ell_1(\tilde{z}_i, \tilde{\alpha}_i)\begin{pmatrix} z_i - \hat{z}_i \\ \alpha_i - \hat{\alpha}_i \end{pmatrix},$$

$$\ell_2(b_i, \gamma_i) - \ell_2(\hat{b}_i, \hat{\gamma}_i) = \nabla\ell_2(\hat{b}_i, \hat{\gamma}_i)^\top \begin{pmatrix} b_i - \hat{b}_i \\ \gamma_i - \hat{\gamma}_i \end{pmatrix} + \frac{1}{2}\begin{pmatrix} b_i - \hat{b}_i \\ \gamma_i - \hat{\gamma}_i \end{pmatrix}^\top \nabla^2\ell_2(\tilde{b}_i, \tilde{\gamma}_i)\begin{pmatrix} b_i - \hat{b}_i \\ \gamma_i - \hat{\gamma}_i \end{pmatrix},$$

for some $(\tilde{z}_i, \tilde{\alpha}_i)$ being the linear combination of $(z_i, \alpha_i)$ and $(\hat{z}_i, \hat{\alpha}_i)$. Note that

$$\nabla\ell_1(\hat{z}_i, \hat{\alpha}_i) = \mathbf{1}_{i \in E}\begin{pmatrix} U^{(23)} \\ 1 \end{pmatrix} - \frac{\exp(U^{(23)\top}\hat{z}_i + \hat{\alpha}_i)}{1 + \exp(U^{(23)\top}\hat{z}_i + \hat{\alpha}_i)}\begin{pmatrix} U^{(23)} \\ 1 \end{pmatrix},$$

$$\nabla^2\ell_1(\tilde{z}_i, \tilde{\alpha}_i) = \frac{\exp(U^{(23)\top}\tilde{z}_i + \tilde{\alpha}_i)}{(1 + \exp(U^{(23)\top}\tilde{z}_i + \tilde{\alpha}_i))^2}\begin{pmatrix} U^{(23)} \\ 1 \end{pmatrix}\begin{pmatrix} U^{(23)} \\ 1 \end{pmatrix}^\top$$

for the $\ell_1$ error, and also

$$\nabla \ell_2(\hat{b}_i, \hat{\gamma}_i) = (x_i - (\hat{\gamma}_i + U^{(12)\top}\hat{b}_i)) \begin{pmatrix} U^{(12)} \\ 1 \end{pmatrix},$$

$$\nabla^2 \ell_2(\tilde{b}_i, \tilde{\gamma}_i) = \begin{pmatrix} U^{(12)} \\ 1 \end{pmatrix} \begin{pmatrix} U^{(12)} \\ 1 \end{pmatrix}^\top.$$

To consider the

$$\Delta_{(\mathcal{Z}_n, B, \alpha, \gamma) - \text{estimation}}$$
$$= \mathbb{E}_{\mathbb{P}_{(E,X,U)}} \left[ \log \left\{ \frac{\frac{d\mathbb{P}_{\mathcal{H}([n], \{E\}, \{X\} | U, \mathcal{Z}_n, B, \alpha, \gamma)}}{d\mu(\mathcal{H}([n], \{E\}, \{X\}))}(E, X, U)}{\frac{d\mathbb{P}_{\mathcal{H}([n], \{\tilde{E}\}, \{\tilde{X}\} | \tilde{U}, \hat{\mathcal{Z}}_n, \hat{B}, \hat{\alpha}, \hat{\gamma})}}{d\mu(\mathcal{H}([n], \{\tilde{E}\}, \{\tilde{X}\}))}(E, X, U)} \right\} \right]$$

$$= \mathbb{E}_{\mathbb{P}_{(E,X,U)}} \left[ \log \left\{ \frac{\frac{d\mathbb{P}_{\mathcal{H}([n], \{E\} | U^{(23)}, \mathcal{Z}_n, \alpha)}}{d\mu(\mathcal{H}([n], \{E\}))}(E, U)}{\frac{d\mathbb{P}_{\mathcal{H}([n], \{\tilde{E}\} | \tilde{U}^{(23)}, \hat{\mathcal{Z}}_n, \hat{\alpha})}}{d\mu(\mathcal{H}([n], \{\tilde{E}\}))}(E, U)} \right\} + \log \left\{ \frac{\frac{d\mathbb{P}_{\mathcal{H}([n], \{X\} | U^{(12)}, B, \gamma)}}{d\mu(\mathcal{H}([n], \{X\}))}(X, U)}{\frac{d\mathbb{P}_{\mathcal{H}([n], \{\tilde{X}\} | \tilde{U}^{(12)}, \hat{B}, \hat{\gamma})}}{d\mu(\mathcal{H}([n], \{\tilde{X}\}))}(X, U)} \right\} \right]$$

$$= \mathbb{E}_{\mathbb{P}_{(E,U)}} \left( \sum_{i \in [n]} \ell_1(z_i, \alpha_i) - \ell_1(\hat{z}_i, \hat{\alpha}_i) \right) + \mathbb{E}_{\mathbb{P}_{(X,U)}} \left( \sum_{i \in [p]} \ell_2(b_i, \gamma_i) - \ell_2(\hat{b}_i, \hat{\gamma}_i) \right),$$

we should analyze the expectation over $(E, X, U)$ by first conditioning on $U$, i.e.,

$$\mathbb{E}_{\mathbb{P}_{(E,U)}} |\ell_1(z_i, \alpha_i) - \ell_1(\hat{z}_i, \hat{\alpha}_i)|$$
$$= \mathbb{E}_{\mathbb{P}_U} \mathbb{E}_{\mathbb{P}_{\mathcal{H}([n], \{E\} | U)}} |\ell_1(z_i, \alpha_i) - \ell_1(\hat{z}_i, \hat{\alpha}_i)|$$
$$\le \mathbb{E}_{\mathbb{P}_U} \left[ \left| \frac{\exp(U^{(23)\top} z_i + \alpha_i)}{1 + \exp(U^{(23)\top} z_i + \alpha_i)} - \frac{\exp(U^{(23)\top} \hat{z}_i + \hat{\alpha}_i)}{1 + \exp(U^{(23)\top} \hat{z}_i + \hat{\alpha}_i)} \right| \begin{pmatrix} U^{(23)} \\ 1 \end{pmatrix}^\top \begin{pmatrix} z_i - \hat{z}_i \\ \alpha_i - \hat{\alpha} \end{pmatrix} \right]$$
$$+ \mathbb{E}_{\mathbb{P}_U} \left[ \frac{1}{2} \frac{\exp(U^{(23)\top} \tilde{z}_i + \tilde{\alpha}_i)}{(1 + \exp(U^{(23)\top} \tilde{z}_i + \tilde{\alpha}_i))^2} \begin{pmatrix} z_i - \hat{z}_i \\ \alpha_i - \hat{\alpha} \end{pmatrix}^\top \begin{pmatrix} U^{(23)} \\ 1 \end{pmatrix} \begin{pmatrix} U^{(23)} \\ 1 \end{pmatrix}^\top \begin{pmatrix} z_i - \hat{z}_i \\ \alpha_i - \hat{\alpha} \end{pmatrix} \right]$$
$$\le (C^2 + 1)^{1/2} \left\| \begin{pmatrix} z_i - \hat{z}_i \\ \alpha_i - \hat{\alpha} \end{pmatrix} \right\|_2 \mathbb{E}_{\mathbb{P}_U} \left| \frac{\exp(U^{(23)\top} z_i + \alpha_i)}{1 + \exp(U^{(23)\top} z_i + \alpha_i)} - \frac{\exp(U^{(23)\top} \hat{z}_i + \hat{\alpha}_i)}{1 + \exp(U^{(23)\top} \hat{z}_i + \hat{\alpha}_i)} \right|$$
$$+ C_1 \exp(\bar{\alpha}_{m,n}) \lambda_{\max} \left( \left[ \mathbb{E}_{\mathbb{P}_U} \begin{pmatrix} U^{(23)} \\ 1 \end{pmatrix} \begin{pmatrix} U^{(23)} \\ 1 \end{pmatrix}^\top \right] \right) \left\| \begin{pmatrix} z_i - \hat{z}_i \\ \alpha_i - \hat{\alpha}_i \end{pmatrix} \right\|_2^2,$$

where we can further expand

$$\mathbb{E}_{\mathbb{P}_U} \left[ \left| \frac{\exp(U^{(23)\top} z_i + \alpha_i)}{1 + \exp(U^{(23)\top} z_i + \alpha_i)} - \frac{\exp(U^{(23)\top} \hat{z}_i + \hat{\alpha}_i)}{1 + \exp(U^{(23)\top} z_i + \hat{\alpha}_i)} \right| \right]$$
$$= \frac{\exp(U^{(23)\top} z_i' + \alpha_i')}{(1 + \exp(U^{(23)\top} z_i' + \alpha_i'))^2} \begin{pmatrix} U^{(23)} \\ 1 \end{pmatrix}^\top \begin{pmatrix} z_i - \hat{z}_i \\ \alpha_i - \hat{\alpha}_i \end{pmatrix}$$
$$\le C_1 \exp(\alpha_{m,n}) \mathbb{E}_{\mathbb{P}_U} \left\| \begin{pmatrix} U^{(23)} \\ 1 \end{pmatrix} \right\|_2 \left\| \begin{pmatrix} z_i - \hat{z}_i \\ \alpha_i - \hat{\alpha}_i \end{pmatrix} \right\|_2.$$

Therefore, by combining Assumption 2, we have

$$\mathbb{E}_{\mathbb{P}_{(E,U)}} |\ell_1(z_i, \alpha_i) - \ell_1(\hat{z}_i, \hat{\alpha}_i)| \lesssim \exp(\bar{\alpha}_{m,n}) \left\| \begin{pmatrix} z_i - \hat{z}_i \\ \alpha_i - \hat{\alpha}_i \end{pmatrix} \right\|_2^2.$$

And also, for the attributes part, we have

$$\mathbb{E}_{\mathbb{P}_{(X,U)}}|\ell_2(b_i,\gamma_i) - \ell_2(\hat{b}_i,\hat{\gamma}_i)|$$

$$= \mathbb{E}_{\mathbb{P}_U}\mathbb{E}_{\mathbb{P}_{X|U^{(12)}}}\left| (x_i - (\hat{\gamma}_i + U^{(12)\top}\hat{b}_i))\begin{pmatrix} U^{(12)} \\ 1 \end{pmatrix}^\top \begin{pmatrix} b_i - \hat{b}_i \\ \gamma_i - \hat{\gamma}_i \end{pmatrix} \right.$$

$$\left. + \frac{1}{2}\begin{pmatrix} b_i - \hat{b}_i \\ \gamma_i - \hat{\gamma}_i \end{pmatrix}^\top \begin{pmatrix} U^{(12)} \\ 1 \end{pmatrix}\begin{pmatrix} U^{(12)} \\ 1 \end{pmatrix}^\top \begin{pmatrix} b_i - \hat{b}_i \\ \gamma_i - \hat{\gamma}_i \end{pmatrix} \right|$$

$$\leq \frac{3}{2}\mathbb{E}_{\mathbb{P}_U}\begin{pmatrix} b_i - \hat{b}_i \\ \gamma_i - \hat{\gamma}_i \end{pmatrix}^\top \begin{pmatrix} U^{(12)} \\ 1 \end{pmatrix}\begin{pmatrix} U^{(12)} \\ 1 \end{pmatrix}^\top \begin{pmatrix} b_i - \hat{b}_i \\ \gamma_i - \hat{\gamma}_i \end{pmatrix}$$

$$\leq \frac{3}{2}\lambda_{\max}\left(\mathbb{E}_{\mathbb{P}_U}\left[\begin{pmatrix} U^{(12)} \\ 1 \end{pmatrix}\begin{pmatrix} U^{(12)} \\ 1 \end{pmatrix}^\top\right]\right)\left\|\begin{pmatrix} b_i - \hat{b}_i \\ \gamma_i - \hat{\gamma}_i \end{pmatrix}\right\|_2^2 \lesssim \left\|\begin{pmatrix} b_i - \hat{b}_i \\ \gamma_i - \hat{\gamma}_i \end{pmatrix}\right\|_2^2.$$

By combining the results above, we have

$$\Delta_{(\mathcal{Z}_n,B,\alpha,\gamma)\text{ estimation}}$$

$$= \mathbb{E}_{\mathbb{P}_{(E,U)}}\left(\sum_{i\in[n]}\ell_1(z_i,\alpha_i) - \ell_1(\hat{z}_i,\hat{\alpha}_i)\right) + \mathbb{E}_{\mathbb{P}_{(X,U)}}\left(\sum_{i\in[p]}\ell_2(b_i,\gamma_i) - \ell_2(\hat{b}_i,\hat{\gamma}_i)\right)$$

$$\leq \sum_{i\in[n]}\mathbb{E}_{\mathbb{P}_{(E,U)}}|\ell_1(z_i,\alpha_i) - \ell_1(\hat{z}_i,\hat{\alpha}_i)| + \sum_{i\in[p]}\mathbb{E}_{\mathbb{P}_{(X,U)}}|\ell_2(b_i,\gamma_i) - \ell_2(\hat{b}_i,\hat{\gamma}_i)|$$

$$\lesssim \exp(\bar{\alpha}_{m,n})\sum_{i=1}^{n}\left\|\begin{pmatrix} z_i - \hat{z}_i \\ \alpha_i - \hat{\alpha}_i \end{pmatrix}\right\|_2^2 + \sum_{i=1}^{p}\left\|\begin{pmatrix} b_i - \hat{b}_i \\ \gamma_i - \hat{\gamma}_i \end{pmatrix}\right\|_2^2$$

$$= \exp(\bar{\alpha}_{m,n})\{\|\hat{Z} - Z\|_F^2 + \|\hat{\alpha} - \alpha\|_2^2\} + \{\|\hat{B} - B\|_F^2 + \|\hat{\gamma} - \gamma\|_2^2\},$$

then by combining Theorem 5, the desired result follows. $\qquad\square$

*Proof of Corollary 1.* First, due to the identification conditions, we have

$$\begin{bmatrix} \hat{\alpha} \\ \hat{\gamma} \end{bmatrix} = \frac{1}{m}\hat{\Theta}\mathbf{1}_m, \quad \begin{bmatrix} \alpha^* \\ \gamma^* \end{bmatrix} = \frac{1}{m}\Theta^*\mathbf{1}_m,$$

thus

$$\left\|\mathbf{1}_m\begin{bmatrix} \hat{\alpha} \\ \hat{\gamma} \end{bmatrix}^\top - \mathbf{1}_m\begin{bmatrix} \alpha^* \\ \gamma^* \end{bmatrix}^\top\right\|_F \leq \left\|\frac{1}{m}\mathbf{1}_m\mathbf{1}_m^\top(\hat{\Theta} - \Theta^*)\right\|_F \leq \|\hat{\Theta} - \Theta^*\|_F,$$

i.e.,

$$\left\|\begin{bmatrix} \hat{\alpha} \\ \hat{\gamma} \end{bmatrix}^\top - \begin{bmatrix} \alpha^* \\ \hat{\gamma}^* \end{bmatrix}^\top\right\|_2 \leq \frac{1}{\sqrt{m}}\|\hat{\Theta} - \Theta^*\|_F = O_p(\delta_{m,n,p}).$$

As a consequence,

$$\|\hat{U}^{(23)}\hat{Z}^\top - U^{(23)*}Z^{*\top}\|_F$$

$$= \|(\hat{\Theta}^H - \Theta^{H*}) - \mathbf{1}_m(\hat{\alpha} - \alpha^*)^\top\|_F \leq \|(\hat{\Theta}^H - \Theta^{H*})\|_F + \|\mathbf{1}_m(\hat{\alpha} - \alpha^*)^\top\|_F$$

$$\leq 2\|(\hat{\Theta}^H - \Theta^{H*})\|_F = O_p(\sqrt{m}\delta_{m,n,p}).$$

Let $\sigma_1, \sigma_2, \cdots, \sigma_{k_2+k_3}$ and $\hat{\sigma}_1, \hat{\sigma}_2, \cdots, \hat{\sigma}_{k_2+k_3}$ be the singular values of $\frac{1}{\sqrt{mn}}U^{(23)*}Z^{*\top}$ and $\frac{1}{\sqrt{mn}}\hat{U}^{(23)}\hat{Z}^\top$ respectively. Further let $e_1, e_2, \cdots, e_{k_2+k_3}$ and $\hat{e}_1, \hat{e}_2, \cdots, \hat{e}_{k_2+k_3}$ be the corresponding left singular vectors. Under the identification constraints, the $k$-th column of $U^{(23)*}$ is $\sqrt{m\sigma_k}e_k$ and the $k$-th column of $\hat{U}^{(23)}$ is $\sqrt{m\hat{\sigma}_k}\hat{e}_k$.

According to Theorem 3 of (Yu et al., 2015), we have

$$\|e_k - \hat{e}_k\|_2 \leq \frac{\sqrt{2}}{\min\{\sigma_k^2 - \sigma_{k+1}^2, \sigma_{k-1}^2 - \sigma_k^2\}}\left\|\frac{1}{\sqrt{mn}}(\hat{U}^{(23)}\hat{Z}^\top - U^{(23)*}Z^{*\top})\right\|_F,$$

and by Assumption 2, there is $\|e_k - \hat{e}_k\|_2 = O_p(\frac{1}{\sqrt{n}}\delta_{m,n,p})$ for any $1 \le k \le k_2 + k_3$.

Also by Weyl's inequality, $|\sigma_k - \hat{\sigma}_k| \le \|\frac{1}{\sqrt{mn}}(\hat{U}^{(23)}\hat{Z}^\top - U^{(23)*}Z^{*\top})\|_{op} = O_p(\frac{1}{\sqrt{n}}\delta_{m,n,p})$. Let $U_k^{(23)*}, \hat{U}_k^{(23)}$ denote the $k$-th column of $U^{(23)*}, \hat{U}^{(23)}$ respectively, we have

$$\|U^{(23)*} - \hat{U}^{(23)}\|_F \le \sum_{k=1}^{k_2+k_3} \|U_k^{(23)*} - \hat{U}_k^{(23)}\|_2 = \sqrt{m}\sum_{k=1}^{k_2+k_3}\|\sigma_k e_k - \hat{\sigma}_k\hat{e}_k\|_2$$

$$\le \sqrt{m}\sum_{k=1}^{k_2+k_3}(|\sqrt{\sigma_k} - \sqrt{\hat{\sigma}_k}| + \sqrt{\sigma_k}\|e_k - \hat{e}_k\|_2)$$

$$\le \sqrt{m}\sum_{k=1}^{k_2+k_3}\left(\frac{|\sigma_k - \hat{\sigma}_k|}{\sqrt{\sigma_k} + \sqrt{\hat{\sigma}_k}} + \sqrt{\sigma_k}\|e_k - \hat{e}_k\|_2\right) = O_p(\sqrt{m/n}\delta_{m,n,p}).$$

Similarly, we can prove that $\|\hat{Z} - Z^*\|_F = O_p(\delta_{m,n,p})$ by considering about the right singular space.

On the other hand, we have

$$\|\hat{U}^{(12)}\hat{B}^\top - U^{(12)*}B^{*\top}\|_F = \|(\hat{\Theta}^X - \Theta^{X*}) - \mathbf{1}_m(\hat{\gamma} - \gamma^*)^\top\|_F$$

$$\le \|(\hat{\Theta}^X - \Theta^{X*})\|_F + \|\mathbf{1}_m(\hat{\gamma} - \gamma^*)^\top\|_F = O_p(\sqrt{m}\delta_{m,n,p}).$$

To consider about the error between $\hat{B} = \begin{bmatrix} \hat{B}_1 & \hat{B}_2 \end{bmatrix}$ and $B^* = \begin{bmatrix} B_1^* & B_2^* \end{bmatrix}$, we construct the error rate for $\hat{B}_1, \hat{B}_2$ respectively. According to the orthogonality between $\hat{U}^{(1)}, \hat{U}^{(2)}$ and $U^{(1)*}, U^{(2)*}$, we have

$$\|\hat{B}_2 - B_2^*\|_F = \|(\hat{U}^{(2)\top}\hat{U}^{(2)})^{-1}\hat{U}^{(2)\top}\hat{U}^{(12)}\hat{B}^\top - (U^{(2)*\top}U^{(2)*})^{-1}U^{(2)*\top}U^{(12)*}B^{*\top}\|_F$$

$$\le \|(\hat{U}^{(2)\top}\hat{U}^{(2)})^{-1}\hat{U}^{(2)\top}\|_2\|\hat{U}^{(12)}\hat{B}^\top - U^{(12)*}B^{*\top}\|_F + \|U^{(12)*}B^{*\top}\|_2$$

$$\|(\hat{U}^{(2)\top}\hat{U}^{(2)})^{-1}\hat{U}^{(2)\top} - (U^{(2)*\top}U^{(2)*})^{-1}U^{(2)*\top}\|_2$$

$$\le \frac{1}{\min_{k=1,\cdots,k_2+k_3}\sqrt{m\hat{\sigma}_k}}\|\hat{U}^{(12)}\hat{B}^\top - U^{(12)*}B^{*\top}\|_F$$

$$+ C_{12}\|(\hat{U}^{(2)\top}\hat{U}^{(2)})^{-1}\hat{U}^{(2)\top} - (U^{(2)*\top}U^{(2)*})^{-1}U^{(2)*\top}\|_F$$

$$= O_p(\delta_{m,n,p}),$$

because the rows of $(\hat{U}^{(2)\top}\hat{U}^{(2)})^{-1}\hat{U}^{(2)\top}$ are $\frac{1}{\sqrt{m\hat{\sigma}_1}}\hat{e}_1^\top, \frac{1}{\sqrt{m\hat{\sigma}_2}}\hat{e}_2^\top, \cdots, \frac{1}{\sqrt{m\hat{\sigma}_{k_2}}}\hat{e}_{k_2}^\top$, and the rows of $(U^{(2)*\top}U^{(2)*})^{-1}U^{(2)*\top}$ are $\frac{1}{\sqrt{m\sigma_1}}e_1^\top, \frac{1}{\sqrt{m\sigma_2}}e_2^\top, \cdots, \frac{1}{\sqrt{m\sigma_{k_2}}}e_{k_2}^\top$. Denote $\hat{U}_k^{(2)}, U_k^{(2)*}$ as the $k$-th row of $\hat{U}^{(2)}, U^{(2)*}, k = 1, 2, \cdots, k_2$. Therefore, with Theorem 3 of (Yu et al., 2015), we have

$$\|(U^{(2)*\top}U^{(2)*})^{-1}U^{(2)*\top} - (\hat{U}^{(2)\top}\hat{U}^{(2)})^{-1}\hat{U}^{(2)\top}\|_F$$

$$\le \sum_{k=1}^{k_2}\|\hat{U}_k^{(2)} - U_k^{(2)*}\|_2 = \frac{1}{\sqrt{m}}\sum_{k=1}^{k_2}\left\|\frac{1}{\sqrt{\sigma_k}}e_k - \frac{1}{\sqrt{\hat{\sigma}_k}}\hat{e}_k\right\|_2$$

$$\le \frac{1}{\sqrt{m}}\sum_{k=1}^{k_2}\left(\left|\frac{1}{\sqrt{\sigma_k}} - \frac{1}{\sqrt{\hat{\sigma}_k}}\right| + \frac{1}{\sqrt{\sigma_k}}\|e_k - \hat{e}_k\|_2\right)$$

$$\le \frac{1}{\sqrt{m}}\sum_{k=1}^{k_2}\left(\frac{|\sigma_k - \hat{\sigma}_k|}{\sqrt{\sigma_k\hat{\sigma}_k}(\sqrt{\sigma_k} + \sqrt{\hat{\sigma}_k})} + \frac{1}{\sqrt{\sigma_k}}\|e_k - \hat{e}_k\|_2\right)$$

$$= O_p(\frac{1}{\sqrt{m^2n}}\delta_{m,n,p}).$$

Therefore, with $\|\hat{B}_2 - B_2^*\|_F$, we have

$$\|\hat{U}^{(2)}\hat{B}_2^\top - U^{(2)*}B_2^{*\top}\|_F \le \|\hat{U}^{(2)} - U^{(2)*}\|_2\|B_2\|_F + \|U^{(2)*}\|_2\|\hat{B}_2^\top - B_2^{*\top}\|_F$$

$$\le \sqrt{pk_2}C_2\|\hat{U}^{(2)} - U^{(2)*}\|_2 + \max_{k\in[k_2]}\sqrt{m\sigma_k}\|\hat{B}_2^\top - B_2^{*\top}\|_F$$

$$= O_p(\sqrt{m}\delta_{m,n,p}),$$

and finally we arrive at

$$\|\hat{U}^{(1)}\hat{B}_1^\top - U^{(1)*}B_1^\top\|_F \leq \|(\hat{U}^{(12)}\hat{B}^\top - U^{(12)*}B^{*\top}) - (\hat{U}^{(2)}\hat{B}_2^\top - U^{(2)*}B_2^\top)\|_F$$
$$\leq \|(\hat{U}^{(12)}\hat{B}^\top - U^{(12)*}B^{*\top})\|_F + \|(\hat{U}^{(2)}\hat{B}_2^\top - U^{(2)*}B_2^\top)\|_F$$
$$= O_p(\sqrt{m}\delta_{m,n,p}).$$

Similarly, we denote $\sigma_1, \sigma_2, \cdots, \sigma_{k_1}$ and $\hat{\sigma}_1, \hat{\sigma}_2, \cdots, \hat{\sigma}_{k_1}$ be the singular values of $\frac{1}{\sqrt{mp}}U^{(1)}B_1^\top$ and $\frac{1}{\sqrt{mp}}\hat{U}^{(1)}\hat{B}_1^\top$ respectively, and further let $f_1, f_2, \cdots, f_{k_1}$ and $\hat{f}_1, \hat{f}_2, \cdots, \hat{f}_{k_1}$ be the corresponding left singular vectors. Then under the identification conditions, the $k-th$ column of $U^{(1)*}$ is $\sqrt{m\delta_k}f_k$, and the $k-th$ column of $\hat{U}^{(1)}$ is $\sqrt{m\hat{\delta}_k}\hat{f}_k$. By applying Theorem 3 in (Yu et al., 2015), we have

$$\|f_k - \hat{f}_k\|_2 \leq \frac{\sqrt{2}}{\min\{\delta_k^2 - \delta_{k+1}^2, \delta_{k-1}^2 - \delta_k^2\}}\left\|\frac{1}{\sqrt{mp}}(\hat{U}^{(1)}\hat{B}_1^\top - U^{(1)*}B_1^\top)\right\|_F = O_p(\frac{1}{\sqrt{p}}\delta_{m,n,p})$$

for any $k = 1, 2, \cdots, k_1$. And by Weyl's inequality, $|\delta_k - \hat{\delta}_k| \leq \|\frac{1}{\sqrt{mp}}(\hat{U}^{(1)}\hat{B}_1^\top - U^{(1)}B_1^\top)\|_{op} = O_p(\frac{1}{\sqrt{p}}\delta_{m,n,p})$. Let $U_k^{(1)*}, \hat{U}_k^{(1)}$ denote the $k$-th colunm of $U^{(1)*}, \hat{U}^{(1)}$ respectively, we have

$$\|U^{(1)*} - \hat{U}^{(1)}\|_F \leq \sum_{k=1}^{k_1}\|U_k^{(1)*} - \hat{U}_k^{(1)}\|_2 = \sqrt{m}\sum_{k=1}^{k_2}\|\delta_k f_k - \hat{\delta}_k\hat{f}_k\|_2$$
$$\leq \sqrt{m}\sum_{k=1}^{k_2}(|\sqrt{\delta_k} - \sqrt{\hat{\delta}_k}| + \sqrt{\delta_k}\|f_k - \hat{f}_k\|_2)$$
$$= \sqrt{m}\sum_{k=1}^{k_2}\left(\frac{|\delta_k - \hat{\delta}_k|}{\sqrt{\delta_k} + \sqrt{\hat{\delta}_k}} + \sqrt{\delta_k}\|f_k - \hat{f}_k\|_2\right) = O_p(\sqrt{m/p}\delta_{m,n,p}).$$

Similarly, we can prove that $\|\hat{B}_1 - B_1^*\|_F = O_p(\delta_{m,n,p})$ by considering about the right singular space. By combining the results together, we have

$$\|\hat{U} - U^*\|_F = O_p\left(\sqrt{\frac{m(n+p)}{np}}\delta_{m,n,p}\right), \quad \|\hat{B} - B^*\|_F = O_p(\delta_{m,n,p}).$$

□

*Proof of Theorem 5.* According to Assumption 2, we first prove that $\|\frac{1}{m}\sum_{j=1}^m u_j^{(2)\prime}u_j^{(i)\prime\top}\|_F = O(\sigma_{m,n,p}^2)$ for both $i = 1, 3$. By directly calculating

$$\|\frac{1}{m}\sum_{j=1}^m u_j^{(2)\prime}u_j^{(i)\prime\top}\|_F = \|\frac{1}{m}\sum_{j=1}^m\{u_j^{(2)} - \frac{1}{m}(\sum_{j=1}^m u_j^{(2)})\}\{u_j^{(i)} - \frac{1}{m}(\sum_{j=1}^m u_j^{(i)})\}^\top\|_F$$
$$= \|\frac{1}{m}\sum_{j=1}^m u_j^{(2)}u_j^{(i)\top} - (\sum_{j=1}^m u_j^{(i)}/m)(\sum_{j=1}^m u_j^{(2)}/m)^\top\|_F$$
$$\leq \|\frac{1}{m}\sum_{j=1}^m u_j^{(2)}u_j^{(i)\top}\|_F + \|\sum_{j=1}^m u_j^{(i)}/m\|_F\|\sum_{j=1}^m u_j^{(2)}/m\|_F = O(\sigma_{m,n,p}^2),$$

there is also $\|\frac{1}{m}U^{(2)\prime\top}U^{(i)\prime}\|_F = O(\sigma_{m,n,p}^2)$ for any $i = 1, 3$.

Denote $(U^*, Z^*, B^*, \alpha^*, \gamma^*)$ as the transformed version of $(U, Z, B, \alpha, \gamma)$ accordng to Remark 5. Then we can analyse the errors as follows.

$$\|\alpha^* - \alpha\|_2 = \|\frac{1}{m}ZU_m^{(23)\top}\mathbf{1}_m\|_2 \leq \|Z\|_2\|\frac{1}{m}\sum_{j=1}^m u_j^{(23)}\|_2 = O(\sqrt{n}\sigma_{m,n,p}),$$

$$\|\gamma^* - \gamma\|_2 = \|\frac{1}{m}BU_m^{(12)\top}\mathbf{1}_m\|_2 \leq \|B\|_2\|\frac{1}{m}\sum_{j=1}^m u_j^{(12)}\|_2 = O(\sqrt{p}\sigma_{m,n,p}),$$

$$\|U^{(23)*} - U_m^{(23)}\|_F \leq \|U^{(23)''} - U_m^{(23)}\|_F + \|U^{(23)''} - U^{(23)*}\|_F$$

$$\leq \|I - G\|_F\|U^{(23)''}\|_F + \|U^{(23)''} - U_m^{(23)}\|_F,$$

$$\|U^{(1)*} - U_m^{(1)}\|_F \leq \|U^{(1)*} - U^{(1)''}\|_F + \|U_m^{(1)} - U^{(1)''}\|_F$$

$$\leq \|I - K\|_F\|U^{(1)''}\|_F + \|U^{(1)} - U^{(1)''}\|_F,$$

$$\|B_1^* - B_1\|_F \leq \|B_1 K^{-\top} - B_1\|_F \leq \|B_1\|_F\|I - K^{-\top}\|_F,$$

$$\|Z^* - Z\|_F \leq \|Z^* - Z'\|_F + \|Z' - Z\|_F = \|Z'\|_F\|I - G^{-\top}\|_F + \|Z' - Z\|_F.$$

Note that $G, K$ can be decomposed into

$$\|G\|_F = \|(Z'^\top Z'/n)^{1/2}\Gamma\mathcal{V}^{-1/4}\|_F$$

$$\leq \|(Z'^\top Z'/n)^{1/2}\|_2\|\Gamma - I\|_F\|\mathcal{V}^{-1/4}\|_F + \|(Z'^\top Z'/n)^{1/2}\|_2\|V^{-1/4}\|_F,$$

$$\|K\|_F = \|(B_1^\top B_1/p)^{1/2}D\mathcal{W}^{-1/4}\|_F$$

$$\leq \|(B_1^\top B_1/p)^{1/2}\|_F\|D - I\|_F\|\mathcal{W}^{-1/4}\|_F + \|(B_1^\top B_1/p)^{1/2}\|_F\|\mathcal{W}^{-1/4}\|_F.$$

We first analyze the error bounds of $G, K$ related terms.

With Assumption 2, we have

$$\left\|\frac{1}{mn}(Z'^\top Z')^{1/2}(U^{(23)''\top}U^{(23)''})(Z'^\top Z')^{1/2} - (Z'^\top Z'/n)^2\right\|_F$$

$$= \left\|\frac{1}{n}(Z'^\top Z')^{1/2}(U^{(23)''\top}U^{(23)''}/m - Z^\top Z/n + Z^\top Z/n - Z'^\top Z'/n)(Z'^\top Z')^{1/2}\right\|_F$$

$$\leq \|(Z'^\top Z'/n)^{1/2}\|_2^2\left[\|U^{(23)''\top}U^{(23)''}/m - Z^\top Z/n\|_F + \|Z^\top Z/n - Z'^\top Z'/n\|_F\right]$$

$$= \|Z'^\top Z'/n\|_2\left[\|U^{(23)''\top}U^{(23)''}/m - Z^\top Z/n\|_F + \|Z^\top Z/n - Z'^\top Z'/n\|_F\right].$$

By Weyl's inequality, we know that the 2-norm of $Z'^\top Z'/n$ can be bounded by both $\|Z^\top Z/n\|_2$ and a permutation term, i.e.,

$$\|Z'^\top Z'/n\|_2 \leq \|Z'^\top Z'/n - Z^\top Z/n\|_2 + \|Z^\top Z/n\|_2,$$

where $\|Z^\top Z/n\|_2 = O(1)$. Note that

$$\frac{1}{n}Z'^\top Z' - \frac{1}{n}Z^\top Z = \frac{1}{n}\begin{pmatrix} Z_2'^\top Z_2' - Z_2^\top Z_2 & (Z_2'^\top - Z_2^\top)Z_3 \\ Z_3^\top(Z_2' - Z_2) & 0 \end{pmatrix}$$

$$= \frac{1}{n}\begin{pmatrix} Z_2^\top Z_3\Gamma_{23}\Gamma_{22} + \Gamma_{22}\Gamma_{23}^\top Z_3^\top Z_2 + \Gamma_{22}\Gamma_{23}^\top Z_3^\top Z_3\Gamma_{23}\Gamma_{22} & \Gamma_{22}\Gamma_{23}^\top Z_3^\top Z_3 \\ Z_3^\top Z_3\Gamma_{23}\Gamma_{22} & 0 \end{pmatrix},$$

where we denote $\Gamma_{22} = (U^{(2)'\top}U^{(2)'})^{-1}$, $\Gamma_{23} = U^{(3)'\top}U^{(2)'}$, hence

$$\|\frac{1}{n}Z'^\top Z' - \frac{1}{n}Z^\top Z\|_F \leq \frac{4}{n}\|Z_2^\top Z_3\Gamma_{23}\Gamma_{22}\|_F + \frac{1}{n}\|\Gamma_{22}\Gamma_{23}^\top Z_3^\top Z_3\Gamma_{23}\Gamma_{22}\|_F$$

$$\leq \frac{4}{n}\|Z_2^\top Z_3\|_F\|\Gamma_{23}\|_F\|\Gamma_{22}\|_2 + \frac{1}{n}\|Z_3^\top Z_3\|_F\|\Gamma_{23}\|_F^2\|\Gamma_{22}\|_2^2$$

$$= O(\sigma_{m,n,p}^2)$$

according to Assumption 2. Also, here we bound $\|\Gamma_{22}\|_2$ by Weyl's inequality, i.e.,

$$\|\Gamma_{22}\|_2 = \sigma_{\max}((U^{(2)'\top}U^{(2)'})^{-1}) = \frac{1}{\sigma_{\min}(U^{(2)'\top}U^{(2)'})} \leq \frac{1}{\sigma_{\min}(U_m^{(2)\top}U_m^{(2)}) - \sigma_{m,n,p}^2} = O(\frac{1}{m}).$$

Therefore, $\|Z'^\top Z'/n\|_2 = O(1)$. Next, we consider about the error bound of $\|U^{(23)''\top}U^{(23)''}/m - Z^\top Z/n\|_F$ by decomposing it into

$$\|U^{(23)''\top}U^{(23)''}/m - Z^\top Z/n\|_F \leq \|U^{(23)''\top}U^{(23)''}/m - U^{(23)'\top}U^{(23)'}/m\|_F$$
$$+ \|U^{(23)'\top}U^{(23)'}/m - Z^\top Z/n\|_F,$$

where the latter is of $O(\sigma^2_{m,n,p})$. For the former, we have

$$U^{(23)''\top}U^{(23)''}/m - U^{(23)'\top}U^{(23)'}/m$$
$$= \frac{1}{m}\begin{pmatrix} 0 & U^{(2)'\top}(U^{(3)''} - U^{(3)'}) \\ (U^{(3)''} - U^{(3)'})^\top U^{(2)'} & U^{(3)''\top}U^{(3)''} - U^{(3)'\top}U^{(3)'} \end{pmatrix}$$
$$= \frac{1}{m}\begin{pmatrix} 0 & -U^{(2)'\top}U^{(3)'} \\ -U^{(3)'\top}U^{(2)'} & -U^{(3)'\top}[U^{(2)'}(U^{(2)'\top}U^{(2)'})^{-1}U^{(2)'\top}]U^{(3)'} \end{pmatrix},$$

whose F-norm can be bounded by considering

$$\|U^{(23)''\top}U^{(23)''}/m - U^{(23)'\top}U^{(23)'}/m\|_F$$
$$\leq \frac{1}{m}\left(2\|U^{(2)'\top}U^{(3)'}\|_F + \|U^{(2)'\top}U^{(3)'}\|_F^2\|\Gamma_{22}\|_F\right) = O\left(\sigma^2_{m,n,p}\right).$$

To sum up, we have

$$\left\|\frac{1}{mn}(Z'^\top Z')^{1/2}(U^{(23)''\top}U^{(23)''})(Z'^\top Z')^{1/2} - (Z'^\top Z'/n)^2\right\|_F = O(\sigma^2_{m,n,p}).$$

Therefore, by Weyl's inequality, we have

$$|\rho_i^2 - \sigma_i((Z'^\top Z'/n)^2)| \leq O(\sigma^2_{m,n,p}),$$

and accordingly,

$$|\rho_i - \sigma_i(Z'^\top Z'/n)| = \frac{|\rho_i^2 - \sigma_i((Z'^\top Z'/n)^2)|}{\rho_i + \sigma_i(Z'^\top Z'/n)} = O(\sigma^2_{m,n,p}),$$

$$|\rho_i^{1/2} - \sigma_i((Z'^\top Z'/n)^{1/2})| = \frac{|\rho_i - \sigma_i(Z'^\top Z'/n)|}{\rho_i^{1/2} + \sigma_i((Z'^\top Z'/n)^{1/2})} = O(\sigma^2_{m,n,p}),$$

$$|\rho_i^{-1/2} - \sigma_i((Z'^\top Z'/n)^{-1/2})| = |\rho_i^{-1/2}||\rho_i^{1/2} - \sigma_i((Z'^\top Z'/n)^{1/2})||\sigma_i((Z'^\top Z'/n)^{-1/2})| = O(\sigma^2_{m,n,p}).$$

Regarding the singular vectors $\Gamma$, we further denote $\Gamma'$ as the singular vector of $(Z'^\top Z'/n)^2$ and apply Theorem 3 from (Yu et al., 2015), then there is

$$\|\Gamma - I\|_F \leq \|\Gamma - \Gamma'\|_F + \|\Gamma' - I\|_F$$
$$= O\left(\left\|\frac{1}{mn}(Z'^\top Z')^{1/2}(U^{(23)''\top}U^{(23)''})(Z'^\top Z')^{1/2} - (Z'^\top Z'/n)^2\right\|_F\right.$$
$$\left. + \|(Z'^\top Z'/n)^2 - (Z^\top Z/n)\|_F\right)$$
$$= O(\sigma^2_{m,n,p}).$$

Regarding the singular value matrix $\mathcal{V}$, we have

$$\|\mathcal{V}^{-1/4}\|_F$$
$$\leq \|\mathcal{V}^{-1/4} - (Z'^\top Z'/n)^{-1/2}\|_F + \|(Z'^\top Z'/n)^{-1/2} - (Z^\top Z/n)^{-1/2}\|_F + \|(Z^\top Z/n)^{-1/2}\|_F$$
$$\leq \sqrt{k_2 + k_3}\max_i|\rho_i^{-1/2} - \sigma_i((Z'^\top Z'/n)^{-1/2})| + \|(Z'^\top Z'/n)^{-1/2} - (Z^\top Z/n)^{-1/2}\|_F$$
$$+ \|(Z^\top Z/n)^{-1/2}\|_F$$
$$= O(1).$$

To sum up, we have

$$\|G\|_F \leq \|(Z'^\top Z'/n)^{1/2}\|_2 \left(\|\Gamma - I\|_F + 1\right)\|\mathcal{V}^{-1/4}\|_F$$
$$= \|(Z'^\top Z'/n)\|_2^{1/2}\left(\|\Gamma - I\|_F + 1\right)\|\mathcal{V}^{-1/4}\|_F = O(1).$$

On the other hand, we should estimate the error bound of $\|I - G^{-T}\|_F$ and $\|G - I\|_F$ in order to bound $\|Z^* - Z\|_F$ and $\|U^* - U\|_F$. Note that

$$\|I - G^{-T}\|_F = \|I - G^{-1}\|_F \leq \|G^{-1}\|_F\|I - G\|_F,$$

where

$$\|G^{-1}\|_F = \|\mathcal{V}^{1/4}\Gamma^\top (Z'^\top Z'/n)^{-1/2}\|_F \leq \|\mathcal{V}^{1/4}\|(\|\Gamma - I\|_F + 1)\|(Z'^\top Z'/n)^{-1/2}\|_F = O(1),$$

$$\begin{aligned}
\|I - G\|_F &= \|I - (Z'^\top Z')^{1/2}\mathcal{V}^{-1/4}\|_F + \|(Z'^\top Z')^{1/2}\|_F\|I - \Gamma\|_F\|\mathcal{V}^{-1/4}\|_F \\
&\leq \|(Z'^\top Z')^{1/2}\|_F\|(Z'^\top Z')^{-1/2} - \mathcal{V}^{-1/4}\|_2 + \|(Z'^\top Z')^{1/2}\|_F\|I - \Gamma\|_F\|\mathcal{V}^{-1/4}\|_F \\
&\leq O(\sigma_{m,n,p}^2).
\end{aligned}$$

Therefore, we reach at the result that

$$\|I - G^{-T}\|_F \leq \|G^{-1}\|_F\|I - G\|_F = O(\sigma_{m,n,p}^2).$$

Similarly, we estimate the error bounds of $\|I - K\|_F$ and $\|I - K^{-\top}\|_F$ by considering about

$$\begin{aligned}
&\|\frac{1}{mp}(B_1^\top B_1)^{1/2}(U^{(1)''\top}U^{(1)''})(B_1^\top B_1)^{1/2} - (B_1^\top B_1/p)^2\|_F \\
&\leq \|\frac{1}{mp}(B_1^\top B_1)^{1/2}(U^{(1)''\top}U^{(1)''} - U^{(1)'\top}U^{(1)'})(B_1^\top B_1)^{1/2}\|_F \\
&\quad + \|\frac{1}{p}(B_1^\top B_1)^{1/2}(U^{(1)'\top}U^{(1)'}/m - B_1^\top B_1/p)(B_1^\top B_1)^{1/2}\|_F \\
&= \frac{1}{m}\|B_1^\top B_1/p\|_2\|U^{(1)''\top}U^{(1)''} - U^{(1)'\top}U^{(1)'}\|_F \\
&\quad + \|B_1^\top B_1/p\|_2\|U^{(1)'\top}U^{(1)'}/m - \mathbb{E}_{\mathbb{P}_U}[U^{(1)}U^{(1)\top}]\|_F,
\end{aligned}$$

where

$$\begin{aligned}
\|U^{(1)''\top}U^{(1)''} - U^{(1)'\top}U^{(1)'}\|_F &= \|U^{(1)'\top}(U^{(2)'}(U^{(2)'\top}U^{(2)'})^{-1}U^{(2)'\top})U^{(1)'}\|_F \\
&\leq \|U^{(1)'\top}U^{(2)'}\|_F^2\|\Gamma_{22}\|_F = O(m\sigma_{m,n,p}^4).
\end{aligned}$$

Therefore, we have

$$\|\frac{1}{mp}(B_1^\top B_1)^{1/2}(U^{(1)''\top}U^{(1)''})(B_1^\top B_1)^{1/2} - (B_1^\top B_1/p)^2\|_F = O(\sigma_{m,n,p}^2).$$

By Weyl's inequality, and note that $\mathcal{W} = \text{diag}(\sigma_1^2, \cdots, \sigma_{k_1}^2)$ is the eigenvalue matrix of $\frac{1}{p}(B_1^\top B_1)^{1/2}(U^{(1)''\top}U^{(1)''})(B_1^\top B_1)^{1/2}$, we have

$$|\delta_i^2 - ((B_1^\top B_1/p)^2)_{ii}| \leq O(\sigma_{m,n,p}^2),$$

$$|\delta_i - (B_1^\top B_1/p)_{ii}| = \frac{|\delta_i^2 - ((B_1^\top B_1/p)^2)_{ii}|}{\delta_i + (B_1^\top B_1/p)_{ii}} = O(\sigma_{m,n,p}^2),$$

$$|\delta_i^{1/2} - ((B_1^\top B_1/p)^{1/2})_{ii}| = \frac{|\delta_i - (B_1^\top B_1/p)_{ii}|}{\delta_i^{1/2} + ((B_1^\top B_1/p)^{1/2})_{ii}} = O(\sigma_{m,n,p}^2).$$

Similarly to our analysis on $G$, we use Theorem 3 in (Yu et al., 2015) and obtain the error bound of singular vectors as

$$\|I - D\|_F = O(\sigma_{m,n,p}^2).$$

Regarding the error bound of $\|I - K\|_F$ and $\|I - K^{-\top}\|_F$, we consider

$$\|\mathcal{W}^{-1/4}\|_F \leq \|\mathcal{W}^{-1/4} - (B_1^\top B_1/p)^{-1/2}\|_F + \|B_1^\top B_1/p\|_F = O(1),$$

$$\|K\|_F \leq \|(B_1^\top B_1/p)^{1/2}\|_F\|\mathcal{W}^{-1/4}\|_F\|D - I\|_F + \|(B_1^\top B_1/p)^{1/2}\|_F\|\mathcal{W}^{-1/4}\|_F = O(1),$$

and then

$$\|I - K\|_F$$
$$= \|I - (B_1^\top B_1/p)^{1/2} D \mathcal{W}^{-1/4}\|_F$$
$$\leq \|I - (B_1^\top B_1/p)^{1/2} \mathcal{W}^{-1/4}\|_F + \|(B_1^\top B_1/p)^{1/2}\|_F \|D - I\|_F \|\mathcal{W}^{-1/4}\|_F$$
$$\leq \|(B_1^\top B_1/p)^{1/2}\|_F \|(B_1^\top B_1/p)^{-1/2} - \mathcal{W}^{-1/4}\|_F + \|(B_1^\top B_1/p)^{1/2}\|_F \|D - I\|_F \|\mathcal{W}^{-1/4}\|_F$$
$$\leq O(\sigma_{m,n,p}^2).$$

Similarly, there is

$$\|K^{-1}\|_F = \|\mathcal{W}^{1/4} D^\top (B_1^\top B_1/p)^{-1/2}\|_F$$
$$\leq \|\mathcal{W}^{1/4}\|_F \|D^\top - I\|_F \|(B_1^\top B_1/p)^{-1/2}\|_F + \|\mathcal{W}^{1/4}\|_F \|(B_1^\top B_1/p)^{-1/2}\|_F$$
$$= O(1),$$

and then
$$\|I - K^{-\top}\|_F \leq \|K^{-1}\|_F \|I - K\|_F = O(\sigma_{m,n,p}^2).$$

Now, we go back to the analysis of error bounds for embeddings and parameters. According to the process of transformation, there is

$$\|U^{(23)*} - U_m^{(23)}\|_F \leq \|U^{(23)\prime\prime} - U_m^{(23)}\|_F + \|U^{(23)\prime\prime} - U^{(23)*}\|_F$$
$$\leq \|I - G\|_F \|U^{(23)\prime\prime}\|_F + \|U^{(23)\prime\prime} - U_m^{(23)}\|_F$$
$$\leq \|I - G\|_F (\|U_m^{(23)}\|_F + \|U^{(23)\prime\prime} - U_m^{(23)}\|_F) + \|U^{(23)\prime\prime} - U_m^{(23)}\|_F,$$

where

$$\|U^{(23)\prime\prime} - U_m^{(23)}\|_F^2$$
$$= \|U^{(2)\prime\prime} - U_m^{(2)}\|_F^2 + \|U^{(3)\prime\prime} - U_m^{(3)}\|_F^2$$
$$\leq \|U^{(2)\prime} - U_m^{(2)}\|_F^2 + (\|U^{(3)\prime\prime} - U^{(3)\prime}\|_F + \|U^{(3)\prime} - U_m^{(3)}\|_F)^2$$
$$\leq \|U^{(2)\prime} - U_m^{(2)}\|_F^2 + \|U^{(3)\prime\prime} - U^{(3)\prime}\|_F^2 + \|U^{(3)\prime} - U_m^{(3)}\|_F^2 + 2\|U^{(3)\prime} - U_m^{(3)}\|_F \|U^{(2)\prime} - U_m^{(2)}\|_F$$
$$\leq \|U^{(2)\prime} - U_m^{(2)}\|_F^2 + \|U^{(3)\prime\top} U^{(2)\prime}\|_F^2 \|\Gamma_{22}\|_F^2 \|U^{(2)\prime}\|_F^2 + \|U^{(3)\prime} - U_m^{(3)}\|_F^2 + 2\|U^{(3)\prime} - U_m^{(3)}\|_F \|U^{(2)\prime} - U_m^{(2)}\|_F$$
$$= O(m\sigma_{m,n,p}^2),$$

thus

$$\|U^{(23)*} - U_m^{(23)}\|_F \leq \|I - G\|_F (\|U_m^{(23)}\|_2 + \|U^{(23)\prime\prime} - U_m^{(23)}\|_F) + \|U^{(23)\prime\prime} - U_m^{(23)}\|_F = O(\sqrt{m}\sigma_{m,n,p}).$$

Similarly, we have

$$\|U^{(1)*} - U_m^{(1)}\|_F \leq \|U^{(1)*} - U^{(1)\prime\prime}\|_F + \|U_m^{(1)} - U^{(1)\prime\prime}\|_F$$
$$\leq \|I - K\|_F \|U^{(1)\prime\prime}\|_F + \|U_m^{(1)} - U^{(1)\prime\prime}\|_F = O(\sqrt{m}\sigma_{m,n,p}).$$

therefore, the overall error bound of hyperlink embedding $U^*$ is

$$\|U^* - U_m\|_F = \sqrt{\|U^{(1)*} - U_m^{(1)}\|_F^2 + \|U^{(23)\prime\prime} - U_m^{(23)}\|_F^2} = O(\sqrt{m}\sigma_{m,n,p}).$$

For the node embeddings, we have the following:

$$\|Z^* - Z\|_F \leq \|Z^* - Z'\|_F + \|Z' - Z\|_F = \|Z'\|_F \|I - G^{-\top}\|_F + \|Z' - Z\|_F,$$

where
$$\|Z' - Z\|_F = \|Z_2' - Z_2\|_F \leq \|(U^{(2)\prime\top} U^{(2)\prime})^{-1} U^{(2)\prime\top} U^{(3)\prime}\|_F \|Z_3\|_F$$
$$\leq \|\Gamma_{22}\|_2 \|\Gamma_{23}\|_F \|Z_3\|_F = O(\sqrt{m}\sigma_{m,n,p}^2),$$

therefore
$$\|Z^* - Z\|_F = O(\sqrt{m}\sigma_{m,n,p}^2).$$

Regarding the parameters $B$, note that

$$\|B_1^* - B_1\|_F \leq \|B_1 K^{-\top} - B_1\|_F \leq \|B_1\|_F \|I - K^{-\top}\|_F = O(\sigma_{m,n,p}^2),$$

and similarly $\|B_2^* - B_2\| = O(\sqrt{m}\sigma_{m,n,p}^2)$ during transformation, we have

$$\|B^* - B\|_F = \sqrt{\|B_1^* - B_1\|_F^2 + \|B_2^* - B_2\|_F^2} = O(\sqrt{m}\sigma_{m,n,p}^2).$$

To sum up, we have proved that for the transformed version $(U^*, Z^*, B^*, \alpha^*, \gamma^*)$, the error bounds are as follows:

$$\|U^* - U_m\|_F = O(\sqrt{m}\sigma_{m,n,p}), \ \|Z^* - Z\|_F = O(\sqrt{m}\sigma_{m,n,p}^2), \ \|B^* - B\|_F = O(\sqrt{m}\sigma_{m,n,p}^2);$$

$$\|\alpha^* - \alpha\|_2 = O(\sqrt{n}\sigma_{m,n,p}), \|\gamma^* - \gamma\|_2 = O(\sqrt{p}\sigma_{m,n,p}).$$

Combine the above result with Corollary 1, then the desired result follows that $\|\hat{\alpha} - \alpha^*\|_2 = O_p(\delta_{m,n,p})$, $\|\hat{U} - U^*\|_F = O_p\left(\sqrt{m(n+p)/(np)}\delta_{m,n,p}\right)$, $\|\hat{Z} - Z^*\|_F = O_p(\delta_{m,n,p})$, $\|\hat{\gamma} - \gamma^*\|_2 = O_p(\delta_{m,n,p})$, $\|\hat{B} - B^*\|_F = O_p(\delta_{m,n,p})$. $\qquad\square$

*Proof of Lemma 2.* Note that $p_{U^{pc}}(u) \geq 0$ according to the definition, then we only need to consider about the integral, i.e.,

$$\int_{\text{supp}(\mathbb{P}_U)} p_{U^{pc}}(u)d\mu(u)$$

$$= \int_{\text{supp}(\mathbb{P}_U)} \{\gamma_{m,n,p}^k \int_{[u^{dis}(u)-\frac{1}{2\gamma_{m,n,p}}, u^{dis}(u)+\frac{1}{2\gamma_{m,n,p}})} p_U(u')d\mu(u')\}d\mu(u)$$

$$= \sum_{u^{dis} \in \mathcal{A}_{C,\gamma_{m,n,p}^{-1}}} \gamma_{m,n,p}^{-k}\{\gamma_{m,n,p}^k \int_{[u^{dis}(u)-\frac{1}{2\gamma_{m,n,p}}, u^{dis}(u)+\frac{1}{2\gamma_{m,n,p}})} p_U(u')d\mu(u')\}$$

$$= \sum_{u^{dis} \in \mathcal{A}_{C,\gamma_{m,n,p}^{-1}}} \int_{[u^{dis}(u)-\frac{1}{2\gamma_{m,n,p}}, u^{dis}(u)+\frac{1}{2\gamma_{m,n,p}})} p_U(u')d\mu(u')$$

$$= \int_{\text{supp}(\mathbb{P}_U)} p_U(u')d\mu(u') = 1,$$

then the result follows.

$\qquad\square$

*Proof of Theorem 6.* According to the mean value theorem and the condition of Lipschitz continuous, we have

$$p_{U^{pc}}(u) = \gamma_{m,n,p}^k \int_{[u^{dis}(u)-\frac{1}{2\gamma_{m,n,p}}, u^{dis}(u)+\frac{1}{2\gamma_{m,n,p}})} p_U(u)d\mu(u)$$

$$= \gamma_{m,n,p}^k \gamma_{m,n,p}^{-k} p_U(c) = p_U(c)$$

for some $c \in [u^{dis}(u) - \frac{1}{2\gamma_{m,n,p}}, u^{dis}(u) + \frac{1}{2\gamma_{m,n,p}})$, Note that $\|u - c\| \leq \sqrt{k}\gamma_{m,n,p}^{-1}$ since they lie in the same hypercube, then we have

$$|p_{U^{pc}}(u) - p_U(u)| = |p_U(c) - p_U(u)| \leq L\|u - c\| \leq L\sqrt{k}\gamma_{m,n,p}^{-1}$$

according to Lipschitz continuity. $\qquad\square$

*Proof of Lemma 3.* Let $C_m(u) := \#\{\text{occurrences of } u \text{ in } \mathcal{U}_m\}$. Then

$$C_m(u) \sim \text{Binomial}(m, p_{U^{dis}}(u)).$$

By Chernoff's bound for binomial random variables, for any $\delta_{m,n,p} < 1$,

$$\mathbb{P}\{|p_{U_m^{dis}}(u) - p_{U^{dis}}(u)| \geq \delta_{m,n,p}\, p_{U^{dis}}(u)\} \leq 2\exp\left(-\frac{\delta_{m,n,p}^2\, m\, p_{U^{dis}}(u)}{3}\right).$$

Let

$$p_{\max} := \max_u p_{U^{dis}}(u), \qquad p_{\min} := \min_u p_{U^{dis}}(u), \qquad \varepsilon_{m,n,p} := \delta_{m,n,p}\, p_{\max}.$$

Since $p_{\max} \geq p_{U^{\mathrm{dis}}}(u) \geq p_{\min}$ for all $u$, we obtain

$$\mathbb{P}\big\{\big|p_{U_m^{\mathrm{dis}}}(u) - p_{U^{\mathrm{dis}}}(u)\big| \geq \varepsilon_{m,n,p}\big\} \leq 2\exp\Big(-\frac{\delta_{m,n,p}^2 \, m \, p_{\min}}{3}\Big) = 2\exp\Big(-\frac{\varepsilon_{m,n,p}^2 \, m \, p_{\min}}{3p_{\max}^2}\Big).$$

Applying a union bound over $u \in \mathcal{A}_{C,\gamma_{m,n,p}^{-1}}$ and noting that $\big|\mathcal{A}_{C,\gamma_{m,n,p}^{-1}}\big| \leq (2C\gamma_{m,n,p})^k$, we have

$$\mathbb{P}\Big\{\exists\, u \in \mathcal{A}_{C,\gamma_{m,n,p}^{-1}} \text{ s.t. } \big|p_{U_m^{\mathrm{dis}}}(u) - p_{U^{\mathrm{dis}}}(u)\big| \geq \varepsilon_{m,n,p}\Big\}$$

$$\leq (2C\gamma_{m,n,p})^k \cdot 2\exp\Big(-\frac{\varepsilon_{m,n,p}^2 \, m \, p_{\min}}{3p_{\max}^2}\Big)$$

$$\leq 2\exp\Big[-\Big\{\frac{\varepsilon_{m,n,p}^2 \, m \, p_{\min}}{3p_{\max}^2} - k\log(2C\gamma_{m,n,p})\Big\}\Big].$$

The failure probability thus converges to $0$ as $m, n, p \to \infty$ whenever

$$\varepsilon_{m,n,p} \gg p_{\max}\sqrt{\frac{k\log(2C\gamma_{m,n,p})}{mp_{\min}}} \quad \text{and} \quad \varepsilon_{m,n,p} < p_{\max}$$

(since $\varepsilon_{m,n,p} = \delta_{m,n,p}p_{\max}$ with $\delta_{m,n,p} < 1$), which concludes the proof. $\qquad\square$

