# OpenReview forum: "ReLaSH: Reconstructing Joint Latent Spaces for Efficient Generation of Synthetic Hypergraphs with Hyperlink Attributes"
_ICLR.cc/2026/Conference — ICLR 2026 Poster_

### Official Review · Reviewer_hHTL · 2025-10-21

**Soundness:** 3
**Presentation:** 3
**Contribution:** 2
**Rating:** 6
**Confidence:** 4

**Summary:**

The authors propose **ReLaSH**, a hypergraph generative model that produces random hypergraphs together with _hyperedge-level features_. The key idea is to construct a **joint latent space** of hyperlinks and their attributes and learn it in a probabilistic, likelihood-based manner.

- **Data-driven latent space learning:** Build a joint latent representation that maximizes the likelihood of observed hyperedges and their associated features.

- **Latent reconstruction and decoding:** Reconstruct this latent distribution via a generator, and decode new samples from it to synthesize realistic hyperedges and their attributes.

**Strengths:**

- **\[S1]** The problem of _hypergraph generation with features_ is timely and important.

- **\[S2]** ReLaSH introduces, to my knowledge, the first _probabilistic and likelihood-based_ joint generative framework for hypergraphs with hyperlink attributes.

**\[S3]** The experiments span datasets with diverse feature modalities (textual, numerical, categorical), showing the model’s potential generality.

**Weaknesses:**

See “Questions” below.

**Questions:**

- **\[Q1]** _Missing references._ It seems that several recent works also tackle hypergraph generation with attributes. Could the authors clarify the novelty of ReLaSH relative to these and, if feasible, include them as baselines?

  - \[r1] Gailhard et al. _Feature-aware Hypergraph Generation via Next-Scale Prediction_, arXiv:2506.01467 (2025).

  - \[r2] Chun et al. _Attributed Hypergraph Generation with Realistic Interplay Between Structure and Attributes_, arXiv:2509.21838 (2025).

  - \[r3] Badalyan et al. _Structure and Inference in Hypergraphs with Node Attributes_, _Nature Communications_ 15 (2024): 7073.

- **\[Q2]** The proposed framework currently handles _hyperedge features_ only, while _node features_ are at least equally important and more common in real applications. Could the authors discuss whether and how ReLaSH could be extended to incorporate node features?

- **\[Q3]** I have concerns about the _evaluation metrics_, which do not appear to be standard. Could the authors justify why these particular choices are reasonable compared with alternatives, and clarify what properties of the generative distribution are expected to be captured if these losses are ideally minimized?

- **\[Q4]** Relatedly, is it conceptually possible to design a model that _directly optimizes_ these metrics (or differentiable surrogates) instead of using them only post hoc? What practical or theoretical challenges would such a design entail?

- **\[Q5]** Several configurations (k = 6, 12, 24, 48; with/without calibration) are reported, but the paper does not discuss how to _choose_ an optimal configuration for new datasets. It would be useful if the authors could provide practical guidelines or an automatic criterion (e.g., based on validation FED or structural RMSE) for selecting the latent dimension and deciding whether to apply calibration.

---

> ### Author Response · Authors · 2025-11-21
> **Author Response to Reviewer hHTL**
>
> Thank you for your positive feedback on our paper. We appreciate your recognition of the novelty of our approach, theory, and the effectiveness of the algorithms. In the following, we address each concern that you raised one by one.
>
> **Q1. Regarding more discussion about recent literature.**
>
> Thank you for providing these relevant references, which help us better position our work in the literature and improve our manuscript through the corresponding discussions. Below, we provide a detailed discussion of these references. We have also incorporated the corresponding references and discussions into our revised manuscript.
>
> Reference [1] proposes a hierarchical coarsening-expansion framework to jointly generate hypergraph topology and node attributes; in generation, their method produces new nodes, node attributes, and hyperlinks, whereas our goal is to generate new hyperlinks and hyperlink attributes on a fixed set of nodes (for example, generating new combinations of symptoms as hyperlinks over a fixed symptom set and corresponding attributes to form new medical records). Additionally, our modeling framework explicitly accounts for special characteristics of hypergraphs, such as degree heterogeneity and sparsity, and comes with clear theoretical guarantees.
>
> Reference [2] (NOAH) focuses on generating synthetic hyperlinks on a fixed set of nodes and node attributes, aiming to realistically capture the interplay between node structure and node attributes. Their algorithm models hyperedge construction as a sequence of probabilistic node attachments driven by attribute affinities, utilizing a core–fringe hierarchy to better reflect real-world patterns. The model generates only the hyperedge structure to reflect the observed node–structure–attribute relationship, which differs from our modeling objective. For a more detailed discussion of NOAH, please refer to our response to W1 for Reviewer MqMV, where we further elaborate on the novelty and distinctions between NOAH and our approach.
>
> Reference [3] introduces node attributes into a community-detection framework for modeling higher-order interactions in hypergraphs. The HyCoSBM model is primarily concerned with probabilistic modeling and inference in observed hypergraph datasets, rather than the generation of synthetic hyperlinks. Its main focus is to leverage node attributes and hyperedge structure jointly for community detection and prediction tasks, rather than the generative modeling task considered in this work.
>
> In addition to the references you mentioned, we also compared our method with state-of-the-art tabular data generation methods, and we kindly refer you to the fourth block of our response to Reviewer ELf7 for more details.
>
> Meanwhile, we have referenced and discussed all these methods in the Introduction of our updated paper.
>
> **Q2. Regarding extending ReLaSH to incorporate node features.**
>
> Thank you for your insightful suggestion regarding the incorporation of node attributes to further enhance the comprehensiveness of our model. We agree that our model can be readily extended to include node features in the generative process.
>
> To integrate node attributes, we can jointly embed the nodes and their corresponding attributes into a joint latent space, similar to our approach for hyperlinks with attributes. Intuitively, the node embeddings $\mathcal{Z}_n$ derived from the hypergraph structure capture latent structural features, while node attributes provide additional explicit information. These attributes can be embedded into the latent space using the same methodology developed for hyperlink attributes. Different ways of incorporating node covariates can be readily applied, since our framework allows general likelihood models that produce latent hyperlink embeddings; introducing node covariates primarily modifies this likelihood component. Furthermore, in applications requiring the generation of both synthetic nodes (and their attributes) and synthetic hyperlinks with attributes, we can model the node embeddings $\mathcal{Z}_n = \{z_1, \dots, z_n\}$ and the embeddings of their attributes as samples from unknown distributions and perform estimation and analysis similar to what we have done for the hyperlink part in our work.
>
> These extensions underscore the flexibility of the ReLaSH framework, whose core principle is to embed structural and attributed data into joint latent spaces and perform reconstruction within a low-dimensional continuous space. We appreciate your valuable suggestion and will consider these directions for future research.

---

> > ### Author Response · Authors · 2025-11-21
> >
> > **Q3. Regarding the evaluation metrics.**
> >
> > Thank you for your valuable insights regarding evaluation metrics. As discussed in our paper, the field of synthetic hypergraph generation currently lacks unified evaluation metrics, as it is relatively new and does not yet benefit from established pre-trained third-party models such as Inception model which are widely used in image generation tasks.
> >
> > Given that our model assumes hyperlinks and attributes follow certain underlying distributions, we focus on statistical similarity error metrics related to moments, including $\Delta_{\mathcal H_{\mathrm{m}}}$, $\Delta_{\mathcal H_{\mathrm{v}}}$, $\Delta_{\mathcal X_{\mathrm{m}}}$, and $\Delta_{\mathcal X_{\mathrm{v}}}$. These metrics are commonly employed in tabular data generative models [4, 5]. Notably, in the context of hypergraphs, these moment-related metrics carry strong statistical significance. For instance, $\Delta_{\mathcal H_{\mathrm{m}}}$ specifically capture differences in node heterogeneity within hypergraphs and hyperlink sparsity, which is a feature usually addressed in the hypergraph modeling literature [6, 7], while $\Delta_{\mathcal H_{\mathrm{v}}}$ captures importance second-moment node co-occurrence information.
> >
> > Additionally, we introduce FED as an extension of the Fréchet Inception Distance (FID) for hypergraph generation tasks. FED provides flexibility and produces results consistent with those of moment-based metrics. While FID is well established for evaluating generative models in vision tasks, its reliance on pre-trained Inception models limits its applicability in domains outside image analysis. To address this limitation, FID has been adapted for various fields: Fréchet Audio Distance (FAD) for music enhancement [8], Fréchet Video Distance (FVD) for generative video models [9], and Fréchet ChemNet Distance (FCD) for AI-generated molecules [10]. Each variant maintains FID’s statistical foundation but is customized to the domain’s unique data characteristics. FED and the embedding machine serve a similar purpose for hypergraph generative modeling.
> >
> > Finally, we recognize the need for further development of evaluation methods adapted to specific domains. This requires not only the effort to train a well-acknowledged third-party model (similar to the Inception model in image generation), but also substantial work in collecting high-quality data. Meanwhile, integrating specific downstream tasks to measure utility and formulating new, standardized metrics for hypergraph generation remain important directions for future work.
> >
> > **Q4. Regarding directly optimizing the proposed error metrics.**
> >
> > Thank you for your insightful question regarding the possibility of directly optimizing the proposed error metrics. As discussed above, our framework is built on the utilization of maximizing the likelihood connecting the abstract hyperlink and attributes to a low-rank joint latent space. The error metrics primarily focus on statistical similarity, reflecting differences in distribution through moment-based calculations. Directly minimizing these metrics would completely ignore the low-rank structure commonly observed in hypergraphs.  In contrast, our approach directly optimizes the fit between the model and the underlying data distribution, encompassing not only moment-based differences but the full distributional and low-rank structure. Moment-related error metrics are essentially summarized partial aspects of the likelihood.
> >
> > Regarding FED, it is a metric built upon an embedding machine, analogous to FID in image generation tasks. While FED (and similar metrics) provides valuable evaluation post hoc, it is challenging to use it as a direct loss function during training. This is because optimizing over such evaluation metrics can introduce non-differentiable components, potential deadlocks, or unwanted dependencies based on the embedding function’s properties.
> >
> > Therefore, while designing models that directly optimize these metrics may be conceptually possible in some cases, it presents significant practical and theoretical challenges, particularly in maintaining consistency with probabilistic modeling and ensuring robust, stable training. Our approach, relying on likelihood-based objectives, provides principled modeling, training, and sampling, while using statistical similarity metrics, including FED, for comprehensive evaluation.

---

> > > ### Author Response · Authors · 2025-11-21
> > >
> > > **Q5. Regarding the choice of optimal configurations.**
> > >
> > > Thank you for your consideration of how to select optimal configurations in our model, which have motivated us to develop such a pipeline and have
> > > significantly improved our paper. This is indeed a more challenging task
> > > than dimension selection in traditional low-rank models, since we need
> > > to determine not only the total size of the latent embedding space
> > > $(k_1 + k_2 + k_3)$, but also each component. Below, we introduce our
> > > pipeline inspired by [1] and referred to as *"hunt then trim" (HTT)*, to achieve this
> > > goal, followed by a discussion and empirical justification of it.
> > >
> > > HTT consists of three steps. In the first and second steps, it hunts for
> > > the latent space dimensions that generate the hypergraph, and then the
> > > dimensions that are orthogonal to those but drive attribute generation.
> > > In the third step, it then trims the dimensions that drive hypergraph
> > > formation but (almost) orthogonal to the attributes. Below is a step-by-step
> > > illustration.
> > >
> > > - **Step 1.** Select $k_2 + k_3$ by cross-validation on the hypergraph.
> > >   For each chosen dimension $k_{23} = k_2 + k_3$, we split the
> > >   population hypergraph into a training set and a testing set, with
> > >   80% of the nodes and attributes used for training and the remaining
> > >   20% for testing. We then perform the embedding procedure on both the
> > >   training and testing sets to estimate the hypergraph embeddings on the
> > >   training set (denote them by $U_{\mathrm{train}}^{(k_{23})}$) and on
> > >   the test set (denote them by $U_{\mathrm{test}}^{(k_{23})}$).
> > >   Next, we calculate the Fréchet distance FD$(k_{23})$ between
> > >   $U_{\mathrm{train}}^{(k_{23})}$ and $U_{\mathrm{test}}^{(k_{23})}$,
> > >   and choose $k_2 + k_3$ as the value of $k_{23}$ that minimizes
> > >   FD$(k_{23})$. With the selected $k_2 + k_3$, we obtain
> > >   $\{\hat{u} _ j^{(23)}\}_{j \in [m]}$ according to the embedding
> > >   procedure.
> > >
> > > - **Step 2.** Let $X = (x_1,\dots,x_m)^\top$ and
> > >   $U^{(23)} = (u^{(23)} _ 1, \dots, u^{(23)} _ m)^\top$. Regress $X$ on
> > >   $U^{(23)}$ via the linear model
> > >   $X = 1_n \mu^\top + U^{(23)} B^\top + E$ with latent components $B$, and obtain fitted values
> > >   $\hat{X}, \hat{U}, \hat{\mu}, \hat{B}$ and residual
> > >   $\hat{E} = X - \hat{X}$. Consider the testing problem
> > >   $H_0: k_1 = l$ versus $H_1: l + 1 \le k_1 \le k_{\max}$ and the test
> > >   statistic
> > >   $r(l) = (\phi_{l+1} - \phi_{k_{\max}+1}) / \phi_{k_{\max}+2}$,
> > >   where $\phi_l$ is the $l$-th largest eigenvalue of
> > >   $\operatorname{Cov}(\hat{E})$.
> > >
> > >
> > >   Under regular assumptions, the following conclusions hold:
> > >
> > >   1. Under $H_0$,
> > >      $r(l) \to (x_l - x_{k_{\max}+1}) / (x_{k_{\max}+1} - x_{k_{\max}+2})$,
> > >      where $x_1, \ldots, x_{k_{\max}+2}$ jointly follow the Tracy–Widom
> > >      distribution induced by the Gaussian orthogonal ensemble.
> > >
> > >   2. Under $H_1$, $r(l) \to \infty$, so asymptotically the testing
> > >      procedure achieves full power.
> > >
> > >   Consequently, for any $l \ge 0$, we can compare the observed value
> > >   of $r(l)$ with that of a value simulated from the joint
> > >   Tracy–Widom distribution [2]. We set $k_{\max}$ large enough and
> > >   determine $k_1$ using a sequential testing procedure. Starting with
> > >   $l = 0$, for each $0 \le l \le k_{\max}$, if the null hypothesis
> > >   $H_0: k_1 = l$ is rejected, we further test the null hypothesis
> > >   $H_0: k_1 = l + 1$ until a null hypothesis is accepted. In the
> > >   special case $l = 0$, the proposed test can be used to test the
> > >   existence of $U^{(1)}$.
> > >
> > > -   Step 3. To distinguish $k_3$ from $k_1$ and $k_2$, that is, to
> > >     identify $u^{(3)}$ separately from $u^{(23)}$, we frame the issue
> > >     within the previously introduced regression problem. Specifically, we incorporate the
> > >     potentially irrelevant factors $u^{(3)}$ into the attribute model.
> > >     This allows us to determine $k_3$ by addressing the following
> > >     multiple testing procedure: for each $1 \leq l \leq k_2 + k_3$, let
> > >     $B_{\cdot, l}$ denote the $l$-th column of $B$.
> > >     We then consider the following hypothesis testing problem:
> > >
> > >     $$H_0: B_{\cdot, l} = 0_{p \times 1} \; \text{versus }H_1: B_{\cdot, l} \neq 0_{p \times 1}$$
> > >     with the test statistics
> > >     $$s(l) = (2 \sum_{j=1}^p V_{jl}^2)^{-1/2} \sum_{j=1}^p (mB_{jl}^2 - V_{jl}),$$
> > >     where
> > >     $V_{jl}$ is  $(U^{(23)\top}U^{(23)})_{ll}^{-1}c_j$ and $c_j$  is the j,j-th entry of $(\mathrm{Cov}(\hat{{E}}))$.
> > >     Under regular assumptions, the following conclusions hold:
> > >
> > >     1.  Under $H_0$, $s(l) \to \mathcal N(0,1)$.
> > >
> > >     2.  Under $H_1$, denote the number of non-zero entries in
> > >         $B_{\cdot, l}$ as $p^{1-\beta}$. If
> > >         $B_{\cdot, l}=\Omega(m^{-1/2} (\log p)^\alpha)$ holds
> > >         uniformly with $\alpha >0$ and $0 < \beta <1/2$, then
> > >         $s(l) \to \infty$. Therefore, asymptotically the testing
> > >         procedure achieves full power.

---

> > > > ### Author Response · Authors · 2025-11-21
> > > >
> > > > Consequently, for each $0 \leq l \leq k_2 + k_3$, we compare the
> > > >     observed value of $s(l)$ to the standard normal distribution. This
> > > >     allows us to infer whether each column of $U^{(23)}$ belongs to
> > > >     $U^{(2)}$ or $U^{(3)}$, thereby enabling simultaneous estimation of
> > > >     $U^{(2)}$, $U^{(3)}$, as well as the corresponding dimensions $k_2$
> > > >     and $k_3$.
> > > >
> > > > To demonstrate the effectiveness of our proposed latent dimension
> > > > selection method, we conducted a simulation study with data size
> > > > $n = p = 500$, $m=5000$ and true latent dimensions
> > > > $k_1 = k_2 = k_3 = 4$. All embeddings and parameters were generated
> > > > according to the simulation settings described in Section B.3 of the
> > > > Appendix. We applied the HTT pipeline to the simulation data across 30
> > > > repetitions, and the results are presented in the following table.
> > > >
> > > > |$k_1$|$k_2$|$k_3$|Freq|
> > > > |---|---|---|---|
> > > > |4|4|4|27|
> > > > |4|3|4|1|
> > > > |4|5|4|2|
> > > >
> > > > Table 4: Result of latent dimension selection across 30 repetitions.
> > > >
> > > > The results indicate that, in most cases, the HTT algorithm
> > > > successfully recovers the true latent dimensions. We further apply this
> > > > algorithm in our real data analyses. With the implementation details in
> > > > Appendix B.4, we apply the HTT algorithm to all three datasets. The
> > > > algorithm identified $k_1=5$, $k_2=0$, $k_3=6$ for the recipe dataset,
> > > > $k_1=7$, $k_2=0$, $k_3=2$ for the patient profile dataset, and $k_1=2$,
> > > > $k_2=7$, $k_3=8$ for the co-citation hypergraph. The error metrics for
> > > > the three datasets are presented in the tables below. Here, a-FED refers
> > > > to a variant of FED that we introduce in response to Reviewer Suk1; a
> > > > detailed description of this metric is provided in Appendix B.5.
> > > >
> > > > |Method|$\Delta\mathcal{H}_v\downarrow$|$\Delta\mathcal{X}_m\downarrow$|$\Delta\mathcal{X}_v\downarrow$|FED$\downarrow$|a-FED$\downarrow$|
> > > > |---|---|---|---|---|---|
> > > > |ReLaSH-(5,0,2)|1.978|2.236|0.894|0.293|0.356|
> > > > |ReLaSH$_c$-(5,0,2)|7.504|2.236|0.894|0.182|0.248|
> > > > |ReLaSH-(5,0,6)|2.129|2.174|**0.820**|**0.003**|**0.048**|
> > > > |ReLaSH$_c$-(5,0,6)|3.583|2.174|**0.820**|0.191|0.258|
> > > > |ReLaSH-(5,0,16)|2.355|**1.533**|1.112|0.766|0.847|
> > > > |ReLaSH$_c$-(5,0,16)|**1.847**|**1.533**|1.112|0.180|0.255|
> > > >
> > > > Table 5: Recipe generation results. Scales of $\Delta\mathcal{H}_v$, $\Delta\mathcal{X}_m$, $\Delta\mathcal{X}_v$, FED, a-FED are $10^{-3},10^{-2},10^{-2},10^{-1},10^{-1}$.
> > > >
> > > > |Method|$\Delta\mathcal{H}_v\downarrow$|$\Delta\mathcal{X}_m\downarrow$|$\Delta\mathcal{X}_v\downarrow$|FED$\downarrow$|a-FED$\downarrow$|
> > > > |---|---|---|---|---|---|
> > > > |ReLaSH-(2,2,2)|1.996|**8.578**|1.887|1.246|0.931|
> > > > |ReLaSH$_c$-(2,2,2)|3.890|**8.578**|1.887|5.481|1.935|
> > > > |ReLaSH-(2,7,8)|1.914|8.817|**1.886**|1.060|0.791|
> > > > |ReLaSH$_c$-(2,7,8)|2.772|8.817|**1.886**|**0.947**|**0.706**|
> > > > |ReLaSH-(8,8,8)|**1.626**|8.608|1.887|1.454|0.871|
> > > > |ReLaSH$_c$-(8,8,8)|2.816|8.608|1.887|6.451|2.042|
> > > >
> > > > Table 6: Co-citation hypergraph generation results. Scales of $\Delta\mathcal{H}_v$, $\Delta \mathcal{X}_m$, $\Delta \mathcal{X}_v$, FED are $10^{-3},10^{-2},10^{-1},10^{-1}$.
> > > >
> > > > |Method|$\Delta\mathcal{H}_v\downarrow$|$\Delta\mathcal{X}_m\downarrow$|$\Delta\mathcal{X}_v\downarrow$|FED$\downarrow$|a-FED$\downarrow$|
> > > > |---|---|---|---|---|---|
> > > > |ReLaSH-(7,0,2)|3.260|**2.989**|**1.435**|0.532|1.084|
> > > > |ReLaSH$_c$-(7,0,2)|27.794|**2.989**|**1.435**|**0.013**|**0.193**|
> > > > |ReLaSH-(7,0,16)|**2.624**|3.681|1.655|11.738|2.047|
> > > > |ReLaSH$_c$-(7,0,16)|6.230|3.681|1.655|9.049|1.574|
> > > >
> > > > Table 7: Patient profile generation results. Scales of $\Delta \mathcal{H}_v$, $\Delta \mathcal{X}_m$, $\Delta \mathcal{X}_v$, FED, a-FED are $10^{-4},10^{-3},10^{-1},10^{-2},10^{-1}$.
> > > >
> > > > For the patient profile dataset, our experiments already include results for ReLaSH-$(7,0,2)$ and ReLaSHc-$(7,0,2)$, with ReLaSHc-$(7,0,2)$ consistently achieving the best performance across all error metrics, which aligns with the chosen latent dimension. For the co-citation hypergraph generation task, ReLaSH-$(2,7,8)$ generally outperforms the other ReLaSH model variants. Accordingly, we have updated the generation samples presented in the paper to showcase synthetic papers generated by ReLaSH-$(2,7,8)$. Regarding the results for the recipe dataset, both ReLaSH-$(5,0,6)$ and ReLaSHc-$(5,0,16)$ demonstrate comparable performance, excelling in different aspects of the error metrics. This suggests that, in real-world applications, it is beneficial to apply the latent dimension selection algorithm as well as experiment with various latent dimension choices for ReLaSH. The synthetic dataset with the lowest error metrics can also be selected as optimal, since real-world data may lack a clear ground truth for latent dimensions. Nevertheless, our model demonstrates robustness, as different choices of latent dimensions can still produce high-quality synthetic results.

---

> ### Comment · Reviewer_hHTL · 2025-11-24
>
> Thanks for the detailed rebuttal.
> I am largely convinced, and I hope the additional discussions and results here can be included in the revised paper.
> I have adjusted my rating accordingly (6 -> 8).

---

> ### Author Response · Authors · 2025-11-24
>
> Thank you for kindly raising our score, and thank you again for your many insightful comments! We have incorporated most of the additional discussions and results into our manuscript and are still working on them; meanwhile, we will keep polishing and checking details before the final deadline. Please do not hesitate to contact us if you have any further questions.
>
> Best,

---

### Official Review · Reviewer_MqMV · 2025-10-26

**Soundness:** 2
**Presentation:** 2
**Contribution:** 2
**Rating:** 4
**Confidence:** 3

**Summary:**

The authors present a framework for hyperedge-attributed hypergraph generation.
They combine various generation techniques that were successful in diverse domains to achive this goal, and demonstrated its effectiveness in certain hypergraph datasets, compared to several baseline methods.

**Strengths:**

S1. The paper is grounded on several theoretical results.

S2. The authors leverage diverse datasets and put the results in the Appendix.

**Weaknesses:**

**W1. [Key novelty]** While the authors present various theoretical results, I cannot understand what is the key novelty of the work. Is this the first work that generates hyperedge attributes? If so, in hypergraphs, changing node to hyperedge, and hyperedge to node, is trivial in many cases (e.g., if node: author and hyperedge: a paper's author list, then node: paper and hyperedge: a set of publications published by an author). There are certain works on hypergraph generative models with features [1], and authors should compare their work with such methods.

**W2. [Implication of theoretical results]** Could the authors present more intuitive results regarding their theories? For instance, Lines 354 - 355 state: "Theorem 3 implies that if m ≍ n ≍ p, the error introduced by estimating (Z, B, α, γ) is asymptotically negligible ..." Does this mean that the model can capture the ground-truth data distributions under certain assumptions?

**W3. [Comparison with tabular generative models]** To my knowledge, the MIMIC 3 dataset is often processed as tabular data. In my opinion, the effectiveness of the proposed method should be compared with the SOTA tabular data generative methods.

**W4. [Dataset construction]** I think the description regarding how the datasets are transformed into hypergraphs is not sufficient. Could the authors elaborate on it?

[1] Attributed Hypergraph Generation with Realistic Interplay Between Structure and Attributes

**Questions:**

See Weakness

---

> ### Author Response · Authors · 2025-11-21
> **Author Response to Reviewer MqMV**
>
> Thank you for for taking the time and effort to review our work, and we truly appreciate your recognition of our theoretical results and your affirmation of the diversity of datasets we apply our methods on. Below, we address each of your comments in sequence.
>
> **W1. Regarding the novelty of our method.**
>
> Thank you for your constructive comment on highlighting the novelty and positioning of our work in the literature. In our revision, we have added a detailed comparison and included the reference [1] you mentioned, along with two methods on modeling attributed hypergraphs [2, 3] and several state-of-the-art tabular data generation methods highlighted in a recent survey [4].
> Below, we discuss these related works in sequence.
>
> For the NOAH method proposed in [1], our goal and methodology are fundamentally different. NOAH utilizes node attributes as conditional information to guide the formation of hyperedges, while both the node set and its attributes are fixed. The model generates only the hyperedge structure to reflect the observed structure–attribute relationship. Swapping the roles of hyperlinks and nodes, applying NOAH to generate synthetic hypergraphs, and then swapping back does not produce new synthetic attributes.
> In contrast, our work targets the explicit generation of realistic hyperedges and their attributes, a substantially different generative objective. We directly model and generate the attribute information associated with hyperedges while preserving the node set. This enables us to generate new hypergraphs whose hyperedge and attribute distributions closely resemble those of the input data, thereby addressing real-world generation tasks such as patient profile generation, as demonstrated in our experiments. For instance, in the patient profile dataset, the disease types (represented as nodes) are fixed, as they correspond to specific medical conditions. It would be unreasonable to interchange these into hyperedges and then attempt to generate additional diseases, since the disease types in our application scenario should remain unchanged.
>
> Reference [2] proposes a hierarchical coarsening-expansion framework to jointly generate hypergraph topology and node attributes; in generation, their method produces new nodes, node attributes, and hyperlinks, whereas our goal is to generate new hyperlinks and hyperlink attributes on a fixed set of nodes (for example, generating new combinations of symptoms as hyperlinks over a fixed symptom set and corresponding attributes to form new medical records).
>
> Reference [3] introduces node attributes into a community detection framework for modeling higher-order interactions in hypergraphs. They study structure and inference in hypergraphs with node attributes, where the primary goal is to understand the observed hypergraph for tasks such as community detection and link prediction, rather than generating synthetic data.
>
> As you noted in W3, tabular data generative methods can be applied to our task, as they handle mixed-type tabular data. A recent survey [4] summarizes advances in tabular data generation. However, these methods do not scale well and ignore the special structural properties of hypergraphs. We have conducted empirical comparisons with these methods; please refer to our response to your W3 for more details.
>
> Moreover, most existing generative modeling approaches for such tasks lack theoretical guarantees. A principled, theoretically grounded generative modeling framework for hypergraphs with hyperlink attributes is therefore greatly needed, and the proposed ReLaSH framework aims to fill this gap.
>
> We have incorporated all these references into the Introduction section of the paper and have conducted empirical comparisons with state-of-the-art tabular data generative methods in the numerical study.
>
> **W.2 Regarding the implications of theoretical results.**
>
> Thank you for your advice on presenting more intuitive results regarding our theoretical guarantees. In Theorem 3 you mentioned, we have further written:
>
> > Consequently, when $m \asymp n \asymp p$, we have  $
> > n^{-1}\Delta_{(\mathcal{Z}_n,B,\alpha,\gamma)\text{-estimation}}
> > = O_p(\log n / n),$
> > thus the error in final generative performance from estimating
> > $\mathcal{Z}_n,B,\alpha,\gamma$ shrinks as fast as $\log n \cdot n^{-1}$.

---

> > ### Author Response · Authors · 2025-11-21
> >
> > We have also added discussions after each assumption in the manuscript and we quote below.
> >
> > After Assumption 1, we have
> > > **Assumption 1 (Hyperlink sparsity).** As $m,n,p \to \infty$,  $\exp(\bar{\alpha}_{m,n}) \gtrsim \log(m \vee n)/(m \wedge n)$.
> > >
> > > When $m \asymp n$, this sparsity scaling is consistent with the sufficient order in Proposition 2.2 and the necessary order in Proposition 2.1 of Wu et al. (2024), up to a logarithmic factor.
> >
> > After Assumption 2, we have
> > > Assumption 2 ensures that the latent embedding space is well-conditioned and identifiable by requiring a non-degenerate, spectrally separated covariance structure, while also guaranteeing that the empirical moments of the embeddings concentrate around their population counterparts to enable consistent estimation. Note that $\exp(\bar{\alpha}_{m,n}) =o(1)$ as $m,n \to \infty$ under hyperlink sparsity. This concentration requirement is easily satisfied and is weaker than the concentration results available for many popular multivariate distributions [5] when $m \asymp n \asymp p$, which cover a wide range of real data scenarios.
> >
> > After Assumption 3, we have
> > > Assumption 3 requires that the learned score network approximates the true score function consistently in an $L_2$ sense. The boundedness assumption of its second moment is automatically satisfied due to the constraint in the embedding algorithm. Together with the $L$-Lipschitz condition, these requirements provide the regularity needed to control sampling and approximation errors in the diffusion dynamics as follows.
> >
> > These assumptions specify under which model conditions our theory applies and ensure that our methods consistently generate synthetic hyperlinks with attributes that are close to the true distribution of hyperlinks with attributes. Of course, it is intractable to verify whether real data are exactly generated from the probabilistic models we study. However, our model class is general and covers a wide range of distributions, and our empirical results demonstrate the strong performance of ReLaSH on real tasks.
> >
> > **W3. Regarding comparisons with other tabular generative models.**
> >
> > **Response.** Thank you for your suggestion to compare our method with the SOTA tabular data generative methods. We have conducted additional empirical studies with more baseline methods and present our results below.
> >
> > As summarized in Table 1 of [4], most state-of-the-art tabular data synthesis models were previously tested on datasets with fewer than 100 features, and their training and sampling processes are generally time- and memory-intensive. Nevertheless, many of these models can handle mixed data types, which apply on our datasets. For our baseline comparisons, we include ForestDiffusion [6], TabPFGen [7], CTAB-GAN [8], CTAB-GAN+ [9], and CTGAN [10], all using their official implementations.
> >
> > To assess the computational resources required, we first trained each method on the recipe hypergraph, which contains 205 features. TabPFGen and CTGAN require approximately twice the runtime of ReLaSH, while CTAB-GAN requires about four times as long. CTAB-GAN+ and ForestDiffusion are the most computationally demanding, requiring several hours for training and generation. The error metrics for these experiments are shown below.
> >
> > |Method|$\Delta\mathcal{H}_v\downarrow$|$\Delta\mathcal{X}_m\downarrow$|$\Delta\mathcal{X}_v\downarrow$|FED$\downarrow$|
> > |---|---|---|---|---|
> > |ReLaSH-(5,0,2)|1.978|2.236|0.894|0.293|
> > |ReLaSH$_c$-(5,0,2)|7.504|2.236|0.894|0.182|
> > |ReLaSH-(5,0,6)|2.129|2.174|0.820|0.003|
> > |ReLaSH$_c$-(5,0,6)|3.583|2.174|0.820|0.191|
> > |ReLaSH-(5,0,16)|2.355|1.533|1.112|0.766|
> > |ReLaSH$_c$-(5,0,16)|1.847|1.533|1.112|0.180|
> > |Gau-Diff|2.375|2.154|4.256|0.802|
> > |RealNVP|2.484|1.146|3.562|0.909|
> > |WGAN|2.208|21.428|1.351|0.907|
> > |VAE|21.587|9.883|5.180|11.553|
> > |CTGAN|2.519|28.799|4.983|0.847|
> > |ForestDiffusion|1.886|8.211|2.073|0.848|
> > |TabPFGen|1.565|1.915|1.205|0.297|
> > |CTAB-GAN|2.552|19.367|3.858|0.925|
> > |CTAB-GAN+|2.488|8.330|3.821|0.898|
> >
> > Table 1: Recipe generation results. Scales of $\Delta\mathcal{H}_v$, $\Delta\mathcal{X}_m$, $\Delta\mathcal{X}_v$, FED are $10^{-3},10^{-2},10^{-1},10^{-1}$.
> >
> > Among the models, TabPFGen has good performance and is comparable to ReLaSH/ReLaSH$_c$, while the other tabular generative models perform similarly to the standard WGAN baseline. However, according to the official implementation, TabPFGen is limited to handling tabular datasets with at most 500 features, whereas both our patient profile and co-citationship datasets exceed 2,000 dimensions—demonstrating the advantages of ReLaSH  in terms of scalability. All the other reviewed baseline models fail to complete training on these larger datasets due to timeout or out-of-memory errors, highlighting the efficiency of ReLaSH for hypergraph data generation tasks.

---

> > > ### Author Response · Authors · 2025-11-21
> > >
> > > **W4. Regarding the construction of our dataset.**
> > >
> > > Thank you for your attention to the details of hypergraph data construction. For a step-by-step description of how we generated the hypergraph data from the original raw dataset, we refer you to Appendix B.4. We have also added new details to this section to further clarify the process.
> > >
> > > [1] Gailhard, Dorian, et al. "Feature-aware Hypergraph Generation via Next-Scale Prediction." arXiv preprint arXiv:2506.01467 (2025).
> > >
> > > [2] Chun, Jaewan, et al. "Attributed Hypergraph Generation with Realistic Interplay Between Structure and Attributes." arXiv preprint arXiv:2509.21838 (2025).
> > >
> > > [3] Badalyan, Anna, Nicolò Ruggeri, and Caterina De Bacco. "Structure and inference in hypergraphs with node attributes." Nature Communications 15.1 (2024): 7073.
> > >
> > > [4] Kim, Dong-Keon, et al. "Generative models for tabular data: A review." Journal of Mechanical Science and Technology 38.9 (2024): 4989-5005.
> > >
> > > [5] Vershynin, Roman. *High-Dimensional Probability: An Introduction with Applications in Data Science*. Cambridge University Press, 2018.
> > >
> > > [6] Jolicoeur-Martineau, Alexia, Kilian Fatras, and Tal Kachman. “Generating and imputing tabular data via diffusion and flow-based gradient-boosted trees.” *AISTATS* (PMLR), 2024.
> > >
> > > [7] Ma, Junwei, et al. “TabPFGen – Tabular Data Generation with TabPFN.” arXiv:2406.05216 (2024).
> > >
> > > [8] Zhao, Zilong, et al. “CTAB-GAN: Effective table data synthesizing.” *Asian Conference on Machine Learning* (PMLR), 2021.
> > >
> > > [9] Zhao, Zilong, et al. “CTAB-GAN+: Enhancing tabular data synthesis.” *Frontiers in Big Data* 6 (2024): 1296508.
> > >
> > > [10] Xu, Lei, et al. “Modeling tabular data using conditional GAN.” *NeurIPS* 32 (2019).

---

> ### Comment · Reviewer_MqMV · 2025-11-21
>
> Dear Authors
>
> Thank you for your detailed responses.
> My main concerns have beed addressed, and therefore, I have increased my score 4 $\rightarrow$ 6.
>
> Please clarify these points in the revised manuscript.

---

> ### Author Response · Authors · 2025-11-21
>
> Dear Reviewer MqMV,
>
> Thank you for your timely response and for kindly raising our score!
>
> We have updated our manuscript and will continue checking and polishing the details before the final deadline. Please do not hesitate to contact us if you have any further questions.
>
> Best,

---

### Official Review · Reviewer_Suk1 · 2025-10-31

**Soundness:** 3
**Presentation:** 3
**Contribution:** 3
**Rating:** 6
**Confidence:** 3

**Summary:**

The paper introduces ReLaSH, a three-stage framework for hypergraph generation. (1) A likelihood-based joint embedding maps the observed hypergraph into a low-dimensional latent space. (2) A distribution-free generator is trained in this space to model and reconstruct the latent distribution. (3) The trained likelihood model decodes sampled latents into hyperlinks and associated attributes to synthesize new hypergraphs. On the theory side, the authors argue, and provide supporting analysis, that operating in the latent space mitigates the curse of ambient dimensionality. Empirically, experiments on co-citation, recipe, and symptom co-occurrence hypergraphs show consistent performance across domains.

**Strengths:**

1. The paper targets hypergraph generation with hyperlink attributes, going beyond structure-only models. The likelihood-based joint formulation explicitly links structure and attributes, which is a well-scoped and meaningful contribution.

2.  Decoupling (i) likelihood-based joint embedding, (ii) distribution-free latent modeling, and (iii) likelihood decoding is clean and practical. Each component is swappable (e.g., alternative generators or attribute heads), which favors reproducibility and future extensions.

3.  The analysis provides an error decomposition and argues why operating in a low-dimensional latent space can mitigate the curse of ambient dimensionality. This gives readers an interpretable lever for understanding where performance gains originate.

4. Experiments on co-citation, recipe, and symptom co-occurrence hypergraphs indicate consistent improvements across heterogeneous domains, suggesting the approach is not tailored to a single benchmark and generalizes reasonably well.

**Weaknesses:**

1. Your FED metric relies on an embedding machine $E^T$ trained on the full population: you first embed the entire population hypergraph to obtain reference parameters, then perform an 80/20 split to choose $k$ by minimizing train/test FID, and finally reuse this $E^T$ to embed real and generated samples for FED. This makes the evaluator and evaluated models non-independent and is a threat to validity. Could you add a fully independent evaluation, for example training $E^T$ on a disjoint subset of the population or using a fixed external encoder such as a frozen autoencoder or a random orthogonal projection to compute FED, and compare the results and sensitivity with your current setup?

2. In the real-data experiments, ReLaSH/ReLaSHc use Forest Diffusion, while Gau-Diff uses a standard diffusion. This inconsistency in score learners may confound the comparison. Could you provide a small matched-architecture ablation?

3. In B.6.1, you select the latent dimension $k$ by minimizing the train/test embedding FID. Is there a clear theoretical connection (e.g., consistency or error bounds) between this selection criterion and your generative likelihood? If the data deviate from the model assumptions, is this criterion still reliable? Could you provide a sensitivity curve (FED, reconstruction error, and downstream metrics vs. $k$) to show that the selected $k$ aligns with overall performance?

**Questions:**

See Weaknesses

---

> ### Author Response · Authors · 2025-11-21
> **Author Response to Reviewer Suk1**
>
> Thank you for taking the time and effort to review our work, and for your positive feedback on our algorithm and theory. Below, we address each of your questions one by one.
>
> **W1. Regarding the independent evaluation of FED.**
>
> **Response.** Thank you for your suggestion to conduct independent evaluations. We have followed your advice and conducted study by splitting the training dataset from the overall population and using the remaining data to train the embedding machine $E^T$, thereby ensuring that the evaluator and evaluated models are independent. In our experiments, this data splitting does not change the final selected rank of the embedding machine, and the new FED results (denoted as adjusted FED, a-FED) are shown below.
>
> **Co-citation hypergraph**
>
> |Method|FED|a-FED|
> |---|---|---|
> |ReLaSH-(2,2,2)|1.246|0.931|
> |ReLaSH\_c-(2,2,2)|5.481|1.935|
> |ReLaSH-(2,7,8)|1.060|0.791|
> |ReLaSH\_c-(2,7,8)|0.947|0.706|
> |ReLaSH-(8,8,8)|1.454|0.871|
> |ReLaSH\_c-(8,8,8)|6.451|2.042|
> |Gau-Diff|5.060|1.503|
> |RealNVP|3.948|1.256|
> |WGAN|1.253|0.700|
> |VAE|1.358|0.904|
>
> Table 1: Co-citation hypergraph generation. Scale of FED is $10^{-1}$.
>
> **Recipe hypergraph**
>
> |Method|FED|a-FED|
> |---|---|---|
> |ReLaSH-(5,0,2)|0.293|0.356|
> |ReLaSH\_c-(5,0,2)|0.182|0.248|
> |ReLaSH-(5,0,6)|0.003|0.048|
> |ReLaSH\_c-(5,0,6)|0.191|0.258|
> |ReLaSH-(5,0,16)|0.766|0.847|
> |ReLaSH\_c-(5,0,16)|0.180|0.255|
> |Gau-Diff|0.802|0.828|
> |RealNVP|0.909|0.997|
> |WGAN|0.907|0.928|
> |VAE|11.553|10.285|
> |CTGAN|0.847|0.865|
> |TabPFGen|0.297|0.303|
> |ForestDiffusion|0.848|0.884|
> |CTAB-GAN|0.925|0.947|
> |CTAB-GAN+|0.898|0.902|
>
> Table 2: Recipe generation. Scales of FED and a-FED are both $10^{-1}$.
>
> **Patient profile hypergraph**
>
> |Method|FED|a-FED|
> |---|---|---|
> |ReLaSH-(7,0,2)|0.532|1.084|
> |ReLaSH\_c-(7,0,2)|0.013|0.193|
> |ReLaSH-(7,0,16)|11.739|2.047|
> |ReLaSH\_c-(7,0,16)|9.049|1.574|
> |Gau-Diff|39.731|4.387|
> |RealNVP|27.685|2.843|
> |WGAN|21.053|2.654|
> |VAE|9.374|1.376|
>
> Table 3: Patient profile generation. Scales of FED and a-FED are $10^{-2}$ and $10^{-1}$, respectively.
>
> For the patient profile and recipe datasets, the FED values under this independent setting are very similar to those in the original version, since the training sets are constructed by random sampling from the population. For the co-citation dataset, the variation in FED is slightly larger because we construct the training hypergraph using the top $m = 2000$ most cited papers and compare with the top $5000$ most cited papers as population. Nevertheless, our method consistently outperforms the baseline methods under these independent evaluation settings. We also observe that, in general, a-FED is slightly higher than the original FED, as reducing the data available to train the embedding machine naturally increases the error.
>
> The idea of evaluating synthetic data via a pre-trained embedding machine $E^T$ is inspired by the Inception Score and Fréchet Inception Distance, both of which use an Inception model pre-trained on a large subset of ImageNet to assess image generative models [2]. While using a Fréchet-style distance may introduce some dependency between the evaluator and the generator, this effect is mitigated when the training set is randomly selected. In hypergraph settings, hyperlink sparsity can limit the effectiveness of splitting the dataset into disjoint training and testing sets. Using a larger training set that includes the data used to train the generative models for the embedding machine may introduce some bias in evalution, but it also significantly improves its ability to capture the characteristics of the full population. We have added more discussion and details on these points in Appendix B.5.

---

> ### Author Response · Authors · 2025-11-21
>
> **W2. Regarding the confounding between ForestDiffusion and Gau-Diff.**
>
> **Response.** Thank you for your question regarding inconsistency between the choice of ForestDiffusion and Gau-Diff. To address this concern, we conducted two additional experiments on all three real-world datasets: (1) applying Gau-Diff to the joint latent space reconstruction step in ReLaSH/ReLaSH\_c, and (2) applying ForestDiffusion directly to the entire dataset. The error metrics for these experiments are presented below. In the tables, “(S.D.)” indicates that standard Gaussian diffusion was used for the joint latent space reconstruction step; otherwise, ForestDiffusion was employed.
>
> **Recipe generation task**
>
> |Method|$\Delta \mathcal{H}_v\downarrow$|$\Delta\mathcal{X}_m\downarrow$|$\Delta \mathcal{X}_v\downarrow$|FED$\downarrow$|
> |---|---|---|---|---|
> |ReLaSH-(5,0,16)|2.355|**1.533**|**1.112**|0.766|
> |ReLaSH\_c-(5,0,16)|**1.847**|**1.533**|**1.112**|**0.180**|
> |ReLaSH-(5,0,16) (S.D.)|2.067|1.832|2.854|0.659|
> |ReLaSH\_c-(5,0,16) (S.D.)|2.462|1.832|2.854|0.473|
> |Gau-Diff|2.375|2.154|4.256|0.802|
> |ForestDiffusion|1.886|8.211|2.073|0.848|
>
> Scales of $\Delta \mathcal{H}_v$, $\Delta\mathcal{X}_m$, $\Delta \mathcal{X}_v$, FED are $10^{-3}$, $10^{-2}$, $10^{-2}$, $10^{-1}$.
>
> **Co-citation hypergraph task**
>
> |Method|$\Delta \mathcal{H}_v\downarrow$|$\Delta\mathcal{X}_m\downarrow$|$\Delta \mathcal{X}_v\downarrow$|FED$\downarrow$|
> |---|---|---|---|---|
> |ReLaSH-(8,8,8)|**1.626**|**8.608**|1.887|**1.454**|
> |ReLaSH\_c-(8,8,8)|2.816|**8.608**|1.887|6.451|
> |ReLaSH-(8,8,8) (S.D.)|1.925|11.836|1.889|2.648|
> |ReLaSH\_c-(8,8,8) (S.D.)|2.816|11.836|1.889|6.027|
> |Gau-Diff|1.672|10.016|**1.824**|5.060|
> |ForestDiffusion|OOM|OOM|OOM|OOM|
>
> Scales of $\Delta \mathcal{H}_v$, $\Delta\mathcal{X}_m$, $\Delta \mathcal{X}_v$, FED are $10^{-3}$, $10^{-2}$, $10^{-1}$, $10^{-1}$.
>
> **Patient profile generation task**
>
> |Method|$\Delta \mathcal{H}_v\downarrow$|$\Delta\mathcal{X}_m\downarrow$|$\Delta \mathcal{X}_v\downarrow$|FED$\downarrow$|
> |---|---|---|---|---|
> |ReLaSH-(7,0,2)|**3.260**|2.989|1.435|0.532|
> |ReLaSH\_c-(7,0,2)|27.794|2.989|1.435|**0.013**|
> |ReLaSH-(7,0,2) (S.D.)|4.085|**2.849**|**1.385**|0.835|
> |ReLaSH\_c-(7,0,2) (S.D.)|17.266|**2.849**|**1.385**|0.548|
> |Gau-Diff|4.268|3.497|1.719|39.731|
> |ForestDiffusion|OOM|OOM|OOM|OOM|
>
> Scales of $\Delta \mathcal{H}_v$, $\Delta\mathcal{X}_m$, $\Delta \mathcal{X}_v$, FED are $10^{-4}$, $10^{-3}$, $10^{-1}$, $10^{-2}$.
>
> The errors on the three generation tasks generally increase slightly when the distribution-free generator used in the latent space reconstruction step is standard Gaussian diffusion rather than ForestDiffusion. This is expected, as we observe empirically that ForestDiffusion achieves lower error in reconstructing the joint latent space. Nonetheless, the ReLaSH pipeline still outperforms directly applying Gau-diff/ForestDiffusion to the whole dataset, again highlighting the effectiveness of the ReLaSH pipeline.
>
> It is also important to note that ForestDiffusion encounters out-of-memory (OOM) errors on the patient profile and co-citation datasets when applied directly to the full tabular data due to their high dimensionality. In contrast, within ReLaSH, both standard diffusion and ForestDiffusion exhibit comparable, fast runtimes. ForestDiffusion is optimized for low-dimensional tabular data, with the largest feature size tested in [1] being 90, whereas the patient profile dataset has 2,237 features and the co-citation dataset has 3,039. By using ForestDiffusion in the joint latent space, we effectively leverage its advantages for reconstructing moderate-dimensional distributions with better accuracy.

---

> > ### Author Response · Authors · 2025-11-21
> >
> > **W3. Regarding the embedding dimensions.**
> >
> > **Response.** We appreciate your comments on the embedding dimensions. In our paper there are two types of latent dimensions: (1) the embedding dimension of the embedding machine used to calculate FED, and (2) the embedding dimensions in ReLaSH treated as tuning parameters. Our response to W1 addresses (1); here we focus on (2).
> >
> > Selecting dimensions in the joint latent space is more challenging than in standard low-rank models, since we must determine not only the total size $(k_1 + k_2 + k_3)$ but also each component. We therefore propose a data-driven pipeline  which we call *hunt then trim* (HTT).
> >
> > HTT consists of three steps. In the first and second steps, it hunts for the latent space dimensions that generate the hypergraph, and then the dimensions that are orthogonal to those but drive attribute generation. In the third step, it then trims the dimensions that drive hypergraph formation but (almost) orthogonal to the attributes. Below is a step-by-step illustration.
> >
> > - **Step 1.** Select $k_2 + k_3$ by cross-validation on the hypergraph.
> >   For each chosen dimension $k_{23} = k_2 + k_3$, we split the
> >   population hypergraph into a training set and a testing set, with
> >   80% of the nodes and attributes used for training and the remaining
> >   20% for testing. We then perform the embedding procedure on both the
> >   training and testing sets to estimate the hypergraph embeddings on the
> >   training set (denote them by $U_{\mathrm{train}}^{(k_{23})}$) and on
> >   the test set (denote them by $U_{\mathrm{test}}^{(k_{23})}$).
> >   Next, we calculate the Fréchet distance FD$(k_{23})$ between
> >   $U_{\mathrm{train}}^{(k_{23})}$ and $U_{\mathrm{test}}^{(k_{23})}$,
> >   and choose $k_2 + k_3$ as the value of $k_{23}$ that minimizes
> >   FD$(k_{23})$. With the selected $k_2 + k_3$, we obtain
> >   $\{\hat{u} _ j^{(23)}\}_{j \in [m]}$ according to the embedding
> >   procedure.
> >
> > - **Step 2.** Let $X = (x_1,\dots,x_m)^\top$ and
> >   $U^{(23)} = (u^{(23)} _ 1, \dots, u^{(23)} _ m)^\top$. Regress $X$ on
> >   $U^{(23)}$ via the linear model
> >   $X = 1_n \mu^\top + U^{(23)} B^\top + E$ with latent components $B$, and obtain fitted values
> >   $\hat{X}, \hat{U}, \hat{\mu}, \hat{B}$ and residual
> >   $\hat{E} = X - \hat{X}$. Consider the testing problem
> >   $H_0: k_1 = l$ versus $H_1: l + 1 \le k_1 \le k_{\max}$ and the test
> >   statistic
> >   $r(l) = (\phi_{l+1} - \phi_{k_{\max}+1}) / \phi_{k_{\max}+2}$,
> >   where $\phi_l$ is the $l$-th largest eigenvalue of
> >   $\operatorname{Cov}(\hat{E})$.
> >
> >
> >   Under regular assumptions, the following conclusions hold:
> >
> >   1. Under $H_0$,
> >      $r(l) \to (x_l - x_{k_{\max}+1}) / (x_{k_{\max}+1} - x_{k_{\max}+2})$,
> >      where $x_1, \ldots, x_{k_{\max}+2}$ jointly follow the Tracy–Widom
> >      distribution induced by the Gaussian orthogonal ensemble.
> >
> >   2. Under $H_1$, $r(l) \to \infty$, so asymptotically the testing
> >      procedure achieves full power.
> >
> >   Consequently, for any $l \ge 0$, we can compare the observed value
> >   of $r(l)$ with that of a value simulated from the joint
> >   Tracy–Widom distribution [2]. We set $k_{\max}$ large enough and
> >   determine $k_1$ using a sequential testing procedure. Starting with
> >   $l = 0$, for each $0 \le l \le k_{\max}$, if the null hypothesis
> >   $H_0: k_1 = l$ is rejected, we further test the null hypothesis
> >   $H_0: k_1 = l + 1$ until a null hypothesis is accepted. In the
> >   special case $l = 0$, the proposed test can be used to test the
> >   existence of $U^{(1)}$.
> >
> > -   Step 3. To distinguish $k_3$ from $k_1$ and $k_2$, that is, to
> >     identify $u^{(3)}$ separately from $u^{(23)}$, we frame the issue
> >     within the previously introduced regression problem. Specifically, we incorporate the
> >     potentially irrelevant factors $u^{(3)}$ into the attribute model.
> >     This allows us to determine $k_3$ by addressing the following
> >     multiple testing procedure: for each $1 \leq l \leq k_2 + k_3$, let
> >     $B_{\cdot, l}$ denote the $l$-th column of $B$.
> >     We then consider the following hypothesis testing problem:
> >
> >     $$H_0: B_{\cdot, l} = 0_{p \times 1} \; \text{versus }H_1: B_{\cdot, l} \neq 0_{p \times 1}$$
> >     with the test statistics
> >     $$s(l) = (2 \sum_{j=1}^p V_{jl}^2)^{-1/2} \sum_{j=1}^p (mB_{jl}^2 - V_{jl}),$$
> >     where
> >     $V_{jl}$ is  $(U^{(23)\top}U^{(23)})_{ll}^{-1}c_j$ and $c_j$  is the j,j-th entry of $(\mathrm{Cov}(\hat{{E}}))$.
> >     Under regular assumptions, the following conclusions hold:
> >
> >     1.  Under $H_0$, $s(l) \to \mathcal N(0,1)$.
> >
> >     2.  Under $H_1$, denote the number of non-zero entries in
> >         $B_{\cdot, l}$ as $p^{1-\beta}$. If
> >         $B_{\cdot, l}=\Omega(m^{-1/2} (\log p)^\alpha)$ holds
> >         uniformly with $\alpha >0$ and $0 < \beta <1/2$, then
> >         $s(l) \to \infty$. Therefore, asymptotically the testing
> >         procedure achieves full power.

---

> ### Author Response · Authors · 2025-11-21
>
> Consequently, for each $0 \leq l \leq k_2 + k_3$, we compare the
> observed value of $s(l)$ to the standard normal distribution. This
> allows us to infer whether each column of $U^{(23)}$ belongs to
> $U^{(2)}$ or $U^{(3)}$, thereby enabling simultaneous estimation of
> $U^{(2)}$, $U^{(3)}$, as well as the corresponding dimensions $k_2$
> and $k_3$.
>
> To demonstrate the effectiveness of our proposed latent dimension
> selection method, we conducted a simulation study with data size
> $n = p = 500$, $m=5000$ and true latent dimensions
> $k_1 = k_2 = k_3 = 4$. All embeddings and parameters were generated
> according to the simulation settings described in Section B.3 of the
> Appendix. We applied the HTT pipeline to the simulation data across 30
> repetitions, and the results are presented in the following table.
>
> |$k_1$|$k_2$|$k_3$|Freq|
> |---|---|---|---|
> |4|4|4|27|
> |4|3|4|1|
> |4|5|4|2|
>
> Table 4: Result of latent dimension selection across 30 repetitions.
>
> The results indicate that, in most cases, the HTT algorithm
> successfully recovers the true latent dimensions. We further apply this
> algorithm in our real data analyses. With the implementation details in
> Appendix B.4, we apply the HTT algorithm to all three datasets. The
> algorithm identified $k_1=5$, $k_2=0$, $k_3=6$ for the recipe dataset,
> $k_1=7$, $k_2=0$, $k_3=2$ for the patient profile dataset, and $k_1=2$,
> $k_2=7$, $k_3=8$ for the co-citation hypergraph. The error metrics for
> the three datasets are presented in the tables below. Here, a-FED refers
> to a variant of FED that we introduce in response to Reviewer Suk1; a
> detailed description of this metric is provided in Appendix B.5.
>
> |Method|$\Delta \mathcal{H}_v\downarrow$|$\Delta\mathcal{X}_m\downarrow$|$\Delta \mathcal{X}_v\downarrow$|FED$\downarrow$|a-FED$\downarrow$|
> |---|---|---|---|---|---|
> |ReLaSH-(5,0,2)|1.978|2.236|0.894|0.293|0.356|
> |ReLaSH$_c$-(5,0,2)|7.504|2.236|0.894|0.182|0.248|
> |ReLaSH-(5,0,6)|2.129|2.174|0.820|0.003|0.048|
> |ReLaSH$_c$-(5,0,6)|3.583|2.174|0.820|0.191|0.258|
> |ReLaSH-(5,0,16)|2.355|1.533|1.112|0.766|0.847|
> |ReLaSH$_c$-(5,0,16)|1.847|1.533|1.112|0.180|0.255|
>
> Table 5: Recipe generation results. Scales of $\Delta \mathcal{H}_v$, $\Delta \mathcal{X}_m$, $\Delta \mathcal{X}_v$, FED, a-FED are $10^{-3},10^{-2},10^{-2},10^{-1},10^_
>
> |Method|$\Delta \mathcal{H}_v\downarrow$|$\Delta\mathcal{X}_m\downarrow$|$\Delta \mathcal{X}_v\downarrow$|FED$\downarrow$|a-FED$\downarrow$|
> |---|---|---|---|---|---|
> |ReLaSH-(2,2,2)|1.996|8.578|1.887|1.246|0.931|
> |ReLaSH$_c$-(2,2,2)|3.890|8.578|1.887|5.481|1.935|
> |ReLaSH-(2,7,8)|1.914|8.817|1.886|1.060|0.791|
> |ReLaSH$_c$-(2,7,8)|2.772|8.817|1.886|0.947|0.706|
> |ReLaSH-(8,8,8)|1.626|8.608|1.887|1.454|0.871|
> |ReLaSH$_c$-(8,8,8)|2.816|8.608|1.887|6.451|2.042|
>
> Table 6: Co-citation hypergraph generation results. Scales of $\Delta \mathcal{H}_v$, $\Delta \mathcal{X}_m$, $\Delta \mathcal{X}_v$, FED are $10^{-3},10^{-2},10^{-1},10^{-1}$.
>
> |Method|$\Delta \mathcal{H}_v\downarrow$|$\Delta\mathcal{X}_m\downarrow$|$\Delta \mathcal{X}_v\downarrow$|FED$\downarrow$|a-FED$\downarrow$|
> |---|---|---|---|---|---|
> |ReLaSH-(7,0,2)|3.260|2.989|1.435|0.532|1.084|
> |ReLaSH$_c$-(7,0,2)|27.794|2.989|1.435|0.013|0.193|
> |ReLaSH-(7,0,16)|2.624|3.681|1.655|11.738|2.047|
> |ReLaSH$_c$-(7,0,16)|6.230|3.681|1.655|9.049|1.574|
>
> Table 7: Patient profile generation results. Scales of $\Delta \mathcal{H}_v$, $\Delta \mathcal{X}_m$, $\Delta \mathcal{X}_v$, FED, a-FED are $10^{-4},10^{-3},10^{-1},10^{-2},10^{-1}$.
>
> For more discussion on these results, we kindly refer you to the fourth block of our response to Reviewer ELf7 (the first reviewer)
>
> **References**
>
> [1] Jolicoeur-Martineau, Alexia, Kilian Fatras, and Tal Kachman. “Generating and imputing tabular data via diffusion and flow-based gradient-boosted trees.” *International Conference on Artificial Intelligence and Statistics (AISTATS)*. PMLR, 2024.
>
> [2] Szegedy, Christian, et al. “Rethinking the Inception architecture for computer vision.” *Proceedings of the IEEE Conference on Computer Vision and Pattern Recognition (CVPR)*, 2016.
>
> [3] Tracy, Craig A., and Harold Widom. “On orthogonal and symplectic matrix ensembles.” *Communications in Mathematical Physics* 177(3) (1996): 727–754.

---

### Official Review · Reviewer_ELf7 · 2025-11-03

**Soundness:** 2
**Presentation:** 2
**Contribution:** 3
**Rating:** 6
**Confidence:** 3

**Summary:**

This paper introduces ReLaSH, a generative framework for hypergraphs with hyperlink attributes. Its core contribution is a novel integration of a likelihood-based joint embedding model with a distribution-free latent space generator. The method first embeds the high-dimensional, sparse hypergraph and its attributes into a structured, low-dimensional latent space, which is then reconstructed and sampled from. This approach effectively circumvents the challenges of high-dimensional generative modeling, is supported by theoretical analysis, and demonstrates superior performance across multiple real-world tasks.

**Strengths:**

1. The proposed joint latent space reconstruction framework is highly innovative. By unifying the embeddings of both hypergraph structure (hyperedges) and attributes into a cohesive low-dimensional space, it effectively circumvents the challenges of direct high-dimensional generation.

2. The method is inherently domain-agnostic, as demonstrated by its seamless and successful application across highly heterogeneous scenarios—from synthetic medical records and recipes to co-citation graphs.

3.ReLaSH is fortified by solid theoretical contributions, including identifiability proofs and a decomposition of the generation error.

**Weaknesses:**

1.The paper proposes a novel latent space partitioning strategy but provides no principled or systematic method for selecting the dimensions k1, k2, and k3 for a given dataset. The experimental setup relies on an ad-hoc equal partitioning scheme or setting k2 = k3 = 0 for specific cases, which lacks justification.

2.The baseline methods compared in the experiments are not up-to-date, with the most recent being from 2020.

3.While the method introduces several key hyperparameters (e.g., k1/k2/k3, $\lambda$) and the appendix includes some empirical studies, a systematic sensitivity analysis is still missing. It remains unclear how the model performance varies with changes in these hyperparameters.

**Questions:**

1.Regarding the selection of k1, k2, and k3:
1.1. Could you provide ablations to demonstrate the effectiveness of the proposed tri-partite partitioning strategy compared to a simpler, unified latent space?
1.2. Please supplement experiments comparing the equal partitioning method with non-equal alternatives to show its (non-)optimality.
1.3. Could you offer practical guidance on how to determine these values for a new, unseen dataset?

2. To strengthen the experimental validation, it is crucial to include more recent and stronger baseline methods to ensure the timeliness and competitiveness of the reported results.

3.The paper would significantly benefit from a systematic sensitivity analysis of its key hyperparameters (e.g., k1/k2/k3, λ, diffusion steps). Please show how the performance metrics trend with variations in these parameters to better understand the model's robustness and tuning requirements.

---

> ### Author Response · Authors · 2025-11-21
> **Author Response to Reviewer ELf7**
>
> Thank you for taking the time and effort to review our paper. We
> appreciate your recognition of the theoretical results, the novelty of
> our method, and the generality of our algorithm. Below, we address
> your questions one by one.
>
> **Q1. Regarding the tri-partite strategy of latent dimensions and the
> selection of latent dimensions.**
>
> We appreciate your question on this aspect. Below are our point-by-point
> responses to your three questions.
>
> **1.1. Question:** "Could you provide ablations to demonstrate the
> effectiveness of the proposed tri-partite partitioning strategy compared
> to a simpler, unified latent space?\"
>
> **Response:** Below, we first discuss the practical reasons for
> incorporating the joint embedding framework, of which a simpler unified
> latent space is a special case, and then present our numerical results
> in response to your question.
>
> -   A unified $k$-dimensional latent space is a specially case of the
>     tri-partite partition with the triple being $(0,k,0)$. The
>     tri-partite partition $(k_1,k_2,k_3)$ is motivated by the
>     factorization of the joint likelihood: $u^{(1)}$ (attribute-only),
>     $u^{(3)}$ (hyperlink-only), and $u^{(2)}$ (shared), allowing us to
>     separate but still couple the hypergraph and its attributes for
>     generative modeling. Using a unified $k$-dimensional latent space
>     would mix attribute-only and hyperlink-only variation and requires
>     stronger assumptions to recover interpretable components; in our
>     framework, the block structure gives clean orthogonality conditions
>     to distinguish latent effects driving the generation of hypergraph
>     and attributes, both separately and jointly.
>
> -   We have conducted ablation studies in which a unified latent space of total dimension
> $k = k_1 + k_2 + k_3$ is employed under the setup $m = n = p = 400$. In this study, the
> tri-partite model demonstrates superior performance, particularly on attribute-related
> metrics and FED, which are notably sensitive to latent-dimension selection, as will be
> discussed in the sensitivity analysis (see response to Q3). We have included these
> results in the appendix and provided a brief summary in the main text. Please refer to
> Section B.3 of the Appendix for more details. Below, we present some main
> results.
>
> | $k_1$ | $k_2$ | $k_3$ | $\Delta\mathcal{H}_m \downarrow$ | $\Delta\mathcal{H}_v \downarrow$ | $\Delta\mathcal{X}_m \downarrow$ | $\Delta\mathcal{X}_v \downarrow$ | FED $\downarrow$ | ReLaSH$_c$ $\Delta\mathcal{H}_v \downarrow$ | ReLaSH$_c$ FED $\downarrow$ |
> |:-----:|:-----:|:-----:|:-------------------------------:|:-------------------------------:|:----------------------:|:----------------------:|:----------:|:-----------------------------------------:|:----------------------------:|
> | 2     | 2     | 2     | 0.0201     | 0.0039    | 0.0511| 0.0265 | 0.1756  | 0.0436                              | 0.2965                       |
> | 0     | 6     | 0     | 0.0209                          | 0.0041                          | 0.0562                 | 0.0384         | 0.6432     | 0.0412                                | 0.5261                       |
>
>   Table 1: Result of setting a unified latent space of total dimension
>   $k=k _ 1+k _ 2+k _ 3$. Each value comes from the mean of 20 repetitions.
>
> | $k_1$ | $k_2$ | $k_3$ | $\Delta\mathcal{H}_m \downarrow$ | $\Delta\mathcal{H}_v \downarrow$ | $\Delta\mathcal{X}_m \downarrow$ | $\Delta\mathcal{X}_v \downarrow$ | FED $\downarrow$ | ReLaSH$_c$ $\Delta\mathcal{H}_v \downarrow$ | ReLaSH$_c$ FED $\downarrow$ |
> |:-----:|:-----:|:-----:|:-------------------------------:|:-------------------------------:|:----------------------:|:----------------------:|:----------:|:-----------------------------------------:|:----------------------------:|
> | 4     | 4     | 4     | 0.0175  | 0.0057 | 0.0514  | 0.0339   | 0.4392     | 0.0356  | 0.4158  |
> | 0     | 12    | 0     | 0.0181  | 0.0050  | 0.0566  | 0.0698 | 1.0164     | 0.0383  | 1.2767  |
>
>   Table 2: Result of setting a unified latent space of total dimension
>   $k=k_1+k_2+k_3$. Each value comes from the mean of 20 repetitions.
>
> | $k_1$ | $k_2$ | $k_3$ | $\Delta\mathcal{H}_m \downarrow$ | $\Delta\mathcal{H}_v \downarrow$ | $\Delta \mathcal{X}_m \downarrow$ | $\Delta \mathcal{X}_v \downarrow$ | FED $\downarrow$ | ReLaSH$_c$ $\Delta\mathcal{H}_v \downarrow$ | ReLaSH$_c$ FED $\downarrow$ |
> |:-----:|:-----:|:-----:|:-------------------------------:|:-------------------------------:|:----------------------:|:----------------------:|:----------:|:-----------------------------------------:|:----------------------------:|
> | 8| 8| 8 | 0.0180  | 0.0077 | 0.0554  | 0.0492  | 1.3333 | 0.0271| 1.3818 |
> | 0| 24 | 0 | 0.0184 | 0.0072     | 0.0625 | 0.0781  | 2.7461| 0.0304 | 2.9824  |
>
>  Table 3: Result of setting a unified latent space of total dimension
>   $k=k_1+k_2+k_3$. Each value comes from the mean of 20 repetitions.

---

> ### Author Response · Authors · 2025-11-21
>
> **1.2. Question.** "Please supplement experiments comparing the equal
> partitioning method with non-equal alternatives to show its
> (non-)optimality."
>
> **Response.** Thanks for the critique on equal partitioning of the joint
> latent space. In fact, our framework allows the three dimensions to be
> chosen flexibly; they can be equal or non-equal. Motivated by your
> question on data-driven selection of the joint latent space dimension in
> Question 1.3, we have developed such a selection pipeline and verified
> its effectiveness through simulation; interestingly, it selects
> non-equal $(k_1, k_2, k_3)$ in our empirical study on real data.
>
> **1.3. Question.** "Could you offer practical guidance on how to
> determine these values for a new, unseen dataset?"
>
> **Response.** We appreciate your thoughtful comments regarding the
> selection of latent space dimensions in the joint latent embedding
> space, which have motivated us to develop such a pipeline and have
> significantly improved our paper. This is indeed a more challenging task
> than dimension selection in traditional low-rank models, since we need
> to determine not only the total size of the latent embedding space
> $(k_1 + k_2 + k_3)$, but also each component. Below, we introduce our
> pipeline inspired by [1] and referred to as *"hunt then trim" (HTT)*, to achieve this
> goal, followed by a discussion and empirical justification of it.
>
> HTT consists of three steps. In the first and second steps, it hunts for
> the latent space dimensions that generate the hypergraph, and then the
> dimensions that are orthogonal to those but drive attribute generation.
> In the third step, it then trims the dimensions that drive hypergraph
> formation but (almost) orthogonal to the attributes. Below is a step-by-step
> illustration.
>
> - **Step 1.** Select $k_2 + k_3$ by cross-validation on the hypergraph.
>   For each chosen dimension $k_{23} = k_2 + k_3$, we split the
>   population hypergraph into a training set and a testing set, with
>   80% of the nodes and attributes used for training and the remaining
>   20% for testing. We then perform the embedding procedure on both the
>   training and testing sets to estimate the hypergraph embeddings on the
>   training set (denote them by $U_{\mathrm{train}}^{(k_{23})}$) and on
>   the test set (denote them by $U_{\mathrm{test}}^{(k_{23})}$).
>   Next, we calculate the Fréchet distance FD$(k_{23})$ between
>   $U_{\mathrm{train}}^{(k_{23})}$ and $U_{\mathrm{test}}^{(k_{23})}$,
>   and choose $k_2 + k_3$ as the value of $k_{23}$ that minimizes
>   FD$(k_{23})$. With the selected $k_2 + k_3$, we obtain
>   $\{\hat{u} _ j^{(23)}\}_{j \in [m]}$ according to the embedding
>   procedure.
>
> - **Step 2.** Let $X = (x_1,\dots,x_m)^\top$ and
>   $U^{(23)} = (u^{(23)} _ 1, \dots, u^{(23)} _ m)^\top$. Regress $X$ on
>   $U^{(23)}$ via the linear model
>   $X = 1_n \mu^\top + U^{(23)} B^\top + E$ with latent components $B$, and obtain fitted values
>   $\hat{X}, \hat{U}, \hat{\mu}, \hat{B}$ and residual
>   $\hat{E} = X - \hat{X}$. Consider the testing problem
>   $H_0: k_1 = l$ versus $H_1: l + 1 \le k_1 \le k_{\max}$ and the test
>   statistic
>   $r(l) = (\phi_{l+1} - \phi_{k_{\max}+1}) / \phi_{k_{\max}+2}$,
>   where $\phi_l$ is the $l$-th largest eigenvalue of
>   $\operatorname{Cov}(\hat{E})$.
>
>
>   Under regular assumptions, the following conclusions hold:
>
>   1. Under $H_0$,
>      $r(l) \to (x_l - x_{k_{\max}+1}) / (x_{k_{\max}+1} - x_{k_{\max}+2})$,
>      where $x_1, \ldots, x_{k_{\max}+2}$ jointly follow the Tracy–Widom
>      distribution induced by the Gaussian orthogonal ensemble.
>
>   2. Under $H_1$, $r(l) \to \infty$, so asymptotically the testing
>      procedure achieves full power.
>
>   Consequently, for any $l \ge 0$, we can compare the observed value
>   of $r(l)$ with that of a value simulated from the joint
>   Tracy–Widom distribution [2]. We set $k_{\max}$ large enough and
>   determine $k_1$ using a sequential testing procedure. Starting with
>   $l = 0$, for each $0 \le l \le k_{\max}$, if the null hypothesis
>   $H_0: k_1 = l$ is rejected, we further test the null hypothesis
>   $H_0: k_1 = l + 1$ until a null hypothesis is accepted. In the
>   special case $l = 0$, the proposed test can be used to test the
>   existence of $U^{(1)}$.

---

> ### Author Response · Authors · 2025-11-21
>
> -   Step 3. To distinguish $k_3$ from $k_1$ and $k_2$, that is, to
>     identify $u^{(3)}$ separately from $u^{(23)}$, we frame the issue
>     within the previously introduced regression problem. Specifically, we incorporate the
>     potentially irrelevant factors $u^{(3)}$ into the attribute model.
>     This allows us to determine $k_3$ by addressing the following
>     multiple testing procedure: for each $1 \leq l \leq k_2 + k_3$, let
>     $B_{\cdot, l}$ denote the $l$-th column of $B$.
>     We then consider the following hypothesis testing problem:
>
>     $$H_0: B_{\cdot, l} = 0_{p \times 1} \; \text{versus }H_1: B_{\cdot, l} \neq 0_{p \times 1}$$
>     with the test statistics
>     $$s(l) = (2 \sum_{j=1}^p V_{jl}^2)^{-1/2} \sum_{j=1}^p (mB_{jl}^2 - V_{jl}),$$
>     where
>     $V_{jl}$ is  $(U^{(23)\top}U^{(23)})_{ll}^{-1}c_j$ and $c_j$  is the j,j-th entry of $(\mathrm{Cov}(\hat{{E}}))$.
>     Under regular assumptions, the following conclusions hold:
>
>     1.  Under $H_0$, $s(l) \to \mathcal N(0,1)$.
>
>     2.  Under $H_1$, denote the number of non-zero entries in
>         $B_{\cdot, l}$ as $p^{1-\beta}$. If
>         $B_{\cdot, l}=\Omega(m^{-1/2} (\log p)^\alpha)$ holds
>         uniformly with $\alpha >0$ and $0 < \beta <1/2$, then
>         $s(l) \to \infty$. Therefore, asymptotically the testing
>         procedure achieves full power.
>
>     Consequently, for each $0 \leq l \leq k_2 + k_3$, we compare the
>     observed value of $s(l)$ to the standard normal distribution. This
>     allows us to infer whether each column of $U^{(23)}$ belongs to
>     $U^{(2)}$ or $U^{(3)}$, thereby enabling simultaneous estimation of
>     $U^{(2)}$, $U^{(3)}$, as well as the corresponding dimensions $k_2$
>     and $k_3$.
>
> To demonstrate the effectiveness of our proposed latent dimension
> selection method, we conducted a simulation study with data size
> $n = p = 500$, $m=5000$ and true latent dimensions
> $k_1 = k_2 = k_3 = 4$. All embeddings and parameters were generated
> according to the simulation settings described in Section B.3 of the
> Appendix. We applied the HTT pipeline to the simulation data across 30
> repetitions, and the results are presented in the following table.
>
> |$k_1$|$k_2$|$k_3$|Freq|
> |---|---|---|---|
> |4|4|4|27|
> |4|3|4|1|
> |4|5|4|2|
>
> Table 4: Result of latent dimension selection across 30 repetitions.
>
> The results indicate that, in most cases, the HTT algorithm
> successfully recovers the true latent dimensions. We further apply this
> algorithm in our real data analyses. With the implementation details in
> Appendix B.4, we apply the HTT algorithm to all three datasets. The
> algorithm identified $k_1=5$, $k_2=0$, $k_3=6$ for the recipe dataset,
> $k_1=7$, $k_2=0$, $k_3=2$ for the patient profile dataset, and $k_1=2$,
> $k_2=7$, $k_3=8$ for the co-citation hypergraph. The error metrics for
> the three datasets are presented in the tables below. Here, a-FED refers
> to a variant of FED that we introduce in response to Reviewer Suk1; a
> detailed description of this metric is provided in Appendix B.5.
>
> |Method|$\Delta\mathcal{H}_v\downarrow$|$\Delta\mathcal{X}_m\downarrow$|$\Delta\mathcal{X}_v\downarrow$|FED$\downarrow$|a-FED$\downarrow$|
> |---|---|---|---|---|---|
> |ReLaSH-(5,0,2)|1.978|2.236|0.894|0.293|0.356|
> |ReLaSH$_c$-(5,0,2)|7.504|2.236|0.894|0.182|0.248|
> |ReLaSH-(5,0,6)|2.129|2.174|**0.820**|**0.003**|**0.048**|
> |ReLaSH$_c$-(5,0,6)|3.583|2.174|**0.820**|0.191|0.258|
> |ReLaSH-(5,0,16)|2.355|**1.533**|1.112|0.766|0.847|
> |ReLaSH$_c$-(5,0,16)|**1.847**|**1.533**|1.112|0.180|0.255|
>
> Table 5: Recipe generation results. Scales of $\Delta\mathcal{H}_v$, $\Delta\mathcal{X}_m$, $\Delta\mathcal{X}_v$, FED, a-FED are $10^{-3},10^{-2},10^{-2},10^{-1},10^{-1}$.
>
> |Method|$\Delta\mathcal{H}_v\downarrow$|$\Delta\mathcal{X}_m\downarrow$|$\Delta\mathcal{X}_v\downarrow$|FED$\downarrow$|a-FED$\downarrow$|
> |---|---|---|---|---|---|
> |ReLaSH-(2,2,2)|1.996|**8.578**|1.887|1.246|0.931|
> |ReLaSH$_c$-(2,2,2)|3.890|**8.578**|1.887|5.481|1.935|
> |ReLaSH-(2,7,8)|1.914|8.817|**1.886**|1.060|0.791|
> |ReLaSH$_c$-(2,7,8)|2.772|8.817|**1.886**|**0.947**|**0.706**|
> |ReLaSH-(8,8,8)|**1.626**|8.608|1.887|1.454|0.871|
> |ReLaSH$_c$-(8,8,8)|2.816|8.608|1.887|6.451|2.042|
>
> Table 6: Co-citation hypergraph generation results. Scales of $\Delta\mathcal{H}_v$, $\Delta \mathcal{X}_m$, $\Delta \mathcal{X}_v$, FED are $10^{-3},10^{-2},10^{-1},10^{-1}$.
>
> |Method|$\Delta\mathcal{H}_v\downarrow$|$\Delta\mathcal{X}_m\downarrow$|$\Delta\mathcal{X}_v\downarrow$|FED$\downarrow$|a-FED$\downarrow$|
> |---|---|---|---|---|---|
> |ReLaSH-(7,0,2)|3.260|**2.989**|**1.435**|0.532|1.084|
> |ReLaSH$_c$-(7,0,2)|27.794|**2.989**|**1.435**|**0.013**|**0.193**|
> |ReLaSH-(7,0,16)|**2.624**|3.681|1.655|11.738|2.047|
> |ReLaSH$_c$-(7,0,16)|6.230|3.681|1.655|9.049|1.574|
>
> Table 7: Patient profile generation results. Scales of $\Delta \mathcal{H}_v$, $\Delta \mathcal{X}_m$, $\Delta \mathcal{X}_v$, FED, a-FED are $10^{-4},10^{-3},10^{-1},10^{-2},10^{-1}$.

---

> ### Author Response · Authors · 2025-11-21
>
> For the patient profile dataset, our experiments already include results for ReLaSH-$(7,0,2)$ and ReLaSHc-$(7,0,2)$, with ReLaSHc-$(7,0,2)$ consistently achieving the best performance across all error metrics, which aligns with the chosen latent dimension. For the co-citation hypergraph generation task, ReLaSH-$(2,7,8)$ generally outperforms the other ReLaSH model variants. Accordingly, we have updated the generation samples presented in the paper to showcase synthetic papers generated by ReLaSH-$(2,7,8)$. Regarding the results for the recipe dataset, both ReLaSH-$(5,0,6)$ and ReLaSHc-$(5,0,16)$ demonstrate comparable performance, excelling in different aspects of the error metrics. This suggests that, in real-world applications, it is beneficial to apply the latent dimension selection algorithm as well as experiment with various latent dimension choices for ReLaSH. The synthetic dataset with the lowest error metrics can also be selected as optimal, since real-world data may lack a clear ground truth for latent dimensions. Nevertheless, our model demonstrates robustness, as different choices of latent dimensions can still produce high-quality synthetic results.
>
> **Q2. Regarding comparisons with more recent and stronger baseline methods.**
>
> **Response.** Thank you for your suggestion to include more recent and stronger baseline methods for comparison. We have conducted additional empirical studies with more baseline methods and present our results below.
>
> To the best of our knowledge, existing generative models designed for hypergraphs do not handle our data generation tasks (please refer to our response to Reviewer hHTL for more details), whereas tabular data generative models can be applied. Therefore, in addition to including well-established models such as GAN, VAE, diffusion models, and RealNVP, we have also incorporated several recent and high-performing tabular data generative models in our comparisons.
>
> As summarized in Table 1 of [3], most state-of-the-art tabular data synthesis models were previously tested on datasets with fewer than 100 features, and their training and sampling processes are generally time- and memory-intensive. Nevertheless, many of these models can handle mixed data types, which apply on our datasets. For our baseline comparisons, we include ForestDiffusion [4], TabPFGen [5], CTAB-GAN [6], CTAB-GAN+ [7], and CTGAN [8], all using their official implementations.
>
> To assess the computational resources required, we first trained each method on the recipe hypergraph, which contains 205 features. TabPFGen and CTGAN require approximately twice the runtime of ReLaSH, while CTAB-GAN requires about four times as long. CTAB-GAN+ and ForestDiffusion are the most computationally demanding, requiring several hours for training and generation. The error metrics for these experiments are shown below.
>
> |Method|$\Delta\mathcal{H}_v\downarrow$|$\Delta\mathcal{X}_m\downarrow$|$\Delta\mathcal{X}_v\downarrow$|FED$\downarrow$|
> |---|---|---|---|---|
> |ReLaSH-(5,0,2)|1.978|2.236|0.894|0.293|
> |ReLaSH$_c$-(5,0,2)|7.504|2.236|0.894|0.182|
> |ReLaSH-(5,0,6)|2.129|2.174|0.820|0.003|
> |ReLaSH$_c$-(5,0,6)|3.583|2.174|0.820|0.191|
> |ReLaSH-(5,0,16)|2.355|1.533|1.112|0.766|
> |ReLaSH$_c$-(5,0,16)|1.847|1.533|1.112|0.180|
> |Gau-Diff|2.375|2.154|4.256|0.802|
> |RealNVP|2.484|1.146|3.562|0.909|
> |WGAN|2.208|21.428|1.351|0.907|
> |VAE|21.587|9.883|5.180|11.553|
> |CTGAN|2.519|28.799|4.983|0.847|
> |ForestDiffusion|1.886|8.211|2.073|0.848|
> |TabPFGen|1.565|1.915|1.205|0.297|
> |CTAB-GAN|2.552|19.367|3.858|0.925|
> |CTAB-GAN+|2.488|8.330|3.821|0.898|
>
> Table 1: Recipe generation results. Scales of $\Delta\mathcal{H}_v$, $\Delta\mathcal{X}_m$, $\Delta\mathcal{X}_v$, FED are $10^{-3},10^{-2},10^{-1},10^{-1}$.
>
> Among the models, TabPFGen has good performance and is comparable to ReLaSH/ReLaSH$_c$, while the other tabular generative models perform similarly to the standard WGAN baseline. However, according to the official implementation, TabPFGen is limited to handling tabular datasets with at most 500 features, whereas both our patient profile and co-citationship datasets exceed 2,000 dimensions—demonstrating the advantages of ReLaSH  in terms of scalability. All the other reviewed baseline models fail to complete training on these larger datasets due to timeout or out-of-memory errors, highlighting the efficiency of ReLaSH for hypergraph data generation tasks.

---

> ### Author Response · Authors · 2025-11-21
>
> **Q3. Regarding sensitivity analysis of key hyperparameters.**
>
> **Response.** Thank you for the suggestion on sensitivity analysis of key hyperparameters. We study each group of hyperparameters as follows.
>
> For the latent dimensions $k_1$, $k_2$, and $k_3$, we conduct a sensitivity analysis with $m = n = p = 400$ and true $k_1 = k_2 = k_3 = 4$. All true embeddings and parameters follow the simulation setup in Section B.3 of the Appendix. We vary the latent dimensions and report mean embedding and generation errors over 20 repetitions in the tables below.
>
> **(a) Varying $k_1 = k_2 = k_3$**
>
> |$k_1$|$k_2$|$k_3$|$\Delta \mathcal{H}_m\downarrow$|$\Delta \mathcal{H}_v\downarrow$|$\Delta\mathcal{X}_m\downarrow$|$\Delta \mathcal{X}_v\downarrow$|FED$\downarrow$|ReLaSH$_c$ $\Delta \mathcal{H}_v\downarrow$|ReLaSH$_c$ FED$\downarrow$|
> |---|---|---|---|---|---|---|---|---|---|
> |2|2|2|0.0186|0.0074|0.0683|0.0522|0.4984|0.0472|0.5527|
> |4|4|4|**0.0175**|**0.0057**|**0.0514**|**0.0339**|**0.4392**|**0.0356**|**0.4158**|
> |6|6|6|0.0181|0.0064|0.0581|0.0449|0.5239|0.0405|0.5204|
>
> **(b) Varying $k_2$ with $k_1 = k_3 = 4$**
>
> |$k_1$|$k_2$|$k_3$|$\Delta \mathcal{H}_m\downarrow$|$\Delta \mathcal{H}_v\downarrow$|$\Delta\mathcal{X}_m\downarrow$|$\Delta \mathcal{X}_v\downarrow$|FED$\downarrow$|ReLaSH$_c$ $\Delta \mathcal{H}_v\downarrow$|ReLaSH$_c$ FED$\downarrow$|
> |---|---|---|---|---|---|---|---|---|---|
> |4|2|4|0.0182|0.0072|0.0639|0.0553|0.4672|**0.0304**|0.4821|
> |4|4|4|**0.0175**|0.0057|**0.0514**|**0.0339**|**0.4392**|0.0356|**0.4158**|
> |4|8|4|0.0179|**0.0053**|0.0576|0.0451|0.4404|0.0398|0.4927|
>
> **(c) Varying $k_3$ with $k_1 = k_2 = 4$**
>
> |$k_1$|$k_2$|$k_3$|$\Delta \mathcal{H}_m\downarrow$|$\Delta \mathcal{H}_v\downarrow$|$\Delta\mathcal{X}_m\downarrow$|$\Delta \mathcal{X}_v\downarrow$|FED$\downarrow$|ReLaSH$_c$ $\Delta \mathcal{H}_v\downarrow$|ReLaSH$_c$ FED$\downarrow$|
> |---|---|---|---|---|---|---|---|---|---|
> |4|4|2|0.0182|0.0069|0.0563|0.0432|0.4782|0.0428|0.5476|
> |4|4|4|**0.0175**|0.0057|0.0514|**0.0339**|**0.4392**|**0.0356**|**0.4158**|
> |4|4|8|0.0180|**0.0048**|**0.0497**|0.0392|0.4625|0.0388|0.4624|
>
> **(d) Varying $k_1$ with $k_2 = k_3 = 4$**
>
> |$k_1$|$k_2$|$k_3$|$\Delta \mathcal{H}_m\downarrow$|$\Delta \mathcal{H}_v\downarrow$|$\Delta\mathcal{X}_m\downarrow$|$\Delta \mathcal{X}_v\downarrow$|FED$\downarrow$|ReLaSH$_c$ $\Delta \mathcal{H}_v\downarrow$|ReLaSH$_c$ FED$\downarrow$|
> |---|---|---|---|---|---|---|---|---|---|
> |2|4|4|0.0179|0.0072|0.0648|0.0582|0.4872|0.0397|0.5623|
> |4|4|4|**0.0175**|**0.0057**|**0.0514**|0.0339|**0.4392**|**0.0356**|**0.4158**|
> |8|4|4|0.0176|0.0063|0.0587|**0.0322**|0.4613|0.0384|0.4729|
>
> Overall, generation performance is best when the true latent dimensions are used. Misspecifying them increases error but still yields comparable performance. The error patterns follow the role of each component: varying $k_3$ (with $k_1,k_2$ fixed) mainly affects hyperlink-related errors, while varying $k_1$ mainly affects attribute-related errors. Changing $k_2$ increases both types of errors, as $u^{(2)}$ is shared by the two parts.
>
> FED is more sensitive than moment-based metrics, likely because of its more complex computation. Nonetheless, moderate deviations in $k_1,k_2,k_3$ do not substantially degrade performance and remain competitive with baselines.
>
> For the tuning parameter $\lambda$, Section B.2 of the Appendix (Figures 15–16) shows that varying $\lambda$ has little impact on estimation error. We mainly use $\lambda$ to stabilize and balance the hyperlink and attribute likelihoods during embedding and to speed up computation. Its choice can also be guided by the theoretical error rates in Theorem 4 (see Remark 3 in Appendix A).
>
> For diffusion steps, we do not fix a preset number. Instead, we use early stopping based on the generator loss: training stops if there is no improvement for 20–30 epochs, a standard patience choice. This works well across settings, avoids manual tuning of training length, and helps prevent overfitting. Further details are in Section B.7 of the Appendix.
>
> Finally, for selecting $k_1,k_2,k_3$ in real data analysis, please see our response to Q1, where we provide an inference-based perspective on choosing latent dimensions.

---

> > ### Author Response · Authors · 2025-11-21
> >
> > **References**
> >
> > [1] Li, Jinming, Gongjun Xu, and Ji Zhu. “High-dimensional Factor Analysis for Network-linked data.” *Biometrika* (2025): asaf012.
> >
> > [2] Tracy, Craig A., and Harold Widom. “On orthogonal and symplectic matrix ensembles.” *Communications in Mathematical Physics* 177(3) (1996): 727–754.
> >
> > [3] Stoian, Mihaela CÄ, Eleonora Giunchiglia, and Thomas Lukasiewicz. “A survey on tabular data generation: Utility, alignment, fidelity, privacy, and beyond.” arXiv:2503.05954 (2025).
> >
> > [4] Jolicoeur-Martineau, Alexia, Kilian Fatras, and Tal Kachman. “Generating and imputing tabular data via diffusion and flow-based gradient-boosted trees.” *AISTATS* (PMLR), 2024.
> >
> > [5] Ma, Junwei, et al. “TabPFGen – Tabular Data Generation with TabPFN.” arXiv:2406.05216 (2024).
> >
> > [6] Zhao, Zilong, et al. “CTAB-GAN: Effective table data synthesizing.” *Asian Conference on Machine Learning* (PMLR), 2021.
> >
> > [7] Zhao, Zilong, et al. “CTAB-GAN+: Enhancing tabular data synthesis.” *Frontiers in Big Data* 6 (2024): 1296508.
> >
> > [8] Xu, Lei, et al. “Modeling tabular data using conditional GAN.” *NeurIPS* 32 (2019).

---

### Author Response · Authors · 2025-12-03
**Comment to Area Chair**

Dear Area Chair,

We are writing regarding the recent incident involving “leaked reviewer/AC identities.” We understand the seriousness of this issue and that it has created significant additional workload for you. Below, we briefly summarize the status of our submission for your convenience; further details are provided in our point-by-point rebuttal.

Our original scores were “**6, 6, 6, 4 (Avg: 5.5)**.” All reviewers acknowledged the novelty and contribution of our paper and provided specific, constructive comments, which we have carefully addressed point by point.

Before the incident occurred, two reviewers had already raised their scores: one from **4 to 6** and another from **6 to 8**, as documented in their responses to our rebuttal. **The remaining two reviewers did not have the opportunity to respond before the discussion phase closed**. Nevertheless, **we have addressed all of their concerns and questions** in our rebuttal, and we have substantially revised the paper to reflect these improvements. We believe these changes significantly enhance both the quality and the contribution of our work.

We sincerely thank you for your time and effort in handling our paper, and we hope that our efforts during the rebuttal phase can be taken into consideration.

Best regards,

---

### Meta-Review · Area_Chair_gyF7 · 2026-01-06

**Summary:**

This paper introduces a generative framework for hypergraphs with hyperlink attributes, which decouple hypergraph generation into three stages: (1) a likelihood-based joint embedding of hypergraph structure and attributes, (2) a distribution-free latent space generator, and (3) likelihood-based decoding back to the original space

The reviewers generally agree on the following strengths.
- The idea of embedding hypergraph structure and hyperlink attributes into a joint latent space is novel and well motivated.
- The proposed method is supported by solid theoretical analysis.
The method demonstrates effectiveness across diverse domains and settings.

The main concerns raised by the reviewers include
- the omission of closely related recent studies and advanced baselines.
- the lack of clear guidance on hyperparameter selection
- the validity and interpretation of the evaluation metrics.

During the rebuttal phase, the authors addressed most of these concerns, and several reviewers explicitly noted that their major concerns had been satisfactorily resolved.

Overall, the paper makes a strong contribution, and the reviewer consensus supports acceptance.

**Reviewer Concerns:**

As far as the AC can see, all concerns have been largely addressed.

**Reviewer Scores:**

It is likely that most reviewers will increase their scores at least slightly. Notably, two reviewers explicitly stated their intention to do so, although this is not the primary basis for the AC’s decision.

---

### Decision · Program_Chairs · 2026-01-26

Accept (Poster)